# Long-term historical trends in air pollutant emissions in Asia: Regional Emission inventory in ASia (REAS) version 3

Junichi Kurokawa[1], Toshimasa Ohara[2]

[1] Asia Center for Air Pollution Research, 1182 Sowa, Nishi-ku, Niigata, Niigata, 950-2144, Japan
[2] National Institute for Environmental Studies, 16-2 Onogawa, Tsukuba, Ibaraki, 305-8506, Japan

*Correspondence to*: Junichi Kurokawa (kurokawa@acap.asia)

**Abstract.** A long-term historical emission inventory of air and climate pollutants in East, Southeast, and South Asia from 1950-2015 was developed as the Regional Emission inventory in ASia version 3 (REASv3). REASv3 provides details of emissions from major anthropogenic sources for each country and its sub-regions and also provides monthly gridded data with $0.25° \times 0.25°$ resolution. The average total emissions in Asia during 1950-1955 and from 2010-2015 (growth rates in these 60 years estimated from the two averages) are as follows: $SO_2$: 3.2 Tg, 42.4 Tg (13.1); $NO_x$: 1.6 Tg, 47.3 Tg (29.1); CO: 56.1 Tg, 303 Tg (5.4); non-methane volatile organic compounds: 7.0 Tg, 57.8 Tg (8.3); $NH_3$: 8.0 Tg, 31.3 Tg (3.9); $CO_2$: 1.1 Pg, 18.6 Pg (16.5) ($CO_2$ excluding biofuel combustion 0.3 Pg, 16.8 Pg (48.6)); $PM_{10}$: 5.9 Tg, 30.2 Tg (5.1); $PM_{2.5}$: 4.6 Tg, 21.3 Tg (4.6); black carbon: 0.69 Tg, 3.2 Tg (4.7); and organic carbon: 2.5 Tg, 6.6 Tg (2.7). Clearly, all the air pollutant emissions in Asia increased significantly during these six decades, but situations were different among countries and regions. Due to China's rapid economic growth in recent years, its relative contribution to emissions in Asia has been the largest. However, most pollutant species reached their peaks by 2015 and the growth rates of other species was found to be reduced or almost zero. On the other hand, air pollutant emissions from India showed an almost continuous increasing trend. As a result, the relative ratio of emissions of India to that of Asia have increased recently. The trend observed in Japan was different from the rest of Asia. In Japan, emissions increased rapidly during the 1950s-1970s, which reflected the economic situation of the period; however, most emissions decreased from their peak values, which were approximately 40 years ago, due to the introduction of control measures for air pollution. Similar features were found in Republic of Korea and Taiwan. In the case of other Asian countries, air pollutant emissions generally showed an increase along with economic growth and motorization. Trends and spatial distribution of air pollutants in Asia are becoming complicated. Datasets of REASv3, including table of emissions by countries and sub-regions for major sectors and fuel types, and monthly gridded data with $0.25° \times 0.25°$ resolution for major source categories are available through the following URL: http://www.nies.go.jp/REAS/.

## 1 Introduction

With an increase in demand for energy, motorization, and industrial and agricultural products, air pollution from anthropogenic emissions is becoming a serious problem in Asia, especially due to its impact on human health. In addition, a

significant increase in anthropogenic emissions in Asia is considered to affect not only the local air quality, but also regional, inter-continental, and global air quality and climate change. Therefore, reduction in air and climate pollutants emissions are urgent issues in Asia (UNEP, 2019). Short-Lived Climate Pollutants (SLCPs), which are gases and particles that contribute to warming and have short lifetimes, have been recently considered to play important roles in the mitigation both air pollution and climate change (UNEP, 2019). SLCPs such as black carbon (BC) and ozone are warming agents, which cause harm to people and ecosystems. A decrease in the emissions of BC and ozone precursors from fuel combustion led to the decrease of other particulate matter (PM) species, such as sulfate and nitrate aerosols. Even though this is a positive step for human health, it has a negative effect on global warming as sulfate and nitrate aerosols act as cooling agents in the troposphere. Therefore, to find effective ways to mitigate both air pollution and climate change, accurate understanding of the current status and historical trends of air and climate pollutants are fundamentally important.

Recently, Hoesly et al. (2018) developed a long-term historical global emission inventory from 1750 to 2014 using the Community Emission Data System (CEDS). This data set is used as input data for the Coupled Model Intercomparison Project phase 6 (CMIP6). The Emission Database for Global Atmospheric Research (EDGAR) also provides global emissions data of both air pollutants and greenhouse gases, with the current version 4.3.2 ranging from the period between 1970-2012 (Crippa et al., 2016). The EDGAR is used as the default data of input emissions for the Task Force on Hemispheric Transport of Air Pollution phase 2 (HTAPv2) (Janssens-Maenhout et al., 2015). For SLCPs, the European Union's Seventh Framework Programme project ECLIPSE (Evaluating the Climate and Air Quality Impact of Short-Lived Pollutants) developed a global emission inventory based on the GAINS model. Current version 5 provides gridded emissions for every five years from 1990 to 2030 and also from 2040 to 2050 (Stohl et al., 2015). However, data from Asia in global emission inventories are generally based on limited country specific information. For the Asian region, several project-based emission inventories are developed, such as Transport and Chemical Evolution over the Pacific (TRACE-P) field campaigns (Streets et al., 2003a, b) and its successor mission, that is Intercontinental Chemical Transport Experiment-Phase B (INTEX-B) (Zhang et al., 2009). Recently, the MIX inventory (mosaic Asian anthropogenic emission inventory) was developed as input emission data sets for the Model Intercomparison Study for Asia (MICS-Asia) Phase 3 by a mosaic of up-to-date regional emission inventories. The MIX inventory is also a component of the HTAPv2 inventory (Li et al., 2017a). For national emission inventories, numerous studies, research papers, and reports have been published. MEIC (Multi-resolution Emission Inventory for China) developed by Tsinghua University is a widely used emission inventory database for China (Zhang et al., 2009; Li et al., 2014; Zheng et al., 2014, Liu et al., 2015) and is included in the MIX inventory. Zhao et al. (2011, 2012, 2013, and 2014) developed recent and projected emission inventories of air pollutants in China. In addition, research papers for regional emission inventories of China were also published recently (Zhu et al., 2018; Zheng et al., 2019a). In the case of India, Garg et al., (2006) developed a historical emission inventory of air pollutants and greenhouse gases from 1985 to 2005. For recent years, Sadavarte and Venkataraman (2014) developed multi-pollutant emission inventories for industry and transport sectors and Pandey et al. (2014) developed the same for domestic and small industry sectors for the same time period, that is 1996-2015. For Japan, several project-based emission data sets were developed, such

as the Japan Auto-Oil Program (JATOP) Emission Inventory-Data Base (JEI-DB) (JPEC 2012a, b, c; 2014), East Asian Air Pollutant Emission Grid Database (EAGrid) (Fukui et al., 2014), and emission data sets for Japan's Study for Reference Air Quality Modeling (J-STREAM) (Chatani et al., 2018). In addition, there are studies for other countries and regions, such as the Clean Air Policy Support System (CAPSS) for Republic of Korea (Lee et al., 2011), Thailand (Thao Pham et al., 2008), Indonesia (Permadi et al., 2017), and Nepal (Jayarathne et al., 2018; Sadavarte et al., 2019). However, these regional and national emission inventories in Asia are available for a limited period, with data of the past missing.

The authors of this study have been devoted in developing the Regional Emission inventory in ASia (REAS) series. First version of REAS (REASv1.1) were developed by Ohara et al. (2007), which accounted for actual emissions during 1980-2003 and projected ones in 2010 and 2020. Kurokawa et al. (2013) updated the inventory in REASv2.1, which focused on the period between 2000-2008 when emissions in China drastically increased. REASv2.1 is used as the default data of the MIX inventory. In this study, a long historical emission inventory in the Asian region from 1950-2015 has been newly developed as REAS version 3 (REASv3). This study provides methodology, results and discussion of REASv3.1. Section 2 gives the basic methodology, including collecting activity data, settings of emission factors and removal efficiencies, and spatial and temporal allocation of emissions to create monthly gridded data sets of REASv3. In Section 3.1, trends in air pollutants emissions in Asia are described in detail and effects of emission controls on emissions in China and Japan are discussed. Spatial and temporal distributions are overviewed in Section 3.2. Section 3.3 compares the results of REASv3.1 with other emission inventories. Uncertainties of REASv3.1 are discussed in Section 3.4. Finally, summary and remarks are presented in Section 4.

## 2 Methodology and data

### 2.1 General description

Table 1 summarizes the general information of REASv3. Major updates from previous versions are as follows:

- Target years are from 1950 to 2015 covering much longer periods than REASv1.1 (1980-2003) and REASv2.1 (2000-2008).
- The long historical data sets of activity data were developed by collecting international and national statistics and related proxy data.
- Emission factors and information of emission controls especially for China and Japan were surveyed from research papers of emission inventories in Asia and related literatures.
- Large power plants constructed after 2008 were added as new point sources.
- Allocation factors for spatial and temporal distribution were updated although several emission inventories developed by other research works were utilized (see Table 2).
- Emissions from Japan, Republic of Korea, and Taiwan were originally estimated except for NMVOC evaporative sources (see Table 2).

REASv3 focuses on the long historical trends of air pollutants emissions in Asia. The start year was chosen to be 1950 as severe air pollution in Japan started from the mid-1950s. For the emission inventory framework, there are two major changes from REASv2.1. One is the target species. REASv3 includes the following major air and climate pollutants: $SO_2$, $NO_x$, CO, non-methane volatile organic compounds (NMVOC), $NH_3$, $PM_{10}$, $PM_{2.5}$, BC, organic carbon (OC), and $CO_2$. However, $CH_4$, and $N_2O$ that were included in REASv2.1 are not in the scope of this version. $CH_4$ is one of important components of SLCP and will be considered in the next version. Another is the target areas. Figure 1 shows the inventory domain of REASv3 which includes East, Southeast, and South Asia. China, India, and Japan have been divided into 33, 17, and 6 regions, respectively to reduce the uncertainties in the spatial distribution. Definition of the sub-regions are the same as for REASv2.1. In REASv3, Central Asia and the Asian part of Russia, which were target areas of REASv2.1 are not included because of the difficulty in collecting necessary data for estimating long historical emissions in these areas. The source categories considered in REASv3 are the same as those in REASv2.1. Major sources include fuel combustion in power plants, industry, transport, and domestic sectors. Non-combustion sources include industrial process, evaporation (NMVOC), and agricultural activities ($NH_3$). However, $NO_x$ emissions from soil as well as from international and domestic aviation and navigation, including fishing ships are exceptions. They were not included in REASv3. The spatial and temporal resolution are the same as those of REASv2.1. Spatial resolution is $0.25° \times 0.25°$, except in the case of large power plants, which are treated as point sources. Temporal resolution is monthly.

In REASv3, most emissions were originally estimated. However, several emission inventories from other research works and officially opened data were utilized as summarized in Table 2. NMVOC emissions in Japan and Republic of Korea from evaporative sources were obtained from the Ministry of the Environment of Japan (MOEJ, 2017) and the National Air Pollutants Emission Service of the National Institute of Environmental Research (available at http://airemiss.nier.go.kr/mbshome/mbs/airemiss/index.do), respectively. For $NH_3$ emissions from agricultural activities, data of base year (2000 and 2005 for Japan and 2000 for others) were obtained from other research works as follows (see Sect. 2.4): REASv2.1 (Kurokawa et al., 2013: JPEC 2012a, b, c; 2014) for Japan and REASv1.1 (Yamaji et al., 2004; Yan et al., 2003) for other counties and regions. In addition, EDGARv4.3.2 were utilized to create grid allocation factors for road transport sector for all species and manure management for $NH_3$ (see Sects. 2.4.1 and 2.6, respectively).

In the following sub-sections, general methodologies and data used in REASv3 are overviewed for stational sources, road transport, agricultural sources, other sources, and spatial and temporal distribution. Details of the methodologies such as data sources and treatments, settings of emission factors and emission controls, and related assumptions are provided in the supplement document entitled "Supplementary information and data to methodology of REASv3" (hereafter, this document is expressed as "the Supplement"). In Sect. S2 of the Supplement, details of frame work of REASv3 including definitions of sub-categories of emission sources, and target countries and sub-regions of China, India, and Japan was provided.

### 2.2 Stationary sources

#### 2.2.1 Basic methodology

Emissions from stationary fuel combustion and industrial processes are traditionally calculated using activity data and emission factors, including the effect of control technologies. In order to increase the accuracy of estimation and to analyze the effects of abatement measures, emissions should be calculated using information on technologies related to emission sources as much as possible. In REASv3, emissions from stationary combustion and industrial processes are estimated based on the following equation:

$$E = \sum_i \sum_j \sum_{k,l} \{A_{i,j} \times F_{i,j,k,l} \times EF_{i,j,k} \times (1 - R_{i,j,l})\} \tag{1}$$

where, $E$ represents emission, $i$ is the type of activity data, $j$ is the type of sector category, $k$ is the type of technology related to emission factor, $l$ stands for the control technology after emission, $A$ is amount of activity data, $EF$ is the emission factor of each technology, $R$ is the removal efficiency of each technology, and $F$ is the fraction rate of activity data for combination of $i$, $j$, $k$, and $l$. When $SO_2$ emissions from combustion sources are estimated using sulfur contents of fuels, $EF_{i,j,k}$ in eq. (1) is

calculated, as follows:

$$EF_{i,j,k} = NCV_{i,j} \times S_{i,j} \times (1 - SR_{i,j,k}) \times 2 \tag{2}$$

where, $NCV$ is the net calorific value of fuel, $S$ is the sulfur content of fuel, and $SR$ is the sulfur retention in ash for combination of $i$, $j$, and $k$. $2$ is a factor to convert the value of S to $SO_2$.

Unfortunately, in the case of Asia, information available on emission factors and removal efficiencies is limited. Even

though there is information on the introduction rates of technologies both for emission factors and removal efficiencies, they are available independently. Therefore, for most cases, an average of the removal efficiencies is calculated using the values of each abatement equipment and its penetration rate. Then, the average removal efficiencies are commonly used to calculate the emission factors of each technology.

Note that several sub-sectors in stationary sources such as coke production and cement industry include both combustion and

non-combustion emission sources. See Sects. S2.4.1 and S2.4.2 of the Supplement for details.

#### 2.2.2 Activity data

Fuel consumption is the core activity data of the emission inventory of air pollutants and greenhouse gases. For most countries, the amount of energy consumption for each fuel type and sector was primarily obtained from the International Energy Agency (IEA) World Energy Balances (IEA, 2017). For China, province-level tables in the China Energy Statistical

Yearbook (CESY) (National Bureau of Statistics of China, 1986, 2001-2017) were used. For countries and regions whose energy data are not included in IEA (2017), fuel consumption data were taken from the United Nations (UN) Energy Statistics Database (UN, 2016) and the UN data, which is a web-based data service of the UN (http://data.un.org/). See Sect. S.3.1.1 of the Supplement for definition of fuel types.

One major obstacle in this study was collecting activity data for the entire target period of REASv3, that is from 1950-2015.
IEA (2017) includes data from Japan during 1960-2015 and those from other countries during 1971-2015; however, for many countries, fuel types and sector categories, the oldest years when data exist are more later than 1971. Furthermore, past data for sectors do not contain as many categories. For example, coal consumption data in detailed sub-categories of the industrial sector existed in Indonesia only after 2000, but corresponding data are only available for industry total before 1999. In this case, relative ratios of fuel consumption in detailed sub-categories to total industry in 2000 were used to distribute the
total industry data to each sub-category for the years before 1999. This procedure is performed for similar cases for all sectors and sometimes for total final consumption. In cases where data did not exist beyond a certain year, fuel consumption data were extrapolated using trends of related data for each sub-category. For example, power generation and amount of industrial products were used to observe trends of fuel consumption in power plant and each industry's sub-category, respectively. Data for long historical trends were obtained from a variety of sources. For example, power generation data and
amounts of major industrial products were obtained from Mitchell (1998) and national and international statistics as well as related literatures were surveyed. See Sect. S3.1.2 of the Supplement for details of data sources of fuel consumption and assumptions to estimate missing historical data. For China, data of CESY for each province were available from 1985 to 2015. During 1950-1984, first, total energy data in China were developed based on IEA (2017) and then, fuel consumption in each province was extrapolated using the total data of China in each fuel type and sector category. See Sect. S3.1.3 of the
Supplement for details of regional fuel consumption data in China. For countries which used Energy Statistics Database, fuel consumption of each fuel and sector was taken from the UN data (available at http://data.un.org/) for the period between 1990-2015 and was extrapolated using the trend of total consumption of each fuel type obtained from the UN Energy Statistics Database.

As described in Section 2.1, India and Japan have 17 and 6 sub-regions, respectively. Therefore, for them, country total data
of IEA (2017) need to be divided for each sub-region. For Japan, energy consumption statistics of each prefecture that were obtained from the Agency for Natural Resources and Energy (available at https://www.enecho.meti.go.jp/statistics/energy_consumption/ec002/results.html) were used as default weighting factors to allocate country total data to the six regions. Similarly, for India, default weighting factors for regional allocation were estimated from TERI (The Energy and Resources Institute) Energy & Environment Data Diary and Yearbook (TERI, 2013,
2018), Annual Survey of Industries (Ministry of Statistics & Programme Implementation, available at http://www.csoisw.gov.in/cms/en/1023-annual-survey-of-industries.aspx), and Census of India (Chandramouli, 2011), among others. In general, details of these weighting factors are less than those of the country's total fuel consumption. In addition, these data are not available for all the years during 1950-2015. Therefore, regional allocation factors for some sectors were developed independently if corresponding proxy data were available. For the power plant sector, generation
capacities of each region and year were calculated as proxy data using the World Electric Power Plants Database (WEPP) (Platts, 2018). For India, traffic volumes (see Section 2.3.1) and amount of industrial production in each region (see the last

paragraph of this section) were used as proxy data. Details of regional fuel consumption data in India and Japan were provided in Sects. S3.1.4. and S3.1.5, respectively.

Similar to REASv2.1, large power plants are treated as point sources in REASv3 and are updated based on REASv2.1
database. Before 2007, power plants that were classified as point sources were the same as those in REASv2.1 and their information, such as generating capacities, and start and retire years were updated using WEPP. During 2000 to 2007, fuel consumption data were the same as that in REASv2.1. In REASv3, power plants whose start years were after 2007 and generation capacities were larger than 300 MW were added as new point sources. Fuel consumption of new power plants were estimated based on relations between fuel consumption amounts and generation capacities of the point data in
REASv2.1. If the (A) total fuel consumption of each power plant in a country is larger than (B) the corresponding data in power plant sector, values of each power plant were adjusted by ratios of (B) per (A). If (B) was larger than (A), differences between (B) and (A) were treated as data of area sources. See also Sect. S3.1.6 of the Supplement for fuel consumption data in power plants.

For emissions from industrial processes, activity data included amount of industrial products. Corresponding data were
mainly obtained from related international statistics and national statistics. For example, iron and steel production data were taken from Steel Statistical Yearbook (World Steel Association, 1978-2016) and data for non-ferrous metals and non-metallic minerals were obtained from the United States Geological Survey (USGS) Minerals Yearbook (USGS, 1994-2015). Brick production data were obtained from a variety of sources, such as Zhang (1997), Maithel (2013), Klimont et al. (2017), and the UN data. For China and India, the authors also used internet database services, namely China Data Online
(https://www.china-data-online.com/) and Indiastat (https://www.indiastat.com/), respectively, which provided both national and regional statistics. The USGS Minerals Yearbook (USGS, 1994-2015) also provided information on plants in each sub-region of China, India, and Japan. Data in the aforementioned statistics were not available for the early years of the target period of REASv3.1. In such cases, data of Mitchell (1998) were used as factors to extrapolate the activity data until 1950. Details of activity data related to industrial production and other transformation were described in Sect. S4.1 of the
Supplement.

### 2.2.3 Emission factors

Setting up of emission factors and removal efficiencies for stationary combustion and industrial processes are difficult procedures, especially for a long historical emission inventory. In this study, emission factors without effects of abatement measures were set, which were used for the entire target period of REASv3. Then, effects of control measures were set
considering their temporal variations, both for abatement measures before emissions such as using low sulfur fuels and low $NO_x$ burners and those after emissions such as flue gas desulfurization (FGD) and electrostatic precipitator (ESP). These settings were done for each country and region based on country and region-specific information. However, such information is still limited, especially in the Asian region. Therefore, default values of unabated emission factors were selected and default removal efficiencies were set to zero. Then, these default values were updated in case information and

literature on each country and region were available. For default emission factors, a majority of settings was continuously used from REASv2.1, but some of them, including effects to control measures (net emission factors) were changed to unabated emission factors. Default emission factors were mainly obtained from Kato and Akimoto (1992) for $SO_2$ and $NO_x$; Bond et al. (2004), Kupiainen and Klimont (2004), and Klimont et al. (2002, 2017) for PM species; the 2006 IPCC Guidelines for National Greenhouse Gas Inventories (IPCC, 2006) for $CO_2$; and the AP-42 (US EPA, 1995), the Global Atmospheric Pollution Forum Air Pollutant Emission Inventory Manual (SEI, 2012), Shrestha et al. (2013), the EMEP/EEA emission inventory guidebook 2016 (EEA, 2016), and other literatures for others.

For country and region-specific settings, in addition to literatures used in REASv2.1 (see Kurokawa et al., 2013), new information, especially for technologies related to settings of emission factors and removal efficiencies was surveyed. Although such information is still limited in Asia, the volume of accessible information on China is relatively large. General information on China in recent years was mainly obtained from Li et al. (2017b) and Zheng (2018). Introduction rates of technologies were obtained from Hua et al. (2016) for cement, Wu et al. (2017) for iron and steel, Huo et al. (2012a) for coke ovens, and Zhao et al. (2013, 2014, and 2015) for a variety of sources. For India, information for technology settings was mainly taken from Sadavarte and Venkataraman (2014), Pandey et al. (2014), Guttikunda and Jawahar (2014), and Reddy and Venkataraman (2002a). For power plants, WEPP database has elements for installed equipment to control $SO_2$, $NO_x$, and PM species which were used for settings of emission factors and removal efficiencies of power plants treated as point sources. However, these data are not available for most power plants, especially in Asia. Therefore, in the case of South and Southeast Asia, a variety of literatures, such as Sloss (2012) and UN Environment (2018) were referred to, to set emission factors and removal efficiencies. For Japan, introduction of control technologies for air pollutants were initiated earlier than other countries in Asia. A lot of domestic reports for air pollution and control technologies in power and industry plants published in Japanese, such as MRI (2015), Shimoda (2016), Suzuki (1990), and Goto (1981) were referred to, to determine emission factors, removal efficiencies, and their temporal variations.

Details of emission factors and settings of emission controls for stationary combustion sources were provided in Sect. S3.2 of the Supplement. Those for stationary non-combustion emissions from industrial production and other transformation sectors were described in Sect. S4.2. Activity data and emission factors of NMVOC from chemical industry were obtained from Sects. S5.1.5 and S5.2.5, respectively. Those for $NH_3$ emissions from industrial production were provided in Sect S8.3.

### 2.3 Road transport

### 2.3.1 Basic methodology

Methodology for road transport sector is the same as that of REASv2.1. Equations to estimate hot and cold start emissions (except for $SO_2$ and $CO_2$) are, as follows:

$$E_{HOT} = \sum_i \{ NV_i \times ADT_i \times EF_{HOTi} \} \qquad (3)$$

where, $E_{HOT}$ is the hot emission, $i$ is the vehicle type, $NV$ is the number of vehicles in operation, $ADT$ is the annual distance traveled, and $EF_{HOT}$ is the emission factor. $SO_2$ emissions are calculated using sulfur contents in gasoline and diesel consumed in road transport sector, assuming sulfur retention in ash is zero. $CO_2$ emissions are estimated by calculating the consumption amounts of fuels (gasoline, diesel, liquefied petroleum gas, and natural gas) and the corresponding emission factors (IPCC, 2006). Details for $SO_2$ and $CO_2$ from road transport were described in S6.2.3 of the Supplement.

Cold start emissions ($E_{COLD}$) are estimated for $NO_x$, CO, $PM_{10}$, $PM_{2.5}$, BC, OC, and NMVOC using the following equation:

$$E_{COLD} = \sum_i \{NV_i \times ADT_i \times EF_{HOTi} \times \beta_i(T) \times F_i(T)\} \qquad (4)$$

where, $\beta$ is the fraction of distance traveled driven with a cold engine or with the catalyst operating below the light-off temperature, and $F$ is the correction factor of $EF_{HOT}$ for cold start emission. $\beta$ and $F$ are functions of temperature $T$ and are taken from EEA (2016) (See Sect. S6.2.1 of the Supplement for additional information of the settings). For Japan, the ratios of cold start and hot emissions for each vehicle type were estimated from the JEI-DB. Then, cold start emissions were calculated by hot emissions and the ratios for each vehicle type. In REASv3, effects of regulations on cold start emissions were ignored and need to be considered in the next version.

For evaporation from gasoline vehicles, emissions ($E_{EVP}$) were estimated using the following equation of Tier 1 of EEA (2016):

$$E_{EVP} = \sum_i \{NV_i \times EF_{EVPi}(T)\} \qquad (5)$$

where, $EF_{EVP}$ is the emission factor as a function of temperature. For Japan, evaporative emissions in 2000, 2005, and 2010 were obtained from the JEI-DB and those between 2000 (2005) and 2005 (2010) were interpolated. For emissions before 2000 and after 2010, emissions from running loss were extrapolated using trends of traffic volume and those from hot soak loss and diurnal breaking loss were extrapolated by trends of vehicle numbers. See Sect. S6.3 of the Supplement for the NMVOC evaporative emissions.

### 2.3.2 Activity data

Basic activity data of road transport sector include number of vehicles in operation for each type. Data on the registered number of vehicles are available in the national statistics of each country and the World Road Statistics (IRF, 1990-2018). If these statistics did not contain data until 1950, the numbers were extrapolated using trends of data for aggregated vehicle categories in Mitchell (1998). For China, data for each sub-region were obtained from China Statistical Yearbook (National Bureau of Statistics of China, 1986-2016) and the China Data Online. Those for India were taken from TERI Energy & Environment Data Diary and Yearbook (TERI, 2013, 2018) and the Indiastat. A problem that was encountered was that registered vehicles were not always in operation. For India, the number of vehicles obtained as registered vehicles were corrected based on Baidya and Borken-Kleefeld (2009) and Prakash and Habib (2018). For other countries, the number of registered vehicles were considered as those in operation due to lack of information. In addition, to estimate emissions, these numbers must be further divided into vehicles based on each fuel type. However, such information is not easily available in

national statistics. In this study, settings of Streets et al. (2003a) and REASv2.1 were used as default and were updated if
national information was available, such as He et al. (2005), Yan and Crookes (2009), Sahu et al. (2014), and Malla (2014). If the number of LPG and CNG vehicles were available only for recent years, data were extrapolated using amounts of fuel consumption in road transport sector in IEA (2017).

Emission factors of road transport sector used in this study were given as emission amounts per traffic volumes. Therefore, annual vehicles kilometer traveled (VKT) per each vehicle type need to be set for each country. We used data of Clean Air Asia (2012) for many countries. Clean Air Asia (2012) includes data for China and India, but data of China were estimated based on Huo et al. (2012b) and those of India were set after Prakash and Habib (2018) and Pandey and Venkataraman (2014). For Japan, the total annual VKT for detailed vehicle types were obtained from reports of Pollutants Release and Transfer Register published by the Ministry of Economy Trade and Industry until 2001 (METI, 2003-2017), which was originally estimated from Road Transport Census of Japan developed by the Ministry of Land, Infrastructure, Transport and Tourism. Before 2001, the total annual VKT was extrapolated using data of more aggregated vehicle categories in the Annual Report of Road Statistics (MLIT, 1961-2016) until 1960 and from the Historical Statistics of Japan (Japan Statistical Association, 2006) until 1950.

Details of number of vehicles and annual vehicles kilometer traveled were described in Sect S6.1.1 of the Supplement.

### 2.3.3 Emission factors

For most countries, road transport is one of major causes of air pollution. In many Asian countries, vehicle emission standards were introduced after the late 1990s and were strengthened in phases (Clean Air Asia, 2014). Therefore, for road vehicles, year-to-year variation of emission factors must be taken into considered for a long historical emission inventory. In REASv3, emission factors of $NO_x$, CO, NMVOC, and PM species for exhaust emissions from road vehicles were estimated by following procedures:

1. Emission factors of each vehicle type in a base year were estimated.
2. Trends of the emission factors for each vehicle type were estimated considering the timing of road vehicle regulations in each country and the regions and the ratios of vehicles production years.
3. Emission factors of each vehicle type during the target period of REASv3 were calculated using those of base years and the corresponding trends.

The information of road vehicle regulations in each country and regions were taken from Clean Air Asia (2014). For the ratios of vehicle production years, due to lack of information, data for Macau derived from Zhang et al. (2016) were used for Hong Kong, Republic of Korea, and Taiwan and those from Japan Environmental Sanitation Center and Suuri Keikaku (2011) for Vietnam were used for other countries and regions. Then, trends of emission factors were estimated using the above data and information with values of Europe and United States standards. Finally, emission factors used to estimate emissions were calculated for each vehicle type. For most countries, the years just before the regulations for road vehicles began were set as base years and no-controlled emission factors that were used in REASv1.1 and REASv2.1 were adopted

for emission factors of the base years. Countries for which information on regulations were not obtained, the no-controlled emission factors were used for the entire target period of REASv3. For China and India, emission factors in 2010 were estimated as base year's data using recently published papers, such as Huo et al. (2012b), Xia et al. (2016), Mishra and Goyal (2014), and Sahu et al. (2014). For Republic of Korea and Taiwan whose emissions were not originally estimated in REASv2.1, emission factors were estimated with high uncertainties based on values of Europe and United States standards, respectively. For Japan, emission factors for each emission standard are available for several vehicle speeds (JPEC, 2012a). Combining these data with information for annual VKT of each vehicle speed, ratios of vehicle ages, and time series of regulation standards, emission of road transport in Japan were calculated. Details of emission factors of exhaust emissions were provided in Sect. S6.2 of the Supplement.

## 2.4 Agricultural sources

REASv3 includes $NH_3$ emissions from manure management and fertilizer application in agricultural sources. Approaches similar to REASv2.1 were adopted to estimate historical emissions and develop monthly gridded data. First, annual emissions of each country and sub-region except for Japan and their gridded data for the year 2000 were selected from REASv1.1 (Yamaji et al., 2004; Yan et al., 2003) as base data. For Japan, corresponding base data were obtained from REASv2.1 (Kurokawa et al., 2013: JPEC 2012a, b, c; 2014) for the year 2000 and 2005. Second, trends of emissions during 1950-2015 were estimated for each country and sub-region. Third, annual emissions for the period were calculated using the trends and base data. Fourth, changes in spatial distribution from base years to target years and monthly variations in each country and sub-region were estimated. Finally, monthly gridded data of emissions were developed for 1950-2015. For Japan, emission data during 2001-2004 were interpolated between those in 2000 and 2005. Details for manure management and fertilizer application are described in Sections 2.4.1 and 2.4.2, respectively.

### 2.4.1 Manure management

Trends in $NH_3$ emissions from manure management of livestock, except for its application as fertilizer, were estimated based on the Tier 1 method of EEA (2016). In this method, emissions are calculated based on the numbers of livestock and the corresponding emission factors. Statistics on the number of animals, such as broilers, dairy cow, and swine are mainly obtained from FAOSTAT (available at http://www.fao.org/faostat/en/) of the Food and Agriculture Organization (FAO) of the UN from the period between 1961 to 2015. For the years before 1960, data were obtained from Mitchell (1998). National statistics were surveyed for data on provinces, states, and prefectures in China, India, and Japan, respectively to develop activity data for each sub-region. Emission factors are obtained from EEA (2016). For spatial distribution, changes in grid allocation for each country and sub-region from the year 2000 were estimated using EDGARv4.3.2 from 1970 to 2012. Grid allocation factors in 1970 and 2012 were used for the period before and after 1970 and 2012, respectively. For temporal variations, monthly allocation factors are estimated as a function of temperature by referring to the monthly variations of

emissions in Japan based on the JEI-DB. Detailed methodologies and data sources for manure management were provided in Sect. S8.1 in the Supplement.

### 355    2.4.2 Fertilizer

In most countries, fertilizer application is the largest source of $NH_3$ emissions. Emission trends after the application of manure and synthetic N fertilizer were estimated using EEA (2016). Manure application is one of the processes of manure management whose emissions trend was calculated based on the number of animals and the corresponding emission factor. For synthetic N fertilizer, trends of total consumption of fertilizer were used in REASv2.1. However, this simple approach

causes uncertainties because emission factors are different among types of fertilizer (EEA, 2016). Therefore, in REASv3, emissions from each N fertilizer, such as ammonium phosphate and urea were estimated separately and trends in total emissions were calculated. For spatial distribution, changes in grid allocation factors for each country and sub-region from the year 2000 were estimated using a historical global N fertilizer application map during 1961-2010, developed by Nishina et al. (2017). Data for 1961 and 2010 were used for the period before 1961 and after 2010, respectively. For seasonal

variations, monthly factors of China and Japan were determined based on Kang et al. (2016) and the JEI-DB, respectively. For other countries, data from Nishina et al. (2017) have monthly application amounts in each grid. However, there are cases that some months have high factors, whereas the others have almost zero. Referring to Janssens-Maenhout et al. (2015), we adopted the conservative way, such that the highest monthly factor was set at 0.2 and the factors of all months were adjusted accordingly. See Sect. S8.2 for details of methodologies and data sources for emissions from fertilizer application.

### 370    2.5 Other sources

NMVOC emissions from evaporative sources are increasing significantly in Asia along with economic growth. Major sources of NMVOC emissions include usage of solvents for dry cleaning, degreasing operations, and adhesive application as well as for paint use. Fugitive emissions related to fossil fuels, such as extraction and handling of oil and gas, oil refinery, and gasoline stations are also important. However, statistics on activity data and information of emission factors for these

sources are often less available than those for fuel combustion and industrial processes. In this study, default activity data and emission factors were obtained from REASv2.1 and were updated if information was available in recently published papers (such as Wei et al. (2011) for China and Sharma et al. (2015) for India). In general, activity data of the past years are not available, and, in such cases, proxy data are prepared for trend factors. For example, population numbers were used for dry cleaning and production numbers of vehicles were used for paint application for automobile manufacturing. GDP was

used for default trend factors. For emission calculation, the same equation for stationary combustion was adopted. Details of activity data and emission factors for non-combustion sources of NMVOC were provided in Sect. S5 of the Supplement.

In addition to agricultural activities, latrines are an important source of $NH_3$, especially in rural areas. Activity data are population numbers in no sewage service areas estimated referring settings of REASv2.1 and emission factor were based on EEA (2016) and SEI (2012). Also, humans themselves are sources of $NH_3$ emissions through perspiration and respiration.

For these sources, population numbers are activity data mainly taken from UN (2018) and emission factors are obtained from EEA (2016). Equation to estimate emission is also the same as that of stationary combustion. Additional data and information for emissions from human and latrines were described in Sects. S8.4 and S8.5, respectively.

In REASv3, aviation and ship emissions including fishing ships are not included, but emissions of fuel combustion in other transport sector (namely, except for aviation, navigation, and road), such as railway and pipeline transport were estimated. Equation (1) is also used for estimating emissions of these sources. See Sect. S7 of the Supplement for additional data and information for other transport sector.

## 2.6 Spatial and temporal distribution

Procedures for developing gridded emission data were the same as those of REASv2.1. Large power plants were treated as point sources, and longitude and latitude of each power plant were provided. Positions of power plants were surveyed based on detailed information, such as names of units, plants, and companies from WEPP (Platts, 2018). These were searched on internet sites, such as Industry About (https://www.industryabout.com/) and Global Energy Observatory (http://globalenergyobservatory.org/). Positions for newly added power plants in REASv3 as well as those in REASv2.1 were surveyed because some of these services were not available when REASv2.1 was developed. For cement, iron, and steel plants (and non-ferrous metal plants in Japan), REASv3 still did not treat them as point sources due to lack of activity data. However, positions, production capacities, start and retire years for large plants were surveyed similar to power plants and used for developing allocation factors for corresponding sub-sectors. For road transport sector, REASv2.1 used coarse grid allocation data of REASv1.1 with $0.5° \times 0.5°$ resolution. Therefore, in REASv3, grid allocation factors for each country and sub-region, except Japan, were updated using gridded emission data of road transport sector of EDGARv4.3.2 during 1970-2012. Before 1970 (after 2012), data for 1970 (2012) were used. For Japan, gridded emission data of the JEI-DB in 2000, 2005, and 2010 were used to develop grid allocation factors. For the year between 2000 (2005) and 2005 (2010), the JEI-DB data were interpolated. For years before 2000 (after 2010), the JEI-DB data for 2000 (2010) were used. For residential sectors, rural, urban, and total population of HYDE 3.2.1 (Klein Goldewijk et al., 2017) with $5' \times 5'$ were used to create allocation factors. Data of HYDE 3.2.1 were available for 1950, 1960, 1970, 1980, 1990, 2000, 2005, 2010, and 2015 and the years between them were interpolated. Spatial distributions of total population were used for grid allocation of all other sources. Detailed methodologies and data sources for grid allocation were provided in Sect. S9.1 in the Supplement.

Methodology to estimate monthly emission data in REASv3 was the same as that of REASv2.1. In general, monthly emissions were estimated by allocating annual emissions to each month using monthly proxy data. Monthly generated power and production amounts of industrial products were used as the monthly allocation factors for power plant sector and corresponding industry sub-sectors, respectively. Basically, monthly factors of REASv2.1 during 2000-2008 were also used in REASv3 and were extended if data existed before (after) 2000 (2008). For the years where surrogate data were unavailable, the data of oldest (newest) year were used before (after) the year. For brick production, monthly allocation factors for Southeast and South Asian countries were estimated referring Maithel et al. (2012) and Maithel (2013). For the

residential sector, monthly variations of emissions were estimated using surface temperature in each grid cell, similar to REASv2.1. Surface temperatures during 1950-2015 were taken from NCEP reanalysis data provided by the NOAA/OAR/ESRL PSD, Boulder, Colorado, USA (https://psl.noaa.gov/data/gridded/data.ncep.reanalysis.html). For Thailand and Japan, most monthly factors were set based on country specific information from Thao Pham et al. (2008) and JPEC (2014), respectively. See Sect. S9.2 of the Supplement for details of monthly variation factors.

## 3 Results and discussion

### 3.1 Trends of Asian and national emissions

Trends in air pollutants emissions from Asia, China, India, Japan, and other countries are described in this section, mainly focusing on $SO_2$, $NO_x$, and BC emissions as they have important roles in both air pollution and climate change. $SO_2$ and $NO_x$ are precursors of sulfate and nitrate aerosols, respectively, which are the major components of secondary $PM_{2.5}$. $NO_x$ is also a precursor of ozone. Furthermore, BC is a major component of primary $PM_{2.5}$. $PM_{2.5}$ and ozone not only harm human health and ecosystems, but influence climate change. BC and ozone have a warming effect on climate change, whereas sulfate and nitrate aerosols have a cooling effect. Note that all the air pollutant emissions from major countries and regions between 1950 to 2015 categorized based on major sectors and fuel types, are provided in the Supplement material (Figs. S1-S12). $CO_2$ emissions in REASv3 include contribution from biofuel combustion unless otherwise indicated.

### 3.1.1 Asia

Table 3 summarizes the national emissions of each species in 2015 and the total emissions from Asia in 1950, 1960, 1970, 1980, 1990, 2000, and from 2010-2015. Figure 2 shows emissions of $SO_2$, $NO_x$, CO, NMVOC, $NH_3$, $CO_2$, $PM_{10}$, $PM_{2.5}$, BC, and OC in China, India, Japan, Southeast Asia (SEA), East Asia other than China and Japan (OEA), and South Asia other than India (OSA) from 1950 to 2015. Average total emissions in Asia during 1950-1955 and 2010-2015 (growth rates in these 60 years estimated from the two averages) are as follows: $SO_2$: 3.2 Tg, 42.4 Tg (13.1); $NO_x$: 1.6 Tg, 47.3 Tg (29.1); CO: 56.1 Tg, 303 Tg (5.4); NMVOC: 7.0 Tg, 57.8 Tg (8.3); $NH_3$: 8.0 Tg, 31.3 Tg (3.9); $CO_2$: 1.1 Pg, 18.6 Pg (16.5) ($CO_2$ excluding biofuel combustion 0.3 Pg, 16.8 Pg (48.6)); $PM_{10}$: 5.9 Tg, 30.2 Tg (5.1); $PM_{2.5}$: 4.6 Tg, 21.3 Tg (4.6); BC: 0.69 Tg, 3.2 Tg (4.7); and OC: 2.5 Tg, 6.6 Tg (2.7). Clearly, all the air pollutant emissions in Asia increased significantly during these six decades. However, this increase was different among the aforementioned species. Growth rates of emissions were relatively large for $SO_2$, $NO_x$, and $CO_2$ because the major sources of these species are power plants, industries, and road transport, for which fuel consumption increased significantly along with economic development in Asia. $SO_2$ increased before the other species because a majority of the emissions were obtained from the combustion of coal, which is easier to obtain than oil and gas. $SO_2$, $NO_x$, and $CO_2$ emissions increased keenly in the early 2000s, along with rapid growth of emissions of these species in China. For $NO_x$, combustion of oil fuels, especially by road vehicles, contributed to a large growth of emissions in the latter half of 1950-2015. Growth rates of NMVOC have also increased recently due to an increase

in the emissions from road vehicles and evaporative sources, such as paint and solvent usage in accordance with economic

growth of Asian countries. On the other hand, rates of growth of CO, $PM_{10}$, $PM_{2.5}$, BC, and OC are relatively small. One

reason is that emissions of these species are mainly from incomplete combustion in low temperature and thus, emissions

from power plants and large industry plants are relatively small. Another reason is that a major source of these species is the

combustion of coal and biofuels in residential sector, which dominated over other sectors in earlier times and were relatively

large even in recent years in Asia. Recently, emissions of these species from industries, including combustion and non-

combustion processes are increasing. In addition, gasoline and diesel vehicles have contributed recently to the growth of CO

and BC emissions, respectively. Agricultural activities, such as manure management of livestock and fertilizer application,

which are major sources of $NH_3$ are rising to support a growing population in Asia. Although the growth rate of $NH_3$

emissions is smaller than other species, it still shows an increasing trend.

Differences in the trends of emissions were also observed on the basis of countries and regions. $SO_2$ and $NO_x$, emissions

from Japan were relatively large in Asia during the 1950s-1970s. Emissions from Japan in 1965 are comparable with and are

larger than those of China for $SO_2$ and $NO_x$, respectively. In 2015, emissions of $SO_2$ and $NO_x$ in Japan decreased largely and

contribute only about 1.5 and 3.8% of Asia's total emissions, respectively. Similar tendencies were also observed in the case

of other species. In 2015, China was the largest contributor of emissions for all the species. Recently, emissions of most

species in China have shown decreasing or stable trends. In the case of $SO_2$, China contributed about 72% of emissions in

2005, but about 49% in 2015. On the other hand, emissions and their relative ratios are increasing in the case of India.

Actually, contribution rates of $SO_2$, $NO_x$, and BC emissions in India increased from 14%, 16%, and 23% in 2005 to 30%,

22%, and 27% in 2015, respectively. Li et al. (2017c) suggested that, in 2016, $SO_2$ emissions in India exceeded those in

China. Recent increase in air pollutants emissions have also been observed in SEA and OSA. On the other hand, emissions

from OEA started to increase slightly later than Japan and then, recently show decreasing trends mainly reflecting trends of

emissions from Republic of Korea and Taiwan.

### 3.1.2 China

Growth rates of all pollutants emissions in China in these 60 years estimated from average during 1950-1955 and 2010-2015

are as follows: 21% for $SO_2$, 54% for $NO_x$, 7.0% for CO, 13% for NMVOC, 4.7% for $NH_3$, 28% for $CO_2$ (105% for $CO_2$

excluding biofuel combustion), 6.8% for $PM_{10}$, 6.1% for $PM_{2.5}$, 5.5% for BC, and 2.7% for OC. It was observed that

emissions of all pollutants increased largely during these six decades, but most species reached their peaks up to 2015 as

shown in Fig. 2. Exceptions to this were NMVOC, $NH_3$, and $CO_2$; however, their growth rates are at least small or almost

zero. Emission trends in China for all the pollutants in each sector and for each fuel type during 1950-2015 were presented in

Figs. S1 and S2, respectively. Figure 3 shows recent trends in actual emissions (solid colored areas) and reduced emissions

by control measures (hatched areas) from each sector for $SO_2$, $NO_x$, and BC during 1990-2015 in China. The reduced

emission by control measures was the difference between emissions calculated without effects of all control measures (such

as FGD, ESP, using low sulfur fuels, regulated vehicles, etc.) and actual emissions. Total $CO_2$ emissions were also plotted

for each panel of Fig. 3 as an indicator of energy consumption. Note that reduced emissions here do not include effects of substitution of fuel types, such as from coal to natural gas.

For $SO_2$, most emissions in China were from coal combustion which controlled trends of total emissions. $SO_2$ emissions in China increased rapidly in the early 2000s, but decreased after 2006 and showed a continuous decline until 2015. Drastic changes in the 2000s were mainly caused by emissions from coal-fired power plants, which increased rapidly along with large economic growth and later decreased due to the introduction of FGD based on the 11[th] Five Year Plan of China. After 2011, control measures for large industry plants started to become effective and as a result, total emissions in 2015 became comparable with those in 1990. Without effects of emission controls, emissions from power plants and industry in 2015 would be 3.7 and 2.6 times higher than those in 2000, respectively. In this study, the emissions in 2015 were estimated to be reduced by about 90% for power plants and 76 % for industry. On the other hand, even without emission controls, $SO_2$ emissions from power plants were almost stable after 2010. The same tendencies were also found in $CO_2$. One considerable reason is an increasing energy supply from nuclear power plants. According to IEA (2017), the total primary energy supply from nuclear power plants increased rapidly recently and those in 2015 were about 2.3 time higher than in 2010.

Similar to $SO_2$, $NO_x$ emissions increased rapidly from the early 2000s, but continued to increase until 2011 and then, started to decline. In the 2000s, low $NO_x$ burner to power plants and regulation of road vehicles were introduced, but their effects were limited. From 2011, introduction of denitrification technologies, such as selective catalytic reduction (SCR) to large power plants and regulations for road vehicles were strengthened based on the 12[th] Five Year Plan of China. Three major drivers of $NO_x$ emissions in China are power plants, industry sector, and road transport. If no emission mitigation was considered, their emissions would be increased by 3.6, 3.0, and 4.7 times from 2000 to 2010, respectively. In 2015, reduction rates of emissions due to emission controls were about 61%, 19%, and 62% for power plants, industry, and road transport respectively. As a result, in 2015, $NO_x$ emissions were about 81% of their peak values in 2011. In 2015, actual $NO_x$ emissions from industry sector were larger than those from power plants and road transport which were comparable each other. Major industries such as iron and steel, chemical and petrochemical, and cement industries were large contributor of $NO_x$ emissions in China.

For BC, emissions also increased from early 2000s, but growth rates were smaller than $SO_2$ and $NO_x$ due to the effects of control equipment in the industrial sector. Actually, trends of BC emissions assuming no emission controls were close to those of $CO_2$ and the BC emissions in 2015 were increased by 2.2 times from 2000. The emissions in 2015 were reduced by about 41% by abatement measures in industry plants and 9% by regulations especially for diesel vehicles. In 2015, large contributors in industry sectors were brick production, coke ovens, and coal combustion in other industry plants. Another reason of relatively small growth rates could be that BC emission factors for coal-fired power plants are originally low. Recently, BC emissions from residential sector as well as industrial sector show decreasing trends. In this study, the reductions in BC emissions in residential sector were mainly caused by a decrease in emissions from biofuel combustion. During 2010 to 2015, consumptions of primary solid biofuels were reduced about 28%, whereas consumption of natural gas and liquefied petroleum gas increased about 62% in the residential sector.

For CO, most emissions in the 1950s were from residential sectors and gradually increased with increasing coal consumption in the industrial sector. CO emissions increased largely in 2000s due to coal combustion and iron and steel production processes. Recently, CO emissions have seen a decline. A major reason for this declining trend is the decrease in biofuel consumption in residential sector and the phasing out of shaft kiln with high CO emission factor in the cement industry.

NMVOC emissions increased significantly from the early 2000s, similar to other species. However, their major sources were different from others. Recent increasing trends are not caused by stationary combustion sources, but by road transport and evaporative sources, such as paint and solvent use. In particular, emissions from non-combustion sources increased largely from 2000 to 2015 (about 3.7 time) and as a result, their contribution rate in 2015 was about 65%. Growth rates of NMVOC emissions tended to slow down around 2015, but emissions increased almost monotonically after the 2000s. $NH_3$ emissions

were mostly from agricultural activities. In China, emissions from fertilizer application showed a significant increase from the early 1970s to the early 2000s. In recent years, $NH_3$ emissions are almost stable. For $PM_{10}$ and $PM_{2.5}$, majority of the emissions are from the industrial sector, followed by residential sector and power plants. Emissions increased largely from the early 1990s mainly due to coal combustion and industrial processes, especially in cement plants. Compared to $SO_2$ and $NO_x$, growth rates of $PM_{10}$ and $PM_{2.5}$ emissions during the early 2000s were small, and later decreased due to the effects of

control equipment in industrial plants. OC emissions were mostly from biofuel combustion in the residential sector. Contributions from the industrial sector has been increasing recently, but total OC emissions have decreased due to reduced usage of biofuels. $CO_2$ emissions were mainly controlled by coal combustion and their trend were similar to those of $SO_2$, $NO_x$, and BC without emission controls as shown in Fig.3. After 2011, $CO_2$ emissions in China were found to be almost stable. As described above, one reason is a trend of emissions from power plants. In addition, emissions from coal

combustion in industry sectors were slightly decreased from 2014 to 2015.

### 3.1.3 India

Growth rates of air pollutants emissions in India based on averaged values during 1950-1955 and 2010-2015 are as follows: 19% for $SO_2$, 23% for $NO_x$, 4.2% for CO, 5.3% for NMVOC, 3.1% for $NH_3$, 8.9% for $CO_2$ (29% for $CO_2$ excluding biofuel combustion), 4.8% for $PM_{10}$, 4.0% for $PM_{2.5}$, 4.8% for BC, and 2.8% for OC. Figures S3 and S4 provide trends of emissions

in India from each sector and fuel type for all the pollutants, respectively, from 1950 to 2015. In general, all the air pollutants show monotonous increase from 1950 to 2015 and growth rates (especially of recent years) are larger for $SO_2$, $NO_x$, NMVOC, and $CO_2$, which is similar to the case of Asia.

Figure 4 shows trends in emission of $SO_2$, $NO_x$, and BC from each fuel type as well as sector with total $CO_2$ emissions during 1950-2015 in India. Clear differences were seen in the structure of emissions in these species. For $SO_2$, large parts of

545 emissions were from coal combustion in power plants and industry sector. $SO_2$ emissions in 2015 were about 3.3 times larger than those in 1990 and contribution rates of the increases from power plants and industry sectors were about 66% and 33%, respectively. Trend so total $NO_x$, emissions were close to those of $SO_2$ and contributions from coal-fired power plants were also large. In addition, for $NO_x$, contribution from road transport especially diesel vehicles were comparable with those

of power plants. Around the year 2005, the contributions from road transport were almost the same or slightly larger than power plants. However, from 2005 to 2015, growth rates of $NO_x$ emissions from power plants were about twice higher than those of road transport emissions. For BC, contributions from the residential sector and biofuel combustion were large, especially in the 1950s-1960s. Contribution rates of residential sector were 73% in 1950 and 38% in 2015, and those of biofuel combustion, which were mainly used in residential sector and some parts are used in industry sector were 86% in 1950 and 45% in 2015. On the other hand, recent increasing trends of BC emissions were also caused by growth of emissions from diesel vehicles and industry sector. From 1990 to 2015, contribution rates of increased emissions from industry, road transport, and residential sectors were 27%, 43%, and 23%, respectively. For recent trends, relative ratios of $SO_2$ emissions from power plants were increased from 43% to 59% during 1990-2015. For $NO_x$, contribution rates from both power plants and road transport were increased and accounted for about 75% of the total emissions in 2015. Even in 2015, about half of the BC emissions were from the residential sector. However, as previously described, recent emission growths were mainly caused by the industrial sector and road transport. These tendencies were similar to Japan and China during their rapid emission growth periods. These features were consistent with trends of $CO_2$ emissions. Before the mid-1980s, majority of $CO_2$ emissions were from biofuel combustion and the trends were close to those of BC. Then, recently, contributions from fossil fuel combustion increased largely and trends of $CO_2$ became close to those of $SO_2$ and $NO_x$, especially after the early 2000s.

Trends and structure of CO emissions were similar to those of BC but contribution rates of the residential sector were larger and those from road transport (mainly from gasoline vehicle) were smaller, as compared to BC. On the other hand, for recent trends, half (51%) of increased emissions during 2005 and 2015 were from industry sector. Similar tendency was also found in OC; however, relative ratios of emissions from residential sector were much larger (about 71% in 2015) and those of industry and road transport sectors were much smaller. For $PM_{10}$ and $PM_{2.5}$, a majority of the emissions was from residential and industrial sectors. Both amounts were almost comparable in $PM_{10}$ and those from residential sectors were larger in $PM_{2.5}$. Different from BC and OC, contributions from coal-fired power plants exist in $PM_{10}$ and $PM_{2.5}$ whose contribution rates in 2015 are about 20% and 13%, respectively. For NMVOC, most emissions were from biofuel combustion before the 1980s. Later, emissions from variety of sources, such as road transport, extraction and handling of fossil fuels, usage of paint and solvents are increasing and are controlling recent trends. For increases of emissions from 1990 to 2015, about 52% were from stationary combustion and road transport and the rest were from stationary non-combustion sectors such as paint and solvent use. Most $NH_3$ emissions are from agricultural activities. Contributions from manure management and fertilizer use were comparable before 1980s. However, emissions from fertilizer application have increased largely which are now determining recent trends.

### 3.1.4 Japan

As described in Sect. 3.1.1, trends of air pollutants emissions in Japan were different from other countries and regions in Asia. The trends from each sector and fuel type during 1950-2015 in Japan were shown in Figs. S5 and S6. Compared to the

rest of Asia, emissions of all species in Japan except $CO_2$ were reduced significantly after reaching peak values. In addition, peak years were mostly 40 years ago (about 1960 for $PM_{10}$, $PM_{2.5}$, and OC, 1970 for $SO_2$ and CO, 1980 for $NO_x$ and $NH_3$, 1990 for BC, and 2000 for NMVOC). Figure 5, similar to Fig. 3, shows trends of actual emissions (solid colored areas) and

reduced emissions by control measures (hatched areas) from each sector for $SO_2$, $NO_x$, and BC during 1950-2015. Total $CO_2$ emissions were also plotted to each panel of Fig. 5. $CO_2$ emissions increased rapidly in the 1960s and have generally continued to increase, but growth rates are much smaller than those in the 1960s reflecting trends of economic status of Japan.

$SO_2$ emissions, especially from power plants and industry sector increased significantly in the 1960s (reflecting the rapid

economic growth) and caused severe air pollutions in Japan. In the 1950s, more than half the emissions were from coal combustion and then, contributions from heavy fuel oil increased rapidly in the 1960s (more than 50% around the peak year). In order to mitigate air pollution, first, regulation of sulfur contents, especially in heavy fuel oil, were strengthened. Then, desulfurization equipment was mainly introduced from the mid-1970s. As a result, about 68%, 84%, and 93% of the $SO_2$ emissions were reduced by regulatory measures in 1975, 1990, and 2015, respectively. Furthermore, although coal

consumption in power plants increased in the 1990s, $SO_2$ emissions almost did not change due to these measures. For trends of $SO_2$ emissions assuming without emission controls and those of $CO_2$, there are clear differences in the 1970s and after the 1980s. The causes of the differences in the 1970s were decrease of heavy fuel oil consumption whose contribution rates on $SO_2$ were much higher than $CO_2$. On the contrary, causes of the differences in the 1980s were increasing consumption of gas and light fuel oil whose sulfur contents were small.

$NO_x$ emissions also increased rapidly from the 1960s mainly by steep increase of traffic volumes and fossil fuel combustion in power and large industry plants. The largest contribution to $NO_x$ emissions during the peak periods was from road transport sector, that is greater than 50% of total emissions. Regulations for road vehicles became effective from the late 1970s but an increase in the number of vehicles partially cancelled the effects. For stationary sources, the number of introduced denitrification equipment increased largely in the 1990s. As a result, $NO_x$ emissions peaked later; furthermore,

reduction rates after the peak were smaller compared to that of $SO_2$. From 1975 to 2015, emissions assuming without emission mitigations would be increased by about 2.0 times for power plants and 2.4 times for road transport. In 2015, by emission abatement equipment for power plants and control measures for road vehicles, the emissions were reduced by 77% and 90%, respectively. As a result, the reduction rate of total $NO_x$ emissions in 2015 was 78%, but it was smaller than $SO_2$ as described above.

For BC, contributing sectors changed during 1950-2015. In the 1950s, most emissions were from industries and residential sectors and their amounts were almost comparable. After the 1960s, both types of emissions declined, but reasons for declines were different. In the 1950s, coal and biofuels, which have large BC emission factors were mainly used in residential sectors. However, these fuels were substituted for cleaner ones, such as natural gas and liquefied petroleum gas which reduced BC emissions significantly. Emissions in industrial sectors decreased gradually after the 1960s due to the

introduction of abatement equipment for PM. Instead, emissions from road transport sector from diesel vehicles increased

from the late 1960s to around 1990. Then, regulations for road vehicles were strengthened and BC emissions were reduced largely from peak values. Before 1986, emission controls for BC were only considered for stationary sources. In 1985, by effects of abatement equipment to power and industrial plants, emissions were reduced by about 58% from those assuming no emission controls. Then, by introducing regulations for diesel vehicles, the reduction rates became about 91% in 2015.

For CO and OC, most emissions in 1950s were from biofuel combustion in the residential sector. CO and NMVOC emissions in road transport increased largely in the 1960s and then decreased gradually, similar to the case of $NO_x$. Recently, a majority of NMVOC emissions were from evaporative sources, such as paint and solvent use. These started to increase from the 1980s and then decreased after 2000. Emissions of CO and OC from the industrial sector showed a similar increase before 1970, whereas OC emissions started to decrease due to control equipment for PM species and CO emissions were

almost stable after 1970. The majority of $NH_3$ emissions in Japan was from agricultural activities, especially manure management; however, contributions from latrines were also large in the past years. Overall, $NH_3$ emissions increased from 1950 to the 1970s but, showed slightly decreasing trends after the 1990s. $PM_{10}$ and $PM_{2.5}$ emission trends were almost the same. The majority of emissions was from the industrial sector, which grew during the 1950s but decreased largely in the 1970s due to the effects of abatement equipment for PM. Contributions from the residential sector were relatively large from

the 1950s to the 1960s. Furthermore, contributions from road transport increased from the 1970s and started to decrease after 1990, similar to BC.

### 3.1.5 Other regions

Similar to India, air pollutant emissions in SEA and OSA tended to increase during these six decades. Figures S7 and S8 (S11 and S12) provide trends for all the air pollutant emissions in SEA (in OSA) for each sector and fuel type, respectively,

from 1950 to 2015. Figures 6 and 7 show emission trends of $SO_2$, $NO_x$, and BC for each sector category and contribution rates of each country from 1950-2015 in SEA and OSA, respectively. Total $CO_2$ emissions were also plotted to upper panels of Figs. 6 and 7.

Contributing sources and their relative ratios in $SO_2$, $NO_x$ and BC emissions are generally close between these regions. For both the regions, major sources of $SO_2$ emissions are power plants and industry sector. For fuel types, contributions from

heavy fuel oil were large in the case of $SO_2$ emissions in OSA and were almost comparable to those of coal in SEA during the 1990s. After 2010, emissions from coal-fired power plants in SEA increased rapidly which were doubled during 2010-2015. On the other hand, in OSA, heavy fuel consumption in power plants increased by 1.8 times from 2005 to 2015 which mainly caused the large increase of $SO_2$ emission. For $NO_x$, majority of the emissions were from road transport, mainly diesel vehicles. This controlled the recent trends in both regions. Contributions from gasoline vehicles were small in OSA,

but relatively large in SEA (about 16% in 2015). On the other hand, $NO_x$ emissions from natural gas vehicles increased from the 2000s in OSA and contribution rates in road transport sector were more than 15% after the late 2000s. Recently, similar to $SO_2$, $NO_x$ emissions from power plants have been increasing by coal and heavy fuel oil combustion in SEA and OSA, respectively. From 2010 to 2015, increases of emissions were mainly caused by power plants in both regions (about 67% for

SEA and 82% for OSA). Although trends are almost stable, emissions from biofuel combustion in the residential sector are relatively large in OSA. BC emissions are mostly from biofuel combustion in the residential sector, especially in OSA. and increased constantly during the period of REASv3. After the late 2000s, BC emissions from road transport show decreasing trends due to effect of emission regulations especially in SEA. Relations between trends of $SO_2$, $NO_x$, BC, and $CO_2$ emissions were similar to the case of India that trends of $CO_2$ were close to those of BC before the 1980s and then those of $SO_2$ and $NO_x$ after the 1990s. In the case of country-wise emissions, currently, the largest contributing countries are Indonesia and Pakistan in SEA and OSA, respectively. In 2015, the second and third highest contributing countries in SEA were Philippines and Vietnam for $SO_2$, Thailand and Philippines for $NO_x$, and Vietnam and Thailand for BC. Relative ratios of $SO_2$ emissions in Thailand were large in the early 1990s but decreased significantly due to the introduction of FGD in large coal-fired power plants. For OSA, the second highest contributing country is Bangladesh; Sri Lanka is ranked third for $SO_2$ and $NO_x$ and Nepal for BC.

Emission trends in OEA from each sector during 1950-2015 were presented for all the air pollutants in Figs. S9 and S10. Emission trends in Republic of Korea and Taiwan were similar to those of Japan. $SO_2$ emissions increased rapidly in the 1970s and reduced largely from their peak values due to the introduction of low sulfur fuels and FGD. $NO_x$ emissions started to increase steeply from the 1980s due to emissions from road vehicles, in addition to those from power and industry plants. Then, $NO_x$ emissions decrease after 2000 due to regulations related to road vehicles and the introduction of control equipment to power plants. However, their rate of decrease was lower than that of $SO_2$. BC emission trends were similar to those of $NO_x$ until around the year 2000, but the ratio of decrease after 2000 is much larger than that of $NO_x$. The differences of reduction rates of emissions between $NO_x$ and BC were caused by effects of emission controls in road transport sector. These features and drivers of trends were generally similar to the case of Japan. For Democratic People's Republic of Korea, emissions of $SO_2$, $NO_x$, $CO_2$, and PM species decreased and those of CO, NMVOC, and $NH_3$ were almost stable recently. The recent decreasing trends were mainly caused by coal consumption amounts in industry sector. For Mongolia, emissions of all the air pollutants, except PM species, show increasing trends recently. The increasing trends were mainly caused by coal-fired power plants for $SO_2$ and $CO_2$, road transport for $NO_x$, CO, NMVOC, and BC, and domestic sector for OC. For $PM_{10}$ and $PM_{2.5}$, due to effects of abatement equipment in power plants, emissions were almost stabilized after 2000. Note that information of these two countries are limited and therefore uncertainties are large.

**3.2 Spatial distribution and monthly variation**

Figure 8 presents the emission map of $SO_2$, $NO_x$, CO, NMVOC, $NH_3$, $PM_{2.5}$, BC, and OC in 1965 and 2015 at $0.25° \times 0.25°$ resolution. Emission maps of $CO_2$ and $PM_{10}$ are presented in Fig. S13. In 1965, high emission grids appeared in industrial areas of Japan, especially for $NO_x$, $SO_2$, and $CO_2$. On the other hand, high emission grids were seen in wide areas in China and India, for CO and PM species, especially OC. This is because emissions of these species were mainly from the residential sector and small industrial plants. In 2015, high emission areas for all species clearly appeared in China and India, especially in the northeastern area, around the Sichuan province, and Pearl River Delta for China and Indo-Gangetic Plain,

around Gujarat, and southern area for India. High emission areas of $SO_2$ and PM species in Japan disappeared or shrinked in 2015 compared to 1965, but still remained in the $NO_x$, CO, NMVOC, and $CO_2$ maps. In SEA, high emission areas were seen in the Java island of Indonesia and around large cities, such as Bangkok (Thailand) and Hanoi (Vietnam). $NH_3$ and OC emissions, whose major sources were agriculture and residential sector, respectively were found in relatively large areas of China, India, and SEA.

As described in Section 2.6, seasonality of emissions is taken into considered for sectors where proxy data for monthly profiles were available or could be estimated. Monthly variations of total emissions of $SO_2$, $NO_x$, BC, and $NH_3$ are shown for China, India, Japan, SEA, OEA, and OSA for the year 2015 in Fig. 9. For $SO_2$ and $NO_x$, monthly variations were generally small. In China, emissions were slightly larger in the second half of the year. Monthly factors of $SO_2$ emissions in OSA were high from December to May and low during July and September due to the timings of brick production. For BC, emissions in winter season were relatively large, especially in China and OEA. This seasonality was mainly determined by fuel consumption in residential sector for the purposes of heating. Therefore, monthly variations of BC emissions were smaller in SEA. For $NH_3$, seasonality of emissions was controlled by the seasonality of emissions from fertilizer application and manure management. In China, Japan, and OEA, peaks of emissions appeared during summertime. Monthly variations of emissions in the whole of SEA were small, but seasonality was different from each country. Finally, it must be noted that monthly variations of emissions in each grid were different to each other because they were determined by monthly profiles of major emission sources in each grid cell.

## 3.3 Comparison with other inventories

In this section, estimated emissions of REASv3 were compared with other global, regional, and national bottom-up inventories and several top-down estimates. Figures 10 and 11 compare the results of REASv3 with other studies for $SO_2$, $NO_x$, and BC emissions in China and India, respectively. For other species, results based on comparison with China are presented in Fig. S14 and those with India are shown in Fig. S15. Furthermore, Figures S16-S19 provide the comparisons of emissions from Japan, SEA, OEA, and OSA, respectively. In Figs. 10, 11, and S14-S19, error bars were plotted in 2015, 1985, and 1955 of emissions in REASv3. These error bars were based on uncertainties estimated in this study for corresponding emissions. See Sect. 3.4 for details about the uncertainties of emissions in REASv3. Note that as described in Sect. 2.1, emissions from domestic and fishing ships are not included in REASv3. Therefore, corresponding data need to be excluded from values of other inventories in the comparisons. This procedure was done for REAS series, EDGARv4.3.2, CEDS, and several research works. For other inventories where emissions from domestic ship were not available independently, total emissions were plotted in the figures. It was confirmed that other sources out of scope of REASv3 such as open biomass burning were not included in the other inventories.

### 3.3.1 China

For long historical trends of $SO_2$ emissions in China, most studies generally agreed with the trends of REASv3 although values of REASv3 during 1995 and 2005 were slightly larger than other inventories. Emissions increased almost monotonically until around 1995 and became stable during the late 1990s. Then, emission increased rapidly from the early 2000s and started to decrease from the late 2000s. However, the decreasing rates were different especially after 2010. Recent rapid decreasing tendency in REASv3 was similar to that of Zheng et al. (2018), but decreasing rates of other studies such as Xia et al. (2016) and Sun et al. (2018) were smaller than REASv3. Values of REASv3 were slightly larger than REASv2.1 during 2000-2005, but the discrepancies were reduced due to a larger decreasing rate of REASv3. For top-down estimates (Qu et al., 2019 [based on retrieval products by National Aeronautics and Space Administration (NASA) standard (SP) and Belgian Institute for Space Aeronomy (BIRA)]; Miyazaki et al., 2020), emission amounts were smaller than most bottom-up inventories, but all top-down results showed large decreasing trends after the late 2000s.

Variability of $NO_x$ emissions among estimations plotted in Fig. 10 was smaller than that of $SO_2$. $NO_x$ emissions in most results increased largely in the 2000s and then decreased or stabilized. Growth rates of Sun et al. (2018) were smaller than others after 2005, but showed similar decreasing trends after 2010. Values of CEDS were slightly larger than other studies. Similar to $SO_2$, values of top-down estimates (Ding et al., 2017 [based on OMI and GOME-2]; Itahashi et al., 2019; Miyazaki et al., 2020) were generally smaller than those of bottom-up results. But, top-down emissions showed similar tendencies that emission increased until the early 2010s and turned to decrease. Trends of Itahashi et al. (2019) where emissions in 2008 of REASv2.1 were used as a priori data were close to those of REASv3.

Compared to $SO_2$ and $NO_x$, relatively large discrepancies were observed in BC emissions among plotted results in Fig. 10. Emissions of REASv3 increased until 1995, slightly decreased during the late 1990s, increased from the early 2000s and then, turned to decrease from the early 2010s. The decreasing rate in the late 1990s of Wang et al. (2012) was much larger than that of REASv3. On the other hand, emissions of Klimont et al. (2017) increased from 1995 to 2000. The majority of results showed increasing trends during the early 2000s, but the following trends were different. Emissions of CEDS increased constantly after 2005, but those of Wang et al. (2012) decreased after 2005 and then started to increase slightly after 2010. BC emissions of both REASv3 and Zheng et al. (2018) decreased from the early 2010s, but the ratio of decrease was larger in Zheng et al. (2018). Values of BC emissions of REASv3 were larger than those of REASv2.1, especially in the early 2000s, but the difference in 2008 was small. For trends and emission amounts of $PM_{10}$ and $PM_{2.5}$, tendencies of relationships among each result were similar. The majority of results showed clear decreasing trends after 2005 except for REASv2.1, EDGARv4.3.1 and Klimont et al. (2017). For OC, most results decreased from 1995 to 2000 and then increase from the early 2000s. After 2005, trends of OC emissions were different among studies.

CO emissions trends were relatively similar among most studies. Increasing rates after the early 2000s are close except for EDGARv4.3.2, but emission amounts of REASv3 were smaller than other studies before 2010. After 2010, the majority of results showed decreasing trends which agreed with top-down estimates (Jiang et al., 2017 [A: MOPITT Column, B:

MOPITT Profile, and C: MOPITT Lower Profile]; Zheng et al., 2019b; Miyazaki et al, 2020). However, before the late 2000s, the trends of CO emissions were much different between bottom-up inventories and top-down results. For NMVOC, most studies showed significant increasing trends after the early 2000s. Compared to bottom-up inventories, top-down estimates of Stavrakou et al. (2017) were almost stable between 2007 and 2012, but increased rapidly after that. Values of REASv3 were generally smaller than others before 2010. Differences among studies of $NH_3$ emissions were large not only in

emission amounts, but also in temporal variations. REAS inventories, CEDS, and EDGARv4.3.2 generally showed increasing trends. On the other hand, trends of MEICv1.2 and Zheng et al. (2018) were almost stable after 2000 and the results of Kang et al. (2016) showed decreasing trends after the mid-2000s. Emissions of REASv3 were also almost stable after 2010.

### 3.3.2 India

For $SO_2$, emissions of most bottom-up inventories showed monotonically increasing trends. However, after the 1990s, two different emission pathways were shown among studies. The growth rates of REASv3 were close to those of Klimont et al. (2013), CEDS (scaled to REASv2.1 for India; Hoesly et al., 2018), Streets et al. (2000), and REASv2.1. On the other hand, the increasing rates of national studies by Garg et al. (2006), Sadavarte and Venkataraman (2014) and Pandey et al. (2014) were smaller than those of REASv3. In 2005, top-down estimates of Qu et al. (2019) were close to results of Sadavarte and

Venkataraman (2014) and Pandey et al. (2014). Another top-down emissions of Miyazaki et al. (2020) were smaller than other inventories. Both bottom-up and top-down emissions after 2005 show increasing trends, but growth rates of bottom-up inventories were higher than those of top-down estimates.

NO$_x$ emissions of REASv3 also increased monotonically during 1950-2015 and the majority of other bottom-up inventories generally agreed with the trends including national studies of Sahu et al. (2012). However, similar to $SO_2$, growth rates of

Venkataraman (2014) and Pandey et al. (2014) were smaller than REASv3 although emission amounts in 2000 and 2005 were almost comparable each other. For the increasing rates, those of top-down estimates of Itahashi et al (2019) using REASv2.1 as a priori emissions were close to those of REASv3. On the other hand, growth rates of another top-down results of Qu et al. (2019) were similar with those of Sadavarte and Venkataraman (2014) and Pandey et al. (2014). Emission amounts of the top-down estimates were much higher than REASv3.

For BC, as in the case of China, discrepancies among studies plotted in Fig. 11 were large. These tendencies were also found in the comparisons of $PM_{10}$, $PM_{2.5}$, and OC emissions provided in Fig. S15. Generally, the majority of bottom-up emission inventories of PM species showed slightly continuous increasing trends and growth rates were smaller than those of $SO_2$ and NO$_x$. On the contrary to the case of $SO_2$ and NO$_x$, emissions of BC and $PM_{2.5}$ of REASv3 were slightly smaller than those of Sadavarte and Venkataraman (2014) and Pandey et al. (2014), but their growth rates were almost comparable.

Amounts and trends of CO emissions compared in Fig. S15 generally agreed well except for REASv1.1 which were much higher than others. Emission increased almost constantly until around 2005 ant then growth rates increased slightly. Values of REASv3 were much smaller than top-down results of Jiang et al., 2017 [A: MOPITT Column, B: MOPITT Profile, and C:

MOPITT Lower Profile] and Miyazaki et al. (2020). However, recent growth rates of REASv3 were close to those of top-down estimates except for Jiang et al. (2017) [C]. For NMVOC, plotted results were generally comparable except for

REASv2.1 and CEDS and indicated increasing trends of emissions. Similar to the case of SO$_2$ and NO$_x$, growth rates of REASv3 were smaller than those of Sadavarte and Venkataraman (2014) and Pandey et al. (2014). For NH$_3$, a comparison of the emissions in Fig. S15 show similar increasing trends. Differences in emission amounts are also relatively small, except for EDGARv4.3.2.

### 3.3.3 Other regions

Comparisons of emissions in Japan between REASv3 and other studies were provided in Fig. S16. For tends of SO$_2$ emissions in Japan, the majority of studies agreed with results of REASv3 that rapid increases in the 1960s, keen decreases in the 1970s, and gradually decreasing trends except for EDGARv4.3.2 and Streets et al. (2000), whose values were lager and smaller, respectively. For NO$_x$, emissions amounts of REASv3 were larger than those of most studies especially before 2000, except for CEDS (scaled to preliminary historical data of REAS for Japan; Hoesly et al., 2018), Kannari et al. (2007),

Zhang et al. (2009) based on Kannari et al (2007) and Fukui et al. (2014). For PM species, the majority of results in Fig. S16 agreed with decreasing trends of REASv3 after 1990. On the other hand, emissions of BC and OC of CEDS increased almost monotonically until their peak around 1990. These tendencies were much different from REASv3. For CO, emission amounts of REASv3 were larger than other results of especially REASv1, EDGARv4.3.2, and CEDS. However, after 2000, emissions and their decreasing trends of other studies were generally comparable to those of REASv3. For NMVOC, results

of REASv3 after 2000 generally agreed well with other studies which showed large decreasing trends except for EDGARv4.3.2 and Zhang et al. (2009) based on Kannari et al. (2007). Trends of NH$_3$ emissions shown in Fig. S16 were similar except for EDGARv4.3.2 before the mid-1990s which showed larger growth rates. Emission amounts of REASv3 were smaller than national inventories by Kannari et al. (2001) and Fukui et al. (2014).

For SEA (see Fig. S17), increasing trends and amounts of SO$_2$ emissions of REASv3 agreed with other results except for

CEDS in the 1990s, Zhang et al. (2009), and Klimont et al. (2013). In CEDS, emissions decreased keenly during the late 1990s. A similar feature was also seen in REASv3 but its rate of decrease was much smaller. For NO$_x$, all results plotted in Fig. S17 indicated monotonically increasing trends of emissions and agreed well until the early 2000s. After that, growth rates of REASv3 became larger than EDGARv4.3.2 and smaller than CEDS (scaled to REASv2.1 for SEA; Hoesly et al., 2018). For BC, REAS series and CEDS showed similar growth rates until around 2005. On the other hand, increasing rates

of Klimont et al. (2017) and EDGARv4.3.2 after 1990 were much smaller and close to those of REASv3 after 2005.

Most results of SO$_2$ emissions in OEA in Fig. S18 show increasing and decreasing trends from the late 1960s and the early 1990s, respectively, although amounts in CEDS from 1970 and 2000 were much smaller. For NO$_x$, all results agreed well until the late 1980s and REASv3, REASv1.1 and EDGARv4.3.2 showed similar increasing trends until around 2000. Emissions of CEDS became almost stable after the late 1980s and started to decrease after 2005. The decreasing rates of

REASv3 and CEDS are close after 2005. On the other hand, emissions of EDGARv4.3.2 were not changed largely after

around 2000. The similar tendencies were shown in the case of SO$_2$. BC emissions of REASv3 and CEDS showed similar trends until 2000. Then, emissions of REASv3 decreased almost monotonically, while those of CEDS were almost stable. Similarly, decreasing rates of EDGARv4.3.2 after 2000 were much smaller than those of REASv3.

For OSA, increasing trends and amounts of SO$_2$ and NO$_x$ emissions were generally similar among studies plotted in Fig. S19.
SO$_2$ emission of Streets et al. (2003a) and Zhang et al. (2009) and NO$_x$ emissions of CEDS (scaled to REASv2.1; Hoesly et al., 2018) were higher than other results. For BC, discrepancies among studies were larger than those of SO$_2$ and NO$_x$, but similar small monotonically growth rates were shown in all results.

### 3.3.4 Relative rations of emissions from each country and region in Asia

Figure 12 compares trends of total emissions in Asia and relative ratios of emissions from China, India, Japan, SEA, OEA,
and OSA among REASv3, CEDS, and EDGARv4.3.2 for SO$_2$, NO$_x$, and BC. Comparisons of other species are presented in Fig. S20. From 1950 to the early 2000s, total SO$_2$ emissions in Asia of all inventories showed similar results. For relative ratios, REASv3 and CEDS values were similar until the mid-2000s. Contributions from Japan were relatively large from 1950 until around 1970 and then, decreased keenly. This was also found in EDGARv4.3.2, but the rate of decrease was smaller than that of REASv3 and CEDS. Then, while emissions of REASv3 decreased largely after the mid-2000s, those of
CDES and EDGARv4.3.2 continued to increase. These discrepancies were mainly due to different trends of emissions from China. Actually, after the mid-2000s, relative ratios of SO$_2$ emissions in China were stable in CEDS and EDGARv4.3.2, but those in REASv3 decreased significantly. Recently, increasing trends of relative ratios of SO$_2$ emissions in India are a common feature in REASv3, EDGARv4.3.2, and CEDS.

For NO$_x$, Asia total emissions of REASv3 and EDGARv4.3.2 were close. Although emissions of CEDS were larger than
REASv3 and EDGARv4.3.2, trends were similar until early 2010. The different trends after 2010 between REASv3 and CEDS were caused by those of emissions in China. For the contributing rates, REASv3 and CEDS generally showed similar temporal variations, although relative ratios of OSA were larger in CEDS. Contribution rates of Japan were large around 1970 and then gradually decreased. Instead, those from China increased almost monotonically until 2010. Similar to the case of SO$_2$, relative ratios of China decreased recently in REASv3, but they were almost stable in CEDS and EDGARv4.3.2. In
addition, contribution rates from India showed gradual increasing trends in all the results

For total Asia emissions of BC, trends of REASv3 and CEDS were similar until the late 1990s, but after 2000 while growth rats of CEDS became larger, emissions of REAS were not changed largely and turned to decrease after 2010. Emission amounts and growth rates of EDGARv4.3.2 were smaller than others until the mid-1990s, but after that the trends were similar to those of REASv3. Compared to SO$_2$ and NO$_x$, temporal variations of relative ratios of BC emissions from each
country and regions were small in all the results. In REASv3, contribution rates of Japan were large before 1970 and then decreased afterwards. On the other hand, in CEDS, contribution rates of Japan after 1970 were larger than those before 1970. After 2000, relative ratios of China in REASv3 were almost stable and showed a marginal decrease after 2011. In CEDS and

EDGARv4.3.2, contribution rates of China increased during the first half of 2000s and then became almost stable. Similar tendencies were seen in OC. Compared to BC, relative ratios of China started to decrease earlier only in REASv3.

For Asia total emissions of CO, although amounts of CEDS were larger than others, trends of all results were close until the early 2000s. After that, REASv3 showed large increases until 2010 and then started to decreased slightly. These tendencies were mainly controlled by emissions in China. Trends of the relative ratios were similar to those of BC. But contribution rates of China in REASv3 increased gradually until the mid-2000s and then decreased, while those in CEDS and EDGARv4.3.2 were almost stable. For NMVOC, total emissions in Asia of REASv3 were smaller than others, but large

increases of emissions were found from the early 1990s. The corresponding feature was shown in contribution rates. Relative ratios of emissions from China in REASv3 increased largely during the 1990s and 2000s. Similar increasing trends were seen in EDGARv4.3.2 but growth rates of REASv3 were much larger. On the other hand, both temporal variations and values of contribution rates of China were relatively small in CEDS.

For NH$_3$, trends of total emissions in Asia of REASv3 were close to EDGARv4.3.2 and slightly larger than CEDS until

around 2000. After that, growth rates of REASv3 were close to CEDS and those of EDGARv4.3.2 became larger. As a result, amounts of total Asia emissions of all inventories became almost the same after 2010. For relative rations of regions, contribution rates of China in REASv3 increased gradually until the mid-2000s and then became almost stable, whereas those in CEDS and EDGARv4.3.2 show slightly decreasing and increasing trends, respectively. In 2015, relative ratios of NH$_3$ emissions from China in REASv3 were between those of EDGARv4.3.2 and CEDS. Compared to EDGARv4.3.2 and

CEDS, contribution rates of NH$_3$ emissions from SEA region were relatively small in REASv3.

### 3.4 Uncertainty

In REASv3, uncertainties in emissions were estimated for each country and region in 1955, 1985, and 2015 using basically the same methodology as that of REASv2.1 (Kurokawa et al., 2013). First, uncertainties in all parameters used to calculate emissions, such as activity data, emission factors, removal efficiencies, and sulfur contents of fuels were estimated in the

range of 2-150%. In estimation of the uncertainties except for activity data, following three causes need to be considered: uncertainties in the data themselves, those caused by selections of the data, and those in settings related to emission controls such as timing of introduction and penetration rates of abatement equipment. In this study, uncertainties in settings of emission controls were explicitly considered only for removal efficiencies. The uncertainties of removal efficiencies were assumed to be zero for emission sources where no emission controls were considered which means that uncertainties caused

by neglecting emission controls were not considered. Furthermore, for emission sources where introduction rates of abatement equipment were small, uncertainties caused by settings of emission controls were assumed to be small. Then, uncertainties in emissions from power plants, industries, road transport, other transport, domestic and other sectors, as well as uncertainties in total emissions were calculated for all the species. The uncertainties of different sub-sectors and activities were combined in quadrature assuming they were independent. On the other hand, for uncertainties of national emissions in

China, India, and Japan, those in their sub-regions were added linearly. Details of the methodology and settings of

uncertainties of each component were described in Sect. S10 of the Supplement. Similar to REASv2.1, uncertainties of emissions that were not originally developed in REASv3 ($NH_3$ emissions from manure management and fertilizer application, and NMVOC evaporative emissions from Japan and Republic of Korea) were not evaluated in this study.

Table 4 summarizes the estimated uncertainties in total emissions of each species for China, India, Japan, SEA, OEA, and
OSA in 1955, 1985, and 2015. Uncertainties in emissions from each sector were provided in the Supplement tables (Table S1 for $SO_2$, $NO_x$, CO, $CO_2$, $PM_{10}$, $PM_{2.5}$, BC, and OC, in Table S2 for NMVOC, and in Table S3 for $NH_3$). For most regions and years, uncertainties for $SO_2$, $NO_x$ and $CO_2$ are smaller than other species. Major emission sources of these species are power plants and large industry sectors. Uncertainties of activity data of these species were assumed to be small because power plants and large industries are critically important for each country and related statistics are expected to be accurate.
In addition, uncertainties of emission factors of combustion at high temperature in power plants and large industries are considered to be small. For $SO_2$ emissions in China, uncertainties in 2015 were estimated to be slightly larger than those in 1985 due to uncertainties for removal efficiencies which were not considered in 1985. The same situation was found in uncertainties of $NO_x$ emissions from power plants in China between 1985 and 2015. Lack of detailed information for changes of technologies such as combustion burners and abatement equipment affect uncertainties of recent emission trends
in Asia. For South and Southeast Asia, uncertainties of $SO_2$ emissions in 1985 were slightly smaller than those in 2015. This is because settings of sulfur contents in fuels were based on surveys conducted in 1990 (Kato and Akimoto, 1992) and thus, the uncertainties in 1985 were assumed to be smaller than those in 2015. In REASv3, information of temporal variations of sulfur contents in fuels both by changes of fuel properties and by low-sulfur fuel regulations was limited which were also causes of uncertainties of emission trends. In general, uncertainties of emissions in REASv3 were smaller in recent years
because activity data of recent years are more accurate. However, detailed surveys for recent changes of technologies and information of emission controls are essential in future studies.

On the other hand, uncertainties of PM species are large compared to other species for most regions and years. For most countries in Asia, a majority of their emissions was from combustion at relatively low temperatures in small industries and residential sectors. Accuracies of activity data and emission factors for these sources are assumed to be low, especially for
biofuel combustion. Therefore, uncertainties of OC emissions mainly from biofuel combustion in Asia are the largest for most regions and years. Uncertainties of $PM_{10}$ are generally smaller than other PM species. This is because for $PM_{10}$ emissions, contribution rates of power plants and industry sectors are generally larger than those of other PM species. For CO and NMVOC, in general, uncertainties of emission factors are assumed to be greater than $SO_2$, $NO_x$, and $CO_2$, but smaller than $PM_{2.5}$, BC and OC. Therefore, uncertainties of total CO and NMVOC emissions are generally between those of
other species. For Southeast and South Asia, uncertainties of CO and NMVOC are comparable to $PM_{10}$ as their relative contribution from biofuel combustion is large.

Uncertainties in emissions from Japan are lesser than those of other countries and regions. This is mainly due to the accuracy of activity data. Accessibility to detailed information in Japan is relatively high compared to other countries in REASv3. In Japan, uncertainties of emission in 1985 were comparable to or slightly smaller than those in 2015. This is because relative

ratios of emissions from road transport whose uncertainties were the smallest in Japan were reduced largely from 1985 to 2015. For China and India, accuracies of emissions are generally improved for most species compared to REASv2.1 using information from recently published literatures of emission inventory of these countries. However, the improvement is not significant due to the lack of country specific information. This situation is almost the same for other countries and regions. Although studies of national emission inventories in Asia are being published, as described in Section 1, information on

technologies related to emissions and their introduction rates is not as easily available. Therefore, continuous efforts to update emission inventories by collecting information of each country and region are essential. For all countries, uncertainties of emissions in 1955 were much larger than those in 2015. This is because most activity data were not obtained directly from statistics, especially in the early half of the target period of REASv3.1. In this study, activity data, which were not available in statistics were extrapolated or assumed using proxy data as described in Section 2. In order to reduce

uncertainties of emissions in long past years, these procedures need to be considered based on detailed information of each country and region during the period.

## 4 Summary and remarks

A long historical emission inventory of major air and climate pollutants in Asia during 1950-2015 was developed as Regional Emission inventory in ASia version 3 (REASv3). Target species were $SO_2$, $NO_x$, CO, NMVOC, $NH_3$, $PM_{10}$, $PM_{2.5}$,

BC, OC, and $CO_2$ and the domain areas included East, Southeast, and South Asia. Emissions from fuel combustion in power plants, industries, transport, and domestic sectors and those from industrial processes were estimated for all the species. In addition, emissions from evaporative sources were included in NMVOC and those from agricultural activities and human physiological phenomenon were considered for NH. REASv3 provides gridded data as well as emissions from each country and sub-region. Spatial resolution is mainly $0.25° \times 0.25°$ and large power plants are treated as point sources. Temporal

resolution is monthly. Emissions were estimated based on information of technologies related to emission factors and removal efficiencies, although available data and literatures are limited in the case of Asia. Activity data for recent years were collected from international and national statistics and those of past years, when detailed information was not available, were extrapolated using proxy data for the target period of REASv3. Details of methodologies such as data sources and treatments, settings of emission factors and emission controls, and related assumptions were provided in the supplement

document entitled "Supplementary information and data to methodology of REASv3".

Total emissions in Asia averaged during 1950-1955 and 2010-2015 (growth rates in these 60 years) are: $SO_2$: 3.2 Tg, 42.4 Tg (13.1); $NO_x$: 1.6 Tg, 47.3 Tg (29.1); CO: 56.1 Tg, 303 Tg (5.4); NMVOC: 7.0 Tg, 57.8 Tg (8.3); $NH_3$: 8.0 Tg, 31.3 Tg (3.9); $CO_2$: 1.1 Pg, 18.6 Pg (16.5); $PM_{10}$: 5.9 Tg, 30.2 Tg (5.1); $PM_{2.5}$: 4.6 Tg, 21.3 Tg (4.6); BC: 0.69 Tg, 3.2 Tg (4.7); and OC: 2.5 Tg, 6.6 Tg (2.7). Clearly, all the air pollutant emissions in Asia increased significantly during these six decades.

However, situations were different among countries and regions. In recent years, the relative contribution of air pollutant emissions from China was the largest along with rapid increase in economic growth, but most species have reached their

peaks and the growth rates of other species have become at least small or almost zero. For $SO_2$ and $NO_x$, introduction of abatement equipment, especially for coal-fired power plants, such as FGD and SCR were considered to be effective in reducing emissions. For PM species, in addition to control equipment in industrial plants, emissions decreased recently due to reduced usage of biofuels. On the other hand, air pollutant emissions from India showed an almost continuous increase. Growth rates were larger for $SO_2$ and $NO_x$, but their structures of emissions were different. Large parts of $SO_2$ emissions were obtained from coal combustion in power plants and industrial sector, and the recent rapid increase of $SO_2$ emission was mainly from coal-fired power plants. For $NO_x$, contribution from road transport especially diesel vehicles were almost comparable with those of power plants. For PM species, a majority of emissions was from the residential sector in the 1950s-1960s and its contribution is still considered to be large. Recent increasing trends were mainly caused by emissions from power and industrial plants and road vehicles. Trends in Japan were much different than those of the whole of Asia. Emissions increased rapidly along with economic growth during the 1950s-1970s, but those of most species were reduced largely from peak values. In addition, peak years were mostly 40 years ago, reflecting the time series of introduction of control measures to mitigate air pollution. Similar features were found in Republic of Korea and Taiwan. For other countries in Asia, emissions of air pollutants generally showed increasing trends along with economic situation and motorization. As described above, trends and spatial distribution of air pollutants in Asia are not simple and are becoming complicated.

Mitigation of air and climate pollutant emissions is an urgent issue in most Asian countries, but the situation is different country-wise. In this study, detailed discussion on effects of emission controls were conducted only for China and Japan due to limitation of information. Therefore, continuous efforts to develop and update emission inventories in Asia based on country specific information are essential especially for countries and regions other than China and Japan. On the other hand, there are inevitable uncertainties in parameters required to develop emission inventories, such as activity data and emission factors. In addition, it is fundamentally impossible to develop a real-time emission inventory because there is a time lag in the publication of basic statistics essential to estimate emissions. Recently, satellite observation data of air pollutants are becoming available at a finer scale for many species, such as $NO_x$, $SO_2$ and $NH_3$. Evaluations and improvements of REASv3 based on these data as well as results of modeling studies, such as inverse modeling are more important next steps. Also, addition of target species, especially $CH_4$, which is one of the key species to mitigate both air pollution and global warming is another important task for future studies.

**Data availability:**

Monthly gridded emission data sets at 0.25° × 0.25° resolution for major sectors from 1950 to 2015 are available from a data download site of REAS. The URL of the site is http://www.nies.go.jp/REAS/. Country and regional emission table data for major sectors during 1950-2015 and those for major fuel types are also provided at the site. Note that datasets of REASv3.1 were released after a publication of Kurokawa et al. (2019) from December 2019. The datasets were revised and the updated

data are available as REASv3.2 together with a publication of this paper. Differences between REASv3.2 and REASv3.1 were presented and discussed in the Supplement document entitled "Differences between REASv3.2 and REASv3.1".

**Author contribution:**

JK and TO conducted the study design. JK contributed to actual works for development of REASv3 such as collecting data and information, settings of parameters, calculating emissions and creating final data sets. JK and TO analyzed and discussed the estimated emissions in REASv3. JK prepared the manuscript with contributions from TO.

**Competing interest:**

The authors declare that they have no conflict of interest.

**Acknowledgements:**

This work was supported by the Environment Research and Technology Development Fund (S-12) of the Environmental Restoration and Conservation Agency of Japan and JSPS KAKENHI Grant Number 19K12303. We appreciate K. Kawashima (Mitsubishi UFJ Research and Consulting Co., Ltd.) and T. Fukui (The Institute of Behavioral Sciences) for 985 their great support in collecting activity data and survey information for settings of parameters. We acknowledge K. Yumimoto (Kyushu University), S. Itahashi (Central Research Institute of Electric Power Industry), T. Nagashima (National Institute for Environmental Studies), and T. Maki (Meteorological Research Institute) for their valuable suggestions to improve REAS. We are grateful to D. Goto (National Institute for Environmental Studies) for his support to update the data download site of REAS. We thank Y. Kiriyama (Asia Center for Air Pollution Research) for his help in drawing figures of 990 gridded emission maps.

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

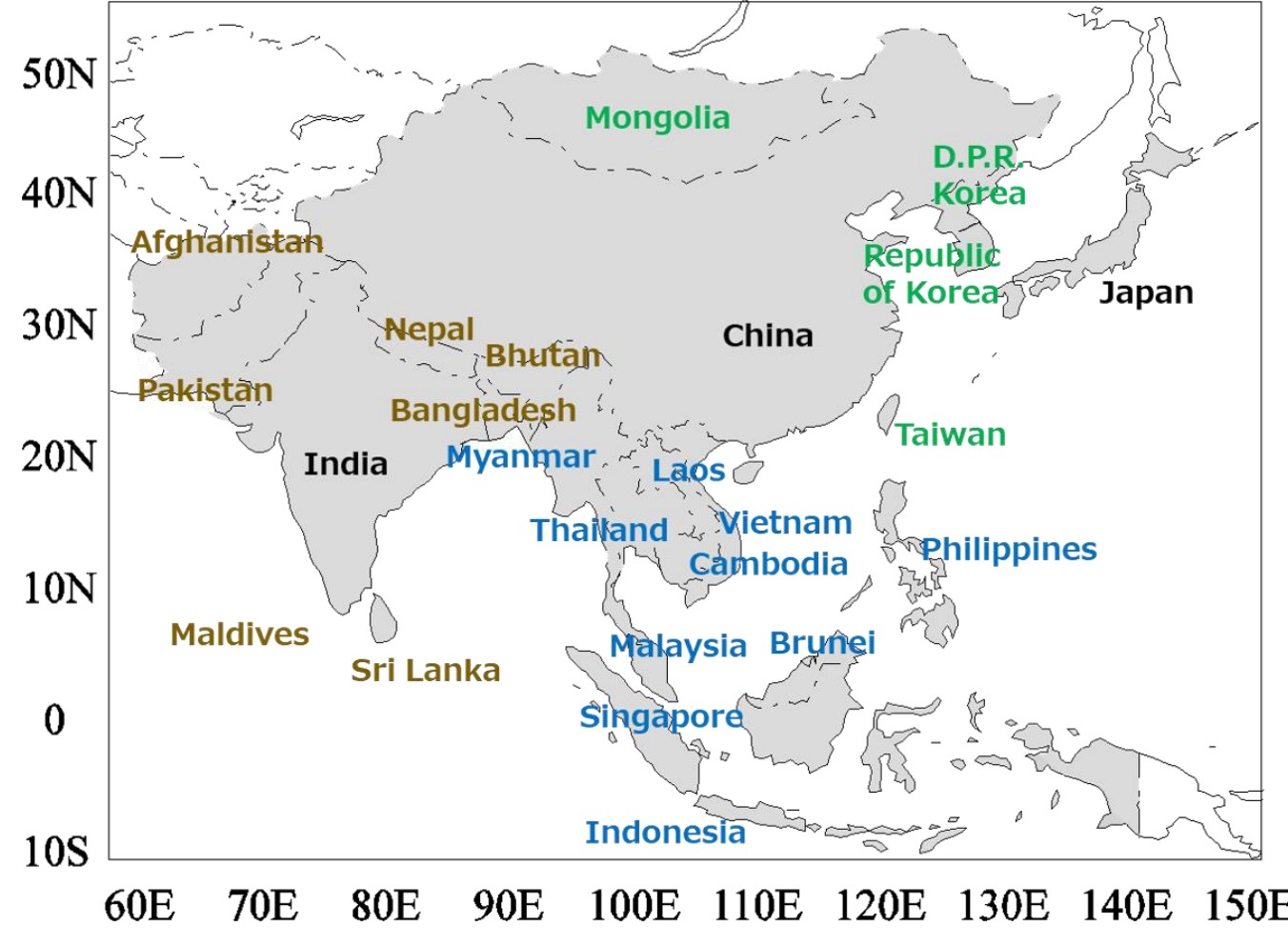

**Figure 1.** Domain and target countries of REASv3. In this paper, countries written in blue, green, and brown characters were defined as SEA (Southeast Asia), OEA (East Asia other than China and Japan), and OSA (South Asia other than India), respectively.

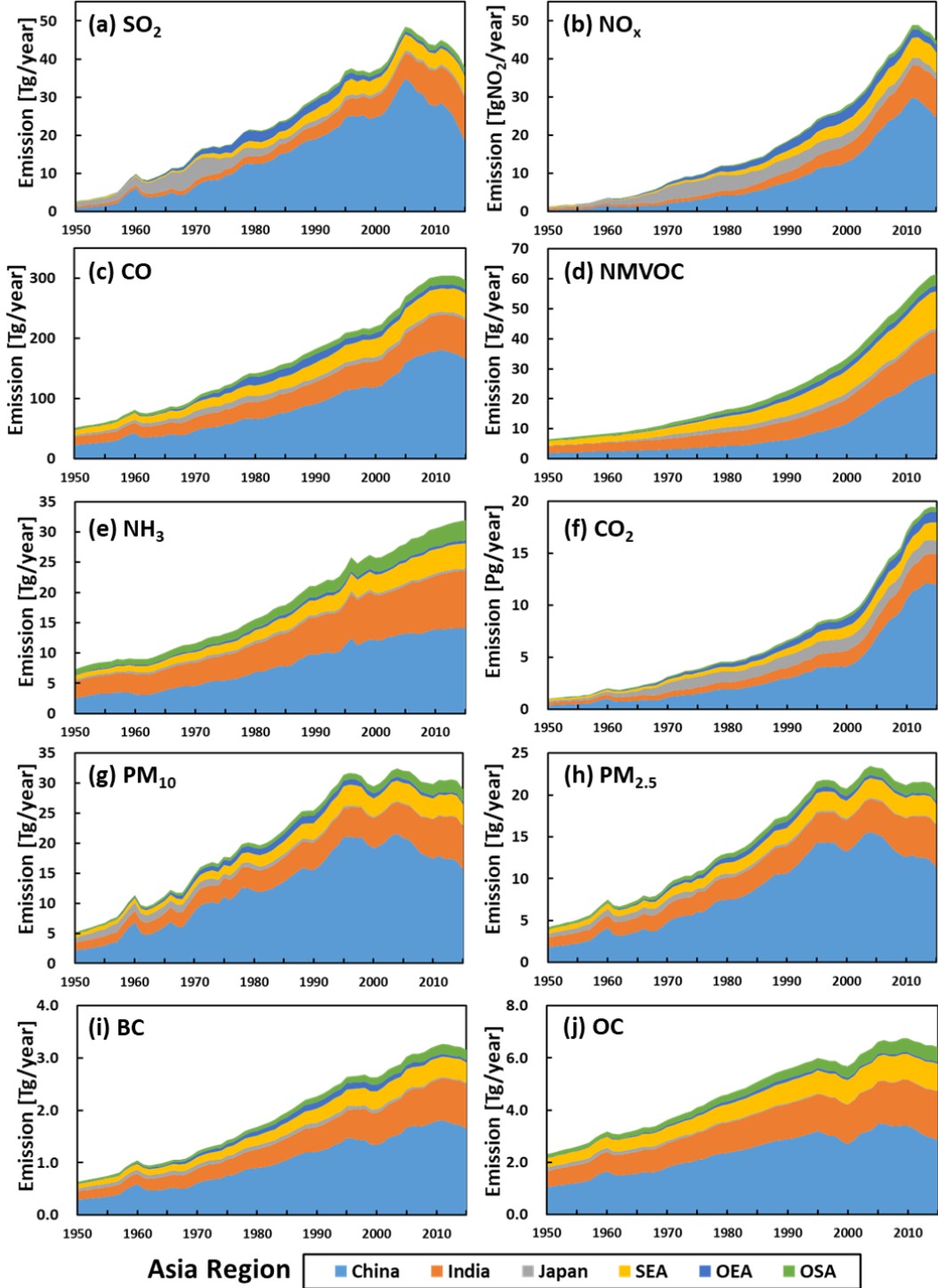

**Figure 2.** Trends of (a) $SO_2$, (b) $NO_x$, (c) CO, (d) NMVOC, (e) $NH_3$, (f) $CO_2$, (g) $PM_{10}$, (h) $PM_{2.5}$, (i) BC, and (j) OC emissions in Asia during 1950-2015 for each region. See Fig. 1 for countries included in SEA, OEA, and OSA.

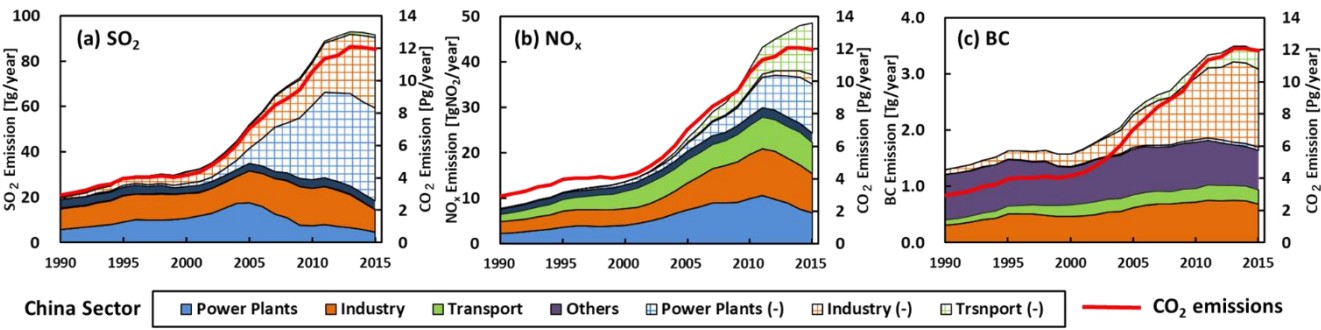

**Figure 3.** Emissions of (a) SO₂, (b) NOₓ, and (c) BC from each major sector in China during 1990-2015. Solid colored areas are actual emissions and hatched ones (-) are reduced emissions due to control measures. Red lines in the panels are total CO₂ emissions.


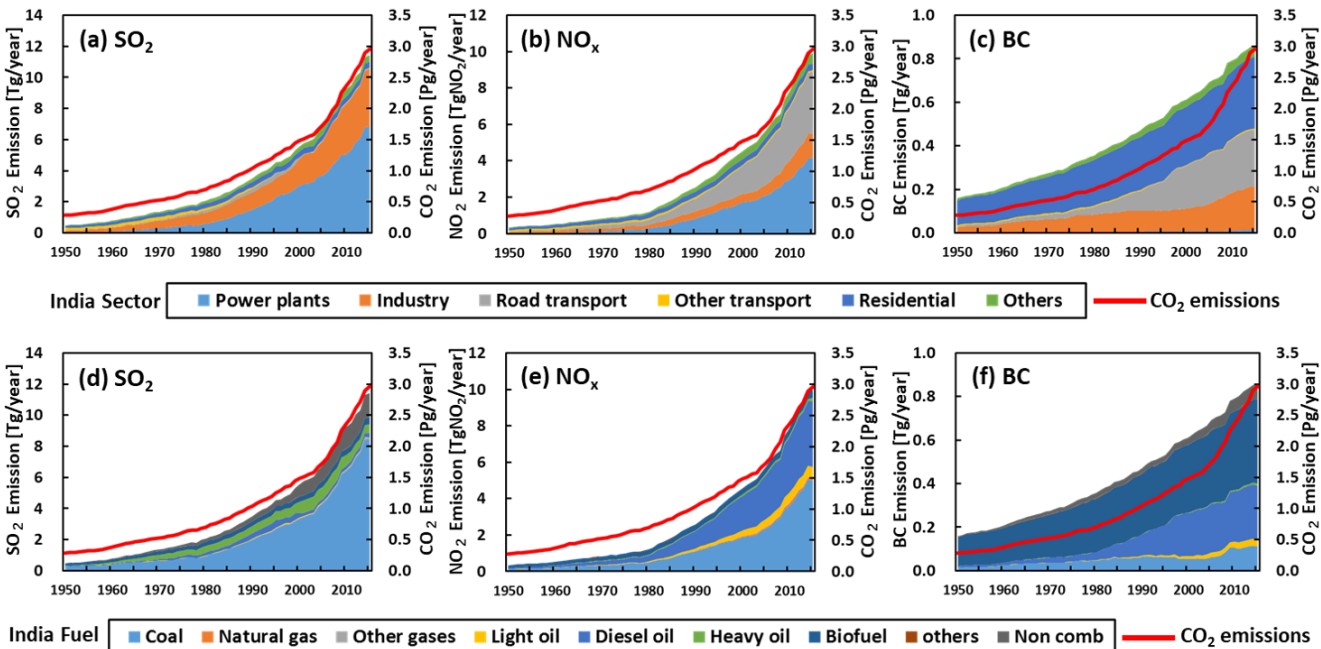

**Figure 4.** Emissions of (a, d) $SO_2$, (b, e) $NO_x$, and (c, f) BC from each major sector category (upper panels) and fuel type (lower panels) in India from 1950 to 2015 (Non comb = Non combustion sources). Red lines in the panels are total $CO_2$ emissions.

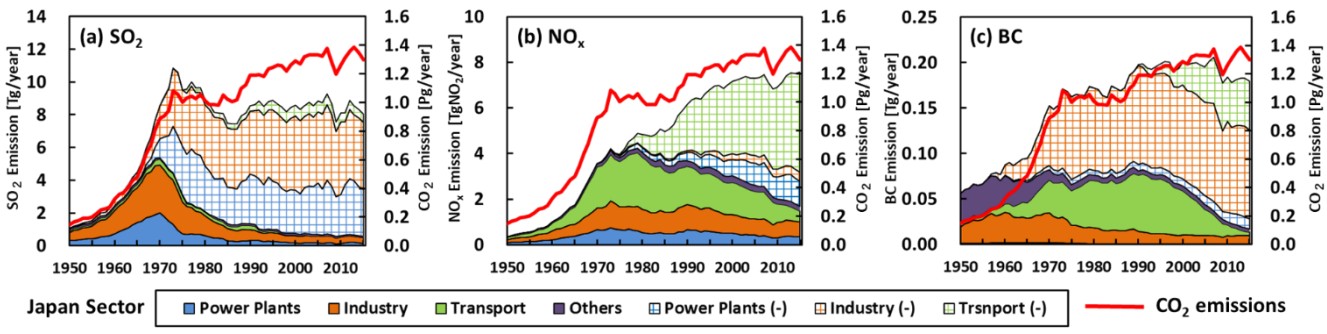


**Figure 5.** Emissions of (a) SO₂, (b) NOₓ, and (c) BC from each major sector in Japan during 1950-2015. Solid colored areas are actual emissions and hatched ones (-) are reduced emissions due to control measures. Red lines in the panels are total CO₂ emissions.

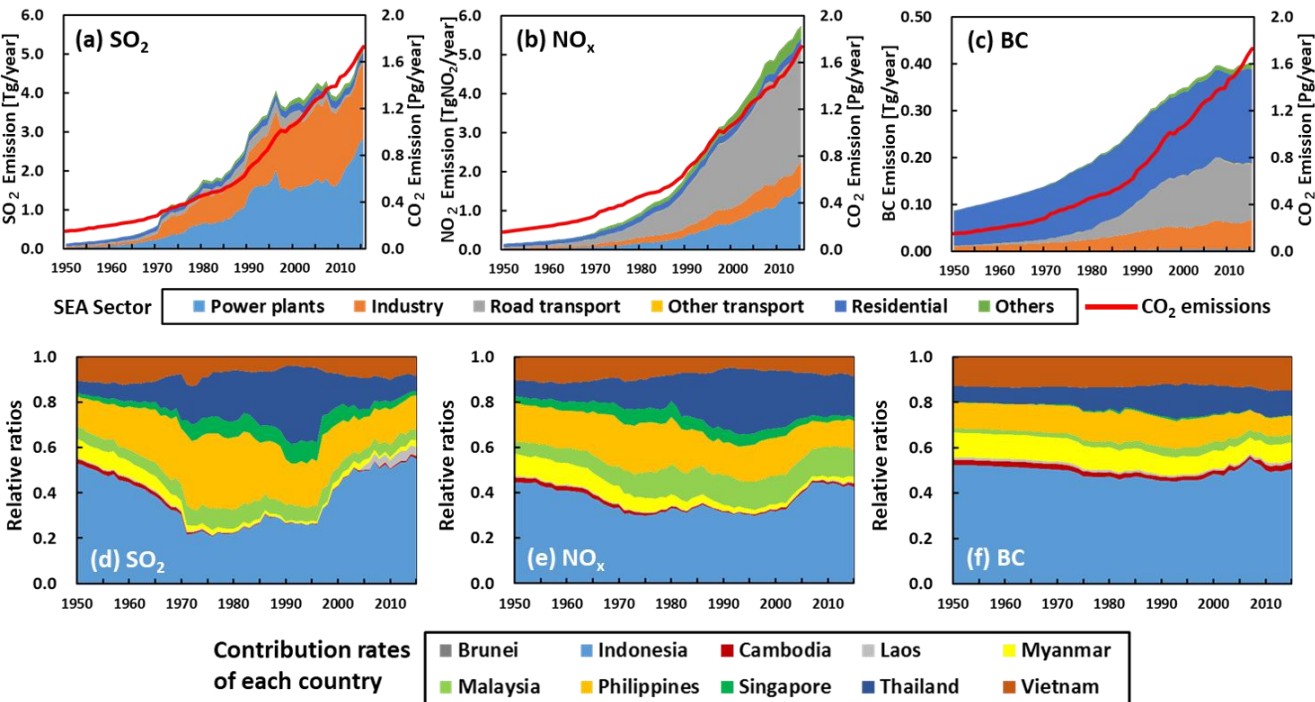

**Figure 6.** Emissions of (a) $SO_2$, (b) $NO_x$, and (c) BC from each major sector in SEA (upper panels) and (d, e, f) relative ratios of emissions from each country in SEA (lower panels) during 1950-2015. Red lines in the upper panels are total $CO_2$ emissions.

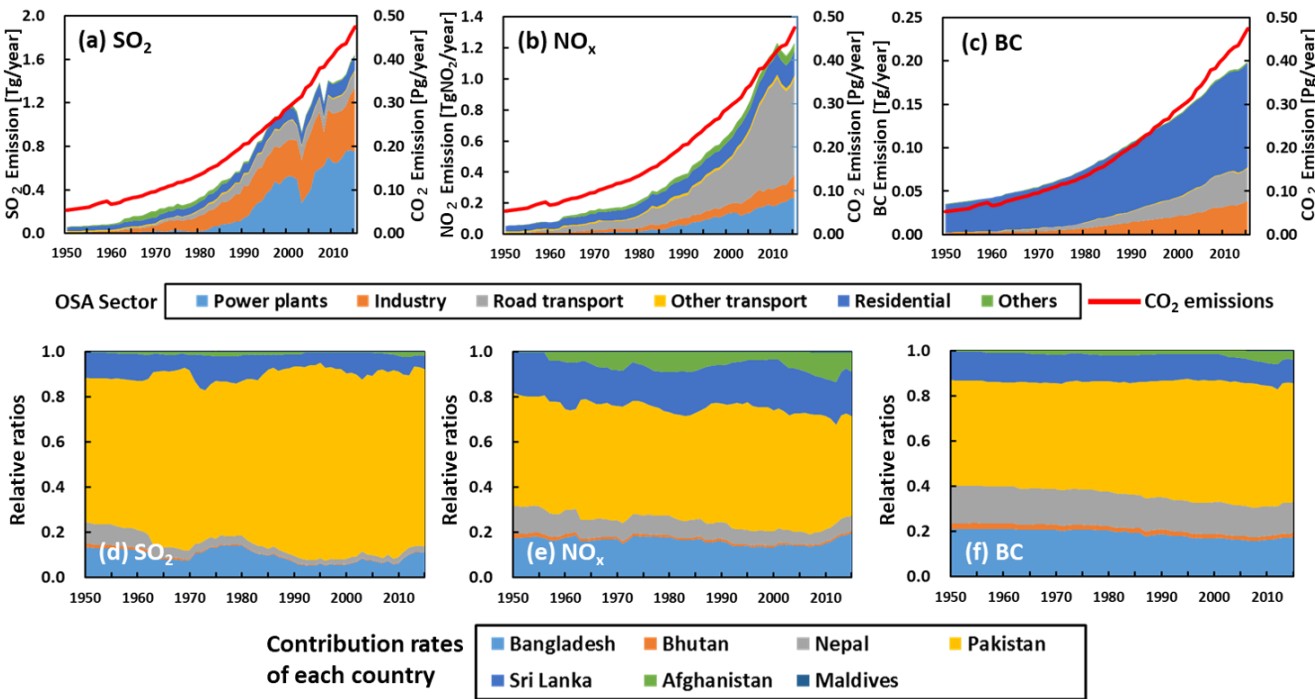

**Figure 7.** Emissions of (a) SO₂, (b) NOₓ, and (c) BC from each major sector in OSA (upper panels) and (d, e, f) relative ratios of emissions from each country in OSA (lower panels) from 1950 to 2015.

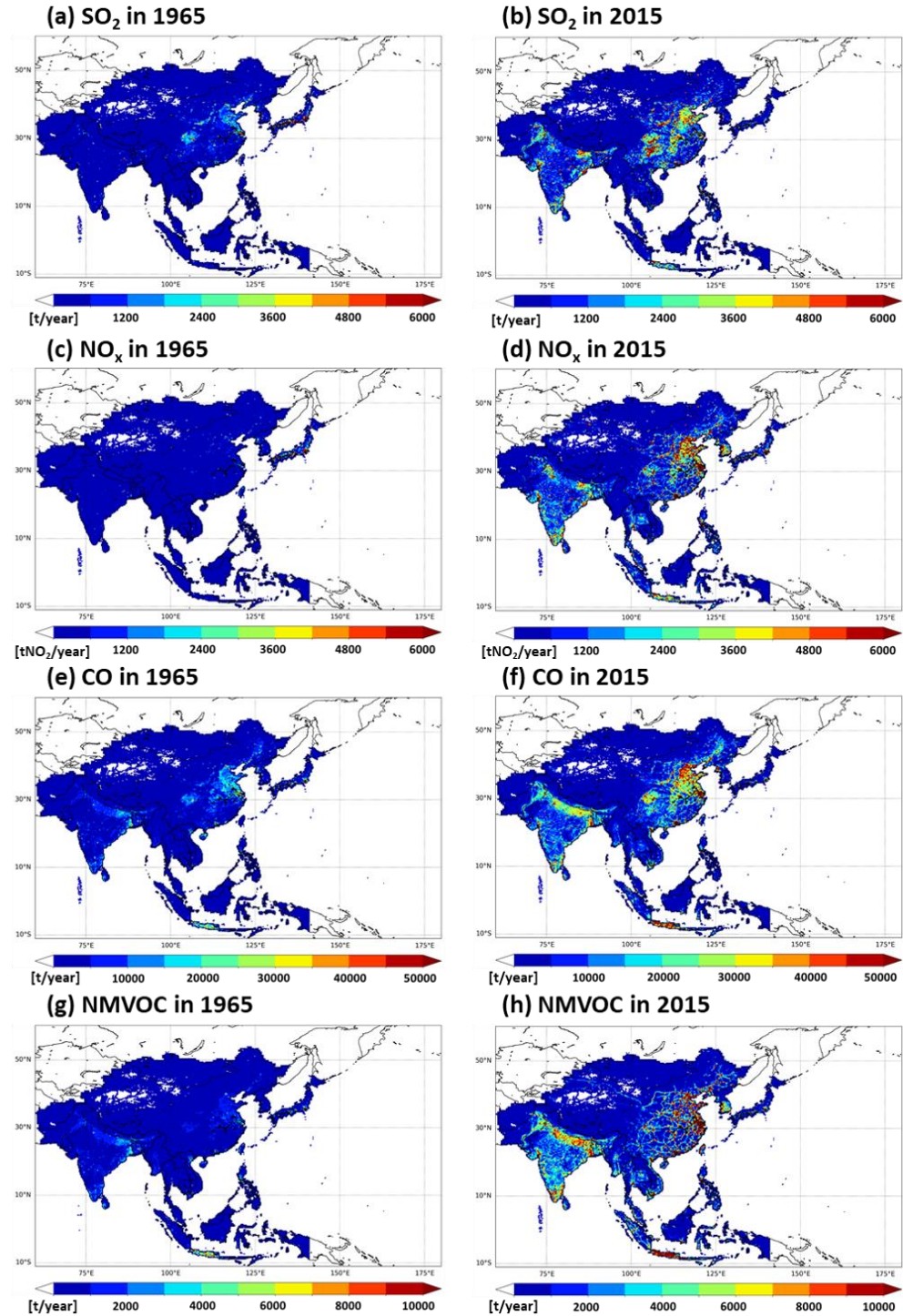

**Figure 8.** Grid maps of annual emissions of (a, b) SO$_2$, (c, d) NO$_x$, (e, f) CO, (g, h) NMVOC, (i, j) NH$_3$, (k, l) PM$_{2.5}$, (m, n) BC, and (o, p) OC in 1965 (left panels) and 2015 (right panels).

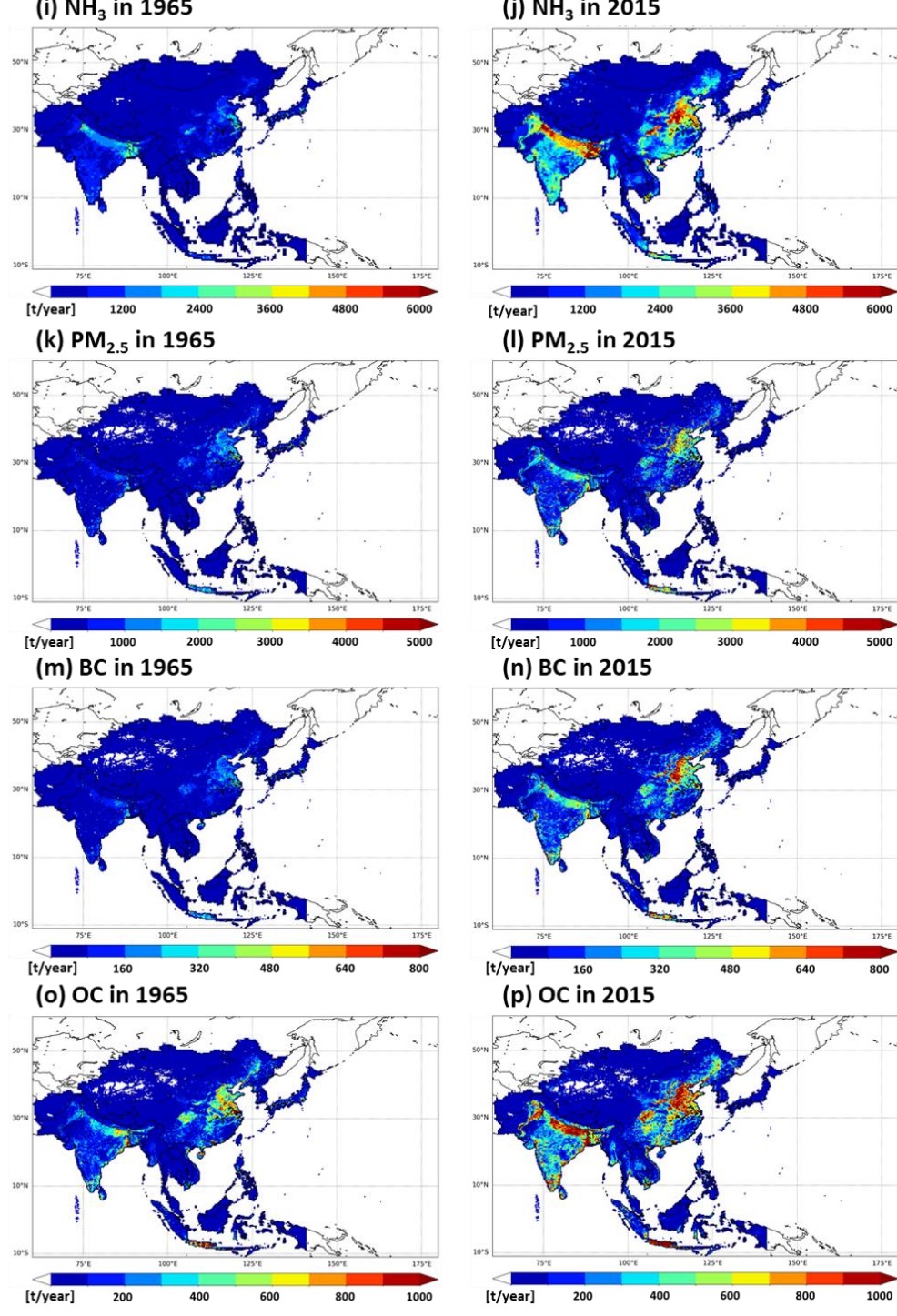


**Figure 8.** Continued.

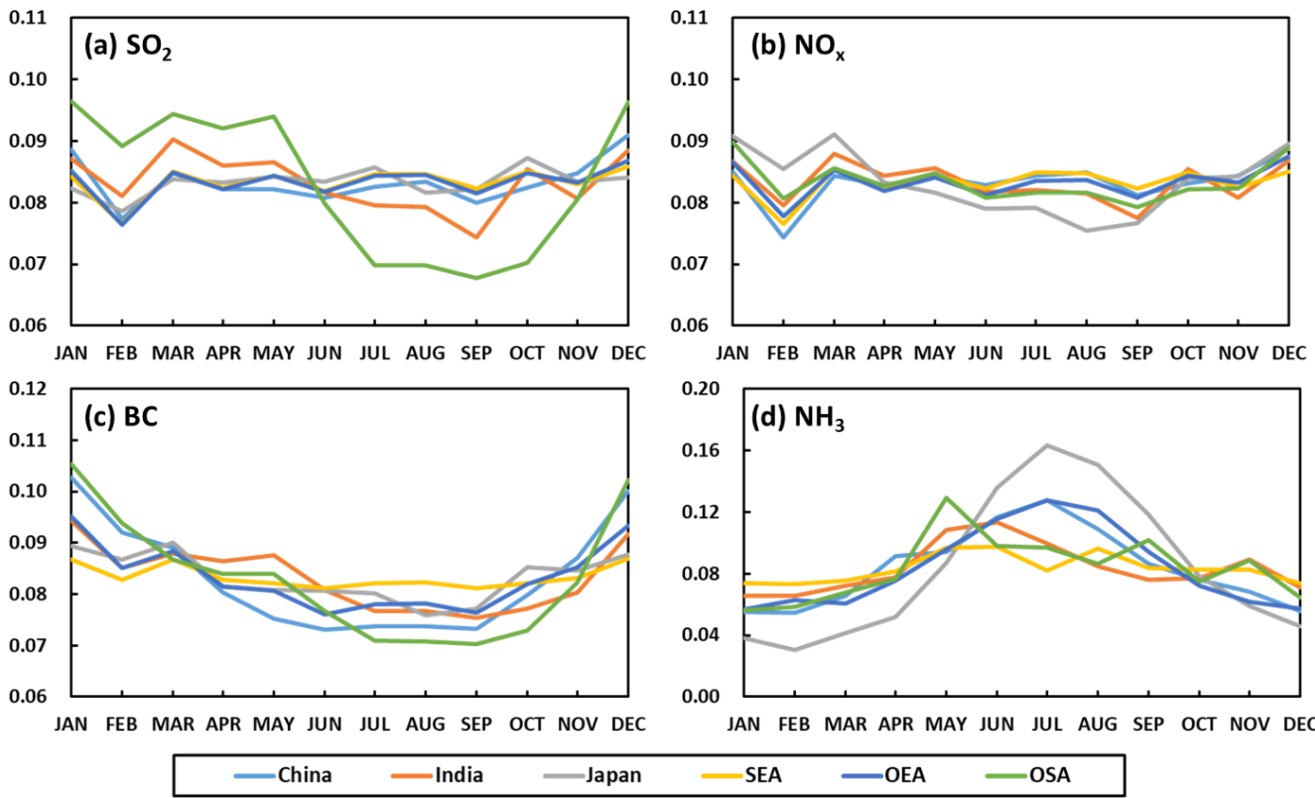

**Figure 9.** Monthly variations of (a) SO$_2$, (b) NO$_x$, (c) BC, and (d) NH$_3$ emissions for each region of Asia in 2015. See Fig. 1 for definitions of SEA OEA, and OSA.


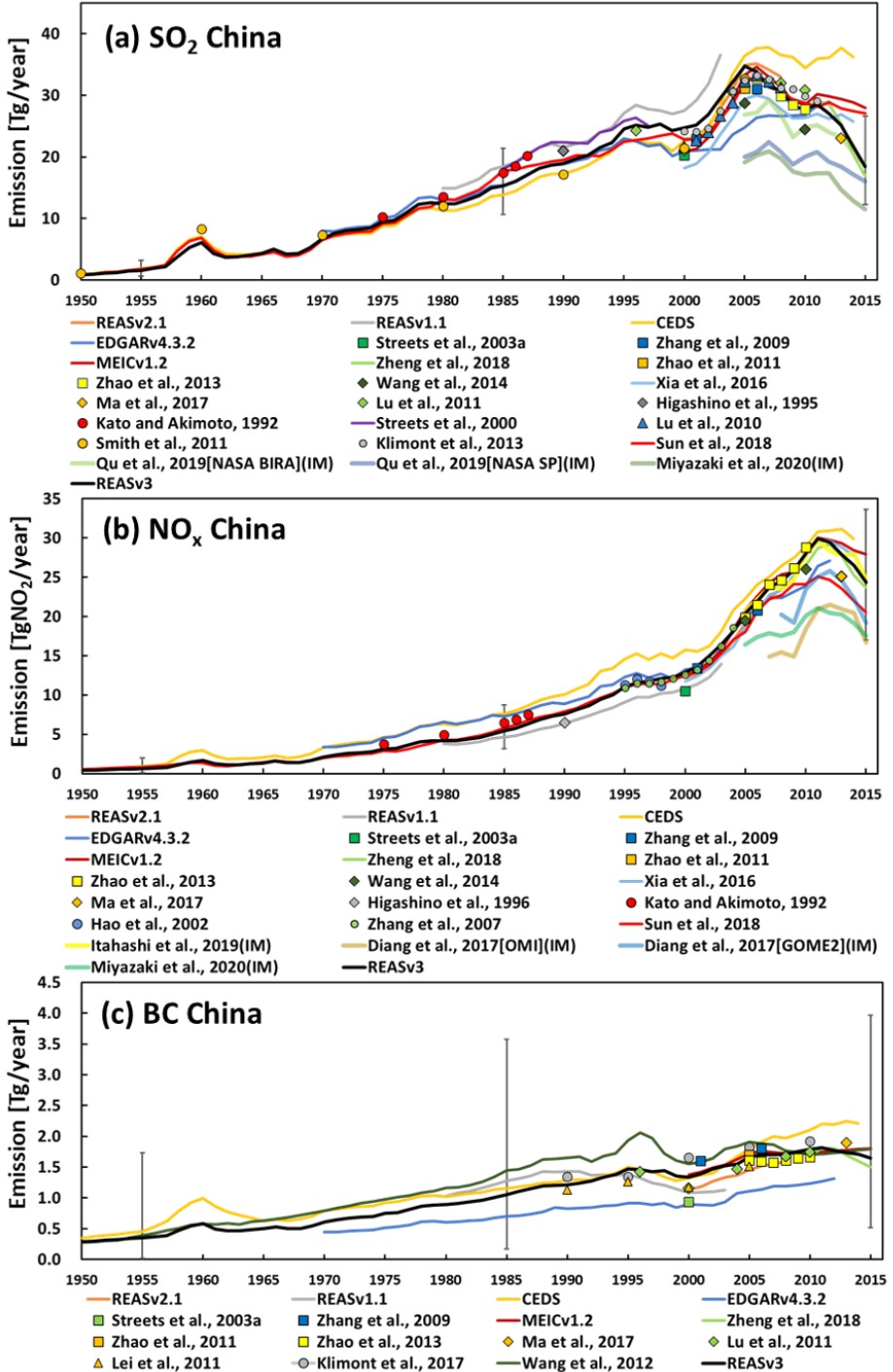

**Figure 10.** Comparison of (a) SO₂, (b) NOₓ, and (c) BC emissions in China between REASv3 and other studies. Note that emissions from domestic and fishing ships were excluded from REAS series, CEDS, EDGARv4.3.2, and Higashino et al. (1996). IM means estimates by inverse modeling. Error bars indicate the uncertainty range of REASv3 in 1955, 1985, and 2015.

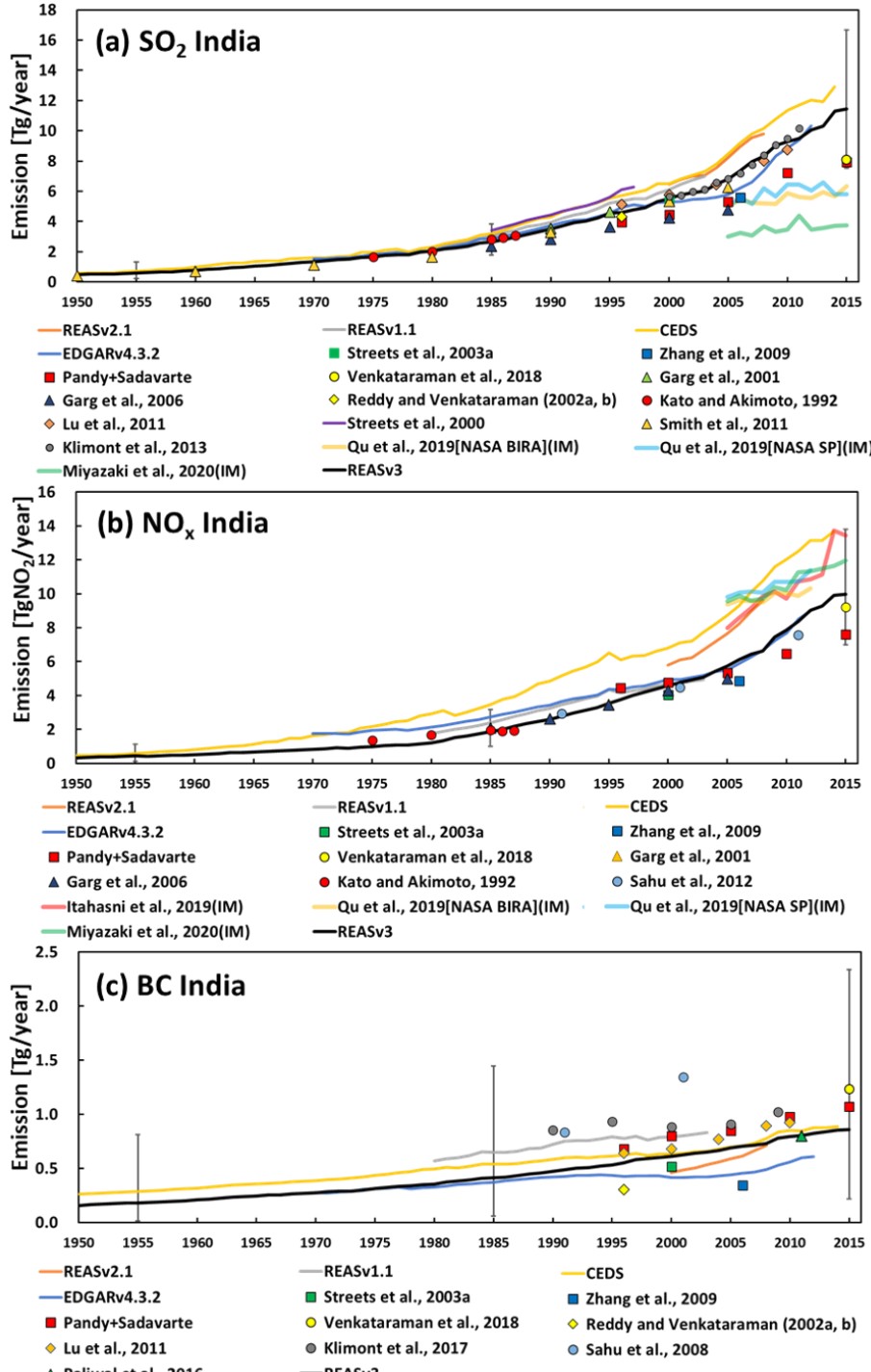


**Figure 11.** Comparison of (a) SO₂, (b) NOₓ, and (c) BC emissions in India between REASv3 and other studies. Note that values of "Pandy+Sadavarte" are calculated from Pandey and Venkataraman (2014) and Sadavarte and Venkataraman (2014). Emissions from domestic and fishing ships were excluded from REAS series, CEDS, EDGARv4.3.2, Garg et al. (2006) and Paliwai et al. (2016). IM means estimates by inverse modeling. Error bars indicate the uncertainty range of REASv3 in 1955, 1985, and 2015.

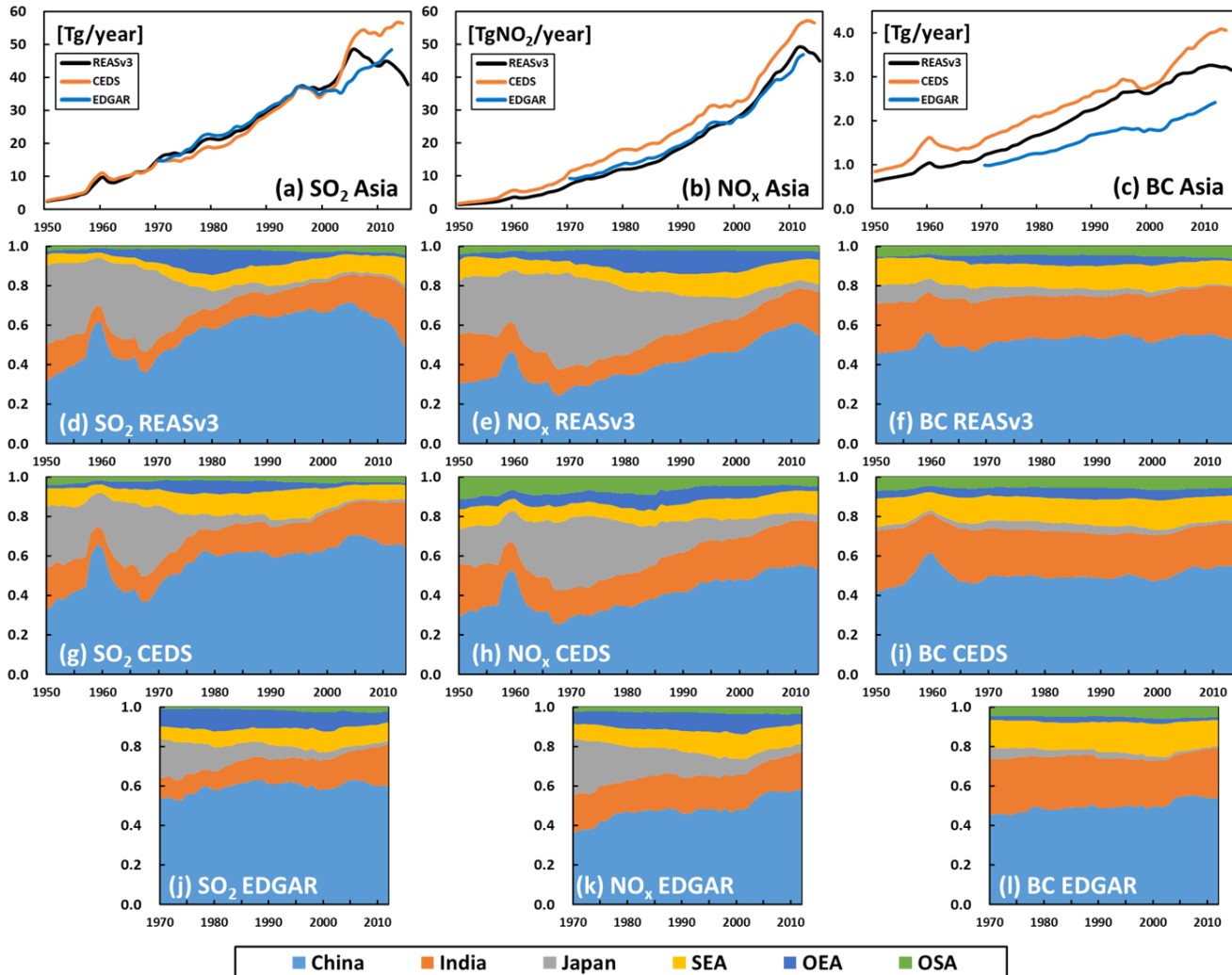

**Figure 12.** Comparison of trends of (a) SO₂, (b) NOₓ and (c) BC emissions in Asia and relative ratios of emissions from China, India, Japan, SEA, OEA, and OSA for (d, g, j) SO₂, (e, h, k) NOₓ, and (f, i, l) BC among (d, e, f) REASv3, (g, h, i) CEDS, and (j, k. l) EDGARv4.3.2. Note that periods of CEDS and EDGARv4.3.2 shown here are during 1950-2014 and 1970-2012, respectively. See Fig. 1 for definitions of SEA, OEA, and OSA.

**Table 1.** General information on REASv3.

| Item | Description |
| --- | --- |
| Species | $SO_2$, $NO_x$, CO, NMVOC, $NH_3$, $CO_2$, $PM_{10}$, $PM_{2.5}$, BC, and OC |
| Years | 1950–2015 |
| Areas | East, Southeast, and South Asia |
| Emission sources | Fuel combustion in power plans, industry, transport, and domestic sectors; Industrial processes; Agricultural activities (fertilizer application and livestock); and Others (fugitive emissions, solvent use, human, etc.) |
| Spatial resolution | 0.25 degree by 0.25 degree |
| Temporal resolution | Monthly |
| Data distribution | http://www.nies.go.jp/REAS/ |

**Table 2.** Emission inventories from other research works and officially opened data utilized in REASv3.

| Other emission inventories and data sources | How utilized in REASv3 |
|---|---|
| VOC Emission Inventory in Japan (MOEJ, 2017) | Evaporative emissions of NMVOC in Japan[a] |
| The National Air Pollutants Emission Service of the National Institute of Environmental Research (http://airemiss.nier.go.kr/mbshome/mbs/airemiss/index.do) | Evaporative emissions of NMVOC in Republic of Korea[a] |
| REASv2.1 (Kurokawa et al., 2013; JPEC 2012a, b, c; 2014) | $NH_3$ emissions from agricultural sources in Japan[b] |
| REASv1.1 (Yamaji et al., 2004; Yan et al., 2003) | $NH_3$ emissions from agricultural sources in countries and regions other than Japan[b] |
| REASv2.1 (Kurokawa et al., 2013; JPEC 2012a, b, c; 2014) | Grid allocation factors for manure management[c] and road transport sectors for Japan[d] |
| EDGARv4.3.1 (Crippa et al., 2016) | Grid allocation factors for manure management[c] and road transport[d] sectors for countries and regions other than Japan |

[a]See Sect. S5.3 of the Supplement. [b]See Sect. 2.4. [c]See Sect. S8.1 of the Supplement. [d]See Sect. 2.6.

**Table 3.** Summary of national emissions in 2015 for each species and total annual emissions in Asia in 1950, 1960, 1970, 1980, 1990, 2000, and 2010-2015 (Gg yr$^{-1}$).

| Country | SO$_2$ | NO$_x$[a] | CO | NMVOC | NH$_3$ | CO$_2$[b] | PM$_{10}$ | PM$_{2.5}$ | BC | OC |
|---|---|---|---|---|---|---|---|---|---|---|
| China | 18404 | 24318 | 165133 | 28189 | 14063 | 11941 (11466) | 15501 | 11342 | 1643 | 2860 |
| India | 11438 | 9969 | 64366 | 14286 | 9505 | 2959 (2290) | 7213 | 5052 | 858 | 1868 |
| Japan | 565 | 1687 | 3877 | 895 | 349 | 1300 (1269) | 129 | 89 | 17 | 13 |
| Korea, D.P.R. | 116 | 200 | 2663 | 134 | 92 | 29 (26) | 106 | 56 | 11 | 18 |
| Korea, Rep of | 336 | 1120 | 1931 | 960 | 170 | 689 (681) | 139 | 114 | 19 | 34 |
| Mongolia | 99 | 127 | 986 | 50 | 139 | 18 (17) | 44 | 20 | 2.9 | 3.2 |
| Taiwan | 124 | 371 | 1027 | 770 | 85 | 281 (279) | 45 | 37 | 6.9 | 7.3 |
| Brunei | 4.0 | 13 | 29 | 43 | 3.8 | 6.1 (6.1) | 7.5 | 2.9 | 0.2 | 0.1 |
| Cambodia | 55 | 61 | 1087 | 212 | 78 | 22 (8.5) | 115 | 69 | 9.0 | 32 |
| Indonesia | 2852 | 2463 | 20517 | 6130 | 1591 | 655 (461) | 1606 | 1160 | 196 | 556 |
| Laos | 201 | 35 | 325 | 66 | 67 | 12 (7.8) | 46 | 25 | 3.6 | 10 |
| Malaysia | 233 | 613 | 1288 | 936 | 163 | 230 (225) | 206 | 119 | 14 | 12 |
| Myanmar | 154 | 121 | 2925 | 867 | 621 | 59 (23) | 184 | 165 | 29 | 98 |
| Philippines | 786 | 767 | 3292 | 898 | 388 | 134 (110) | 284 | 183 | 38 | 61 |
| Singapore | 87 | 89 | 76 | 302 | 6.4 | 46 (46) | 81 | 62 | 1.2 | 0.5 |
| Thailand | 341 | 1137 | 5436 | 1543 | 542 | 320 (250) | 522 | 363 | 49 | 125 |
| Vietnam | 436 | 507 | 6078 | 1552 | 747 | 250 (198) | 587 | 362 | 59 | 146 |
| Afghanistan | 24 | 97 | 404 | 93 | 251 | 9.4 (8.0) | 18 | 14 | 6.9 | 4.4 |
| Bangladesh | 171 | 305 | 2755 | 704 | 883 | 110 (77) | 519 | 287 | 40 | 102 |
| Bhutan | 3.3 | 6.8 | 269 | 55 | 9.5 | 4.7 (0.6) | 29 | 19 | 3.0 | 10 |
| Maldives | 3.1 | 4.1 | 9.4 | 3.7 | 0.4 | 0.8 (0.8) | 0.2 | 0.2 | 0.1 | 0.0 |
| Nepal | 42 | 64 | 2381 | 533 | 321 | 40 (7.0) | 207 | 161 | 26 | 89 |
| Pakistan | 1310 | 573 | 8576 | 2031 | 1772 | 273 (161) | 1310 | 841 | 105 | 324 |
| Sri Lanka | 92 | 187 | 1382 | 374 | 103 | 37 (20) | 135 | 98 | 19 | 49 |
| Asia[c] 1950 | 2540 | 1339 | 51804 | 6551 | 7310 | 1005 (262) | 5089 | 4162 | 630 | 2308 |
| Asia[c] 1960 | 9880 | 3639 | 81220 | 8461 | 8968 | 2016 (1125) | 11405 | 7487 | 1040 | 3185 |
| Asia[c] 1970 | 15287 | 7470 | 100368 | 11599 | 11579 | 3117 (2076) | 14770 | 9217 | 1221 | 3629 |
| Asia[c] 1980 | 21425 | 12080 | 142102 | 16432 | 15632 | 4550 (3288) | 19900 | 13060 | 1680 | 4602 |
| Asia[c] 1990 | 29721 | 18481 | 182418 | 22670 | 21035 | 6595 (5105) | 25427 | 17542 | 2264 | 5574 |
| Asia[c] 2000 | 37074 | 27782 | 219516 | 33498 | 25775 | 9083 (7536) | 29461 | 20758 | 2626 | 5682 |
| Asia[c] 2010 | 43635 | 46368 | 302562 | 52711 | 30621 | 17055 (15213) | 29880 | 21220 | 3233 | 6757 |
| Asia[c] 2011 | 45003 | 48868 | 304900 | 55136 | 30878 | 18047 (16237) | 30540 | 21559 | 3266 | 6652 |
| Asia[c] 2012 | 44227 | 48962 | 304396 | 57285 | 31283 | 18496 (16698) | 30414 | 21526 | 3254 | 6587 |
| Asia[c] 2013 | 42725 | 47561 | 304484 | 58971 | 31559 | 19200 (17427) | 30649 | 21627 | 3227 | 6485 |
| Asia[c] 2014 | 40864 | 46970 | 302718 | 60801 | 31770 | 19447 (17666) | 30469 | 21475 | 3219 | 6478 |
| Asia[c] 2015 | 37876 | 44835 | 296809 | 61627 | 31950 | 19423 (17639) | 29034 | 20644 | 3155 | 6422 |

[a]Gg-NO$_2$ yr$^{-1}$.
[b]Tg yr$^{-1}$. Values in parentheses are CO$_2$ emissions excluding biofuel combustion.
[c]Asia in this table include all target countries and sub-regions in REASv3.

**Table 4.** Uncertainties [%] of emissions in China, India, Japan, SEA, OEA, and OSA in 1955, 1985, and 2015. See Fig. 1 for definitions of SEA OEA, and OSA.

| | $SO_2$ | $NO_x$ | CO | NMVOC | $NH_3$ | $CO_2$ | $PM_{10}$ | $PM_{2.5}$ | BC | OC |
|---|---|---|---|---|---|---|---|---|---|---|
| **1955** | | | | | | | | | | |
| China | ±85 | ±167 | ±291 | ±277 | ±174 | ±133 | ±253 | ±315 | ±334 | ±365 |
| India | ±96 | ±122 | ±265 | ±295 | ±161 | ±116 | ±257 | ±294 | ±277 | ±314 |
| Japan | ±59 | ±62 | ±157 | ±135 | ±141 | ±49 | ±94 | ±117 | ±170 | ±270 |
| SEA | ±134 | ±153 | ±260 | ±272 | ±169 | ±126 | ±291 | ±307 | ±323 | ±317 |
| OEA | ±73 | ±88 | ±146 | ±184 | ±148 | ±59 | ±120 | ±157 | ±157 | ±262 |
| OSA | ±70 | ±112 | ±272 | ±270 | ±168 | ±110 | ±219 | ±281 | ±310 | ±345 |
| **1985** | | | | | | | | | | |
| China | ±36 | ±53 | ±157 | ±150 | ±139 | ±39 | ±101 | ±129 | ±182 | ±250 |
| India | ±40 | ±60 | ±196 | ±212 | ±135 | ±58 | ±160 | ±201 | ±191 | ±259 |
| Japan | ±30 | ±31 | ±44 | ±50 | ±93 | ±14 | ±72 | ±71 | ±53 | ±67 |
| SEA | ±40 | ±56 | ±185 | ±162 | ±141 | ±56 | ±157 | ±191 | ±218 | ±259 |
| OEA | ±48 | ±70 | ±72 | ±78 | ±113 | ±27 | ±80 | ±82 | ±88 | ±102 |
| OSA | ±36 | ±44 | ±144 | ±137 | ±134 | ±33 | ±108 | ±137 | ±176 | ±248 |
| **2015** | | | | | | | | | | |
| China | ±40 | ±35 | ±73 | ±76 | ±82 | ±19 | ±83 | ±94 | ±111 | ±193 |
| India | ±41 | ±35 | ±136 | ±115 | ±111 | ±27 | ±120 | ±151 | ±133 | ±233 |
| Japan | ±34 | ±32 | ±45 | ±63 | ±103 | ±13 | ±68 | ±74 | ±58 | ±100 |
| SEA | ±46 | ±38 | ±124 | ±86 | ±115 | ±25 | ±125 | ±155 | ±161 | ±232 |
| OEA | ±38 | ±60 | ±67 | ±63 | ±94 | ±19 | ±69 | ±85 | ±82 | ±168 |
| OSA | ±40 | ±34 | ±87 | ±73 | ±93 | ±19 | ±96 | ±112 | ±124 | ±211 |