# Peer review of "Long-term historical trends in air pollutant emissions in Asia: Regional Emission inventory in ASia (REAS) version 3"

_Atmospheric Chemistry and Physics, 2019_

## Referee Comment (RC1) · Anonymous Referee #1 · 26 Jan 2020

I've reviewed the paper "Long-term historical trends in air pollutant emissions in Asia: Regional Emission inventory in ASia (REAS) version 3.1" by Kurokawa and Ohara. This is an important contribution to the literature, coverring a region where emissions are changing rapidly, as mentioned by the authors. The authors should also be commended for providing summary data on-line.

As detailed below, however, significantly more detail needs to be provided in terms of definitions and methodology. The present paper gives a general overview of the methodology, but the resulting emissions are still too much of a "black box" otherwise. I would like to re-iterate that this is important and useful work, but I believe additional

documentation is required as discussed below.

Section 2 "Methodology and data" provides a reasonable overview of the methodology, but this needs to be supplemented with much more detail in the supplementary information. There are numerous places throughout this section where a general procedure is described, but then no details are provided. This additional detail is needed is both to satisfy the general scientific publishing principle that work must be, in principle, reproducible, but also so that readers and users of the data can better interpret these results. This information should largely be in the supplementary material.

Some specific points in this regard:

Definitions of the sectors that are included within REAS needs to be provided in much more detail. The EDGAR inventory, for example, provides a fairly comprehensive list of sectors (although even here not all sources are included - REAS includes human waste sources of NH3, for example, while EDGAR does not. This is an example of why it is important to have comprehensive documentation of the methodology and definitions). It is important for users of this data to have this information available.

What is needed is a detailed table with sub-sector definitions. For example Table S3 in Janssens-Maenhout et al 2019 and Tables A1 and A2 in Hoesly et al. 2018.

It is not clear the extent to which non-combustion emissions are included in REAS (I presume they are), and which sectors are included and how they are estimated. For example see the list of non-combustion sectors from EDGAR noted above. Are all of these included for all species in REAS, or just some? (Industrial process emission sources are mentioned in the text, but that is not very specific. Some of the specific non-combustion sources that should be discussed (whose level of inclusion varies between inventories) include: emissions from coke production, non-ferrous metal smelters, refineries, agricultural NOx, open residential waste burning (and waste burning in dumps), and so on.

For these sectors, clarify if just combustion-related emissions are considered, or also if process (non-combustion) emissions. This is particularly important for sectors that have both combustion-related and non-combustion emissions such as refining, coke production, etc.

Similar to sectors, the fuel categories used in the calculation need to be discussed as well. While the table "3.2 Fuel types" in the document "Brief description about table data v3.1.pdf" gives an overview, this is not a complete definition of each fuel type. To give just one example, it is not clear what category LPG is in. Since the authors use the IEA energy statistics, what would be most useful is a correspondence between the fuels detailed in IEA and the aggregate fuels reported by the authors.

Given that multiple data sources are used for fuel consumption, comparability of definitions and data time series between multiple sources used for some countries should be addressed.

The authors mention that fuel use by sector is extrapolated back into time before the point where detailed data is used by constant sector shares. This is an important assumption, so the year at which this assumption comes into force should be listed in the supplement. (For example, given the greater data availability for Japan, perhaps fuel use by sector is available over all time periods.)

For some countries and time periods fuel consumption in non power plant transformation sectors, particularly coal coke production, is large. How this and other non-combustion or feedstock use of fuels is dealt with over time should be discussed.

Overall, the data sources for assumptions for both activity data (including associated information such as fuel use by vehicle type) and emission factors are unclear. What would be helpful is a comprehensive table (or, likely set of tables), that list this information by sector and country (and species, where relevant). It appears that there are specific data sources for some countries, while for others more generic assumptions are used (in which cases, those countries could be grouped together).

[Figure]

For road vehicles, the relationship between the "tentative emission factors" and final emission factors in section 2.3.3 Emission factors is unclear.

One of the signifiant contributions of this work is the consideration of emission controls. The data sources and assumptions for these calculations by country should be described in greater detail (again, largely in the supplement). This will provide the readers with important context in terms of how reliable this information might be (e.g., presumably more reliable for Japan than for some other countries!)

At minimum, figures for countries other than Japan and China (e.g. Figure 3, and 5) should be provided in the supplement (at least for those countries with non-zero levels of control). Ideally numerical values would be provided as well.

The development of emission control assumptions for industry is not described in the text. This is an important sector for air pollutions, so further information should be provided. For example, it is not clear to me what BC or SO2 controls in China represent going back as far as 1990. (Very recently SO2 controls were mandated in industrial boilers, but my understanding is that these controls were only for very recent years.)

Also, are emission controls considered for only industrial combustion emissions? Or also for industrial process emissions. Note that technology changes can have a very significant impact on emissions from industrial process technologies (e.g., metal smelters, coke production plants, refineries, etc.). The assumptions here should be discussed. (As noted above, it is not clear the extent to which emissions from these sources/sectors, other than combustion emissions, are included.)

It is unclear what is meant by "In REASv3.1, aviation and ship emissions are not included", except then it is said "emissions from domestic shipping including fishing ships were roughly calculated for comparison with other inventories (see Section 3.3)." Clarify. I assume this means that international shipping emissions are not included at all. Does this mean that domestic shipping and fishing emissions are included? What are the data sources for these? (Presumably largely IEA?) So are these emissions

included in the REAS3.1 totals? Or are they just included for comparison with other inventories? P

Note the substantial literature on shipping emissions, which notes that shipping fuel use, particularly into the past, is inconsistently and incompletely reported in general.

I note that with these additional details in the supplement, it will likely be possible to streamline the main text somewhat to make the paper more readable.

Section "3.1 Trends of Asian and national emissions" is a bit tedious with the reporting of numerical results. Given that the numerical results are available on-line, the detailed recitation of numerical values in the text is not necessary and detracts from reading the manuscript. A more general overview of the trends and drivers with fewer numerical values would be more useful.

Line 595 "As described in Section 2.5". I believe this is Section 2.6?

In "3.3 Comparison with other inventories" it should be mentioned if the sectoral coverage of the inventories compared are the same? Different sectoral coverage can lead to artificial differences in such comparisons.

Note that for many emissions and countries in Asia the CEDS data was calibrated to REAS2.1 - hence the similarity in results.

This section is long and difficult to read. The extremely long paragraphs should be broken up and, where possible, streamlined.

The uncertainty calculation is a valuable and important portion of the paper. However, the specific uncertainty assumptions used need to be provided (e.g. in supplemental information), It would be useful to provide some equations here.

How do these assumptions vary by sector and fuel? Are these assumptions constant across countries? And across time? How is uncertainty in emission control percentages handled? One important assumption is how uncertainty was combined across

fuels and sectors is a critical part of the methodology and needs to be described (e.g. are independent uncertainties assumed, or is some correlation assumed?).

Janssens-Maenhout, G., Crippa, M., Guizzardi, D., Muntean, M., Schaaf, E., Dentener, F., Bergamaschi, P., Pagliari, V., Olivier, J. G. J., Peters, J. A. H. W., van Aardenne, J. A., Monni, S., Doering, U., Petrescu, A. M. R., Solazzo, E., and Oreggioni, G. D.: EDGAR v4.3.2 Global Atlas of the three major greenhouse gas emissions for the period 1970–2012, Earth Syst. Sci. Data, 11, 959–1002, https://doi.org/10.5194/essd-11-959-2019, 2019.

Hoesly, R. M., Smith, S. J., Feng, L., Klimont, Z., Janssens-Maenhout, G., Pitkanen, T., Seibert, J. J., Vu, L., Andres, R. J., Bolt, R. M., Bond, T. C., Dawidowski, L., Kholod, N., Kurokawa, J.-I., Li, M., Liu, L., Lu, Z., Moura, M. C. P., O'Rourke, P. R., and Zhang, Q.: Historical (1750–2014) anthropogenic emissions of reactive gases and aerosols from the Community Emissions Data System (CEDS), Geosci. Model Dev., 11, 369–408, https://doi.org/10.5194/gmd-11-369-2018, 2018.

---

## Referee Comment (RC2) · Anonymous Referee #2 · 16 Feb 2020

The authors developed a new version of the REAS emission inventory, presented the trends in Asian air pollutant emissions, and analyzed the regional and sectoral drivers. The comparison with the up-to-date regional inventories presents broadly consistent emission trends, and the uncertainties in the REAS v3.1 estimates are quantified according to the errors in each parameter. This is quite important work, because the REAS inventory has been widely used in the modeling of climate and of air quality. This new version extended the emission time series to 1950-2015 and made necessary updates in both the methods and the data input. My major concern is that the method part is not well structured and is very difficult to follow, and the comparison with the previous emissions data lacks the top-down inversion estimates, which should

be included.

Major comments:

1) Method. The method section should put more focus on the new features of the new REAS version 3.1 compared to the last version 2.1. Please summarize the new data development process and give a detailed table to show the new methods developed and the new data sources used in the REAS v3.1. Part of the REAS v2.1 emissions data are directly adopted by the REAS v3.1, such as the agricultural sources in Japan, which should be described clearly in this table. The REAS inventory relies on plenty of other emission inventories to provide the emissions data or the spatial proxies used in the emission distribution. The data dependencies across different inventories would better be clarified specifically in a new table, which would benefit the users of different inventories.

2) Data sources. The manuscript briefly describes the sources of the input data, but the values of parameters are not given. I understand that it is difficult to present all the detailed input data of a large-scale emission inventory. However, knowing the exact values of some key parameters can help the audience understand the drivers of emissions changes. I suggest the authors present some key parameters that determine the curve of emission changes, show their values, and discuss why such values are adopted (e.g., due to more stringent emission legislations). I noticed that the authors used many proxy data to calculate the "trend factors" when the activity data of the past years are not available. This method needs to be justified. Please show the relationship between the proxy data and the associated activity data using the historical values when they are both available.

3) Results. The results section mainly focuses on the emissions of $SO_2$, $NO_x$, and BC. Please add $CO_2$ in each plot of the results to reflect the energy consumption trends. It is difficult to understand the drivers of emission changes from the text now. Please quantitatively estimate the contributions of the energy consumption growth and of the

air pollution control progresses on the emission changes over each region discussed in Sect. 3. For the comparison with other inventories, the authors only compared their emission results with other bottom-up emission inventories, while did not consider top-down emissions data constrained by satellite observations that have developed very fast in recent years. In my opinion, different bottom-up emission inventories commonly share the same sources of input data, which are not completely independent of each other. It would be better to evaluate the long-term emission trends with top-down information from previous literature. For the uncertainty assessment, I cannot understand why the uncertainties of CO2 emissions are so large, particularly ±28% for China and ±23% for Japan, which are much higher than the typical uncertainty range (±10%) of country CO2 emissions.

Minor comments:

1) Line 47 on Page 2. The GAINS model not the GANS model.

2) Line 318 on Page 10. For spatial distribution not the special distribution.

3) Lines 354 and 355 on Page 12. Please clarify how the information of large plants is used for developing allocation factors for corresponding emission source sectors.

4) The caption of Figure 3. During 1990-2015 not 1950-2015.

5) Figures 10 and 11. The colors of some curves are close to each other and are difficult to distinguish. And please also add the uncertainty range of REAS v3.1 in the plots.

---

## Author Response (AR1)

Dear Editor,

The authors would like to appreciate Editor for taking your precious time to handle our manuscript. First, I'd like to explain about revisions of datasets in the manuscript:

1. Both Referees #1 and #2 pointed out necessity of providing details of the methodology. Authors agreed the comment and created a new supplement of the manuscript entitled "Supplementary information and data related to methodology of REASv3" (hereafter "the Supplement").

2. For development of the Supplement, we thoroughly checked the data and system of REASv3.1 (a version of ACPD paper) and found several points needed to be revised including trivial errors. Base on the results of the checks, revisions of the data and system were conducted. All values, tables, and figures in the manuscript including supplementary tables and figures were revised using the updated data.

3. Although discussions and conclusions of the manuscript were generally not influenced by the revisions, there were discrepancies in some species, and countries and regions between REASv3.1 and the revised data (hereafter tentatively named as REASv3.2). Therefore, we prepared another new supplemental document showing the differences between REASv3.2 and REASv3.1.

4. For distribution of the revised data, considering the possibility of additional modification during the revision processes, we would like to take the following processes:

   ➢ We did not use the detailed version number (REASv3.1), but used REAS version 3 (REASv3) in the revised main manuscript including the title. The detailed version number were described only in the "Data availability" section.

   ➢ The tentative data during the revision processes will not be opened in the download site of REAS.

   ➢ When the revision process has been completed, the final version will be opened at the REAS download site as REASv3.2.

Above points were also described in the author comments for Referees #1 and #2.

The structure of this document is as follows:

(1) Comments, author's response, and author's changes in manuscript related to Referee #1
(2) Comments, author's response, and author's changes in manuscript related to Referee #2
(3) The revised main manuscript where changed parts were yellow highlighted
(4) The revised main manuscript with track changes
(5) Revision of the supplementary materials

Sincerely Yours,

Jun-ichi Kurokawa

Asia Center for Air Pollution Research

kurokawa@acap.asia

TEL: +81-25-263-9558

FAX +81-25-263-0567

(1) Comments, author's response, and author's changes in manuscript related to Referee #1

**No. 1**

*Referee comments*

*I've reviewed the paper "Long-term historical trends in air pollutant emissions in Asia: Regional Emission inventory in ASia (REAS) version 3.1" by Kurokawa and Ohara. This is an important contribution to the literature, coverring a region where emissions are changing rapidly, as mentioned by the authors. The authors should also be commended for providing summary data on-line. As detailed below, however, significantly more detail needs to be provided in terms of definitions and methodology. The present paper gives a general overview of the methodology, but the resulting emissions are still too much of a "black box" otherwise. I would like to re-iterate that this is important and useful work, but I believe additional documentation is required as discussed below.*

Author's response to the Referee comments

One major point which was pointed out by both Referees #1 and #2 are necessity of providing details of the methodology in the manuscript. We totally agree the indications and created a new supplement of the manuscript entitled "Supplementary information and data related to methodology of REASv3" which provides detailed descriptions for the framework, activity data, emission factors, emission controls and other settings adopted in REASv3 including definition of sectors, data sources, treatment of the data, related assumptions, etc. (Hereafter, referred as "the Supplement")

For development of the Supplement, we thoroughly checked the data and system of REASv3.1 (a version of the ACPD paper) and found several points which should be revised including trivial errors in the data and system. Based on the results of the checks, revisions of the data and system were conducted including correction of the errors. In general, discussion and conclusions of the manuscript were not influenced by the revision. However, for some species, countries and regions, there were discrepancies between REASv3.1 and the revised one which is tentatively named as REASv3.2. Therefore, we prepared another supplemental document showing the differences between REASv3.2 and REASv3.1 and causes of the discrepancies entitled "Differences between REASv3.2 and REASv3.1". For distribution of the revised data, considering the possibility of additional modification during the revision processes, we would like to take the following processes:

- We did not use the detailed version number (REASv3.1), but used REAS version 3 (REASv3) in the revised main manuscript including the title. The detailed version number were described only in the "Data availability" section.
- The tentative data during the revision processes will not be opened in the download site of REAS.
- When the revision process has been completed, the final version will be opened at the REAS

download site as REASv3.2.

Author's changes in manuscript
- As described in the author's response, following two new supplemental documents were created
  - Supplementary information and data related to methodology of REASv3
  - Differences between REASv3.2 and REASv3.1
- All figures and tables in the main manuscript and supplement were recreated by the updated datasets which were also explained in the author's response.
- "version 3" and "REASv3" were used instead of "version 3.1" and "REASv3.1", respectively. The differences of versions were described in "Data availability" and the new supplement was introduced there.
- Numbers such as emission amounts and growth rates in abstract, Sects 3.1, and 4 were revised as seen in yellow highlighted parts in abstract, Sects 3.1, and 4 in (3) and corresponding track changes in (4). Line numbers (Page numbers) including corresponding revisions in the revised main manuscript are as follows: L11-14 (P1); L437-439 (P14); L460, L463-465, L471-472 (P15); L499, L511-512 (P16); L535-536 (P17); L549-551, L554-555, L565, L569 (P18); L588 (P19); and L927-930 (P29).

**No. 2**

*Referee comments*

*Section 2 "Methodology and data" provides a reasonable overview of the methodology, but this needs to be supplemented with much more detail in the supplementary information. There are numerous places throughout this section where a general procedure is described, but then no details are provided. This additional detail is needed is both to satisfy the general scientific publishing principle that work must be, in principle, reproducible, but also so that readers and users of the data can better interpret these results. This information should largely be in the supplementary material.*

Author's response to the Referee comments

As describe above, we developed the Supplement providing detailed information and explanations related to Sect. 2 "Methodology and data". In order to avoid making the Sect. 2 long, detailed descriptions were not added to the main manuscript, but appropriate parts in the Supplement were indicated in Sect. 2. Following revisions were conducted for the main manuscript related to the Supplement:

- Sect. 2.1 (General description) was fully revised also referring comments from Referee #2, including addition of a new table (Table 2 entitled "Emission inventories from other research

works and officially opened data utilized in REASv3."). In Sect. 2.1 of the revised main manuscript, the Supplement was introduced.

➢ In Sect. 2.2.1, Sects. S2.4.1 and S2.4.2 of the Supplement were cited for descriptions for combustion and non-combustion sources.

➢ In Sect. 2.2.2, Sects. S3.1.1-6, and S4.1 of the Supplement were cited for definition of fuel types and details of activity data for stationary sources, including fuel consumption, industrial production, and other transformation.

➢ In Sect. 2.2.3, Sects. S3.2, S4.2, S5.1.5, S5.2.5, and S8.3 of the Supplement were cited for emission factors and emission controls for stationary combustion, industrial production, other transformation sector.

➢ In Sect. 2.3.1, Sects. S6.2.1, S6.2.3, and S6.3 of the Supplement were cited for additional information about methodology of road transport sector.

➢ In Sect. 2.3.2, Sect. S6.1.1 of the Supplement was cited for number of vehicles and annual vehicles kilometer traveled. In addition, wrong citations of references in the previous main manuscript were corrected as follows:

✧ L246 of the previous manuscript: Road Transport Yearbook (Morth, 2003-2017) was changed to TERI Energy & Environment Data Diary and Yearbook (TERI, 2013, 2018).

✧ L249 of the previous manuscript: Pandey and Venkataraman (2014) was deleted.

✧ L252-253 of the previous manuscript: "In this study, settings of REASv2.1 were used as default and were updated if new information was available, such as Pandey and Venkataraman (2014), Sahu et al. (2014) and Mishra and Goyal (2014)." was revised as "In this study, settings of Streets et al. (2003a) and REASv2.1 were used as default and were updated if national information was available, such as He et al. (2005), Yan and Crookes (2009), Sahu et al. (2014), and Malla (2014).".

➢ Sect. 2.3.3 for emission factors of road transport was fully revised and Sect. S6.2 of the Supplement was cited.

➢ In Sect. 2.4.1, Sect. S8.1 of the Supplement was cited for methodologies and data sources for manure management sector for $NH_3$.

➢ In Sect. 2.4.2, Sect. S8.2 of the Supplement was cited for methodologies and data sources for fertilizer application sector for $NH_3$.

➢ In Sect. 2.5, Sects. S5, S7, S8.4, and S8.5 of the Supplement were cited for activity data and emission factors for non-combustion sources of NMVOC, $NH_3$, and other transport sector.

➢ In Sect. 2.6, Sects. S9.1 and S9.2 of the Supplement were cited for methodologies and data sources for grid allocation and monthly variation factors.

➢ In Sect. 3.4, Sect. S10 of the Supplement was cited for methodologies and settings of uncertainties of each component.

Author's changes in manuscript

- Changes of Sect 2.1 are described in (2).
- Sect. 2.3.3 was fully revised.
- In Sects. 2.2-2.5, descriptions related to citing the Supplement were added as seen in yellow highlighted parts in (3). Line numbers (Page numbers) including corresponding parts in the revised main manuscript are as follows: L149-150, L157-L158 (P5); L167-171, L171-172, L174-175 (P6); L192-193, L202-203, L214-215 (P7); L247-L250 (P8); L261, L266, L268-269, L276-277 (P9); L303 (P10); L353-354, L369, L380-381 (P12); L386-387, L388-391, L410 (P13); and L422 (P14); L872-873 (P27); and L924-926 (P29).
- In Sects. 2.2-2.5, incorrect citations including by typos were corrected as seen in yellow highlighted parts in (3) and corresponding track changes in (4). Line numbers (Page numbers) including corresponding revisions in the revised main manuscript are as follows: L176, L186 (P6); L231, L242, L245 (P8); L283-284, L286 (P9); L289-290 (P10); L346, L347 (P11); L377 (P12); L385, L396, L417 (P13); L420 (P14).

**No. 3**

*Referee comments*

*Some specific points in this regard:*

*Definitions of the sectors that are included within REAS needs to be provided in much more detail. The EDGAR inventory, for example, provides a fairly comprehensive list of sectors (although even here not all sources are included - REAS includes human waste sources of NH3, for example, while EDGAR does not. This is an example of why it is important to have comprehensive documentation of the methodology and definitions). It is important for users of this data to have this information available.*

*What is needed is a detailed table with sub-sector definitions. For example Table S3 in*
*Janssens-Maenhout et al 2019 and Tables A1 and A2 in Hoesly et al. 2018.*

*It is not clear the extent to which non-combustion emissions are included in REAS (I presume they are), and which sectors are included and how they are estimated. For example see the list of non-combustion sectors from EDGAR noted above. Are all of these included for all species in REAS, or just some? (Industrial process emission sources are mentioned in the text, but that is not very specific. Some of the specific non-combustion sources that should be discussed (whose level of inclusion varies between inventories) include: emissions from coke production, non-ferrous metal smelters, refineries, agricultural NOx, open residential waste burning (and waste burning in dumps), and so on.*

*For these sectors, clarify if just combustion-related emissions are considered, or also if process*

*(non-combustion) emissions. This is particularly important for sectors that have both combustion-related and non-combustion emissions such as refining, coke production, etc.*

Author's response to the Referee comments

Tables of sub-sectors included in REASv3 are provided in Sect. S2 of the Supplement. For combustion sources, sub-sector categories are compared with IEA code. For some sectors such as iron, steel, and coke production, as suggested, relationships between combustion and non-combustion emissions are also complicated in REASv3. The details are described in Sects. S3 and S4 of the Supplement. In addition, for non-combustion sources of NMVOC and $NH_3$, details are described in Sects. S5 and S8 as well as Sect. S2 in the Supplement. Related descriptions were added to Sect. 2.1 of the revised main manuscript as described above.

Author's changes in manuscript

- The framework of REASv3 including tables of sub-sectors were provided in Sect. S2 of the Supplement and related description was added to Sect. 2.1 of the revised main manuscript (L125-136 (P4)).

- For combustion and non-combustion sectors, general descriptions were written in Sect. S2 and details were described in Sects. S3 (Combustion sources) and S4 (Industrial process and other transformation) of the Supplement. Related description was added to Sect. 2.2.1 of the revised main manuscript (L149-150 (P5)).

- Non-combustion sources of NMVOC and $NH_3$ were also described in Sects. S5 and S8 of the Supplement which were cited in Sects. 2.2.3, 2.4.1, and 2.5 of the revised main manuscript (L249-250 (P8); L353-354, L369, L380-381 (P12)).

**No. 4**

*Referee comments*

*Similar to sectors, the fuel categories used in the calculation need to be discussed as well. While the table "3.2 Fuel types" in the document "Brief description about table data v3.1.pdf" gives an overview, this is not a complete definition of each fuel type. To give just one example, it is not clear what category LPG is in. Since the authors use the IEA energy statistics, what would be most useful is a correspondence between the fuels detailed in IEA and the aggregate fuels reported by the authors.*

*Given that multiple data sources are used for fuel consumption, comparability of definitions and data time series between multiple sources used for some countries should be addressed.*

*The authors mention that fuel use by sector is extrapolated back into time before the point where detailed data is used by constant sector shares. This is an important assumption, so the year at*

*which this assumption comes into force should be listed in the supplement. (For example, given the greater data availability for Japan, perhaps fuel use by sector is available over all time periods.)*

Author's response to the Referee comments

List of detailed fuel types and definition of aggregated categories used in the main manuscript and supplement are provided in Sect. S3.1.1 and Table 3.1 in the Supplement. Data sources of fuel consumption and assumptions to estimate missing historical data are also provided for each country in Sect. S3.1.2 and Table 3.2 in the Supplement document. Related descriptions were added to Sect. 2.1 of the main manuscript as described above.

Author's changes in manuscript
- Definition of fuel type was provided in Table 3.1 in Sect. S3.1.1 of the Supplement and related description was added to Sect. S2.2.1 of the revised main manuscript (L149-150 (P5)).
- Details of data sources, treatments, and related assumptions for development of historical data were also provided in Sect. S3.1.2 and Table 3.2. Related description was added to Sect. 2.2.2 of the revised main manuscript (L171-172 (P6)).

**No. 5**

*Referee comments*

*For some countries and time periods fuel consumption in non-power plant transformation sectors, particularly coal coke production, is large. How this and other non-combustion or feedstock use of fuels is dealt with over time should be discussed.*

Author's response to the Referee comments

For fuel consumption including input amounts of coal for coke ovens and crude oil for oil refinery, data sources and assumptions for estimating missing historical data were described in Sect. S3 of the Supplement document. The historical data and assumptions for coke production amounts were described in Sect. S4 of the Supplement.

Author's changes in manuscript
- Details of data sources and treatments of fuel consumption were provided in Sect. 3 of the Supplement. Fuel data for coke ovens and oil refinery were described in Table 3.3 and Sect. 4.1.8 of the Supplement.

**No. 6**

*Referee comments*

*Overall, the data sources for assumptions for both activity data (including associated information such as fuel use by vehicle type) and emission factors are unclear. What would be helpful is a comprehensive table (or, likely set of tables), that list this information by sector and country (and species, where relevant). It appears that there are specific data sources for some countries, while for others more generic assumptions are used (in which cases, those countries could be grouped together).*

Author's response to the Referee comments

As described above, details of activity data, emission factors, and emission controls adopted in REASv3 were described in the Supplement document for all sources and species. When country-specific settings were adopted, related data and information were also described in the Supplement.

Author's changes in manuscript

- The situations of available data and information were different among countries and regions. Therefore, considering comments from both Referee #1 and #2, we created the Supplement which provides details of activity data, emission factors, settings of emission controls, etc.

**No. 7**

*Referee comments*

*For road vehicles, the relationship between the "tentative emission factors" and final emission factors in section 2.3.3 Emission factors is unclear.*

Author's response to the Referee comments

Thank you for pointing out the problem. We agree that explanations and expressions in Sect. 2.3.3 were unclear and inappropriate. In addition, we also found some wrong citations. In Sect. S6 of the Supplement, detailed data and information for road transport sector were provided. We revised the Sect. 2.3.3 and cited Sect. S6 of the Supplement, as described above.

Author's changes in manuscript

- Sect. 2.3.3 was fully revised.

**No. 8**

*Referee comments*

*One of the signifiant contributions of this work is the consideration of emission controls. The data sources and assumptions for these calculations by country should be described in greater detail*

*(again, largely in the supplement). This will provide the readers with important context in terms of how reliable this information might be (e.g., presumably more reliable for Japan than for some other countries!)*

*At minimum, figures for countries other than Japan and China (e.g. Figure 3, and 5) should be provided in the supplement (at least for those countries with non-zero levels of control). Ideally numerical values would be provided as well.*

*The development of emission control assumptions for industry is not described in the text. This is an important sector for air pollutions, so further information should be provided. For example, it is not clear to me what BC or SO2 controls in China represent going back as far as 1990. (Very recently SO2 controls were mandated in industrial boilers, but my understanding is that these controls were only for very recent years.)*

*Also, are emission controls considered for only industrial combustion emissions? Or also for industrial process emissions. Note that technology changes can have a very significant impact on emissions from industrial process technologies (e.g., metal smelters, coke production plants, refineries, etc.). The assumptions here should be discussed. (As noted above, it is not clear the extent to which emissions from these sources/sectors, other than combustion emissions, are included.)*

Author's response to the Referee comments

We appreciate the important comments. In Sects. S3 and S4 of the Supplement document, details of settings and assumptions for emission controls both for power plants and industry sectors adopted in REASv3 were described for all countries and regions. However, except for China and Japan, available data and information were limited. For emission controls in industrial processes, the same settings for combustion emissions were adopted except for China where information on some sectors was available from studies for emission inventories of China. Considering the above status, in this manuscript, detailed discussions on effects of emission controls using the figures (like Figs. 3 and 5) were conducted only for China and Japan. Further surveys of local information of emission controls and related abatement technologies are necessary especially for countries and regions other than China and Japan and detailed discussions are important tasks in future studies. These points were emphasized in Sect. 4 of the revised main manuscript.

Author's changes in manuscript

● Details for settings of emission controls were described in Sects. S3 and S4 of the Supplement. and related description was added to Sect. 2.2.1 of the revised main manuscript (L247-249 (P8)).

● Due to reasons described in the author's response, detailed discussion on effects of emission controls using the figures were focused to China and Japan in this study and those for other

countries and regions are important issues in future studies. Related descriptions were added to Sects. 1 and 4 (L78-79 (P3); L949-951 (P30)).

**No. 9**

*Referee comments*

*It is unclear what is meant by "In REASv3.1, aviation and ship emissions are not included", except then it is said "emissions from domestic shipping including fishing ships were roughly calculated for comparison with other inventories (see Section 3.3)." Clarify. I assume this means that international shipping emissions are not included at all. Does this mean that domestic shipping and fishing emissions are included? What are the data sources for these? (Presumably largely IEA?) So are these emissions C4 included in the REAS3.1 totals? Or are they just included for comparison with other inventories?*

*Note the substantial literature on shipping emissions, which notes that shipping fuel use, particularly into the past, is inconsistently and incompletely reported in general.*

Author's response to the Referee comments

In the first manuscript, we included roughly estimated emissions of domestic and fishing ships just for comparisons with other inventories in Sect. 3.3. However, we reconsidered that including roughly estimated ship emissions for the comparison with other inventories was not appropriate. In the revised main manuscript, we did not add any shipping emissions to REASv3 for the comparison with other studies. This means that in REASv3, emissions from both international and domestic aviation and navigation including fishing ships are totally out of scope. This was also clarified in the revised main manuscript.

Author's changes in manuscript

- In REASv3, international and domestic aviation and navigation including fishing ships are out of scope. This was clarified in Sects. 2.1 (L108-109 (P4)) and 3.3 (L703-704 (P22)) in the revised main manuscript.

**No. 10**

*Referee comments*

*I note that with these additional details in the supplement, it will likely be possible to streamline the main text somewhat to make the paper more readable.*

Author's response to the Referee comments

We agree the suggestion. The corresponding sections in the Supplement were indicated in the

main manuscript to provide details for data sources, their treatment, settings and related assumptions as described above.

Author's changes in manuscript

- This is a general response to the general comment. Corresponding revisions were described above.

**No. 11**

*Referee comments*

*Section "3.1 Trends of Asian and national emissions" is a bit tedious with the reporting of numerical results. Given that the numerical results are available on-line, the detailed recitation of numerical values in the text is not necessary and detracts from reading the manuscript. A more general overview of the trends and drivers with fewer numerical values would be more useful.*

Author's response to the Referee comments

We appreciate the comments. For the indicated numerical results, we left data of total Asia in the abstract, the first paragraph of Sects. 3.1.1. and 4 to provide general status in Asia. For China and India, results of growth rates in these 60 years were left because these were key features in Asia. For other countries and regions, the indicated numerical values for all species were deleted. In Sects. 3.1.2-3.1.5, more descriptions for features of trends and their drivers were added. On the other hand, Referee #2 gives a following comment: "Please quantitatively estimate the contributions of the energy consumption growth and of the air pollution control progresses on the emission changes over each region discussed in Sect. 3.". Considering the comment, some quantitative discussions were added for major points of trends and their drivers. For effects of emission controls, as explained above, detailed discussions were conducted only for China and Japan.

Author's changes in manuscript

- For the numerical results (reports of emissions of each air pollutant), different revisions were done for different countries and regions as follows:
  - ➤ Asia: The numerical results in abstract (L11-14 (P1)), Sects. 3.1.1 (L437-439 (P14)), and 4 (L927-930 (P29)) were not deleted to provide general status in Asia. But values were revised as mentioned first.
  - ➤ China and India: The results of growth rates in these 60 years were left (China: L470-472 (P15), India: L534-536 (P17)) considering that they were key features in Asia. Values were also revised.
  - ➤ Other countries and regions: Corresponding results were deleted in the revised main

manuscript. (In the ACPD paper: L502-505 (P16) for Japan, L546-553 (P16) for Southeast Asia and South Asia other than India, L571-575 (P16) for East Asia other than China and Japan).

- New discussions on features of trends and their drivers including quantitative evaluations were added to descriptions in Sects. 3.1.2-3.1.5. Line numbers (Page numbers) including corresponding parts in the revised main manuscript are as follows:
  - ➢ Sect. 3.1.2: L481, L486-491, L495-499, L499-502, L504-507, L508 (P16); L519-520, L529-532 (P17).
  - ➢ Sect. 3.1.3: L542-544, L546-L548 (P17); L552-553, L558-561, L563-564, L566, L568, L571-573 (P18).
  - ➢ Sect. 3.1.4: L577-578 (P18); L582-585, L590-596, L602-606, L610, L614-617 (P19).
  - ➢ Sect. 3.1.5: L630, L638-640, L642-646 (P20); L648-651, L660, L663-665, L667, L668-670 (P21).

**No. 12**

*Referee comments*

*Line 595 "As described in Section 2.5". I believe this is Section 2.6?*

Author's response to the Referee comments

Thank you for pointing out the typo. It was corrected.

Author's changes in manuscript

- The pointed out part (P22L684 in the revised main manuscript) was corrected.

**No. 13**

*Referee comments*

*In "3.3 Comparison with other inventories" it should be mentioned if the sectoral coverage of the inventories compared are the same? Different sectoral coverage can lead to artificial differences in such comparisons.*

*Note that for many emissions and countries in Asia the CEDS data was calibrated to*

*REAS2.1 - hence the similarity in results.*

*This section is long and difficult to read. The extremely long paragraphs should be*

*broken up and, where possible, streamlined.*

Author's response to the Referee comments

First, we agree the comment that the Sect. 3.3 is too long and should be streamlined. In the

revised main manuscript, we divided the contents to 4 sub-sections: China, India, Other regions, and Relative ratios of emissions from each country and region in Asia and descriptions in these sections were revised. In addition, comparison of emissions in total Asia among REASv3, CEDS, and EDGARv4.3.2 were added to Figs. 12 and S20. For the relationship between CEDS and REASv2.1, the information was added appropriate parts of the main manuscript. For sector categories, as described above, emissions from aviation and navigation were not included in REASv3. Therefore, if it was possible, corresponding emissions were subtracted from total emissions of other inventories. But, unfortunately, there were many inventories where no independent emission data of domestic navigation were available. In this study, such inventories were also included in the comparisons with the notices in the figure captions. These procedures were also mentioned in the revised main manuscript. In addition, based on a comment from Referee #2, we included following top-down emission data.

> Ding, J., Miyazaki, K., van der A, R. J., Mijling, B., Kurokawa, J.-I., Cho, S., Janssens-Maenhout, G., Zhang, Q., Liu, F., and Levelt, P. F.: Intercomparison of $NO_x$ emission inventories over East Asia, Atmos. Chem. Phys., 17, 10125–10141, https://doi.org/10.5194/acp-17-10125-2017, 2017.

> Itahashi, S., Yumimoto, K., Kurokawa, J., Morino, Y., Nagashima, T., Miyazaki, K., Maki, T., and Ohara, T.: Inverse estimation of $NO_x$ emissions over China and India2005–2016: contrasting recent trends and future perspectives, Environ. Res. Lett., 14, 124020, https://doi.org/10.1088/1748-9326/ab4d7f, 2019.

> Jiang, Z., Worden, J. R., Worden, H., Deeter, M., Jones, D. B. A., Arellano, A. F., and Henze, D. K.: A 15-year record of CO emissions constrained by MOPITT CO observations, Atmos. Chem. Phys., 17, 4565–4583, https://doi.org/10.5194/acp-17-4565-2017, 2017.

> Miyazaki, K., Bowman, K., Sekiya, T., Eskes, H., Boersma, F., Worden, H., Livesey, N., Payne, V. H., Sudo, K., Kanaya, Y., Takigawa, M., and Ogochi, K.: An updated tropospheric chemistry reanalysis and emission estimates, TCR-2, for 2005–2018, Earth Syst. Sci. Data Discuss., https://doi.org/10.5194/essd-2020-30, in review, 2020.

> Qu, Z., Henze, D. K., Li, C., Theys, N., Wang, Y., Wang, J., Wang, W., Han, J., Shim, C., Dickerson, R. R., and Ren, X.: $SO_2$ emission estimates using OMI $SO_2$ retrievals for 2005–2017, J. Geophys. Res. Atmos., 124, 8336-8359, https://doi.org/10.1029/2019JD030243, 2019.

> Stavrakou, T., Muller, J. F., Bauwens, M., De Smedt, I.: Sources and long-term trends of ozone precursors to Asian Pollution, Air Pollution in Eastern Asia: an integrated perspective, eds. Bouarar, I., Wang, X., Brasseur, G., Springer international Publishing, 167–189, https://doi.org/10.1007/978-3-319-59489-7-8, 2017.

> Zheng, B., Chevallier, F., Yin, Y., Ciais, P., Fortems-Cheiney, A., Deeter, M. N., Parker, R. J., Wang, Y., Worden, H. M., and Zhao, Y.: Global atmospheric carbon monoxide budget 2000–

2017 inferred from multi-species atmospheric inversions, Earth Syst. Sci. Data, 11, 1411–1436, https://doi.org/10.5194/essd-11-1411-2019, 2019.

Furthermore, we added following two bottom-up historical emission inventories of China:

➢ Sun, W., Shao, M., Granier, C., Liu, Y., Ye, C. S., and Zheng, J. Y.: Long-term trends of anthropogenic $SO_2$, $NO_x$, CO, and NMVOCs emissions in China, Earth's Future, 6, 1112-1133, https://doi.org/10.1029/2018EF000822, 2018.

➢ Wang, R., Tao, S., Wang, W., Liu, J., Shen, H., Shen, G., Wang, B., Liu, X., Li, W., Huang, Y., Zhang, Y., Lu, Y., Chen, H., Chen, Y., Wang, C., Zhu, D., Wang, X., Li, B., Liu, X., and Ma, J.: Black Carbon Emissions in China from 1949 to 2050, Environ. Sci. Technol., 46, 7595-7603, https://doi.org/10.1021/es3003684, 2012.

Author's changes in manuscript

● Sect. 3.3 were fully revised. The section was divided into 4 independent sub-sections: 3.3.1 China, 3.3.2 India, 3.3.3 Other regions, and 3.3.4 Relative rations of emissions from each country and region in Asia. Furthermore, comparison of total emissions in Asia among REASv3, CEDS, and EDGARv4.3.2 were added to Figs. 12 and S20.

● Relationships between CEDS and REASv2.1 were pointed out in the revised main manuscript (L754 (P24); L786, L800 (P25); L812 (P26))

● Descriptions for problems of sector coverage especially about domestic navigation were added to Sect. 3.3 of the revised main manuscript (L703-708 (P22)) and notes were added to captions of Figs. 10 and 11 of the revised main manuscript and Figs. S14-S19 in the supplementary figures.

**No. 14**

*Referee comments*

*The uncertainty calculation is a valuable and important portion of the paper. However, the specific uncertainty assumptions used need to be provided (e.g. in supplemental information), It would be useful to provide some equations here.*

*How do these assumptions vary by sector and fuel? Are these assumptions constant across countries? And across time? How is uncertainty in emission control percentages handled? One important assumption is how uncertainty was combined across fuels and sectors is a critical part of the methodology and needs to be described (e.g. are independent uncertainties assumed, or is some correlation assumed?).*

Author's response to the Referee comments

We appreciate valuable comments for uncertainties. First, we realized that in the first

manuscript, uncertainties in settings of emission controls such as timing of introduction and penetration rates of abatement equipment were not considered. Therefore, we revisited the settings and assumptions for uncertainties of removal efficiencies. Because it is difficult to assume the corresponding uncertainties in each year of the target period of REASv3, we decided to analyze the uncertainties of emissions in REASv3 focusing in the years 1955, 1985, and 2015. Uncertainties for all target years of REASv3 will be analyzed in future studies. Details of methodology including equations, settings of uncertainties of each component, and related assumptions were described in Sect. S10 of the Supplement. In addition, as described in "General Reply", we thoroughly checked the data and system of REASv3 which include those for estimation of uncertainties and also found several points need to be revised including trivial errors. By the revisions, uncertainties of $SO_2$ became lager and those of $CO_2$ became smaller compared to previous results. Corresponding descriptions in Sect. 3.4 of the main manuscript were revised.

For combination of uncertainties across emission sources, in this study, it was assumed that the uncertainties were independent.

Author's changes in manuscript

- Descriptions about the settings of uncertainties including removal efficiencies and combination of different emission sources were added to the revised main manuscript (L860-868, L870-872 (P27)). In addition, details of methodology and assumption of settings were provided in the Supplement and cited in the revise main manuscript (L872-873 (P27)).

- As described in the Author's response to the Referee comments, settings and assumptions for uncertainties of removal efficiencies were reconsidered and several errors were corrected. Furthermore, emissions themselves were updated in the revised main manuscript. Therefore, all uncertainties were recalculated focusing in the years 1955, 1985, and 2015 and related discussions in the following parts were revised: L878-887, L892-893, L894-897, L899-902 (P28).

(2) Comments, author's response, and author's changes in manuscript related to Referee #2

**No. 1**

*Referee comments*

*The authors developed a new version of the REAS emission inventory, presented the trends in Asian air pollutant emissions, and analyzed the regional and sectoral drivers. The comparison with the up-to-date regional inventories presents broadly consistent emission trends, and the uncertainties in the REAS v3.1 estimates are quantified according to the errors in each parameter. This is quite important work, because the REAS inventory has been widely used in the modeling of climate and of air quality. This new version extended the emission time series to 1950-2015 and made necessary updates in both the methods and the data input. My major concern is that the method part is not well structured and is very difficult to follow, and the comparison with the previous emissions data lacks the top-down inversion estimates, which should*

Author's response to the Referee comments

One major point which was pointed out by both Referees #1 and #2 are necessity of providing details of the methodology in the manuscript. We totally agree the indications and created a new supplement of the manuscript entitled "Supplementary information and data related to methodology of REASv3" which provides detailed descriptions for the framework, activity data, emission factors, emission controls and other settings adopted in REASv3 including definition of sectors, data sources, treatment of the data, related assumptions, etc. (Hereafter, referred as "the Supplement")

For development of the Supplement, we thoroughly checked the data and system of REASv3.1 (a version of the ACPD paper) and found several points which should be revised including trivial errors in the data and system. Based on the results of the checks, revisions of the data and system were conducted including correction of the errors. In general, discussion and conclusions of the manuscript were not influenced by the revision. However, for some species, countries and regions, there were discrepancies between REASv3.1 and the revised one which is tentatively named as REASv3.2. Therefore, we prepared another supplemental document showing the differences between REASv3.2 and REASv3.1 and causes of the discrepancies entitled "Differences between REASv3.2 and REASv3.1". For distribution of the revised data, considering the possibility of additional modification during the revision processes, we would like to take the following processes:

● We did not use the detailed version number (REASv3.1), but used REAS version 3 (REASv3) in the revised main manuscript including the title. The detailed version number were described only in the "Data availability" section.

● The tentative data during the revision processes will not be opened in the download site of REAS.

- When the revision process has been completed, the final version will be opened at the REAS download site as REASv3.2.

Author's changes in manuscript
- The same changes described in "Author's change in manuscript" for Referee Comments No.1 of Referee #1.

**No. 2**

*Referee comments*

*1) Method. The method section should put more focus on the new features of the new REAS version 3.1 compared to the last version 2.1. Please summarize the new data development process and give a detailed table to show the new methods developed and the new data sources used in the REAS v3.1. Part of the REAS v2.1 emissions data are directly adopted by the REAS v3.1, such as the agricultural sources in Japan, which should be described clearly in this table. The REAS inventory relies on plenty of other emission inventories to provide the emissions data or the spatial proxies used in the emission distribution. The data dependencies across different inventories would better be clarified specifically in a new table, which would benefit the users of different inventories.*

Author's response to the Referee comments

The major new feature of REASv3 is a development of the long-term period emission inventory in Asia. A lot of database, statistics, literatures, and information were corrected, surveyed and processed for the development. On the other hand, for estimation of emissions, more detailed information such as for abatement technologies and regulations for road vehicles were taken into considered compare to previous versions of REAS, but basic methodologies themselves were based on traditional ones. Therefore, providing a new table seems to be too much for the new features, but instead, important points and updates from previous versions of REAS were summarized in bullet point format to emphasize them in Sect. 2.1 of the revised main manuscript. In addition, as described above, we developed the Supplement describing details of REASv3 including processing of historical data and appropriate parts of the Supplement were cited in Sect. 2 of the revised main manuscript as described in the Reply for 2). For data dependencies across different inventories, a new table was created (as Table 2) providing other inventories utilized in REASv3 with descriptions how the datasets were utilized.

Author's changes in manuscript
- Sect. 2.1 was fully revised and Table 2 was newly added. (Therefore, the numbers of Tables 2 and 3 in ACPD papers were shifted to 3 and 4, respectively.)

- Table S1 of the ACPD paper for countries and sub-regions included in REASv3.1 was moved to Table 2.3 in Sect. S2 of the Supplement which provides Framework of REASv3. The corresponding description was added to Sect. 2.1 of the revised main manuscript (L125-126 (P4)). For numbers of supplementary tables, Tables S2-S4 in the supplement of ACPD paper were shifted to Tables S1-S3 in the revised supplementary tables.

**No. 3**

*Referee comments*

*2) Data sources. The manuscript briefly describes the sources of the input data, but the values of parameters are not given. I understand that it is difficult to present all the detailed input data of a large-scale emission inventory. However, knowing the exact values of some key parameters can help the audience understand the drivers of emissions changes. I suggest the authors present some key parameters that determine the curve of emission changes, show their values, and discuss why such values are adopted (e.g., due to more stringent emission legislations). I noticed that the authors used many proxy data to calculate the "trend factors" when the activity data of the past years are not available. This method needs to be justified. Please show the relationship between the proxy data and the associated activity data using the historical values when they are both available.*

Author's response to the Referee comments

As described above, we developed the Supplement providing details of REASv3 including values of emission factors and removal efficiencies for major sources, data sources and treatment of activity data with assumptions for estimating missing historical data. Appropriate parts of the Supplement were indicated in Sect. 2 of the revised main manuscript. The revisions conducted in the main manuscript related to the Supplement were as follows:

➢ Sect. 2.1 (General description) was fully revised also referring comments from Referee #2, including addition of a new table (Table 2 entitled "Emission inventories from other research works and officially opened data utilized in REASv3."). In Sect. 2.1 of the revised main manuscript, the Supplement was introduced.

➢ In Sect. 2.2.1, Sects. S2.4.1 and S2.4.2 of the Supplement were cited for descriptions for combustion and non-combustion sources.

➢ In Sect. 2.2.2, Sects. S3.1.1-6, and S4.1 of the Supplement were cited for definition of fuel types and details of activity data for stationary sources, including fuel consumption, industrial production, and other transformation.

➢ In Sect. 2.2.3, Sects. S3.2, S4.2, S5.1.5, S5.2.5, and S8.3 of the Supplement were cited for emission factors and emission controls for stationary combustion, industrial production, other transformation sector.

➢ In Sect. 2.3.1, Sects. S6.2.1, S6.2.3, and S6.3 of the Supplement were cited for additional information about methodology of road transport sector.

➢ In Sect. 2.3.2, Sect. S6.1.1 of the Supplement was cited for number of vehicles and annual vehicles kilometer traveled. In addition, wrong citations of references in the previous main manuscript were corrected as follows:

✧ L246 of the previous manuscript: Road Transport Yearbook (Morth, 2003-2017) was changed to TERI Energy & Environment Data Diary and Yearbook (TERI, 2013, 2018).

✧ L249 of the previous manuscript: Pandey and Venkataraman (2014) was deleted.

✧ L252-253 of the previous manuscript: "In this study, settings of REASv2.1 were used as default and were updated if new information was available, such as Pandey and Venkataraman (2014), Sahu et al. (2014) and Mishra and Goyal (2014)." was revised as "In this study, settings of Streets et al. (2003a) and REASv2.1 were used as default and were updated if national information was available, such as He et al. (2005), Yan and Crookes (2009), Sahu et al. (2014), and Malla (2014).".

➢ Sect. 2.3.3 for emission factors of road transport was fully revised and Sect. S6.2 of the Supplement was cited.

➢ In Sect. 2.4.1, Sect. S8.1 of the Supplement was cited for methodologies and data sources for manure management sector for $NH_3$.

➢ In Sect. 2.4.2, Sect. S8.2 of the Supplement was cited for methodologies and data sources for fertilizer application sector for $NH_3$.

➢ In Sect. 2.5, Sects. S5, S7, S8.4, and S8.5 of the Supplement were cited for activity data and emission factors for non-combustion sources of NMVOC, $NH_3$, and other transport sector.

➢ In Sect. 2.6, Sects. S9.1 and S9.2 of the Supplement were cited for methodologies and data sources for grid allocation and monthly variation factors.

➢ In Sect. 3.4, Sect. S10 of the Supplement was cited for methodologies and settings of uncertainties of each component.

Author's changes in manuscript

● The same changes described in "Author's change in manuscript" for Referee Comments No.2 of Referee #1.

**No. 4**

*Referee comments*

*3) Results. The results section mainly focuses on the emissions of SO2, NOx, and BC. Please add CO2 in each plot of the results to reflect the energy consumption trends. It is difficult to understand the drivers of emission changes from the text now. Please quantitatively estimate the contributions of*

*the energy consumption growth and of the air pollution control progresses on the emission changes over each region discussed in Sect. 3.*

Author's response to the Referee comments

First, the curves of $CO_2$ emissions were added to each panel of $SO_2$, $NO_x$, and BC emissions. Then, we added some quantitative discussion on drivers of emission changes for major points of trends in Sects. 3.1.2-3.1.5. However, for emission controls, as seen in Sects. S3 and S4 of the Supplement, available data and information were limited except for China and Japan. Therefore, in this manuscript, detailed discussions on effects of emission controls were conducted focusing on China and Japan. Further surveys of local information of emission controls and related abatement technologies are necessary especially for countries and regions other than China and Japan and detailed discussion are important tasks in future studies. These points were emphasized in Sect. 4 of the revised main manuscript.

Author's changes in manuscript
- The curves of total $CO_2$ emissions were added to Figs. 3-7. Descriptions related to the $CO_2$ emissions were added to Sects. 3.1.2-3.1.5 as follows: L479 (P15); L489-L491, L504-505 (P16); L530-532, L540 (P17); L558-561 (P18); L582-585, L592-596 (P19); L631-634 (P20); L649-651 (P21).
- Descriptions related to quantitative discussions on drivers of emission changes for major points of trends were also added to Sects. 3.1.2-3.1.5 as follows: L486-488, L496-499, L504-506 (P16); L519-520, L542-544, L547-548 (P17); L552-553, L563-564, L571-573 (P18); L590-592, L602-606, L614-616 (P19); L638-640, L642-646 (P20)

**No. 5**

*Referee comments*

*For the comparison with other inventories, the authors only compared their emission results with other bottom-up emission inventories, while did not consider topdown emissions data constrained by satellite observations that have developed very fast in recent years. In my opinion, different bottom-up emission inventories commonly share the same sources of input data, which are not completely independent of each other. It would be better to evaluate the long-term emission trends with top-down information from previous literature.*

Author's response to the Referee comments

We agree with importance of comparison of bottom-up emission inventories with top-down emissions data. The following data were plotted to the figures for comparisons of inventories and

discussed:

- Ding, J., Miyazaki, K., van der A, R. J., Mijling, B., Kurokawa, J.-I., Cho, S., Janssens-Maenhout, G., Zhang, Q., Liu, F., and Levelt, P. F.: Intercomparison of $NO_x$ emission inventories over East Asia, Atmos. Chem. Phys., 17, 10125–10141, https://doi.org/10.5194/acp-17-10125-2017, 2017.

- Itahashi, S., Yumimoto, K., Kurokawa, J., Morino, Y., Nagashima, T., Miyazaki, K., Maki, T., and Ohara, T.: Inverse estimation of $NO_x$ emissions over China and India2005–2016: contrasting recent trends and future perspectives, Environ. Res. Lett., 14, 124020, https://doi.org/10.1088/1748-9326/ab4d7f, 2019.

- Jiang, Z., Worden, J. R., Worden, H., Deeter, M., Jones, D. B. A., Arellano, A. F., and Henze, D. K.: A 15-year record of CO emissions constrained by MOPITT CO observations, Atmos. Chem. Phys., 17, 4565–4583, https://doi.org/10.5194/acp-17-4565-2017, 2017.

- Miyazaki, K., Bowman, K., Sekiya, T., Eskes, H., Boersma, F., Worden, H., Livesey, N., Payne, V. H., Sudo, K., Kanaya, Y., Takigawa, M., and Ogochi, K.: An updated tropospheric chemistry reanalysis and emission estimates, TCR-2, for 2005–2018, Earth Syst. Sci. Data Discuss., https://doi.org/10.5194/essd-2020-30, in review, 2020.

- Qu, Z., Henze, D. K., Li, C., Theys, N., Wang, Y., Wang, J., Wang, W., Han, J., Shim, C., Dickerson, R. R., and Ren, X.: $SO_2$ emission estimates using OMI $SO_2$ retrievals for 2005–2017, J. Geophys. Res. Atmos., 124, 8336-8359, https://doi.org/10.1029/2019JD030243, 2019.

- Stavrakou, T., Muller, J. F., Bauwens, M., De Smedt, I.: Sources and long-term trends of ozone precursors to Asian Pollution, Air Pollution in Eastern Asia: an integrated perspective, eds. Bouarar, I., Wang, X., Brasseur, G., Springer international Publishing, 167–189, https://doi.org/10.1007/978-3-319-59489-7-8, 2017.

- Zheng, B., Chevallier, F., Yin, Y., Ciais, P., Fortems-Cheiney, A., Deeter, M. N., Parker, R. J., Wang, Y., Worden, H. M., and Zhao, Y.: Global atmospheric carbon monoxide budget 2000–2017 inferred from multi-species atmospheric inversions, Earth Syst. Sci. Data, 11, 1411–1436, https://doi.org/10.5194/essd-11-1411-2019, 2019.

Furthermore, we added following two bottom-up historical emission inventories of China:

- Sun, W., Shao, M., Granier, C., Liu, Y., Ye, C. S., and Zheng, J. Y.: Long-term trends of anthropogenic $SO_2$, $NO_x$, CO, and NMVOCs emissions in China, Earth's Future, 6, 1112-1133, https://doi.org/10.1029/2018EF000822, 2018.

- Wang, R., Tao, S., Wang, W., Liu, J., Shen, H., Shen, G., Wang, B., Liu, X., Li, W., Huang, Y., Zhang, Y., Lu, Y., Chen, H., Chen, Y., Wang, C., Zhu, D., Wang, X., Li, B., Liu, X., and Ma, J.: Black Carbon Emissions in China from 1949 to 2050, Environ. Sci. Technol., 46, 7595-7603, https://doi.org/10.1021/es3003684, 2012.

Author's changes in manuscript

- For comparison with other inventories, results of inverse modeling were added to Figs. 10 ($SO_2$ and $NO_x$ for China), 11 ($SO_2$ and $NO_x$ for India), S14 (CO and NMVOC for China) and S15 (CO for India). Discussions with inverse modeling results were added to Sect. S3.3 as follows: L716-719, L723-726, L740-743, L744-754 (P23); L756-759, L763-766, L773-776 (P24)

- Furthermore, two bottom-up emission inventories were added for China. (Sun et al. (2018) for $SO_2$ and $NO_x$ in Fig. 10 and for CO and NMVOC in Fig. S14; Wang et al. (2012) for BC in Fig. 10)

**No. 6**

*Referee comments*

*For the uncertainty assessment, I cannot understand why the uncertainties of CO2 emissions are so large, particularly ±28% for China and ±23% for Japan, which are much higher than the typical uncertainty range (±10%) of country CO2 emissions.*

Author's response to the Referee comments

Thank you for pointing out the issue. First, from comments of Referee #1, we realized that in the first manuscript, uncertainties in settings of emission controls such as timing of introduction and penetration rates of abatement equipment were not considered. Therefore, we revisited the settings and assumptions for uncertainties of removal efficiencies. Details of methodology including equations, settings of uncertainties of each component, and related assumptions were described in Sect. S10 of the Supplement. In addition, as described in "General Reply", we thoroughly checked the data and system of REASv3 which include those for estimation of uncertainties and found several points need to be revised including trivial errors. By the revisions, uncertainties of $SO_2$ became lager and those of $CO_2$ became smaller compared to previous results. Corresponding descriptions in Sect. 3.4 were revised.

For $CO_2$, in the revisiting process, we found errors in settings of uncertainties of $CO_2$ emission factors for fossil fuel combustion. After the correction of errors, as described above, uncertainties of $CO_2$ emissions became lower than those in first manuscript. For Japan, the updated uncertainties are ±13% from ±23% in the first manuscript. However, for China, even after the correction of errors, the updated uncertainties (±19%) were still higher than ±10%. One reason is that high uncertainties were assumed for emission factors of biofuel combustion (50%). Another considerable reason is that in REASv3, uncertainties in fossil fuel consumption data were assumed to be higher than those of OECD countries except for Japan, Republic of Korea, and Taiwan. For example, uncertainties in coal consumption in power plants, small industries, and residential sectors in China were assumed to

be 10%, 15%, and 20%, respectively.

Author's changes in manuscript

- The changes related to uncertainties are the same as for "Author's change in manuscript" for Referee Comments No.14 of Referee #1.
- Discussions on problems in uncertainties for $CO_2$ emissions described in the authors response were not included in the revised main manuscript because corresponding discussions are considered to be a bit too specific for this manuscript.

**No. 7**

*Referee comments*

*1) Line 47 on Page 2. The GAINS model not the GANS model.*

Author's response to the Referee comments

Thank you for pointing out the typo. It was corrected.

Author's changes in manuscript

- The corresponding type was corrected (P2L47).

**No. 8**

*Referee comments*

*2) Line 318 on Page 10. For spatial distribution not the special distribution.*

Author's response to the Referee comments

Thank you for pointing out the typo. It was corrected.

Author's changes in manuscript

- The corresponding type was corrected (P12L362).

**No. 9**

*Referee comments*

*3) Lines 354 and 355 on Page 12. Please clarify how the information of large plants is used for developing allocation factors for corresponding emission source sectors.*

Author's response to the Referee comments

In Sect. S9 of the Supplement, how to utilize the information of large plants is explained. It was

referred in the main manuscript.

Author's changes in manuscript
● Details were described in Sect. S9.1 of the Supplement and the description related to citing the Supplement was added to Sect. 2.6 (L410 (P13)).

**No. 10**

*Referee comments*

*4) The caption of Figure 3. During 1990-2015 not 1950-2015.*

Author's response to the Referee comments

Thank you for pointing out the typo. It was corrected.

Author's changes in manuscript
● The corresponding type was corrected (the caption of Fig.3).

**No. 10**

*Referee comments*

*5) Figures 10 and 11. The colors of some curves are close to each other and are difficult to distinguish. And please also add the uncertainty range of REAS v3.1 in the plots.*

Author's response to the Referee comments

Thank you for pointing out the issue. For curves which were difficult to be identified, colors or line thicknesses were changed. For uncertainties, as described above, we revisited the settings and assumptions for uncertainties of removal efficiencies. Because it is difficult to assume the corresponding uncertainties in each year of the target period of REASv3, we decided to analyze the uncertainties of emissions in REASv3 focusing in the years 1955, 1985, and 2015. For the uncertainty ranges, error bars in 1955, 1985, and 2015 were added to the plots of comparisons. Uncertainties for all target years of REASv3 will be analyzed in future studies.

Author's changes in manuscript
● Colors, shapes, thickness, etc. of plotted data in Figs. 10, 11, and S14-19 were reconsidered and revised.
● Error bars based on uncertainties ranges of REASv3 were added to data in 1955, 1985, and 2015 for all panels in Figs. 10, 11, and S14-S19. Related descriptions were added to Sect. S3.3 (L701-703 (P22)) and captions of Figs. 10, 11, and S14-S19.

(3) The revised main manuscript where changed parts were yellow highlighted

From next pages, the revised main manuscript where changed parts were yellow highlighted is provided. In addition to the revisions based on comments from Referees #1 and #2 described in (1) and (2), some additional revisions were done mainly for correction of typos and English problems as follows:

- "control measures" were used instead of "regulations and laws" in abstract (L21 (P1)) and Sect. 4 (L945 (P30)).
- "the Republic of Korea" was changed to "Republic of Korea" in several points: L22 (P1); L658 (P21); L875 (P27); and L945 (P30)).
- "the REAS series" was changed to "the Regional Emission inventory in ASia (REAS) series" in Sect.1 (L70 (P3)).
- "PM" was changed to "PM species" in several points: L240 (P8); L621 (P20); and L666, L668 (P21).
- Acronyms SEA (Southeast Asia), OEA (East Asia other than China and Japan), and OSA (South Asia other than India) were introduced in Sect. 3.1.1 (P14L435-436) and corresponding changes were done in the following points: L466-467 (P15); L630, L631, L633, L637, L638, L642 (P20); L652, L680 (P21); L683, L686, L691, L693 (P22); and L876-877 (P28). In addition, for readers' conveniences, colors of country names in Fig.1 were changed to indicate SEA, OEA, and OSA. Furthermore, for captions of figures and tables in the revised main manuscript and supplementary materials, SEA, OEA, and OSA were used and the description "See Fig. 1 for countries included in SEA, OEA, and OSA" was added to them.
- Other small revisions such as correction of grammatical problems and English expressions were done in the following points: L19 (P1); L266, L267 (P9); L412 (P13); L430 (P14); L482, L483, L513 (P16); L523 (P17); L549, L551, L569 (P18); L588, L608, L609 (P19); L622, L624, L625, L627, L631, L633, L636, L637 (P20); L657, L675, L676, L680 (P21); L682, L685, L689, L690,. L691, L692, L693, L694 (P22); L859 (P27); L877 (P28); L918-919, L941 (P29); L956 (P30)

[revised manuscript text omitted]

(5) Revision of the supplementary materials

For the ACPD paper, one supplement including both supplementary figures and tables was provided. In the revised manuscript, as mentioned above, following two new supplements were created:

- Supplementary information and data related to methodology of REASv3
- Differences between REASv3.2 and REASv3.1

Then, we divided the first supplement into figure and table parts and also prepared the title page. Finally, five files are provided as supplements of the revised main manuscript.

For the supplementary figures, revisions from the ACPD version are as follows:

- All figures were recreated using the updated data (REASv3.2).
- Following revisions were done in Figs. S14-S19:
  - Results of inverse modeling and two bottom-up inventories were added and corresponding references were added.
  - Error bars indicating the uncertainty range in 1955, 1985, and 2015 were added.
  - Markers and lines in the figures were reconsidered and revised.
- Comparisons of total emissions in Asia among REASv3, CEDS, and EDGARv4.3.2 were added to Figs. 12 and S20.

For the supplementary tables, revisions from the ACPD version are as follows:

- Table for target countries and regions (Table S1 for the ACPD version) was moved to the new supplement for methodology (Table 2.3 in the Supplement) and thus, Tables S2-S4 in the ACPD version were shifted to Tables S1-S3.
- All data in Tables S1-S3 in the new supplementary tables were recalculated using the updated data (REASv3.2).

From next page, the five supplementary materials are provided as follows:

- Title page
- Supplementary information and data related to methodology of REASv3
- Supplementary figures
- Supplementary tables
- Differences between REASv3.2 and REASv3.1

*Supplement of*

**Long-term historical trends in air pollutant emissions in Asia: Regional Emission inventory in ASia (REAS) version 3**

**Junichi Kurokawa and Toshimasa Ohara**

5    *Correspondence to*: Junichi Kurokawa (kurokawa@acap.asia)

- Supplementary information and data related to methodology of REASv3

- Supplementary figures

- Supplementary tables

- Differences between REASv3.2 and REASv3.1

*Supplement of*

**Long-term historical trends in air pollutant emissions in Asia: Regional Emission inventory in ASia (REAS) version 3**

**Junichi Kurokawa and Toshimasa Ohara**

*Correspondence to*: Junichi Kurokawa (kurokawa@acap.asia)

**Supplementary information and data related to methodology of REASv3**

**Contents**

**S1. Introduction**

This document provides detailed information related to methodologies of Regional Emission inventory in ASia (REAS) version 3 (hereafter REASv3 in this document) developed as a supplementary material of the main manuscript entitled "Long-term historical trends in air pollutant emissions in Asia: Regional Emission inventory in ASia (REAS) version 3". In this document, first and second versions of REAS are often cited and expressed as REASv1 (Ohara et al., 2007) and REASv2 (Kurokawa et al., 2013), respectively. The framework of REASv3 such as target species, countries and regions, and emission sources was summarized in Sect. 2. Sects. 3, 4, 5, 6, and 7 provide details of activity data and emission factors including settings of emission controls for stationary combustion, industrial production, non-combustion sources of NMVOC, road transport, and other transport, respectively. The details related to methodology for non-combustion sources of NH$_3$ were given in Sect. 8. Grid allocation and monthly variation factors for spatial and temporal distribution were described in Sect. 9. In Sect. 10, details of methodology and settings for estimation of uncertainties were provided.

Note that this document is for REASv3.2 which is an updated version of REASv3.1 (Kurokawa et al., 2019). The differences between REASv3.2 and REASv3.1 and causes of the discrepancies were provided in another document entitled "Differences between REASv3.2 and REASv3.1" developed as an additional supplement of the main manuscript.

**S2. Framework of REASv3**

**S2.1 Target species**

Target species of REASv3 are summarized in Table 2.1. In REASv3, NMVOC species were divided into 19 chemical species categories as presented in Table 2.2. Codes of each species used in emission tables and gridded data of REASv3 are also provided in the tables.

**Table 2.1.** Target species of REASv3.

| Species code | Species |
|---|---|
| SO2 | Sulfur dioxide |
| NOX | Nitrogen oxides (as $NO_2$) |
| CO_ | Carbon monoxide |
| NMV | Non-methane volatile organic compounds |
| NH3 | Ammonia |
| CO2 | Carbon dioxide |
| PM10_ | Primary $PM_{10}$ |
| PM2.5 | Primary $PM_{2.5}$ |
| BC_ | Black carbon |
| OC_ | Primary Organic carbon |

**Table 2.2.** NMVOC species categories defined in REASv3.

| Species number code | NMVOC species |
|---|---|
| 01 | Ethane |
| 02 | Propane |
| 03 | Butanes |
| 04 | Pentanes |
| 05 | Other Alkanes |
| 06 | Ethylene |
| 07 | Propene |
| 08 | Terminal Alkenes |
| 09 | Internal Alkenes |
| 10 | Acetylene |
| 11 | Benzene |
| 12 | Toluene |
| 13 | Xylenes |

| 14 | Other Aromatics |
| 15 | Formaldehyde |
| 16 | Other Aromatics |
| 17 | Ketones |
| 18 | Halocarbons |
| 19 | Others |
| 20 | Total |

**S2.2 Target years**

Target years of REASv3 are 1950-2015 (each year). In future updated versions, the oldest target year is basically fixed, but data in later years (after 2016) are planned to be added.

**S2.3 Target countries and regions**

Table 2.3 provides list of countries and sub-regions included in the inventory domain of REASv3. Codes of region, countries, and sub-regions used in the main manuscript, emission tables and gridded data of REASv3 are also provided in the table.

**Table 2.3.** Region, country, and sub-region included in the inventory domain of REASv3 with codes used in the main manuscript and files of emission tables and gridded data provided from the REAS website (https://www.nies.go.jp/REAS/).

| Region name/ Region code | Country name: Sub-region name | Country and sub-region code CCCRR CCC: Country code RR: Sub-region code |
| --- | --- | --- |
| China/ CHN | China: Whole Country | CHNWC |
| | China: Beijing | CHNBJ |
| | China: Tianjin | CHNTJ |
| | China: Hebei | CHNHE |
| | China: Shanxi | CHNSX |
| | China: Inner Mongolia | CHNNM |
| | China: Liaoning | CHNLN |
| | China: Jilin | CHNJL |
| | China: Heilongjiang | CHNHL |

| | | |
|---|---|---|
| | China: Shanghai | CHNSH |
| | China: Jiangsu | CHNJS |
| | China: Zhejiang | CHNZJ |
| | China: Anhui | CHNAH |
| | China: Fujian | CHNFJ |
| | China: Jiangxi | CHNJX |
| | China: Shandong | CHNSD |
| | China: Henan | CHNHA |
| | China: Hubei | CHNHB |
| | China: Hunan | CHNHN |
| | China: Guangdong | CHNGD |
| | China: Guangxi | CHNGX |
| | China: Hainan | CHNHI |
| | China: Chongqing | CHNCQ |
| | China: Sichuan | CHNSC |
| | China: Guizhou | CHNGZ |
| | China: Yunnan | CHNYN |
| | China: Tibet | CHNXZ |
| | China: Shaanxi | CHNSN |
| | China: Gansu | CHNGS |
| | China: Qinghai | CHNQH |
| | China: Ningxia | CHNNX |
| | China: Xinjiang | CHNXJ |
| | China: Hong Kong | CHNHK |
| | China: Macau | CHNMC |
| India/ IND | India: Whole Country | INDWC |
| | India: Andhra Pradesh | INDAP |
| | India: Bihar, Jharkhand | INDBJ |
| | India: North East (Arunachal Pradesh/Assam/Manipur/ Meghalaya/Mizoram/Nagaland/Sikkim/Tripura) | INDAN |
| | India: Gujarat | INDGU |
| | India: Haryana | INDHA |
| | India: Karnataka/Goa | INDKG |
| | India: Kerala | INDKE |
| | India: Madhya Pradesh/Chhattisgarh | INDMC |

| | India: Maharashtra | INDMA |
|---|---|---|
| | India: Orissa | INDOR |
| | India: Punjab/Chandigarh | INDPU |
| | India: Rajasthan | INDRA |
| | India: Tamil Nadu | INDTN |
| | India: Utter Pradesh/Uttaranchal | INDUU |
| | India: West Bengal | INDWB |
| | India: Himachal Pradesh/Jammu and Kashmir | INDHJ |
| | India: Delhi | INDDE |
| Japan/ JPN | Japan: Whole Country | JPNWC |
| | Japan: Hokkaido-Tohoku (Hokkaido/Aomori/Iwate/ Miyagi/Akita/Yamagata/Fukukshima) | JPNHT |
| | Japan: Kanto (Ibaraki/Tochigi/Gunma/Saitama/Chiba/ Tokyo/Kanagawa) | JPNKN |
| | Japan: Chubu (Niigata/Toyama/Ishikawa/Fukui/ Yamanashi/Nagano/Gifu/Shizuoka/Aichi) | JPNCB |
| | Japan: Kinki (Mie/Shiga/Kyoto/Osaka/Hyogo/Nara/ Wakayama) | JPNKK |
| | Japan: Chugoku-Shikoku (Tottori/Shimane/Okayama/ Hiroshima/Yamaguchi/Tokushima/Kagawa/Ehime/Kochi) | JPNCS |
| | Japan: Kyushu-Okinawa (Fukuoka/Saga/Nagasaki/ Kumamoto/Oita/Miyazaki/Kagoshima/Okinawa) | JPNKO |
| Other East Asia / OEA | Democratic People's Republic of Korea, Whole Country | PRKWC |
| | Republic of Korea, Whole Country | KORWC |
| | Mongolia: Whole Country | MNGWC |
| | Taiwan: Whole Country | TWNWC |
| Southeast Asia/ SEA | Brunei: Whole Country | BRNWC |
| | Cambodia: Whole Country | KHMWC |
| | Indonesia: Whole Country | IDNWC |
| | Laos: Whole Country | LAOWC |
| | Malaysia: Whole Country | MYSWC |
| | Myanmar: Whole Country | MMRWC |
| | Philippines: Whole Country | PHLWC |
| | Singapore: Whole Country re | SGPWC |
| | Thailand: Whole Country | THAWC |

| | Vietnam: Whole Country | VNMWC |
|---|---|---|
| Other South Asia/ OSA | Afghanistan: Whole Country | AFGWC |
| | Bangladesh: Whole Country | BGDWC |
| | Bhutan: Whole Country | BTNWC |
| | Maldives: Whole Country | MDVWC |
| | Nepal: Whole Country | NPLWC |
| | Pakistan: Whole Country | PAKWC |
| | Sri Lanka: Whole Country | LKAWC |

**S2.4 Target emission sources**

**S2.4.1 Combustion sources**

Table 2.4 provides list of sub-sector categories of combustion sources defined in REASv3. Aggregated sector categories used in the main manuscript and emission tables of REASv3 are presented as "Sector code". IEA codes show relationships between sub-sector categories of REASv3 and the International Energy Agency (IEA) World Energy Balances (IEAWEB) (IEA, 2017). Fuel types defined in REASv3 are provided in Sect S3.1.1. See Sects. S3, S6, and S7 for details of stationary combustion, road transport, and other transport sectors, respectively.

Several emission sources related to transformation sectors except for power plants were included in Table 2.4. Sources categorized as energy sectors in IEAWEB are only considered as combustion sources. For coke ovens (not as the energy sector), emissions were estimated based on coal input for $SO_2$ and $NO_x$ and coke production for CO, NMVOC, $CO_2$, and PM species. In REASv3, for coke ovens as energy transformation sectors, contributions from both combustion and non-combustion processes were included in the emissions. In other words, their emissions were not estimated separately. Similarly, the following sources include both combustion and non-combustion emissions which were not estimated separately:

● Charcoal production plants

● Manufacture of other solid fuels

● Gas works

In addition, CO emissions from pig iron, crude steel, and sinter production for all countries, those from brick production except for China, Japan, Republic of Korea, and Taiwan, emissions of PM species from sinter and pig iron production for China, and those from brick production for all countries estimated based on their production amounts include contributions from both combustion and non-combustion sources (not estimated separately).

**Table 2.4.** Sub-sector categories of combustion sources considered in REASv3 with sector codes used in the main manuscript and emission tables of REASv3 and IEA codes showing relationships between sub-sector categories of REASv3 and the IEAWEB.

| Sector code | Sub-sector category | IEA code |
|---|---|---|
| Power Plants/ PP | Power plants (point sources/area sources) | MAINELEC/AUTOELEC/ MAINCHP/AUTOCHP/ MAINHEAT/AUTOHEAT/ THEA/TBOILER/TELE |
| | Power plants (energy) | EPOWERPLT |
| Industry/ IND | Coke ovens | TCOKEOVS |
| | Charcoal production plants | TCHARCOAL |
| | Manufacture of other solid fuels | TPATFUEL/TBKB/TNONSPEC |
| | Coke ovens (energy) | ECOKEOVS |
| | Charcoal production plants (energy) | ECHARCOAL |
| | Manufacture of other solid fuels (energy) | EMINES/EPATFUEL/EBKB/ ENONSPEC |
| | Petroleum refineries (energy) | EREFINER |
| | Manufacture of other liquid fuels (energy) | EOILGASEX/ECOALLIQ/EGTL |
| | Gas works | TGASWKS |
| | Gas works (energy) | EGASWKS |
| | Manufacture of other gaseous fuels (energy) | ELNG/EGTL |
| | Chemical and petrochemical industry | CHEMICAL |
| | Iron and steel industry | IRONSTL |
| | Blast furnace | TBLASTFUR |
| | Blast furnace (energy) | EBLASTFUR |
| | Non-ferrous metal industry | NONFERR |
| | Cement industry | NONMET |
| | Lime industry | |
| | Brick industry | |
| | Other non-metallic minerals industries | |
| | Construction industry | CONSTRUC |
| | Transport equipment industry | TRANSEQ |
| | Machinery industry | MACHINE |
| | Mining and quarrying industry | MINING |
| | Food and tobacco industry | FOODPRO |

| | Paper, pulp and printing industry | PAPERPRO |
|---|---|---|
| | Wood and wood products industry | WOODPRO |
| | Textile and leather industry | TEXTILES |
| | Other industries | INONSPEC |
| Road transport/ ROAD | Road transport | ROAD |
| Other transport/ OTRA | Rail | RAIL |
| | Pipeline transport | PIPLINE |
| | Other transport*[1] | TRNONSPE |
| Residential/ RESI | Residential | RESIDENT |
| Other domestic/ ODOM | Commercial and public services | COMMPUB |
| | Agriculture*[2] | AGRICULT |
| | Others | ONONSPEC |

*[1]Aviation and navigation (both for domestic and international) are not included.

*[2]Forestry is included, but fishing is not included.

**S2.4.2 Non-combustion sources: Industrial production and other transformation**

Table 2.5 provides list of sub-sector categories of non-combustion sources defined in REASv3 with target species and notes for each sub-sector category. See Sect. S4 for details of industrial processes and other transformation. See Sects. S5 and S8 for industrial processes related to NMVOC and $NH_3$, respectively. Note that, as described in Sect S2.4.1, non-combustion emissions from coke production, those of CO from pig iron, crude steel, and sinter productions (for all countries and regions) and from brick production (except for China, Japan, Republic of Korea, and Taiwan), and those of PM species from sinter and pig iron production (for China) and from brick production (for all countries) were not estimated separately. For these sources, estimated emission in REASv3 include contributions from both combustion and non-combustion processes.

**Table 2.5.** Sub-sector categories of non-combustion sources from industrial production and other transformation considered in REASv3.

| Sub-sector category | Target species | Notes |
|---|---|---|
| Pig iron production | CO, PM species | Iron and steel industry |
| Crude steel production | CO, NMVOC, PM species | |

| | | |
|---|---|---|
| Sinter production | CO, PM species | |
| Rolled steel production | NMVOC | |
| Copper production | $SO_2$, PM species | Non-ferrous metal industry |
| Zinc production | $SO_2$, PM species | |
| Lead production | $SO_2$, PM species | |
| Almina production | $SO_2$, PM species | |
| Aluminium production | $SO_2$, PM species | |
| Cement production | $CO_2$, PM species | Non-metallic minerals industry |
| Lime production | $CO_2$, PM species | |
| Brick production | PM species | |
| Sulphuric acid production | $SO_2$ | Inorganic chemicals industry |
| Carbon black production | NMVOC, PM species | |
| Ethylene production | NMVOC | Organic chemicals industry |
| Polyethylene production | NMVOC | |
| Styrene production | NMVOC | |
| Polystyrene production | NMVOC | |
| Polyvinylchloride production | NMVOC | |
| Propylene production | NMVOC | |
| Polypropylene production | NMVOC | |
| Polyvinylchloride processing | NMVOC | |
| Polystyrene processing | NMVOC | |
| Bread production | NMVOC | Other industries considered for NMVOC |
| Beer production | NMVOC | |
| Asphalt production | NMVOC | |
| Pulp and paper production | NMVOC | |
| Ammonia | $NH_3$ | Synthetic fertilizer industry considered for $NH_3$ |
| Ammonium nitrate | $NH_3$ | |
| Urea | $NH_3$ | |
| Coke production | CO, NMVOC, $CO_2$, PM species | Manufacture of solid fuels |
| Petroleum refineries | $SO_2$, NMVOC, PM species | Manufacture of liquid fuels For NMVOC, contributions were included in extraction processes. See Sect. S2.4.3. |

**S2.4.3 Non-combustion sources of NMVOC**

Non-combustion sources for NMVOC emissions considered in REASv3 are extraction processes, solvent use, industrial processes, waste disposal and evaporative emissions from road vehicles. Sub-categories of extraction processes and solvent use are summarized in Tables 2.6 and 2.7. Definitions of the sub-sectors are the same as with those of Klimont et al. (2002a). See Table 2.5, Sect. 5.1.7 and Sect. S6.3 for industrial processes, waste disposal, and evaporative emissions from road vehicles, respectively. See Sect. S5 for details of non-combustion sources of NMVOC.

**Table 2.6.** Sub-sector categories of extraction processes considered in REASv3.

| Sub-category |
| --- |
| Gas production |
| Gas distribution |
| Crude oil production |
| Crude oil handling |
| Petroleum refineries[a] |
| Service station |
| Transport and depots |

a. Except for NMVOC, contributions were included in industrial processes. See Sect. S2.4.2.

**Table 2.7.** Sub-sector categories of solvent use considered in REASv3.

| Sub-category |
| --- |
| Dry cleaning |
| Decreasing operation |
| Vehicle treatment |
| Domestic use of solvents |
| Asphalt blowing |
| Paint production |
| Ink production |
| Tire production |
| Synthetic rubber production |
| Textile industry |
| Preservation of wood |
| Adhesive application |
| Printing[a] |
| Paint application[b] |

a. Contributions from following activities were included: packing offset printing, publication, and screen printing were included. b. Contributions from following purposes were included: architecture, domestic usage, automobile manufacture, vehicle refinishing, and other industrial application.

**S2.4.4 Non-combustion sources of NH₃**

Non-combustion sources for $NH_3$ emissions considered in REASv3 are manure management of livestock, fertilizer application, industrial processes, human, and latrines as summarized in Table 2.8. See Sect. S8 for details of non-combustion sources of $NH_3$.

**Table 2.8.** Sub-sector categories of non-combustion sources of $NH_3$ considered in REASv3.

| Sub-category |
| --- |
| Manure management[a] |
| Fertilizer application[b] |
| Industrial processes[c] |
| Human[d] |
| Latrines |

a. Contributions from manure management including housing, storage and yards were included. Those from manure applied to soils were included in fertilizer application. b. Contributions from both synthetic fertilizer and animal manure used as fertilizer were included. c. See Sect. S2.4.2. d. Contributions from perspiration and respiration were included.

**S2.5 Spatial and temporal resolution**

In REASv3, only large power plants are treated as point sources and gridded data of other emission sources are provided with a horizontal resolution of 0.25° × 0.25°. For temporal resolution, monthly emissions are estimated in REASv3 by allocating annual emissions to each month using monthly proxy data. Details of methodologies and data used for spatial and temporal allocation are described in Sect. S9.

Table 2.9 provides sub-sector categories included in aggregated sector codes for gridded data in REASv3.

**Table 2.9.** Sector codes for gridded data in REASv3 and sub-sector categories included in each code.

| Sector categories code | Sub-sector categories included in each sector code |
|---|---|
| POWER_PLANTS_POINT | Power plants (points) in Table 2.4 |
| POWER_PLANTS_NON-POINT | Power plants (area sources and energy) in Table 2.4 |
| INDUSTRY | Combustion sources of industry sector in Table 2.4 |
| | Non-combustion sources of industrial production and other transformation sector in Table 2.5 |
| ROAD_TRANSPORT | Road transport sector in Table 2.4 |
| | Evaporative NMVOC emissions from road vehicles described in Sect. S6.3 |
| OTHER_TRANSPORT | Other transport sector in Table 2.4 |
| DOMESTIC | Residential and other domestic sectors in Table 2.4 |
| EXTRACTION | NMVOC emissions from extraction processes in Table 2.6 |
| SOLVENTGS | NMVOC emissions from solvent use in Table 2.7 |
| WASTE | NMVOC emissions from waste disposal described in Sect. S5.1.7 |
| MANURE_MANAGEMENT | $NH_3$ emissions from manure management described in Sect. S8.1 |
| FERTILIZER | $NH_3$ emissions from fertilizer application described in Sect. S8.2 |
| MISC | $NH_3$ emissions from human and latrines described in Sects. S8.4 and S8.5. |

**S3. Stationary combustion**

**S3.1 Activity data**

**S3.1.1 Definition of fuel types**

Table 3.1 describes fuel types considered in stationary combustion sources of REASv3. Emissions of air pollutants were estimated individually for each fuel type. In Fig. 4 of the main manuscript and Figs. S2, S4, S6, S8, S10, and S12 of the supplement, fuel types are aggregated to several categories. Definition of the categories are also provided in Table 3.1. For each fuel type, definitions are mostly the same as those of the International Energy Agency (IEA) World Energy Balances (IEAWEB) (IEA, 2017). Exceptions are "Raw coal", "Cleaned coal", "Other washed coal", and "Other coking products" which are defined only for China in the China Energy Statistical Yearbook (CESY) (National Bureau of Statistics of China, 1986, 2001-2017). Definition of "Bituminous coal", "Kerosene", and "Diesel oil" in Table 3.1. is the same as that of "Other bituminous coal", "Other kerosene", and "Gas/diesel oil excl. biofuels" of IEAWEB, respectively. For hard (brown) coal, if there is no detailed information, corresponding fuel type is considered as "Bituminous coal" ("Lignite"). For other fuel types, emissions from combustion were ignored in REASv3.

**Table 3.1.** List of detailed fuel types considered in REASv3 and definition of aggregated categories used in the main manuscript and the supplement.

| Aggregated categories (code) | Aggregated categories (description) | Detailed fuel types |
|---|---|---|
| COAL | Primary coal | Coking coal |
| | | Anthracite |
| | | Bituminous coal |
| | | Raw coal |
| | | Cleaned coal |
| | | Other washed coal |
| | | Sub-bituminous coal |
| | | Lignite |
| DC | Secondary coal | Coke oven coke |
| | | Gas coke |
| | | Coal tar |
| | | Patent fuel |
| | | Brown coal briquettes (BKB) |
| | | Other coking products |
| NGAS | Natural gas | Natural gas |
| OGAS | Other gas fuels | Gas works gas |
| | | Coke oven gas |
| | | Blast furnace gas |
| | | Other recovered gases |
| LF | Light oil fuels | Refinery gas |
| | | Liquefied petroleum gas (LPG) |
| | | Natural gas liquids |
| | | Motor gasoline |
| | | Naphtha |
| | | Kerosene |
| MD | Diesel oil | Diesel oil |
| HF | Heavy oil fuels | Crude oil |
| | | Heavy fuel oil |
| | | Petroleum coke |
| | | Other oil products |
| BF | Biofuel | Fuelwood |
| | | Crop Residue |

| | | | Animal waste |
|---|---|---|---|
| | | | Biogas |
| | | | Biogasoline |
| | | | Biodiesels |
| | | | Charcoal |
| OTH | Other fuels | | Municipal waste (renewable) |
| | | | Municipal waste (non-renewable) |
| | | | Industrial waste |

**S3.1.2 Data sources of fuel consumption and assumptions to estimate missing historical data**

In REASv3, fuel consumption data were primarily obtained from IEAWEB, CESY, the United Nations (UN) Energy Statistics Database (UN, 2016), and UN data, which is a web-based data service of the UN (http://data.un.org/). However, all these sources do not include data for the entire target period of REASv3, that is from 1950-2015. Furthermore, past data for sectors do not contain as many categories. In this sub-section, data sources and assumptions for estimating missing historical data used in REASv3 are summarized in Table 3.2 including how to distribute total or sub-total data to detailed sub-sectors and how to extrapolate data until 1950. Note that descriptions for fuel consumption data in transport sector are also included in this sub-section.

**Table. 3.2.** Data sources and assumptions for estimating missing historical data used in REASv3 for each country and region.

**(a) China**

| Data sources and treatments | • Fuel consumption for each region except for Tibet, Hong Kong and Macau were obtained from CESY during 1985-2015 and those before 1984 were extrapolated to 1950 using data for whole China during 1950-2015. See Sect. S3.1.3 for regional fuel consumption data in China.

• Data of whole country were taken from IEAWEB during 1971-2015 and extrapolated to 1950. Those of Tibet were taken from REASv2 (based on GAINS ASIA at that time) during 2000-2008 and extrapolated using data of whole country. See (n) and (o) of this sub-section for Hong Kong and Macau, respectively. |
|---|---|
| Assumptions for estimating missing historical data | • Assumptions for modifying IEAWEB during 1971-2015 are as follows:
  ➢ Energy industry own use sector:
    ✦ Data of bituminous coal and natural gas before 1989 were distributed to sub-sectors based on relative ratios of fuel consumption data in 1990.
    ✦ Fuel consumption data of coke oven gas in 1990 were extrapolated to 1980 using trends of coke oven gas production in IEAWEB during 1980-1990 and then, extrapolated to 1971 based on trends of coke oven coke production in IEAWEB during 1971-1980.
  ➢ Industry sector:
    ✦ Data of coking coal, gas works gas, coke oven gas, refinery gas, and LPG/other bituminous coal and crude oil/natural gas, other kerosene, diesel oil, and heavy fuel oil before 1989/1984/1979 were distributed to sub-sectors based on relative ratios in 1990/1985/1980.
    ✦ Fuel consumption data of coke oven gas in 1980 were extrapolated to 1971 using trends of coke oven gas production in IEAWEB during 1971-1980.
  ➢ Transport sector:
    ✦ Data of diesel oil before 1989 were distributed to road transport, domestic navigation, and agriculture/forestry based on relative ratios of corresponding fuel consumption in 1990.
• See "Assumption for data extrapolation" in this sub-section how to extrapolate the data of IEAWEB to 1950. |

**(b) India**

| Data sources and treatments | • Data of whole country were taken from IEAWEB during 1971-2015 and extrapolated to 1950.
• See Sect. S3.1.4 for regional fuel consumption data in India. |
|---|---|
| Assumptions for estimating missing historical data | • No major modifications were done for IEAWEB during 1971-2015.
• See "Assumption for data extrapolation" in this sub-section how to extrapolate the data of IEAWEB to 1950. |

**(c) Japan**

| Data sources and treatments | • Data of whole country were taken from IEAWEB during 1960-2015 and extrapolated to 1950.
• See Sect. S3.1.5 for regional fuel consumption data in Japan. |
|---|---|
| Assumptions for estimating missing historical data | • Assumptions for modifying IEAWEB during 1960-2015 are as follows:
  ➤ Industry sector:
    ✧ Data of hard coal and coke oven coke/natural gas and LPG/crude oil/heavy fuel oil before 1974/1981/1965/1969 were distributed to sub-sectors based on relative ratios of fuel consumption data in 1975/1982/1966/1970.
  ➤ Residential and other sectors:
    ✧ Data of heavy fuel oil before 1969 were distributed to sub-sectors based on relative ratios in 1970.
  ➤ Other kerosene and diesel oil:
    ✧ Data of total final consumption before 1969 were distributed to sub-sectors based on relative ratios in 1970.
• See "Assumption for data extrapolation" in this sub-section how to extrapolate the data of IEAWEB to 1950 except for following procedures:
  ➤ Consumption of hard coal, brown coal, patent fuel, coke oven coke, gas works gas, natural gas, and primary solid biofuels in residential sector were extrapolated to 1950 using the Historical Statistics of Japan (Japan Statistical Association, 2006).
  ➤ Consumption of primary solid biofuels in paper, pulp and printing industry before 1981 were extrapolated to 1950 based on trends of production amounts of paper and pulp in Japan (Economy, Trade and Industry Statistics Association, 1998). |

**(d) Republic of Korea**

| Data sources and treatments | ● Data of whole country were taken from IEAWEB during 1971-2015 and extrapolated to 1950. |
|---|---|
| Assumptions for estimating missing historical data | ● Assumptions for modifying IEAWEB during 1971-2015 are as follows:
 ➢ Industry sector:
  ✧ Data of coke oven coke/other kerosene, diesel oil, and heavy fuel oil/natural gas before 2001/1980/1992 were distributed to sub-sectors based on relative ratios of fuel consumption data in 2002/1981/1993.
 ➢ Transport and other sectors:
  ✧ Data of diesel oil and heavy fuel oil before 1980 were distributed to sub-sectors based on relative ratios in 1981.
 ➢ Residential and other sectors:
  ✧ Data of primary solid biofuels before 1989 were distributed to sub-sectors based on relative ratios in 1990.
 ● See "Assumption for data extrapolation" in this sub-section how to extrapolate the data of IEAWEB to 1950. |

**(e) Taiwan**

| Data sources and treatments | ● Data of whole country were taken from IEAWEB during 1971-2015 and extrapolated to 1950. |
|---|---|
| Assumptions for estimating missing historical data | ● Assumptions for modifying IEAWEB during 1971-2015 are as follows:
 ➢ Residential and other sectors:
  ✧ Data of diesel oil/heavy fuel oil before 1979/1981 were distributed to sub-sectors based on relative ratios of fuel consumption data in 1980/1982.
 ● See "Assumption for data extrapolation" in this sub-section how to extrapolate the data of IEAWEB to 1950. |

**(f) Indonesia**

| Data sources and treatments | ● Data of whole country were taken from IEAWEB during 1971-2015 and extrapolated to 1950. |
|---|---|
| Assumptions for estimating missing historical data | ● Assumptions for modifying IEAWEB during 1971-2015 are as follows:
➢ Industry sector:
  ✧ Data of other bituminous coal and sub-bituminous coal before 1999 were distributed to sub-sectors based on relative ratios of consumption data of sub-bituminous coal in 2000.
  ✧ Data of natural gas/diesel oil and heavy fuel oil before 1980/1988 were distributed to sub-sectors based on relative ratios of fuel consumption data in 1981/1989.
  ✧ Fuel consumption data of primary solid biofuels in 1990 were extrapolated to 1971 using trends of primary solid biofuels consumption data in the other sector in IEAWEB during 1971-1990.
➢ Transport, residential and other sectors:
  ✧ Data of heavy fuel oil after 2000 were distributed to sub-sectors based on relative ratios in 1999.
● See "Assumption for data extrapolation" in this sub-section how to extrapolate the data of IEAWEB to 1950. |

**(g) Myanmar**

| Data sources and treatments | ● Data of whole country were taken from IEAWEB during 1971-2015 and extrapolated to 1950. |
|---|---|
| Assumptions for estimating missing historical data | ● Assumptions for modifying IEAWEB during 1971-2015 are as follows:
➢ Industry sector:
  ✧ Data of other bituminous coal/diesel oil before 2010/2011 were distributed to sub-sectors based on relative ratios of fuel consumption data in 2011/2012.
● See "Assumption for data extrapolation" in this sub-section how to extrapolate the data of IEAWEB to 1950. |

**(h) Philippines**

| Data sources and treatments | • Data of whole country were taken from IEAWEB during 1971-2015 and extrapolated to 1950. |
|---|---|
| Assumptions for estimating missing historical data | • Assumptions for modifying IEAWEB during 1971-2015 are as follows:
  ➢ Industry sector:
    ✧ Data of diesel oil and heavy fuel oil before 1979 were distributed to sub-sectors based on relative ratios of fuel consumption data in 1980.
• See "Assumption for data extrapolation" in this sub-section how to extrapolate the data of IEAWEB to 1950. |

**(i) Singapore**

| Data sources and treatments | • Data of whole country were taken from IEAWEB during 1971-2015 and extrapolated to 1950. |
|---|---|
| Assumptions for estimating missing historical data | • Assumptions for modifying IEAWEB during 1971-2015 are as follows:
  ➢ Residential and other sectors:
    ✧ Data of natural gas before 2005 were distributed to sub-sectors based on relative ratios of fuel consumption data in 2006.
• See "Assumption for data extrapolation" in this sub-section how to extrapolate the data of IEAWEB to 1950. |

**(j) Thailand**

| Data sources and treatments | • Data of whole country were taken from IEAWEB during 1971-2015 and extrapolated to 1950. |
|---|---|
| Assumptions for estimating missing historical data | • Assumptions for modifying IEAWEB during 1971-2015 are as follows:
  ➢ Industry sector:
    ✧ Data of other bituminous coal/natural gas before 1988/2001 were distributed to sub-sectors based on relative ratios of fuel consumption data in 1989/2002.
• See "Assumption for data extrapolation" in this sub-section how to extrapolate the data of IEAWEB to 1950. |

**(k) Vietnam**

| Data sources and treatments | • Data of whole country were taken from IEAWEB during 1971-2015 and extrapolated to 1950. |
|---|---|
| Assumptions for estimating missing historical data | • Assumptions for modifying IEAWEB during 1971-2015 are as follows:
  ➢ Industry
    ✧ Data of anthracite, diesel oil and heavy fuel oil during 1980-2009 were distributed to sub-sectors based on relative ratios of corresponding fuel consumption data in 2010.
    ✧ Data of natural gas before 2009 were distributed to sub-sectors based on relative ratios in 2010.
    ✧ Data of other bituminous coal and lignite before 2009 were distributed to sub-sectors based on relative ratios of anthracite consumption data in 2010.
    ✧ Data of other bituminous coal and sub-bituminous coal after 2011 were distributed to sub-sectors based on relative ratios of anthracite consumption data in corresponding years of 2011-2015.
  ➢ Hard coal, diesel oil, and heavy fuel oil
    ✧ Data of total final consumption before 1979 were distributed to sub-sectors based on relative ratios in 1980.
• See "Assumption for data extrapolation" in this sub-section how to extrapolate the data of IEAWEB to 1950. |

**(l) Mongolia**

| Data sources and treatments | • Data of whole country were taken from IEAWEB during 1985-2015 and extrapolated to 1950. |
|---|---|
| Assumptions for estimating missing historical data | • No major modifications were done for IEAWEB during 1985-2015.
• See "Assumption for data extrapolation" in this sub-section how to extrapolate the data of IEAWEB to 1950. |

**(m) Cambodia**

| Data sources and treatments | • Data of whole country were taken from IEAWEB during 1995-2015 and extrapolated to 1950. |
|---|---|
| Assumptions for estimating missing historical data | • No major modifications were done for IEAWEB during 1995-2015.
• See "Assumption for data extrapolation" in this sub-section how to extrapolate the data of IEAWEB to 1950. |

**(n) Hong Kong, Democratic People's Republic of Korea, Brunei, Malaysia, Bangladesh, Nepal, Pakistan, and Sri Lanka**

| Data sources and treatments | • Data of whole country were taken from IEAWEB during 1971-2015 and extrapolated to 1950. |
|---|---|
| Assumptions for estimating missing historical data | • No major modifications were done for IEAWEB during 1995-2015.
• See "Assumption for data extrapolation" in this sub-section how to extrapolate the data of IEAWEB to 1950. |

**(o) Macau, Laos, Afghanistan, Bhutan, and Maldives**

| Data sources and treatments | • Data of whole country were taken from UN data during 1990-2015 and extrapolated to 1950. |
|---|---|
| Assumptions for estimating missing historical data | • No major modifications were done for UN data during 1990-2015.
• Data before 1990 were extrapolated to 1950 using trends of fuel consumption estimated using UN Energy Statistics Database as follows:
Consumption = Production + Import – Export + Changes in stocks
• Biofuel consumption data before 1970 were extrapolated to 1950 using trends of population numbers. |

**Assumption for data extrapolation**

As described above, fuel consumption data before 1959 and 1970 were not included in IEAWEB for Japan and other countries, respectively. The missing historical fuel consumption data were estimated by extrapolation using trends of related data for each sub-sector. Trend factors used in REASv3 are summarized in Table 3.3.

**Table. 3.3.** Trend factors for extrapolating fuel consumption data to 1950 in each sub-sector.

| Sub-sectors | Trend factors and data sources |
|---|---|
| Power plants including energy sector | • Trend factors: Amounts of generated power for all fuel types
• Data sources:
  ➢ Each region of China: China Data Online
  ➢ Other countries and regions: Mitchell (1998) |
| Coke ovens and blast furnace including energy sector | • Trend factors: Amounts of pig iron production for all fuel types
• Data sources: See Sect. S4.1.1 |
| Charcoal production plants | • Trend factors: Amounts of charcoal production for all fuel types
• Data sources: Data after 1961 were obtained from FAOSTAT |

| | |
|---|---|
| | (http://www.fao.org/faostat/en) and trends between 1950 and 1960 were assumed based on Fernandes et al. (2007). |
| Petroleum Refineries including energy sector | • Trend factors: Amounts of total crude oil consumption for all fuel types
• Data sources: Total crude oil consumption was estimated using Mitchell (1998) as follow: Consumption = Production + Import – Export |
| Iron and steel | • Trend factors: Total amounts of pig iron and crude steel production for all fuel types
• Data sources: See Sect. S4.1.1 |
| Non-ferrous metals | • Trend factors: Total amounts of copper, lead, zinc, and primary aluminum production for all fuel types
• Data sources: See Sect. S4.1.2 |
| Non-metallic minerals industry (cement, lime, and brick) | • Trend factors: Amounts of cement production for all fuel types
• Data sources: See Sect. S4.1.3 |
| Railway | • Trend factors: Length of railway line for all fuel types
• Data sources: Mitchell (1998) |
| Road transport | • Trend factors: Total annual mileages of vehicles for each fuel type
• Data sources: See Sect. S6.1.1 |
| Others | • Trend factors and data sources:
  ➤ Coal fuels except for coke fuels: Total coal consumption estimated using Mitchell (1998) as follows: Consumption = Production + Import – Export
  ➤ Coke fuels and gas fuels except for natural gas: The same trends as those for coke ovens
  ➤ Natural gas: Total natural gas consumption estimated using Mitchell (1998)
  ➤ Oil fuels: The same trends as those for petroleum refineries
  ➤ Biofuels: See Sect. S3.1.8
  ➤ Charcoal: The same trends as those for charcoal production plants
  ➤ Other fuels: Fuel consumption data were not extended to 1950. |

**S3.1.3 Regional fuel consumption data in China**

REASv3 used CESY for fuel consumption data of regions in China defined in Table 2.1 except for Hong Kong and Macau. However, in CESY, only total data are available in industry and transport sectors which need to be distributed to sub-sectors. In REASv3, weighting factors for the distribution were prepared for each region. Basic methodology and data used for the weighting factors are described briefly in this sub-section. Note that all motor gasoline listed in both industry and transport sectors of CESY are assumed to be consumed in road transport sector based on IEAWEB.

**Industry sector**

For most regions, total consumption data in industry sector were divided into sub-sectors based on weighting factors prepared using energy data in statistical yearbook of each region. Availabilities of detailed data for the weighting factors are different among regions and summarized in Table 3.4 except for Shanghai, Jiangsu, Zhejiang, Shandong, Hainan and Sichuan where no energy data are available in statistical yearbook of each region.

**Table. 3.4.** Data sources and treatments of weighting factors for each region to distribute total fuel consumption in industry sector to each sub-sector.

| Regions | Data sources and treatments |
|---------|------------------------------|
| Beijing | ● Data of major fuel types were taken from Beijing Statistical Yearbook.
● For the year when statistics are not available, data in 2001/2005/2007/2010/2014 were used before 2000/for 2004/for 2008/for 2011/for 2015. |
| Tianjin | ● Data of major fuel types were taken from Tianjin Statistical Yearbook.
● For the year when statistics are not available, data in 2001/2010/2013 were used before 2000/for 2011/after 2012. |
| Hebei | ● Consumption of main energy sources were taken from Hebei Statistical Yearbook and used for all fuel types.
● For the year when statistics are not available, data in 2005/2010/2013 were used before 2004/for 2011/after 2012. |
| Shanxi | ● Data of coal, coke, and diesel oil were taken from Shanxi Statistical Yearbook. For other fuels, weighting factors were based on data of REASv2 (based on GAINS ASIA at that time). |

| | |
|---|---|
| | • For the year when Shanxi Statistical Yearbook are not available, data in 2000/2010/2013/2014 were used before 1999/for 2011/for 2012/for 2015. For REASv2 (available during 2000-2008), data in 2000/2008 were used before 1999/after 2009. |
| Inner Mongolia | • Data of major fuel types were taken from Inner Mongolia Statistical Yearbook.

• For the year when statistics are not available, data in 2001/2007/2010/2013 were used before 2000/for 2006/for 2011/after 2012. |
| Liaoning | • Data of major fuel types were taken from Liaoning Statistical Yearbook.

• For the year when statistics are not available, data in 2001/2010/2013 were used before 2000/for 2011/after 2012. |
| Jilin | • Data of major fuel types were taken from Jilin Statistical Yearbook.

• For the year when statistics are not available, data in 2000/2002/2005/2010/2013 were used before 1999/for 2001/for 2004/for 2011/after 2012. |
| Heilongjiang | • Data of major fuel types were taken from Heilongjiang Statistical Yearbook.

• For the year when statistics are not available, data in 2005/2010/2013 were used before 2004/for 2011/after 2012. |
| Shanghai | See descriptions below this table. |
| Jiangsu | See descriptions below this table. |
| Zhejiang | See descriptions below this table. |
| Anhui | • Data of major fuel types were taken from Anhui Statistical Yearbook.

• For the year when statistics are not available, data in 2000/2002/2010/2013 were used before 1999/for 2001/for 2011/after 2012. |
| Fujian | • Data of major fuel types were taken from Fujian Statistical Yearbook.

• For the year when statistics are not available, data in 2001/2010/2013 were used before 2000/for 2011/after 2012. |
| Jiangxi | • Data of major fuel types were taken from Jiangxi Statistical Yearbook.

• For the year when statistics are not available, data in 2000/2010/2013 were used before 1999/for 2011/after 2012. |
| Shandong | See descriptions below this table. |

| Henan | • Data of major fuel types were taken from Henan Statistical Yearbook. |
| | • For the year when statistics are not available, data in 2001/2010/2013 were used before 2000/for 2011/after 2012. |
| Hubei | • Data of coal and diesel oil were taken from Hubei Statistical Yearbook. For other fuels, weighting factors were based on data of REASv2 (based on GAINS ASIA at that time). |
| | • For the year when Hubei Statistical Yearbook are not available, data in 2000/2010/2013 were used before 1999/for 2011/after 2012. For REASv2 (available during 2000-2008), data in 2000/2008 were used before 1999/after 2009. |
| Hunan | • Data of major fuel types were taken from Hunan Statistical Yearbook. |
| | • For the year when statistics are not available, data in 2001/2005/2010/2013 were used before 2000/for 2004/for 2011/after 2012. |
| Guangdong | • Data of coal were taken from Guangdong Statistical Yearbook. For other fuels, weighting factors were based on data of REASv2 (based on GAINS ASIA at that time). |
| | • For the year when Guangdong Statistical Yearbook are not available, data in 2000/2010/2013/2014 were used before 1999/for 2011/for 2012/for 2015. For REASv2 (available during 2000-2008), data in 2000/2008 were used before 1999/after 2009. |
| Guangxi | • Data of total energy consumption were taken from Guangxi Statistical Yearbook for all fuel types. |
| | • For the year when statistics are not available, data in 1995/2000/2014 were used before 1997/for 1998 and 1999/for 2015. |
| Hainan | See descriptions below this table. |
| Chongqing | • Data of major fuel types were taken from Chongqing Statistical Yearbook. |
| | • For the year when statistics are not available, data in 2001/2010/2013 were used before 2000/for 2011/after 2012. |
| Sichuan | See descriptions below this table. |
| Guizhou | • Data of major fuel types were taken from Guizhou Statistical Yearbook. |
| | • For the year when statistics are not available, data in 2000/2010/2014 were used before 1999/for 2011/for 2015. |
| Yunnan | • Data of coal, coke, and oil were taken from Yunnan Statistical |

| | |
|---|---|
| | Yearbook. For other fuels, weighting factors were based on data of REASv2 (based on GAINS ASIA at that time).
● For the year when Yunnan Statistical Yearbook are not available, data in 2000/2013 were used before 1999/after 2014. For REASv2 (available during 2000-2008), data in 2000/2008 were used before 1999/after 2009. |
| Tibet | ● Fuel consumption data were not from CESY. (See Sect. S3.1.2) |
| Shaanxi | ● Data of coal, coke, and diesel oil were taken from Shaanxi Statistical Yearbook. For other fuels, weighting factors were based on data of REASv2 (based on GAINS ASIA at that time).
● For the year when Shanxi Statistical Yearbook are not available, data in 2002/2005/2010/2013 were used before 2001/for 2004/for 2009 and 2011/after 2014. For REASv2 (available during 2000-2008), data in 2000/2008 were used before 1999/after 2009. |
| Gansu | ● Data of major fuel types were taken from Gansu Statistical Yearbook.
● For the year when statistics are not available, data in 2001/2010/2013/2014 were used before 2000/for 2011/for 2012/for 2015. |
| Qinghai | ● Data of coal were taken from Qinghai Statistical Yearbook. For other fuels, weighting factors were based on data of REASv2 (based on GAINS ASIA at that time).
● For the year when Qinghai Statistical Yearbook are not available, data in 2001/2010/2013 were used before 2000/for 2011/after 2014. For REASv2 (available during 2000-2008), data in 2000/2008 were used before 1999/after 2009. |
| Ningxia | ● Data of major fuel types were taken from Ningxia Statistical Yearbook.
● For the year when statistics are not available, data in 2000/2010/2013 were used before 1999/for 2011/after 2014. |
| Xinjiang | ● Data of major fuel types were taken from Xinjiang Statistical Yearbook.
● For the year when statistics are not available, data in 2001/2007/2009/2013 were used before 2000/for 2008/for 2010/after 2014. |
| Hong Kong | Fuel consumption data were not from CESY. (See Sect. S3.1.2) |
| Macau | Fuel consumption data were not from CESY. (See Sect. S3.1.2) |

For Shanghai, Jiangsu, Zhejiang, Shandong, Hainan and Sichuan, weighting factors were assumed based on sub-sector level fuel consumption data developed using the China total data described in Sect. S3.1.2 and related regional data as follows:

- Weighting factors to distribute fuel consumption in whole China to each region were prepared for each sub-sector and commonly used for all fuel types. The weighting factors for each sub-sector used in REASv3 are as follows:
  - ➤ Amounts of steel production in each region (see Sect. S4.1.1) were used for iron and steel sub-sector.
  - ➤ Total amounts of copper, lead, zinc, and primary aluminum production in each region (see Sect. S4.1.2) were used for non-ferrous metals sub-sector.
  - ➤ Amounts of cement production in each region (see Sect. S4.1.3) were used for non-metallic minerals sub-sector in IEAWEB. (Fuel consumption in non-metallic minerals were further distributed to cement, lime, and brick sub-sectors in REASv3. See Sect. S3.1.7.)
  - ➤ Amounts of coal production in each region taken from China Data Online were used for coal mines (in energy sector) and mining and quarrying sub-sectors.
  - ➤ Amounts of paper and paperboard production in each region taken from China Data Online were used for paper, pulp and prints sub-sector.
  - ➤ Amounts of textile production in each region (see Sect. S5.1.2) were used for textile and leather sub-sector.
  - ➤ GDP of each region taken from China Data Online were used for other sectors.
- Using the China total data and the weighting factors, the tentative regional fuel consumption data (TRFCD) were developed. Then, the fuel consumption ratio of each sub-sector to industry sector total was calculated for Shanghai, Jiangsu, Zhejiang, Shandong, Hainan and Sichuan using the TRFCD of each region. Finally, fuel consumption in industry sector of each region in CESY was distributed to sub-sectors using the corresponding ratio. When categories of fuel types are different between the TRFCD and CESY, following procedures were adopted:
  - ➤ For raw coal, cleaned coal, and other washed coal in CESY, the ratio for total of anthracite, coking coal and other bituminous coal in the TRFCD were used.
  - ➤ For other coking products and other petroleum products in CESY, the ratio for coke oven coke and heavy fuel oil in the TRFCD were used, respectively.

**Transport sector**

For transport sector, no detailed data are available even in statistical yearbook of each region. Therefore, weighting factors for each region were assumed in the similar procedure for industry sector as follows:

- As mentioned in the first paragraph of this sub-section, all motor gasoline consumption (including those in industry sector) is distributed to road transport sector.
- All solid coal fuels are assumed to be used in railway sector.
- Natural gas consumption before and after 1995 was distributed to pipeline transport and road transport sectors, respectively.
- All heavy fuel oil consumption is distributed to domestic navigation sector.
- For diesel oil, using the same methodology for industry sector, diesel oil consumption data in road transport, railway, and domestic navigation sectors in each region were developed and then, weighting factors were assumed. For regional diesel oil consumption data, those in railway and domestic navigation sectors were taken from REASv2 (based on GAINS ASIA at that time) during 2000-2008 and data in 2000 and 2008 were used before 1999 and 2009, respectively. See Sect. S6.1.2 for diesel oil consumption in each region in road transport sector.
- Consumption of all other fuels is distributed to non-specified transport sector.
- Assumptions of motor gasoline, solid coal fuels, natural gas and heavy fuel oil described above were based on IEAWEB.

**S3.1.4 Regional fuel consumption data in India**

As defined in Table 2.1, REASv3 has 17 sub-regions for India. Therefore, fuel consumption data of country total based on IEAWEB need to be divided for each sub-region. Table 3.5 provides weighting factors used to allocate country total data to the 17 sub-regions.

**Table. 3.5.** Weighting factors for allocating country total fuel consumption data to the 17 sub-regions in India.

| Sectors and fuel types | Weighting factors and data sources |
|---|---|
| Power plants including energy sector | • Weighting factors: Total generation capacities in each region
• Data sources: World Electric Power Plants Database (Platts, 2018) |
| Iron and steel | • Weighting factors: Amounts of crude steel production for all fuel types
• Data sources: See Sect. S4.1.1 |
| Non-ferrous metals | • Weighting factors: Total amounts of copper, lead, zinc, and primary |

| | |
|---|---|
| | aluminum production for all fuel types
● Data sources: See Sect. S4.1.2 |
| Non-metallic minerals industry (cement, lime, and brick) | ● Weighting factors: Amounts of cement production for all fuel types
● Data sources: See Sect. S4.1.3 |
| Road | ● See Sect. S6.1 |
| Rail | ● Weighting factors: Length of railway line open for all fuel types
● Data sources: Factors after 2005 were estimated from TERI (2013, 2018) and those in 2005 were used before 2004. |
| Biofuels | ● See Sect. S3.1.8 |
| Industry and energy sectors (default) | ● Weighting factors and data sources:
➢ Factors for LPG, motor gasoline, kerosene, diesel oil, heavy fuel oil, and naphtha after 1998 were estimated from TERI (2013, 2018) and those in 1998 were used before 1997.
➢ Factors for other fuels after 1999 were estimated from "Fuel Consumed" in Annual Survey of Industries (Ministry of Statistics & Programme Implementation, http://www.csoisw.gov.in/cms/en/1023-annual-survey-of-industries.aspx) and those in 1999 were used before 1998. |
| Residential and other domestic sectors | ● Weighting factors and data sources:
➢ Factors for kerosene and LPG after 1983 were estimated from TERI (2013, 2018) and those in 1983 were used before 1982.
➢ Data of LPG were also used for natural gas and for other fuels, those of kerosene were used. |

**S3.1.5 Regional fuel consumption data in Japan**

REASv3 has 6 sub-regions for Japan as defined in Table 2.1 and the same as the case of India, fuel consumption data of country total based on IEAWEB need to be divided for each sub-region. Table 3.6 provides weighting factors used to allocate country total data to the 6 sub-regions.

**Table 3.6.** Weighting factors for allocating country total fuel consumption data to the 6 sub-regions in Japan.

| Sectors and fuel types | Weighting factors and data sources |
|---|---|
| Power plants including energy sector | • Weighting factors: Total generation capacities in each region
• Data sources: World Electric Power Plants Database (Platts, 2018) |
| Non-ferrous metals | • Weighting factors: Total amounts of copper, lead, zinc, and primary aluminum production for all fuel types
• Data sources: See Sect. S4.1.2 |
| Road | • See Sect. S6.1 |
| Others | • Weighting factors:
  ➢ Factors for each sector and fuel type during 1990-2015 were estimated using energy consumption statistics of each prefecture in corresponding years of 1990-2015.
  ➢ Factors in 1990 were used for those before 1989.
• Data sources:
  ➢ Website of the Agency for National Resources and Energy https://www.enecho.meti.go.jp/statistics/energy_consumption/ec002/results.html (in Japanese) |

**S3.1.6 Fuel consumption in power plants**

**General methodology**

In REASv3, power plants with following criteria were treated as point sources:
- Power plants which were treated as point sources in REASv2 (see Kurokawa et al., 2013).
- Power plants which entered commercial operation after 2008 and whose total generating capacities of units in each power plant were larger than 300MW.

Then, fuel consumption in power plants sector was estimated as follows:

1) Fuel consumption in each power plant (point source) was estimated. (see "Fuel consumption in each power plant" below)

2) (A) Total of the fuel consumption in each power plant was calculated in each country and region.

3) If (A) was larger than (B) fuel consumption in total power plant sector in a corresponding country and region, data of each power plants prepared in 1) were adjusted by the ratio of (B) to (A). In this case, fuel consumption of power plants as area sources was assumed to be zero.

4) IF (A) was smaller than (B), the value of (B) minus (A) was assumed to be fuel consumption in area sources. In this case, there is no change for the data of each power plant developed in 1).

**Fuel consumption in each power plant**

In REASv2, power plants whose annual $CO_2$ emissions in the Carbon Monitoring for Action (CARMA) Database (Wheeler and Ummel, 2008) were more than 1 Mt in 2000 and/or 2007 were treated as point sources. Before 2007, REASv3 used the same power plants as point sources with some revisions for such as generation capacities, fuel types, etc. using the updated World Electric Power Plants Database (Platts, 2018). For fuel consumption, data between 2000 and 2007 were basically the same as those in REASv2. Before 2000, fuel consumption of each power plant in operation was assumed to be the same as that in 2000 which will be adjusted based on total fuel consumption in power plants sector as described in "General methodology" above. (Note that power plants which were constructed and retired before 2000 were not considered in REASv3.) After 2008, REASv3 included power plants which entered commercial operation after 2008 as new point sources based on the WEPP (see also "General methodology" above). Although major information was available including fuel types used in each power plant, there are no data of fuel consumption in the WEPP. Thus, in REASv3, annual fuel consumption per generation capacity for each fuel type was estimated first using data in 2000 and 2007 for each country. The data were estimated for power plants which started operation before 1999 and after 2000, separately. Then, using the generation capacities data obtained from the WEPP, fuel consumption in each power plant was estimated.

**S3.1.7 Fuel consumption in non-metallic minerals**

REASv3 defined cement, lime, brick, and non-specified sub-sectors in the non-metallic minerals category in stationary combustion sources. However, energy statistics used in REASv3 including IEAWEB and regional statistical yearbook of China provide fuel consumption in total non-metallic minerals industry which needs to be distributed to each sub-sector.

In REASv3, all primary coal fuels were assumed to be used in cement, lime, and brick production. For China, Hua et al. (2016), Wang et al. (2012), and Streets et al. (2006) give coal consumption in cement (1980-2012), brick (1950-2015), and lime (2001) industries, respectively. Using these data and production amounts of cement, lime and brick, coal consumption per unit of production of cement, lime, and brick was estimated, respectively. Then, coal consumption data in non-metallic minerals in each region were distributed to each sub-sector based on production amounts of cement, lime, and brick in each region and corresponding coal consumption per united of production. Similarly, Maithel (2013) provides coal consumption in cement and brick industries in Pakistan

during 2001-2010 and with production amounts of cement and brick, fuel consumption in non-metallic minerals industry were distributed to each sub-sector. For other countries, due to lack of information, averaged coal consumption per unit of production of cement, lime, and brick for China was used for other East and Southeast Asian countries. For other countries in South Asia, averaged coal consumption per unit of production of cement and brick for Pakistan and that of lime for China was used. Then, with production data of cement, lime, and brick, fuel consumption in non-metallic minerals were distributed to each sub-sector. See Sects. S4.1.3, S4.1.4, and S4.1.5 for production data of cement, lime, and brick, respectively.

For other fuels, in REASv3, coke oven coke and heavy fuel oil were assumed to be used in cement industry and others including gas fuels and diesel oil were allocated to the non-specified sub-sector.

**S3.1.8 Biofuels**

**China**

CESY provides biofuel consumption data of fuelwood, crop residue, and biogas in each region during 1998-2007 which were used in REASv3. Before 1997, data were extended to 1980 using trends of each fuel consumption data in REASv1 and then extended to 1950 based on trends of biofuel consumption in East Asia obtained from Fernandes et al. (2007). After 2007, fuelwood, crop residue, and biogas consumption in total China were extrapolated to 2015 using trends of primary solid biofuels consumption in IEAWEB. Then, consumption of each fuel in each region in 2007 were tentatively extrapolated to 2015 using trends of rural population numbers in each region. Finally, fuelwood, crop residue, and biogas consumption in total China estimated during 2008-2015 were distributed to each region using the tentatively extrapolated data in each region.

**India**

Primary solid biofuels in IEAWEB were assumed to be total of fuelwood, crop residue and animal waste in India during 1971-2015. Before 1970, the primary solid biofuels consumption was extrapolated to 1950 using trends of biofuel consumption in South Asia obtained from Fernandes et al. (2007). Then, relative ratios of fuelwood, crop residue, and animal waste consumption in 17 sub-regions to consumption of the primary solid biofuels in total India were calculated for 1990 and 2010 using data in Streets and Waldhoff (1998) and Census of India 2011 (Chandramouli, 2011), respectively and interpolated between 1991 and 2009. Before 1989 and after 2011, the ratios of 1990 and 2010 were assumed to be constant, respectively. Finally, fuel consumption of fuelwood, crop residue, and animal waste in each sub-region during 1950-2015 were calculated.

**Japan**

Primary solid biofuels consumption in IEAWEB were assumed to be fuelwood consumption in Japan during 1982-2015. Before 1981, as described in Sect. S3.1.2, fuel consumption in residential and paper, pulp and printing industry sectors was extrapolated to 1950 using the Historical Statistics of Japan (Japan Statistical Association, 2006) and trends of production amounts of paper and pulp in Japan, respectively.

**Macau, Laos, Afghanistan, Bhutan, and Maldives**

See Sect. S3.1.2 for methodology and data sources. Only fuelwood and charcoal were included for this group.

**Other countries**

Primary solid biofuels data in IEAWEB were assumed to be total of fuelwood, crop residue and animal waste consumption in each country and extrapolated to 1950 using trends of biofuel consumption in East or Southeast or South Asia obtained from Fernandes et al. (2007). For distribution to each fuel type, consumption ratios of fuelwood, crop residue, and animal waste in 1990 obtained from Streets and Waldhoff (1998) were used during 1950-2015.

**S3.2 Emission factors and settings of emission controls**

**S3.2.1 SO$_2$**

**Sulfur contents in fuels**

In REASv3, default settings were taken from those of REASv1 during 1980-2000 generally based on RAINS ASIA at that time, Streets et al. (2000), Kato and Akimoto (1992) and Kato et al. (1991). For countries using default settings, data in 1980 and 2000 were used before 1979 and after 2001, respectively. For China, India, Japan, Republic of Korea, and Taiwan, additional country-specific settings were considered as described in Table 3.7.

**Table 3.7.** Settings and assumptions of sulfur contents in fuels for China, India, Japan, Republic of Korea, and Taiwan.

| Countries | Settings and assumptions |
|---|---|
| China | • Coal:
  ➢ During 1985-2000: Data were taken from REASv1 based on Kato and Akimoto (1992) in 1985 and China Coal Industry Yearbook 2002 (State Administration for Coal Safety, 2003) in 1990 and 1995. In 2000, data in 1995 were adjusted so that the national average sulfur contents were 1.08% after Lu et al. (2010). Data in other years were interpolated.
  ➢ During 2001-2005: Data were taken from REASv2 where settings of power plants in 2005 were based on Zhao et al. (2008) and national average sulfur contents were adjusted to 1.02% after Lu et al. (2010). Data between 2000 and 2005 were interpolated.
  ➢ Before 1984 and after 2006, settings in 1980 and 2005 were used, respectively.
• Oil
  ➢ Before 1985, data were obtained from Kato et al. (1991) and those in 1995 were based on information from Tsinghua University (1.5% for heavy fuel oil and 0.58%, 0.35%, and 0.163% for diesel oil in north, northeast, and other areas, respectively) for REASv1. Data between 1986-1994 were interpolated and after 1996, data in 1995 were used. |
| India | • Data were taken from REASv1 based on Reddy and Venkataraman (2002) for coal, heavy fuel oil, and light fuels and Kato et al. (1991) for others. The same data were used for the entire target period of REASv3. |
| Japan | • Coal: Data during 1960-1996 were taken from Li and Dai (2000). The value in 1960 was 1.06% and gradually decreased to 0.60% in 1996. It was assumed that the value was reduced by 10% from 1996 to 2010 referring a report of MOEJ (2012). Data between 1996 and 2010 were interpolated and those in 1960 and 2010 were used before 1959 and after 2011, respectively.
• Heavy fuel oil and crude oil: Settings during 1965-2010 for power plants were based on Iwaya (2013). Those for industry were based on Kato et al. (1991), Streets et al. (2000), and Imura et al. (1999). Data |

| | in 1965 and 2010 were used before 1964 and after 2011, respectively. |
| | |
| | ➢ Heavy fuel oil for power plants: The values before 1965 were 2.6% and decreased almost constantly to 0.80% in 1975. Then the values were gradually decreased to 0.75% in 1990 and the values was used after 1990. |
| | ➢ Heavy fuel oil for industry: The values before 1965 were 2.60% and assumed to be decreased gradually to 1.4% in 1975, 1.1% in 1985, and 1.0% in 2000. The values after 2000 were assumed to be constant. |
| | ➢ Crude oil for power plants: The value before 1966 were 2.8% and decreased almost linearly to 0.20% in 1975. After 1975, values were between 0.15% and 0.20%. |
| | ● Diesel: Settings were based on regulations of diesel oil in Japan as follows: 1.2% before 1975, 0.50% during 1976-1991, 0.20% during 1992-1996, 0.05% during 1997-2003, and 0.01% after 2004. |
| Republic of Korea and Taiwan | ● Data during 1980-2000 were taken from REASv1 based on Kato et al. (1991), RAINS ASIA, and Streets et a. (2000) and those in 1975 were obtained from Kato et al. (1991). Data between 1976-1981 were interpolated and those in 1975 and 2000 were used before 1974 and after 2001, respectively. |

**Emission factors**

SO$_2$ emissions from coal and oil fuels were calculated using sulfur contents in fuels and ratios of sulfur emitted as SO$_2$. Settings of REASv3 were taken from REASv1 and REASv2 based on Kato and Akimoto (1992), Kato et al. (1991) and RAINS ASIA as follows:

- Power plants (point sources): 0.95
- Power plants (area sources)): 0.90 for Japan, Republic of Korea, and Taiwan; 0.775 for other countries and regions.
- Industry sector: 0.775
- Coke ovens: 0.0685
- Iron and steel: 0.1483
- Transport sector: 0.775
- Domestic sector: 0.60
- Coke oven coke for all sectors: 0.885
- Oil fuels for all sectors: 1.0

For coke ovens, activity data are coal input and it is considered that the estimated $SO_2$ emissions include both combustion and non-combustion sources.

For gas fuels such as coke oven gas and blast furnace gas, light fuels such as LPG, and other fuels except for primary biofuels such as charcoal and municipal wastes, emission factors were derived from Kato and Akimoto (1991). Those for fuelwood and crop residue were taken from Garg et al. (2001) and those for animal waste were from Gadi et al. (2003).

In cement plants, effects of absorption of $SO_2$ by cements need to be considered. In REASv3, the absorption rates for China were obtained from Li et al. (2017) and those for other countries were based on Kato et al. (1991).

**Settings of emission controls**

Settings and assumptions for reduction of $SO_2$ emissions from combustion sources by abatement equipment adopted in REASv3 are summarized in Table 3.8. For other sources not described in Table 3.8, no emission controls were considered.

**Table 3.8.** Settings and assumptions of emission controls of $SO_2$

| Countries | Settings and assumption |
|---|---|
| China | • Power plants: Effects of flue-gas desulfurization (FGD) were considered after 2000 as follows:
 ➢ Settings during 2000-2008 were taken from REASv2 based on national introduction rates of FGD from Lu et al. (2010) and those of each province from Zhao et al. (2008).
 ➢ After 2008, increases of penetration of FGD were assumed referring Liu et al. (2015) and Li et al. (2017). In 2015, the introduction rates were assumed to be 100% in power plants considered as point sources and 90% for other power plants.
 • Industry: Effects of FGD were roughly assumed as follows:
 ➢ Referring Li et al. (2017), it was assumed that regulations started from (A) Beijing and Shanghai, then (B) Shandong, Hebei, and Guangdong, and finally (C) other provinces.
 ➢ Regulations of industrial boiler were strengthened after 2014 referring Zheng et al. (2018).
 ➢ For (A), it was assumed that introduction of FGD started from 2000 and penetration rates in 2010 were 40% which is a setting for China in 2020 in Business-as-usual scenario of Wang et al. |

| | |
|---|---|
| | (2014). For the penetration rates, linear trends were assumed during 2000-2013.

➢ For (B) and (C), it was assumed that penetration of FGD started 2 and 4 years after (A), respectively and reduction effects were assumed to be smaller than (A) by 10% and 15%, respectively.

➢ In 2015, reduction rates of $SO_2$ emissions were assumed to be 75%, 63%, and 52% for (A), (B), and (C), respectively. |
| Japan | ● Power plants: Referring MRI (2015), Kato et al. (1991), and MOEJ (2000), effects of FGD were considered after 1968 as follows:

➢ In 1990 and after 2000, introduction rates of FGD in power plants as point sources were assumed to be 95% and 100%, respectively. Trends of the introduction rates during 1968 and 1990 were assumed based on MOEJ (2000) and those between 1990 and 2000 were interpolated.

➢ For introduction rates of FGD in power plants as area sources, it was assumed to be 95% after 2000 and the trends before 1990 were estimated based on those of point sources.

● Other sectors: Referring Kato et al. (1991), reduction rates of $SO_2$ emissions were assumed as follows:

➢ For large industries including sulphuric acid plants, 80% of reduction rates of power plants as area sources were adopted.

➢ For other industries, reduction rates were assumed to be 50% of large industries.

➢ For commercial and public services, 50% of reduction rates of other industries were adopted. |
| Republic of Korea | ● Effects of FGD were roughly assumed as follows:

➢ Power plants: Referring Ebata et al. (1997) and Wang et al. (2014), it is assumed that introduction of FGD was from 1990. The penetration rates in power plants as point sources in 2000, 2005, and 2010 were 90%, 97%, and 98%, respectively. Data between 1990, 2000, 2005, and 2010 were interpolated and data in 2010 were used after 2011. Effects of FGD on power plants as area sources were assumed to be 5% lower than point sources.

➢ Industry: It was assumed that introduction of FGD started from 1990 and penetration rates of FGD were 80% and 85% in 2005 and 2010, respectively based on Wang et al. (2014). Data between |

| | |
|---|---|
| | 1990, 2005, and 2010 were interpolated and data in 2010 were used after 2011. |
| Taiwan | ● Effects of FGD were roughly assumed as follows:

➢ Power plants: Due to lack of information, the same reduction rates of Republic of Korea were adopted after 1995. But according to Ebata et al. (1997), introduction of FGD started earlier than Republic of Korea. It was assumed that penetration rates in 10% and 30% in 1980 and 1990, respectively and data between 1980, 1990, and 1995 were interpolated.

➢ Industry: Similar to power plants, the same reduction rates of Republic of Korea were adopted after 2000 and it was assumed that introduction of FGD started from 1985. Data between 1985 and 2000 were interpolated. |
| Thailand | ● Effects of FGD were assumed as follows:

➢ Power plants as point sources: Referring UN Environment (2018), reduction rates were assumed for four power plants as follows: Mae Moh (0.8-0.97 in 1978-2015), BLCP Power (0.84 from 2006), National Power Supply (0.75 from 1999), and GHECO-One (0.952 from 2012). |
| Other countries | ● Effects of FGD were assumed as follows:

➢ Power plants as point sources: Reduction rets (0.7-0.9) were assumed if units have information of installed FGD equipment in World Electric Power Plants Database (Platts, 2018).

➢ Countries which have power plants with FGD and number of such power plants in 2015 (in parentheses) in REASv3 were as follows: India (10), Indonesia (5), Laos (1), Malaysia (4), Vietnam (10), and Sri Lanka (2). |

**S3.2.2 NO$_x$**

**Default emission factors**

Table 3.9 summarized default emission factors used in REASv3 for fuel combustion in power plants, industry and residential sectors. Specific settings for coke ovens, iron and steel industry, cement industry, and emission controls were described below the table.

**Table 3.9.** Default emission factors of $NO_x$ from fuel combustion in power plants, industry and residential sectors. Unit is t/PJ expressed as $NO_2$.

| Fuel type | Power plants | Industry | Residential |
|---|---|---|---|
| Hard coal[h] | 345[a] | 260[e] | 78[g] |
| Raw coal[i] | See Table 3.10. | 203[f] | 61.1[g] |
| Cleaned coal[i] | | 162[f] | 48.5[g] |
| Other washed coal[i] | | 509[f] | 153[g] |
| Sub-bituminous coal | 524[a] | A | B |
| Lignite | 433[a] | A | B |
| Coke oven coke[j] | 345 | 260 | 78 |
| Natural gas | 105[b] | 53[b] | 37[b] |
| Gas works gas | 10.5[b] | 7.4[b] | 5.25[b] |
| Coke oven gas | 77.8[b] | 55[b] | 38[b] |
| Blast furnace gas | 10.5[b] | 7.4[b] | 38[b] |
| LPG | 79[b] | 56[b] | 33[b] |
| Kerosene | 485[b] | 167[b] | 25[b] |
| Diesel oil | 632[b] | 222[b] | 74[b] |
| Crude oil | 249[b] | 145[b] | 49[b] |
| Heavy fuel oil | 249[b] | 145[b] | 49[b] |
| Fuelwood | 45[c] | | |
| Crop residue | 91.1[c] | | |
| Animal waste | 91.1[c] | | |
| Charcoal | 100[d] | | |

a. AP-42 (US EPA, 1995). b. Kato and Akimoto (1992). c. Streets and Waldhoff (1998), d. Revised 1996 IPCC guidelines (IPCC, 1997). e. Estimated based on ratios of emission factors between power plants and industry in Kato and Akimoto (1992). f. Estimated referring Zhang et al. (2007). g. 30% of emission factors of industry were adopted based on Kato and Akimoto (1992). h. Emission factors were commonly used for coking coal, anthracite and bituminous coal. i. Only defined for China. j. Emission factors for hard coal were adopted. A. Estimated based on ratios of emission factors between power plants and industry in Kato and Akimoto (1992) considering differences of net calorific values. B. 30% of emission factors of industry were adopted.

**Coke ovens**

For coal input to coke ovens, emission factor was 1.0 t/kt taken from Kato and Akimoto (1992). It is considered that $NO_x$ emissions estimated using this emission factor include contributions from both combustion and non-combustion processes.

**Iron and steel industry**

In iron and steel industry, emission factors for cokes, coke oven gas, and blast furnace gas were taken from Kato and Akimoto (1992) as follows:

- Coke oven coke: 4.0 t/kt for China and 2.5 t/kt for other countries
- Coke oven gas: 141 t/PJ
- Blast furnace gas: 76.4 t/PJ

For other fuel types, default emission factors were used.

**Cement industry**

For China, emission factors of coal combustion in each cement kiln type were obtained from Lei et al. (2011) as follows: 15.3 t/kt for precalciner kilns, 18.5 t/kt for other rotary kilns, and 1.7 t/kt for shaft kilns. Coal consumption in each cement kiln type were estimated based on Lei et al. (2011) and Hua et al. (2016). For other fuel types, default emission factors in industry were used.

For Japan, $NO_x$ emissions were not estimated based on fuel consumption, but using amount of cement production in each kiln type. Emission factors (t/kt of clinker produced) were taken from AP-42 (US EPA, 1995) as follows: 3.7 for wet process kilns, 3.0 for long dry process kilns, 2.4 for preheater process kilns and 2.1 for preheater/precalciner kilns. Ratio of clinker to cement was assumed to be 0.85 based on Cement handbook (Japan Cement Association, 2019). (See Sect. S4.1.3 for production data by different kiln types.)

For other countries and regions, default emission factors in industry were used for all fuel types.

**Settings of emission controls**

Settings and assumptions for reduction of $NO_x$ emissions from combustion sources by abatement equipment adopted in REASv3 are summarized in Table 3.10. For other sources not described in Table 3.10, no emission controls were considered.

**Table 3.10.** Settings and assumptions of emission controls of NO$_x$

| Countries | Settings and assumption |
|---|---|
| China | ● Power plants
➢ Referring Zhang et al. (2007) and Liu et al. (2015), emission factors [t/PJ] for coal fired power plants were assumed considering effects of low-NO$_x$ burner based on capacity and years as follows:
  ✧ 227: Larger than 300 MW or equal to 300 MW after 1995
  ✧ 300: Smaller than 300 MW but equal to or larger than 100 MW after 1997.
  ✧ 393: Equal to 300 MW before 1995 or Smaller than 300 MW but equal to or larger than 100 MW before 1997.
  ✧ 360: Less than 100 MW.
  ✧ 300: Power plants as area sources (no information of capacity) before 2000. The values were assumed to be decreased by 10% until 2010 and by 15% until 2015.
➢ Penetration rates of selective catalytic reduction (SCR: efficiency 73%) and selective non-catalytic reduction (SNCR: efficiency 30%) for each province in 2011 were taken from Chen et al. (2014). Referring Chen et al. (2014), Li et al. (2017), and Zheng et al. (2018), national introduction rates were assumed to be 12%, 18%, and 75% in 2010, 2011, and 2015 and reduction rates for as point sources were estimated. For area sources, 50% of reduction rates of point sources were adopted.
● Industry
➢ Referring Li et al. (2017), effects of De-NO$_x$ system were considered for precalciner kilns in cement plants and penetration rates were roughly assumed to be 0% in 2010, 50% in 2014 and 90% in 2015.0 |
| Japan | ● Power plants: Referring MRI (2015), JMF and ICETT (2003), and MOEJ (2000), effects of low-NO$_x$ burner and SCR were considered as follows:
➢ Effects of low-NO$_x$ burner were considered after 1970 and reduction efficiencies were assumed to be 15%, 35%, and 50% in 1975, 1980, and after 2005, respectively. Data between 1970, 1975, 1980, and 2005 were interpolated. |

| | |
|---|---|
| | <li>Effects of SCR were considered after 1974 and introduction rates in coal, oil, and gas power plants as point sources were assumed to be 80%, 40%, and 72% in 2002 and 90%, 45%, and 80% after 2010, respectively. Trends of the introduction rates during 1974 -2002 were assumed based on MOEJ (2000) and reduction rates during 2002-2010 were interpolated. For power plants as area sources, reduction rates were assumed to be 85% of point sources.</li><li>Industry: Effects of low-$NO_x$ burner and SCR were roughly assumed referring MRI (2015) and Kato et al. (1991) as follows:</li><li>It was assumed that trends of introduction rates of low-$NO_x$ burner were the same as for those of power plants, but reduction efficiencies were 50% of those for power plants as point sources.</li><li>For large industries such as cement, iron and steel, it was assumed that trends of penetration rates of SCR were the same as those of power plants, but reduction efficiencies were 50% of those for power plants as point sources. For other industries, reduction rates were assumed to be 50% of those for large industries.</li> |
| Republic of Korea/Taiwan | <li>For power plants, introduction rates of low-$NO_x$ burner were 84% and 86% in 2005 and 2010, respectively and those of SCR (SNCR) were 56% (5%) and 68% (5%) in 2005, and 2010, respectively based on Wang et al. (2014). It was roughly assumed that low-$NO_x$ burner, SCR, and SNCR were installed from 1990 and their penetration rates in 2015 were 90%, 73%, and 5%, respectively. Reduction rates between 1990, 2005, 2010, and 2015 were interpolated.</li><li>Due to lack of information, the same settings for Republic ok Korea were adopted to Taiwan.</li> |
| Others | <li>Effects of low-$NO_x$ burner and De-$NO_x$ system were assumed as follows:</li><li>Power plants as point sources: Reduction rets (0.7-0.9) were assumed if units have information of installed FGD equipment in World Electric Power Plants Database (Platts, 2018).</li><li>Countries which have power plants with De-$NO_x$ equipment and number of such power plants in 2015 (in parentheses) in REASv3 were as follows: India (11), Indonesia (5), Malaysia (6), Philippines (4), Singapore (4), Thailand (9), Vietnam (4), Pakistan (1), and Sri Lanka (2).</li> |

**S3.2.3 CO**

**Default emission factors**

Table 3.11 summarized default emission factors used in REASv3 for fuel combustion in power plants, industry and residential sectors. Specific settings for coal combustion and, iron and steel industry, cement and other non-metallic minerals industries were described below the table.

**Table 3.11.** Default emission factors of CO from fuel combustion in power plants, industry and residential sectors. Unit is t/PJ.

| Fuel type | Power plants | Industry | Residential |
|---|---|---|---|
| Hard coal[e] | 20[a] | See "Emission factors for coal combustion" below. | |
| Raw coal[f] | 20[a] | | |
| Cleaned coal[f] | 20[a] | | |
| Other washed coal[f] | 20[a] | | |
| Sub-bituminous coal | 20[a] | | |
| Lignite | 20[a] | | |
| Coke oven coke | 20[a] | 150[a] | 2000[a] |
| Natural gas | 20[a] | 30[a] | 50[a] |
| Gas works gas | 20[a] | 150[a] | 150[a] |
| Coke oven gas | 20[a] | 150[a] | 150[a] |
| Blast furnace gas | 20[a] | 150[a] | 150[a] |
| LPG | 15[a] | 10[a] | 326[a] |
| Kerosene | 15[a] | 15[a] | 179[a] |
| Diesel oil | 15[a] | 15[a] | 20[a] |
| Crude oil | 15[a] | 15[a] | 20[a] |
| Heavy fuel oil | 15[a] | 15[a] | 20[a] |
| Fuelwood | 255.5[b] | 2555[c] | 5110[d] |
| Crop residue | 354.5[b] | 3545[c] | 7090[d] |
| Animal waste | 330[b] | 3300[c] | 6600[d] |
| Charcoal | 400[b] | 4000[a] | 7000[a] |

a. The global atmospheric pollution forum air pollutant emission inventory manual (SEI, 2012). b. Emission factors of power plants were assumed to be 10% of industry sector. c. Emission factors of industry sector were assumed to be 50% of residential sector. d. Streets and Waldhoff (1999). e. Emission factors were commonly used for coking coal, anthracite and bituminous coal. f. Only defined for China.

**Emission factors for coal combustion**

(a) Industry sector except for cement and other non-metallic minerals industries

Due to lack of information of detailed boiler and furnace types in industry sub-sectors in each country, CO emission factors of industry sector were roughly assumed in REASv3 as follows:

- 5.75 t/kt: average of emission factors for fluidized bed furnace and automatic stoker boiler based on AP-42 (US EPA, 1995).
  - Default emission factors for Japan, Republic of Korea, and Taiwan
  - Emission factors for large industries in China
- 18.6 t/kt: Emission factors for other industries in China estimated referring Streets et al. (2006) and data for fluidized bed furnace, automatic stoker, and hand-feed stoker in AP-42 (US EPA, 1995).
- 8.5 t/kt: Emission factors based on automatic stoker in AP-42 (UE EPA, 1995) were adopted for large industries in other countries.
- 66.25 t/kt: Emission factors based on average of automatic stoker and hand-feed stoker in AP-42 (UE UPA, 1995) for other industries in other countries.
- It was assumed that emission factors in China were decreased by 25% from 2000 to 2015 linearly assuming improvement in combustion efficiency.

(b) Residential sector

Emission factors for China, India, and other countries were assumed as follows:

- 75 t/kt for China obtained from Streets et al. (2006) for stove in residential sector.
- 275 t/kt for India taken from Pandey et al. (2014) for traditional stove in residential sector.
- 2.61 kt/PJ for other countries as default emission factor derived from the global atmospheric pollution forum air pollutant emission inventory manual (SEI, 2012)

**Coke production and iron and steel industry**

In REASv3, CO emissions from coke production and iron and steel industry were also estimated using production amounts of coke oven coke, sinter, pig iron, and crude steel (see Sects. S4.2.1 and S4.2.8). CO emission factors for coal consumption in coke ovens, those for coal and coke fuels in blast furnace, and coke furls and gas fuels in iron and industry sectors were assumed to be zero assuming their contributions were included in the emissions estimated based on production amounts described in Sects S4.2.1 and S4.2.8. These mean that CO emissions from combustion sources in coke production and iron and steel industry were not estimated separately in REASv3.

**Cement industries**

For China, emission factors of coal combustion in each cement kiln type were obtained from Lei et al. (2011) as follows: 17.8 t/kt for precalciner kilns, 17.8 t/kt for other rotary kilns, and 155.7 t/kt for shaft kilns. Coal consumption in each cement kiln type were estimated based on Lei et al. (2011) and Hua et al. (2016). For other fuel types, default emission factors in industry were used.

For Japan, CO emissions were not estimated based on fuel consumption, but using amount of cement production in each kiln type. Emission factors (t/kt of clinker produced) were taken from AP-42 (US EPA, 1995) as follows: 0.06 for wet process kilns, 0.11 for long dry process kilns, 0.49 for preheater process kilns and 1.8 for preheater/precalciner kilns. Ratio of clinker to cement was assumed to be 0.85 based on Cement handbook (Japan Cement Association, 2019). (See Sect. S4.1.3 for production data by different kiln types.)

For other countries and regions, 63.8 t/kt were used for emission factors for coal consumption in cement industry based on average of emission factors for precalciner kilns, other rotary kilns, and shaft kilns taken from AP-42 (US EPA, 1995). For other fuel types, default emission factors in industry were used.

**Other non-metallic minerals industries**

For lime industry, 155.7 t/kt were commonly used for coal combustion in all countries and default emission factors were used for other fuel types. For brick industry, 150 t/kt were used for coal combustion in China and default emission factors were adopted for Japan, Republic of Korea, and Taiwan. For other countries, emissions from brick industry were not estimated based on fuel combustion, but using amount of brick production. Emission factor 2.0 t/kt of brick produced was assumed based on Weyant et al. (2014) (See Sect. S4.2.5). For other sources, default emission factors were used.

**S3.2.4 PM species**

**Default emission factors**

Tables 3.12-14 summarized default emission factors of $PM_{10}$, $PM_{2.5}$, BC, and OC used in REASv3 for fuel combustion in power plants, industry and residential sectors (Note that emissions of PM species from gas fuels were neglected in REASv3). Specific settings for biofuels, iron and steel industry, cement and other non-metallic minerals industries were described below the table.

**Table 3.12.** Default emission factors of $PM_{10}$, $PM_{2.5}$, BC, and OC from fuel combustion in power plants. Unit is t/kt.

| Fuel type | $PM_{10}$ | $PM_{2.5}$ | BC | OC |
|---|---|---|---|---|
| Hard coal[f] | 12.0[a] | 5.08[c] | 0.072[a] | 0.0[a] |
| Raw coal[g] | 46.0[b] | 12.0[b] | 0.024[b] | 0.0[b] |
| Cleaned coal[g] | 46.0[b] | 12.0[b] | 0.024[b] | 0.0[b] |
| Other washed coal[g] | 46.0[b] | 12.0[b] | 0.024[b] | 0.0[b] |
| Sub-bituminous coal | 29.0[a] | 9.3[c] | 0.174[a] | 0.0[a] |
| Lignite | 29.0[a] | 9.3[c] | 0.174[a] | 0.0[a] |
| Coke oven coke[h] | 12.0 | 5.08 | 0.072 | 0.0 |
| Diesel oil | 0.49[a] | 0.186[d] | 0.147[a] | 0.0441[a] |
| Crude oil[i] | 1.1 | 0.775 | 0.088 | 0.033 |
| Heavy fuel oil | 1.1[a] | 0.775[d] | 0.088[a] | 0.033[a] |
| Fuelwood | 2.2[e] | 1.79[e] | 0.11[e] | 0.44[e] |
| Crop residue[j] | 2.2 | 1.79 | 0.11 | 0.44 |
| Animal waste[j] | 2.2 | 1.79 | 0.11 | 0.44 |
| Charcoal | 4.1[e] | 3.32[e] | 0.205[e] | 0.82[e] |

a. Bond et al. (2004). b. Lei et al. (2011). c. $PM_{2.5}/PM_{10}$ ratios were estimated based on AP-42 (US UPA, 1995). d. $PM_{2.5}/PM_{10}$ ratios were estimated based on Klimont et al. (2002b). e. Emission factors of $PM_{10}$, BC, and OC for fuelwood and charcoal were taken from Bond et al. (2004). $PM_{2.5}/PM_{10}$ ratios were estimated based on the global atmospheric pollution forum air pollutant emission inventory manual (SEI, 2012). f. Emission factors were commonly used for coking coal, anthracite and bituminous coal. g. Only defined for China. h. Emission factors for hard coal were adopted. i. Emission factors for heavy fuel oil were adopted. j. Emission factors for fuelwood were adopted.

**Table 3.13.** Default emission factors of $PM_{10}$, $PM_{2.5}$, BC, and OC from fuel combustion in industry sector. Unit is t/kt.

| Fuel type | $PM_{10}$ | $PM_{2.5}$ | BC | OC |
|---|---|---|---|---|
| Hard coal[f] | 4.2[a] | 1.79[c] | 0.84[a] | 0.168[a] |
| Raw coal[g] | 7.21[b] | 2.17[b] | 0.412[b] | 0.0868[b] |
| Cleaned coal[g] | 7.21[b] | 2.17[b] | 0.412[b] | 0.0868[b] |
| Other washed coal[g] | 7.21[b] | 2.17[b] | 0.412[b] | 0.0868[b] |
| Sub-bituminous coal | 17.0[a] | 7.23[c] | 0.85[a] | 1.7[c] |
| Lignite | 17.0[a] | 7.23[c] | 0.85[a] | 1.7[c] |
| Coke oven coke[h] | 4.2 | 1.79 | 0.84 | 0.168 |
| Kerosene | 0.9[a] | 0.341[d] | 0.117[a] | 0.09[a] |
| Diesel oil | 0.49[a] | 0.186[d] | 0.147[a] | 0.0441[a] |
| Crude oil[i] | 1.1 | 0.775 | 0.088 | 0.033 |
| Heavy fuel oil | 1.1[a] | 0.775[d] | 0.088[a] | 0.033[a] |
| Fuelwood | 6.1[e] | 4.95[e] | 0.555[e] | 3.22[e] |
| Crop residue[j] | 6.1 | 4.95 | 0.555 | 3.22 |
| Animal waste[j] | 6.1 | 4.95 | 0.555 | 3.22 |
| Charcoal | 4.1[e] | 3.32[e] | 0.205[e] | 0.82[e] |

a. Bond et al. (2004). b. Estimated based on Lei et al. (2011) and Streets et al. (2006). c. $PM_{2.5}/PM_{10}$ ratio was estimated based on the global atmospheric pollution forum air pollutant emission inventory manual (SEI, 2012). OC/BC ratio was assumed based on ABC Emission Inventory Manual (Shrestha et al., 2013). d. $PM_{2.5}/PM_{10}$ ratios were estimated based on Klimont et al. (2002b). e. Emission factors of $PM_{10}$, BC, and OC for fuelwood and charcoal were taken from Bond et al. (2004). $PM_{2.5}/PM_{10}$ ratios were estimated based on the global atmospheric pollution forum air pollutant emission inventory manual (SEI, 2012). f. Emission factors were commonly used for coking coal, anthracite and bituminous coal. g. Only defined for China. h. Emission factors for hard coal were adopted. i. Emission factors for heavy fuel oil were adopted. j. Emission factors for fuelwood were adopted.

**Table 3.14.** Default emission factors of $PM_{10}$, $PM_{2.5}$, BC, and OC from fuel combustion in residential sector. Unit is t/kt.

| Fuel type | $PM_{10}$ | $PM_{2.5}$ | BC | OC |
|---|---|---|---|---|
| Hard coal[i] | 7.4[a] | 4.49[a] | 1.02[a] | 2.15[a] |
| Raw coal[j] | 8.82[b] | 6.86[b] | 1.56[b] | 3.29[b] |
| Cleaned coal[j] | 8.82[b] | 6.86[b] | 1.56[b] | 3.29[b] |

| | | | |
|---|---|---|---|
| Other washed coal[j] | 8.82[b] | 6.86[b] | 1.56[b] | 3.29[b] |
| Sub-bituminous coal | 4.6[c] | 2.79[c] | 0.636[c] | 1.334[c] |
| Lignite | 4.6[c] | 2.79[c] | 0.636[c] | 1.334[c] |
| Coke oven coke[k] | 7.4 | 4.49 | 1.02 | 2.15 |
| LPG | 0.52[d] | 0.197[d] | 0.0676[d] | 0.052[d] |
| Kerosene | 0.9[d] | 0.341[d] | 0.117[d] | 0.09[d] |
| Diesel oil | 0.49[d] | 0.186[d] | 0.147[d] | 0.0441[d] |
| Crude oil[l] | 1.1 | 0.775 | 0.088 | 0.033 |
| Heavy fuel oil | 1.1[d] | 0.775[d] | 0.088[d] | 0.033[d] |
| Fuelwood | 5.76[e], 4.80[f] | 5.58[e], 4.60[f] | 1.12[e], 0.85[f] | 4.46[e], 3.20[f] |
| Crop residue | 7.21[e], 6.01[f] | 6.98[e], 5.75[f] | 1.05[e], 0.95[f] | 3.98[e], 3.70[f] |
| Animal waste | 9.8[g] | 9.8[g] | 0.4[g] | 3.1[g] |
| Charcoal | 4.1[h] | 3.32[h] | 0.205[h] | 0.82[h] |

a. Estimated based on $PM_{10}$ emission factors for residential sectors in Bond et al. (2004) and ratios of $PM_{2.5}$, BC, and OC to $PM_{10}$ in Lei et al. (2011). b. Estimated based on emission factors for stove in Lei et al. (2011). c. Emission factor for $PM_{10}$ derived from Bond et al. (2004) and ratios of $PM_{2.5}$, BC, and OC to $PM_{10}$ were from those for hard coal. d. Bond et al. (2004) for $PM_{10}$, BC, and OC and $PM_{2.5}/PM_{10}$ ratios were estimated based on Klimont et al. (2002b). e. Estimated based on Lei et al. (2011) and used for East Asian countries. f. Estimated based on Pandy et al. (2014) and used for Southeast and South Asian countries. g. Estimated based on Pandy et al. (2014) and commonly used for all countries. h. Emission factors of $PM_{10}$, BC, and OC were taken from Bond et al. (2004). $PM_{2.5}/PM_{10}$ ratios were estimated based on the global atmospheric pollution forum air pollutant emission inventory manual (SEI, 2012). i. Emission factors were commonly used for coking coal, anthracite and bituminous coal. j. Only defined for China. Values were gradually decreased from 1990 until their two third by 2005 referring Lei et al. (2011). k. Emission factors for hard coal were adopted. l. Emission factors for heavy fuel oil were adopted.

**Coke production and iron and steel industry**

The same as for CO, in REASv3, emissions of PM species from coke ovens were also estimated base on production amounts of coke oven coke (see Sect. S4.2.8). Emission factors of PM species for coal consumption in coke ovens were assumed to be zero assuming their contribution were included in the emissions estimated based on production amounts of coke described in Sect. S4.2.8. For China, emissions of PM species from iron and steel production were also estimated base on

production amounts of sinter, pig iron, and crude steel (see Sect. S4.2.1). It was assumed that emission factors for sinter and pig iron production obtained from Lei et al (2011) include emissions from coal combustion. Therefore, emission factors of PM species for coal combustion in iron and steel industry were assumed to be zero for China.

**Cement industry**

Emissions of PM species in China and Japan were not estimated based on fuel consumption, but using amount of cement production in each kiln type. For China, emission factors (t/kt of cement produced) of $PM_{10}$/$PM_{2.5}$/BC/OC were estimated based on Hua et al. (2016) and Lei et al. (2011) as follows: 44.8/19.2/0.115/0.192 for precalciner kilns, 37.3/14.9/0.0894/0.149 for other rotary kilns, and 8.9/3.2/0.0192/0.032 for shaft kilns. For Japan, emission factors of $PM_{10}$/$PM_{2.5}$/BC/OC (t/kt of clinker produced) were taken from AP-42 (US EPA, 1995) and Kupiainen and Klimont (2004) as follows: 15.6/4.55/0.0273/0.0455 for wet process kilns, 35.9/15.4/0.0924/0.154 for long dry process kilns, 54.6/23.4/0.140/0.234 for preheater process kilns and preheater/precalciner kilns. Ratio of clinker to cement was assumed to be 0.85 based on Cement handbook (Japan Cement Association, 2019). (See Sect. S4.1.3 for production data by different kiln types.). For other countries and regions, default emission factors in industry were used for all fuel types. See Sect. S4.2.3 for non-combustion emissions from cement production.

**Brick industry**

Emissions of PM species from brick production were not estimated based on fuel combustion, but using amount of brick production. Emission factors of $PM_{10}$/$PM_{2.5}$/BC/OC were assumed referring Lei et al. (2011), Weyant et al. (2014), and Klimont et al. (2017) as follows:
- China: 0.71/0.27/0.108/0.0945 t/kt of brick produced
- Japan, Republic of Korea, and Taiwan: 0.473/0.18/0.002/0.0035 t/kt of brick produced
- Other countries: 0.5/0.19/0.15/0.007 t/kt of brick produced

**Settings of emission controls**

Settings and assumptions for reduction of emissions of PM species from combustion sources by abatement equipment adopted in REASv3 are summarized in Table 3.15. For other sources not described in Table 3.15, no emission controls were considered.

**Table 3.15.** Settings and assumptions of emission controls of PM species

| Countries | Settings and assumption |
|---|---|
| China | ● Power plants
➢ Effects of control technologies by cyclones, wet scrubbers, electrostatic precipitators (ESP), and fabric filters during 1990-2015 were estimated based on their penetration rates in Lei et al. (2011) and Zhao et al. (2014).
➢ Reduction rates of $PM_{10}/PM_{2.5}$ were assumed to be 0.84/0.62, 0.92/0.78, and 0.98/0.94, and in 1990, 2000, and 2015, respectively. It was assumed that reduction rates before 1970 were zero and the values between 1970 and 1990 were interpolated.
● Industry
➢ Iron and steel industry: See Sect. S4.2.1
➢ Coke ovens: See Sect. S4.2.8.
➢ Non-ferrous metals industry: See Sect. S4.2.2
➢ Cement industry: See Sect. S4.2.3.
➢ Lime industry: See Sect. S4.2.4.
➢ Brick industry: See Sect. S4.2.5.
➢ Other industries: Due to lack of information, reduction rates were roughly assumed as follows: Reduction rates of $PM_{10}$ and $PM_{2.5}$ in 1990 were 0.55 and 0.25 referring settings of cement industry. Those in 2015 were 0.77 and 0.53 referring Wang et al. (2014) for settings of industry in 2010. It was assumed that reduction rates before 1980 were zero and the values between 1980, 1990, and 2015 were interpolated. |
| India | ● Due to lack of information, referring Sadavarte and Venkataraman (2014), Pandey et al. (2014), Guttikunda and Jawahar (2014), and Reddy and Venkataraman (2002), reduction rates of $PM_{10}/PM_{2.5}$ for power plants and industries during 1980-2015 were roughly assumed as follows: |

| | |
|---|---|
| | <li>Power plants: 0.0/0.0, 0.45/0.40, 0.85/0.81, and 0.87/0.85 in 1980, 1985, 2000, and 2015, respectively. Values between 1980, 1985, 2000, and 2015 were interpolated.</li><li>Iron and steel and cement industries: 0.0/0.0, 0.47/0.46, and 0.85/0.83 in 1980, 1995, and 2015, respectively. Values between 1980, 1995, and 2015 were interpolated.</li><li>Other industries: 0.0/0.0, 0.40/0.30, and 0.45/0.40 in 1980, 1995, and 2015, respectively. Values between 1980, 1995, and 2015 were interpolated.</li> |
| Japan | <li>Referring MRI (2015) and other literatures such as Shimoda (2016), Suzuki (1990) and Goto (1981), following assumptions were considered for control equipment of PM species:<li>Introduction of control equipment for power plants was expanded from 1957.</li><li>Introduction of bag filter was expanded from 1960.</li><li>From 1968, installation of ESP in power plants became mandatory.</li><li>Introduction of high quality ESP was expanded from 1975.</li><li>Regulations for PM species were strengthened from 1995.</li></li><li>Based on above assumption, reduction rates of $PM_{10}/PM_{2.5}$ for power plants were assumed as follows: 0.37/0.27, 0.9/0.88, and 0.995/0.99 in 1960, 1975, and after 2000, respectively. It was assumed that reduction rates before 1956 were zero and the values between 1950, 1960, 1975, and 2000 were interpolated.</li><li>For industry, reduction rates of $PM_{10}/PM_{2.5}$ after 2000 were assumed to be 0.99/0.985 for iron and steel and cement industries and 0.98/0.96 for other industries. Trends between 1950 and 2000 were assumed to be the same as for those of power plants.</li> |
| Republic of Korea/Taiwan | <li>Power plants: Based on Wang et al. (2014), reduction rates of $PM_{10}$ and $PM_{2.5}$ after 2005 were assumed to be 0.985 and 0.97, respectively. Referring Ebata et al. (1997), it was assumed that penetration rates of control equipment of PM species in 1990 were already high. Reduction rates in 1990 were assumed to be 0.9 and 0.88 for $PM_{10}$ and $PM_{2.5}$, respective and zero before 1970. Values between 1970, 1990, and 2005 were interpolated.</li><li>Industry: Based on Wang et al. (2014), reduction rates of $PM_{10}/PM_{2.5}$</li> |

| | |
|---|---|
| | in 2005 and in 2010 were assumed to be 0.944/0.905 and 0.948/0.910, respectively. It was roughly assumed that reduction rates of $PM_{10}$/$PM_{2.5}$ in 2015 were 0.968/0.935, respectively and zero before 1970. Values between 1970, 2005, 2010, and 2015 were interpolated.
 • Due to lack of information, the same settings for Republic of Korea were adopted. |
| Thailand | • Power plants: Referring Thao Pham et al. (2008), reduction rates of $PM_{10}$ and $PM_{2.5}$ in 2000 were assumed to be 0.84 and 0.80, respectively. For trends of reduction rates, it was roughly assumed that reduction rates of $PM_{10}$ and $PM_{2.5}$ were increased to 0.90 and 0.88 in 2015, respectively and zero before 1980. Values between 1980, 2000, and 2015 were interpolated.
 • Industry: Referring Thao Pham et al. (2008), for iron and steel and cement industries, reduction rates of $PM_{10}$ and $PM_{2.5}$ in 2005 were assumed to be 0.82 and 0.80, respectively. For trends of reduction rats, it was roughly assumed that reduction rates of $PM_{10}$ and $PM_{2.5}$ in 2015 were 0.85 and 0.83, respectively and zero before 1980. Values between 1980, 2000, and 2015 were interpolated. |
| Others | • Due to lack of information, settings of Thailand during 1980-2005 were adopted for those of Indonesia, Malaysia, Myanmar, Philippines, Vietnam and Mongolia during 1990-2015 and the same settings of Thailand were used for Singapore.
 • For Laos and Sri Lanka, reduction rates of 0.95/0.92 for $PM_{10}$/$PM_{2.5}$ were used for large power plants equipped with ESP based on information from World Electric Power Plants Database (Platts, 2018), |

**S3.2.5 Other species and sources**

**NMVOC**

Emission factors for fossil fuel combustion were taken from REASv2 based on Wei et al. (2008) for East Asian countries and the global atmospheric pollution forum air pollutant emission inventory manual (SEI, 2012) for Southeast and South Asian countries. For fuelwood, crop residue, and animal waste, emission factors were estimated as follows:

- Fuelwood
  - 3.13 t/kt based on Wei et al. (2008) for East Asian countries
  - 15.9 t/kt based on Sharma et al. (2015) for Southeast and South Asian countries
- Crop residue
  - 8.36 t/kt based on Wei et al. (2008) for East Asian countries
  - 13.3 t/kt based on Sharma et al. (2015) for Southeast and South Asian countries
- Animal waste
  - 10.4 t/kt based on Sharma et al. (2015) for all countries
- Charcoal
  - 100 t/PJ taken from IPCC (1997) for all countries

Emission factors described above were for total NMVOC. In REASv3, total NMVOC emissions were allocated to 19 NMVOC species categories defined in Sect. S2.1. The speciation was conducted based on speciation profiles for each sub-sector and fuel type provided by D. G. Streets (private communication) generally based on Klimont et al. (2002a) used for REASv1 and REASv2. The speciation profiles were commonly used for all countries and periods.

**NH$_3$**

Emission factors for fossil fuel combustion were taken from REASv1 based on EMEP/CORINAIR Emission Inventory Guidebook (EEA, 1996). For biofuel, 1.29 t/kt for fuelwood and 0.97 t/kt for charcoal were obtained from ABC Emission Inventory Manual (Shrestha et al., 2013). Due to lack of information, the emission factor for fuelwood was adopted to crop residue and animal waste.

**CO$_2$**

Emission factors for fuel combustion except for fuelwood, crop residue, and animal wastes were obtained from 2006 IPCC Guidelines for National Greenhouse Gas Inventories (IPCC, 2006). Default emission factors were used except for those of coal combustion in China where lower values were adopted referring Guan et al. (2012). Emission factors for fuelwood, crop residue, and animal wastes were 83.1, 87.0, and 76.9 kt/PJ derived from Streets and Waldhoff (1999).

**Agriculture**

For emissions from fuel combustion in agriculture sub-sector, emission factors of industry sector were used except for following settings for diesel oil referring Bond et al. (2004) and ABC Emission Inventory Manual (Shrestha et al., 2013):

- 50.3 t/kt for NO$_x$
- 16.0 t/kt for CO
- 2.0 t/kt for PM$_{10}$
- 1.72 t/kt for PM$_{2.5}$
- 1.14 t/kt for BC
- 0.36 t/kt for OC

**Charcoal production**

Activity data to estimate emissions from charcoal production as energy transformation sectors is wood input. Fuelwood consumption data developed based on methodologies described in Sect. S3.1 were used. Emission factors of NO$_x$, CO, and NMVOC were taken from Revised 1996 IPCC guidelines (IPCC, 1997) and those of others were based on Akagi et al. (2011).

**S4. Stationary non-combustion: Industrial production and other transformation**

Descriptions for evaporative NMVOC emissions and NH$_3$ emissions from non-combustion sources are provided in Sects. S5 and S8, respectively.

**S4.1 Activity data**

**S4.1.1 Iron and steel production**

Activity data to estimate non-combustion emissions from iron and steel production industry in REASv3 are production amounts of pig iron, crude steel, sinter, and hot rolled products. National total production of pig iron, crude steel, and hot rolled products were obtained from Steel Statistical Yearbook (World Steel Association, https://www.worldsteel.org/steel-by-topic/statistics/steel-statistical-yearbook.html) during 1968-2015 and extrapolated to 1950 using trends of pig iron and crude steel production in Mitchell (1998). For crude steel, production data by each process, oxygen-blown converter, electric furnace, and open-hearth furnace were separately obtained. Sinter production data were taken from Steel Statistical Yearbook during 1977-1992. For China, sinter production data were available during 2000-2015 and those between 1992 and 2000 were interpolated. Then, missing data between 1950 and 2015 were estimated based on trends of pig iron production in each country.

For regional distribution in China, production amounts of steel during 1950-2015 and pig iron during 1983-2015 in each region were available in China Data Online and China Statistical Yearbook (National Bureau of Statistics of China, 1986-2016), respectively. Pig iron data before 1982 were extrapolated for each region using the trends of steel production in China Data Online. Then, using the steel data, production amounts of crude steel and hot rolled products in China total were distributed to each region. Similarly using the regional pig iron data, sinter and pig iron production amounts in whole China were distributed to each region. For India, ratios of crude steel production in 17 sub-regions were estimated using Minerals Yearbook (United States Geological Survey (USGS)) and Indiastat during 2000-2015. Using the regional data, production amounts of pig iron, crude steel, singer, and hot rolled products in India total were distributed to each sub-region. For Japan, ratios of steel production amounts in 6 sub-regions during 2003 and 2011 were estimated using statistics of major factories (https://www.japanmetaldaily.com/statistics/crudemateworks/details/index.html) and production data of pig iron, crude steel, singer, and hot rolled products in India total were distributed to each sub-region.

**S4.1.2 Non-ferrous metal production**

In REASv3, non-combustion emissions from copper, zinc, lead, and aluminum production were considered in non-ferrous metal production processes. Activity data were production amounts of primary copper, zinc, lead, alumina, aluminum, and secondary aluminum obtained from Minerals Yearbook during 1960-2015 (USGS) and extrapolated to 1950 using trends of corresponding production data in Mitchell (1998). For China, India, and Japan, national total data need to be distributed to each sub-region. Weighting factors for the distribution were estimated during 1995-2015 using annual generation capacities of major plants in Minerals Yearbook (USGS). Before 1994, the weighting factors for 1995 were used.

**S4.1.3 Cement production**

Activity data for non-combustion emissions from cement industry are production amounts of cement. For China, regional data were basically available in China Data Online during 1950-2015. However, not all regions had complete data during the period and sometimes interpolation and extrapolation procedures were necessary. Therefore, in REASv3, regional data were used for weighting factors to distribute national total data of cement production to each sub-region. For Japan, national cement production during 1990-2015 were obtained from Minerals Yearbook (USGS) and extrapolated to 1950 using trends of corresponding data in the Historical Statistics of Japan (Japan Statistical Association, 2006). For the distribution to each sub-region, first, weighting factors in 2004 and 2018 were estimated using production amounts by major cement plants. Then, those during 2005-2015 were interpolated and data in 2004 were used before 2003. In addition to total amounts, production data by different kiln types were available in China (Hua et al., 2016) and Japan (Japan Cement Association, http://www.jcassoc.or.jp/cement/2eng/index.html). For other countries, national total production during 1960-2015 were obtained from Minerals Yearbook (USGS). For extrapolation to 1950, in REASv3, trends of national $CO_2$ emissions from cement production taken from CDIAC (Carbon Dioxide Information Analysis Center) (Marland et al., 2008). For regional data in India, weighting factors during 1984 and 2009 were estimated using regional production data in TERI Energy & Environment Data Diary and Yearbook (TERI, 2013, 2018). Before 1983 and after 2010, data in 1984 and 2009 were used, respectively.

**S4.1.4 Lime production**

Activity data for non-combustion emissions from lime industry are production amounts of lime. Data were obtained from Minerals Yearbook during 1960-2015 (USGS) and were extrapolated to 1950 using trends of cement production estimated in REASv3.

**S4.1.5 Brick production**

Activity data for non-combustion emissions from brick industry are production amounts of brick. However, unlike the other products in non-metallic minerals industry, brick production data were not available in most international and national statistics. For Japan, national production data during 1950-2007 were taken from Hiragushi (2009) and Japan Statistical Yearbook (Statistics Bureau, 2010-2018) and were distributed to 6 sub-regions using total fuel consumption in non-metallic minerals sector. For other countries, first, default data were prepared taken from REASv2 and GAINS ASIA at that time during 1990-2015 and extrapolated to 1950 using trends of cement production in each country. For China, Vietnam, Bangladesh, India, and Pakistan, national production data in 1990, 2000, 2005, and 2010 were obtained from Klimont et al. (2017) and interpolated during 1990-2010 and extrapolated to 2015 using trends of the default data. For China, data between 1980-1990 were extrapolated based on trends of production in Zhang (1997) and those before 1980 were extrapolated using trends of the default data. For regional distribution, fuel consumption data in brick production in each region (see Sects. S3.1.3 and S3.1.7) were used for weighting factors. For India, data between 1983-1990 were extrapolated based on trends of production in Industrial Commodity Statistical Yearbook taken from UN data, which is a web-based data service of the UN (http://data.un.org/) and those before 1983 were extrapolated using trends of the default data. For regional distribution, common weighting factors during 1950-2015 were estimated based on Maithel et al. (2012). For Vietnam, Bangladesh, and Pakistan, data before 1990 were extrapolated using trends of the default data. For Nepal, production data in 2006 were obtained from Maithel (2013) and extrapolated during 1950-2015 using trends of the default data. For Rep. of Korea, Indonesia, Myanmar, the default data were used during 1990-2015 and before 1990, data were extrapolated to 1985 using trends of production in Industrial Commodity Statistical Yearbook and then extended to 1950 using trends of the default data. For other countries, the default data were directly used.

**S4.1.6 Sulphuric acid production**

Activity data to estimate non-combustion emissions from sulphuric acid plants are amounts of total sulphuric acid production in each country and region. For China, national total production data during 1950-2015 were obtained from China Data Online and distributed to each region using regional data during 1983-2015 in China Statistical Yearbook (National Bureau of Statistics of China, 1986-2016). Before 1983, data in 1983 were used as weighting factors for the regional distribution. For Japan, national production data were taken from statistics provided by the Sulphuric Acid Association of Japan (http://www.ryusan-kyokai.org/) during 1983-2015 and extrapolated to 1950 using trends of sulphuric acid production in Mitchell (1998). Weighting factors for regional distribution were estimated using annual generation capacities of major plants in 2015 in Minerals Yearbook (USGS). For other countries, national total production data were provided by the Sulphuric Acid Association of Japan during 1980-2015 and extrapolated to 1950 using trends of sulphuric acid production in Mitchell (1998). For India, national total data were distributed to 17 sub-regions using data of REASv2 during 2000-2008 based on GAINS ASIA at that time. For the weighting factors, data in 2000 and 2008 were used before 2000 and 2008, respectively.

**S4.1.7 Carbon black production**

In REASv3, non-combustion emissions from carbon black production were only considered for China, India, Japan, and, Rep. of Korea. Similar to brick production, default data were prepared taken from REASv2 and GAINS ASIA at that time during 1990-2015 and extrapolated to 1950 using GDP in each country and region. For GDP, regional data in China during 1950-2015 were obtained from China Data Online. For other countries, data during 1970-2015 were derived from UN data, which is a web-based data service of the UN (http://data.un.org/) and extrapolated to 1960 using OECD Data (https://data.oecd.org/gdp/gross-domestic-product-gdp.htm) and then extrapolated to 1950 using trends of total population.

For China, national total production in 2010 were obtained from Wei et al. (2011) and were extrapolated during 1950-2015 and distributed to each region using the default data as weighting factors. For India, national production data during 1983-2003 were taken from Industrial Commodity Statistical Yearbook taken from the UN data and similar to China, the data were extrapolated during 1950-2015 and distributed to each region using the default data. For Japan and Rep. of Korea, national production data during 1964-2014 were obtained from Mineral Yearbook (USGS) and extrapolated during 1950-2015 and data in Japan were distributed to 6 sub-regions using the default data.

**S4.1.8 Other transformation sectors**

**Coke ovens**

In REASv3, activity data to estimate emissions from coke ovens as energy transformation sectors are coal input for $SO_2$ and $NO_x$ and coke production for CO, NMVOC, $CO_2$, and PM species. Coal consumption was taken from data developed based on methodologies described in Sect. S3.1. For coke production, national data were obtained from the International Energy Agency (IEA) World Energy Balances (IEA, 2017) during 1960-2015 for Japan and 1971-2015 for other countries. The data were extrapolated to 1950 based on trends of pig iron production before 1959 and 1970 for Japan and other countries, respectively. For China, regional production data during 1990-2015 were available in the China Energy Statistical Yearbook (CESY) (National Bureau of Statistics of China, 1986, 2001-2017) and used to distribute national total production data to each sub-region. Before 1990, data in 1990 were used. For India and Japan, weighting factors for the regional distribution were based on regional pig iron production data in each country.

**Petroleum refineries**

Activity data to estimate emissions from petroleum refineries as energy transformation sectors is crude oil input. Consumption data of crude oil developed based on methodologies described in Sect. S3.1 were used.

**S4.2 Emission factors and settings of emission controls**

**S4.2.1 Iron and steel production**

**Emission factors**

In REASv3, emissions of CO, NMVOC, $CO_2$, and PM species were estimated using production amounts of sinter, pig iron, crude steel, and rolled steel. Default emission factors are summarized in Table 4.1 and emission factors of PM species for China are provided in Table 4.2. Note that emission factors of CO for all countries and those of PM species for China include contributions from both combustion and non-combustion emissions. (See also Sects. S3.2.3 and S3.2.4.)

**Table 4.1.** Default emission factors of CO, NMVOC, $CO_2$, $PM_{10}$, $PM_{2.5}$, BC, and OC from production of sinter, pig iron, crude steel, and rolled steel. It was assumed that both combustion and non-combustion emissions are included in emission factors of CO. Unit is t/kt-produced.

| | Sinter | Pig iron | Crude steel/ OHF[a] | Crude steel/ BOF[a] | Crude steel/ EF[a] | Rolled steel |
|---|---|---|---|---|---|---|
| CO | 22.0[b] | 40.5[c] | 34.5[d] | 69.0[b] | 9.0[b] | - |
| NMVOC[e] | - | - | 0.055 | 0.055 | 0.055 | 0.025 |
| $CO_2$[f] | - | - | - | - | 80.0 | - |
| $PM_{10}$[g] | 1.555 | 0.490 | 8.760 | 14.63 | 10.18 | - |
| $PM_{2.5}$[g] | 0.691 | 0.300 | 6.330 | 10.45 | 7.550 | - |
| BC[h] | 0.005 | 0.018 | - | - | - | - |
| OC[h] | 0.026 | - | - | 2.090 | 0.180 | - |

a. OHF: Open-hearth furnace, BOF: Basic oxygen furnace, and EF: Electric furnace. b. AP-42 (US EPA, 1995), c. Streets et al. (2006), d. 50% of BOF was adopted. e. Klimont et al. (2002a). f. IPCC (2006). g. Klimont et al. (2002b). h. Kupiainen and Klimont (2004).

**Table 4.2.** Emission factors of $PM_{10}$, $PM_{2.5}$, BC, and OC from production of sinter, pig iron, crude steel, and rolled steel for China. It was assumed that both combustion and non-combustion emissions are included (except for emission factors of PM species for crude steel production). Unit is t/kt-produced.

| | Sinter | Pig iron | Crude steel/ OHF[a] | Crude steel/ BOF[a] | Crude steel/ EF[a] | Rolled steel |
|---|---|---|---|---|---|---|
| CO[b] | 22.00 | 40.50 | 27.10[d] | 54.20 | 9.000 | - |
| $PM_{10}$[c] | 6.050 | 9.650 | 19.10 | 14.63 | 8.120 | - |
| $PM_{2.5}$[c] | 2.620 | 6.000 | 13.80 | 10.45 | 6.020 | - |
| BC[c] | 0.0262 | 0.600 | 0.138 | - | - | - |
| OC[c] | 0.131 | 0.120 | 0.690 | 2.090 | 0.120 | - |

a. OHF: Open-hearth furnace, BOF: Basic oxygen furnace, and EF: Electric furnace. b. Streets et al. (2006). c. Lei et al. (2011). d. 50% of BOF was adopted.

For CO, the gas from blast furnace and basic oxygen furnace is collected and recycled in modern factories (Streets et al., 2006) and in REASv1, corresponding CO emissions in Japan were neglected. In REASv3, following settings were roughly assumed:

- China: Emission factors in Table 4.2 were used during 1950-2000 and 50% of the value was adopted in 2015. Emission factors between 2000 and 2015 were interpolated.
- Japan: Default emission factors were used before 1960 and 10% of the value was adopted in

1990. Emission factors between 1960 and1990 were interpolated.

- Republic of Korea and Taiwan: Default emission factors were used before 1975 and 10% of the value was adopted in 2005. Emission factors between 1975 and 2005 were interpolated.

**Settings of emission controls**

For iron and steel production, emission controls were only considered for PM species. Settings and assumptions for reduction of emissions in China by abatement equipment adopted in REASv3 are summarized in Table 4.3. For other countries, the same settings for combustion emissions in iron and steel industry were adopted. (See Table 3.15 in Sect. S3.2.4.)

**Table 4.3.** Settings and assumptions of emission controls of PM species for iron and steel production in China.

| Countries | Settings and assumption |
|---|---|
| China | <li>Referring Wu et al. (2017), reduction rates of $PM_{10}/PM_{2.5}$ for sinter production, pig iron, BOF, and EF in 2000, 2005, 2010, and 2015 were assumed as follows</li><li>Sinter: 0.780/0.592, 0.892/0.809, 0.946/0.916, and 0.956/0.939</li><li>Pig iron: 0.850/0.715, 0.910/0.844, 0.954/0.936, and 0.961/0.945</li><li>BOF: 0.850/0.715, 0.870/0.758, 0.955/0.937, and 0.959/0.943</li><li>EF: 0.782/0.568, 0.834/0.678, 0.900/0.815, and 0.977/0.968</li><li>It was assumed that reduction rates were zero in 1980 and values between 1980, 2000, 2005, 2010, and 2015 were interpolated.</li> |

**S4.2.2 Non-ferrous metal production**

In REASv3, emissions of $SO_2$, $PM_{10}$, and $PM_{2.5}$ were estimated using production amounts of copper, zinc, lead, and aluminum.

**$SO_2$**

Default emission factors were taken from Kato and Akimoto (1992) as follows:

- Copper: 2.0 kt/kt- produced
- Zinc: 1.0 kt/kt-produced
- Lead: 0.32 kt/kt-produced

In some countries, $SO_2$ emitted from non-ferrous metal plants were collected and used for

materials of sulphuric acid. In that case, the amounts of collected $SO_2$ need to be reduced from $SO_2$ emissions calculated by default emission factors. In REASv3, amounts of sulphuric acid produced using $SO_2$ collected from non-ferrous metal plants were obtained from the Sulphuric Acid Association of Japan based on reports of International Fertilizer Industry Association, the British Sulphur Cooperation Limited, Sulphuric Acid Notebook of Japan, and Kato et al. (1991). In addition, the same reduction rates of $SO_2$ by emission control equipment for non-ferrous metal industry were adopted.

**$PM_{10}$ and $PM_{2.5}$**

Default emission factors t/kt-produced were obtained from Lei et al. (2011) for China and Klimont et al. (2002b) for other countries as follows:
China:
- Copper, Zinc, and Lead: 276.0 for $PM_{10}$ and 246.0 for $PM_{2.5}$
- Aluminum (primary): 26.51 for $PM_{10}$ and 18.28 for $PM_{2.5}$
- Aluminum (secondary): 6.98 for $PM_{10}$ and 5.20 for $PM_{2.5}$

Other countries:
- Copper, Zinc, and Lead: 13.8 for $PM_{10}$ and 12.3 for $PM_{2.5}$
- Aluminum (primary): 27.26 for $PM_{10}$ and 18.5 for $PM_{2.5}$
- Aluminum (secondary): 6.97 for $PM_{10}$ and 5.195 for $PM_{2.5}$

For emission controls, the same settings for combustion emissions in industry sectors were adopted except for China. (See Table 3.15 in Sect. S3.2.4.) For China, reduction rates were assumed as follows:
- Referring Zhao et al. (2014), reduction rates of $PM_{10}$/$PM_{2.5}$ in 2010 and 2015 were 0.910/0.882 and 0.945/0.906, respectively and values between 2010 and 2015 were interpolated.
- Trends of reduction rates between 1980 and 2010 were assumed to be the same as settings for combustion emissions in other industries. (See Table 3.15 in Sect. S3.2.4.)

**S4.2.3 Cement production**

In REASv3, emissions of $CO_2$ and PM species for all countries and those of $NO_x$ and CO for Japan were estimated using production amounts of cement. For emission of $NO_x$ and CO in Japan and those of PM species in China and Japan, emission factors for combustion emissions were described in Sects. S3.2.2, S3.2.3 and S3.2.4, respectively. In this sub-section, emission factors for non-combustion emissions were described.

Default emission factor of $CO_2$ was 0.52 t/t-clinker produced based on IPCC (2006). Clinker to

cement ratios were roughly assumed as follows:

- China: 0.72 before 2005 and 0.6 in 2015 based on Gao et al. (2017). Values between 2005 and 2015 were interpolated.

- India: 0.83 before 1990 and 0.77 after 2005 based on Barcelo (2014). Values between 1990 and 2015 were interpolated.

- Japan: 0.85 base on Cement handbook (Japan Cement Association, 2019)

- Others: 0.9 before 1990 and 0.85 after 2005 based on Barcelo (2014). Values between 1990 and 2015 were interpolated.

For PM species, default emission factors of $PM_{10}$, $PM_{2.5}$, BC, and OC t/kt-produced were assumed as follows:

- China: 34.3, 9.8, 0.0588, and 0.098 were taken from Hua et al. (2016) and Lei et al. (2011).

- Others: 16.0, 4.64, 0.0278, and 0.0464 were derived from AP-42 (US EPA, 1995) and Lei et al. (2011).

For emission controls, the same settings for combustion emissions in cement industry were adopted except for China. (See Table 3.15 in Sect. S3.2.4.) For China, reduction rates were assumed as follows:

- Referring Hua et al. (2016), reduction rates of $PM_{10}$/$PM_{2.5}$ during 1980-2012 were estimated for each year. Values were 0.565/0.218, 0.586/0.250, 0.746/0.527, and 0.973/0.916 in 1980, 1990, 2000, and 2012, respectively.

- It was roughly assumed that reduction rates of $PM_{10}$/$PM_{2.5}$ in 2015 were 0.98/0.97 and zero in 1975. Values between 1975 and 1980 and those between 2010 and 2015 were interpolated.

**S4.2.4 Lime production**

In REASv3, emissions of $CO_2$ and PM species were estimated using production amounts of lime. Default emission factors of $CO_2$ were taken from IPCC (2006) and those of PM species were derived from Klimont et al. (2002b) and Kupiainen and Klimont (2004) as follows:

- $CO_2$: 750 t/kt-produced

- $PM_{10}$: 12.0 t/kt-produced

- $PM_{2.5}$: 1.4 t/kt-produced

- BC: 0.028 t/kt-produced

- OC: 0.014 t/kt-produced

For emission controls of PM species, the same settings for combustion emissions in industry sectors were adopted except for China. (See Table 3.15 in Sect. S3.2.4.) For China, reduction rates were assumed as follows:

- Referring Zhao et al. (2014), reduction rates of $PM_{10}$/$PM_{2.5}$ in 2010 and 2015 were 0.766/0.670

and 0.782/0.697, respectively and values between 2010 and 2015 were interpolated.

- Trends of reduction rates between 1985 and 2010 were assumed to be the same as settings between 1980 and 2005 for combustion emissions in other industries. (See Table 3.15 in Sect. S3.2.4.)

**S4.2.5 Brick production**

In REASv3, emissions of CO and PM species were estimated using production amounts of brick.

For CO, note that emissions in China, Japan, Republic of Korea, and Taiwan were estimated using fuel consumption as described in Sect. S3.2.3. For other countries, emissions were estimated with production amounts of brick and emission factor 2.0 t/kt-produced was taken from Weyan et al. (2014).

For PM species, default emission factors of $PM_{10}$, $PM_{2.5}$, BC, and OC t/kt-produced were assumed as follows:

- China: 0.71, 0.27, 0.108, and 0.0945 were taken from Lei et al. (2011).
- Japan, Republic of Korea, and Taiwan: Emission factors of tunnel kiln 0.4773, 0.18, 0.002, and 0.0035 were obtained from Klimont et al. (2017).
- Others: Emission factors of Bull's trench kiln 0.5, 0.19, 0.15, and 0.007 were based on Weyant et al. (2014).

For emission controls of PM species, the same settings for combustion emissions in industry sectors were adopted except for China. (See Table 3.15 in Sect. S3.2.4.) For China, reduction rates were assumed as follows:

- Referring Zhao et al. (2014), reduction rates of $PM_{10}$/$PM_{2.5}$ in 2010 and 2015 were 0.425/0.208 and 0.362/0.143, respectively and values between 2010 and 2015 were interpolated.
- Trends of reduction rates between 1985 and 2010 were assumed to be the same as settings for combustion emissions in other industries. (See Table 3.15 in Sect. S3.2.4.)

**S4.2.6 Sulphuric acid production**

In REASv3, emissions of $SO_2$ were estimated using production amounts of sulphuric acid. Default emission factors were taken from Kato et al. (1991) as follows:

- 20.0 t/kt-produced for China, Japan, Republic of Korea, and Taiwan
- 33.0 t/kt-produced for other countries.

For emission controls, the same settings for combustion emissions in large industries were adopted for Japan, Republic of Korea, and Taiwan and those for other industries were applied for China. For other countries, no emission controls were considered.

**S4.2.7 Carbon black production**

In REASv3, emissions of NMVOC and PM species were estimated using production amounts of carbon black. Default emission factor of NMVOC was taken from Klimont et al. (2002a) and those of PM species were derived from Klimont et al. (2002b) and Kupiainen and Klimont (2004) as follows:

- NMVOC: 90 t/kt-produced
- $PM_{10}$: 1.60 t/kt-produced
- $PM_{2.5}$: 1.44 t/kt-produced
- BC: 1.10 t/kt-produced
- OC: 0.00 t/kt-produced

For emission controls of PM species, the same settings for combustion emissions in industry sectors were adopted for all countries. (See Table 3.15 in Sect. S3.2.4.)

**S4.2.8 Other transformation sectors**

**Coke ovens**

In REASv3, emissions of CO, NMVOC, $CO_2$, and PM species were estimated using production amounts of coke oven coke.

For CO, emission factors were taken from Streets et al. (2006) as follows:
- 1.6 t/kt-produced for machinery coke ovens
- 15.6 t/kt-produced for indigenous coke ovens

Production amounts of coke oven coke in different technologies were only considered for China. Ratios of production amounts between machinery and indigenous coke ovens in each province in 2005 and 2006 were taken from China Industrial Economy Statistics Yearbook (National Bureau of Statistics, 2006-2007) and were extrapolated based on national ratios during 1990-2011 obtained from Huo et al. (2012a). It was roughly assumed that ratios of machinery coke ovens in 1970 were zero and gradually increased from 2011 to 2015. Data between 1970 and 1990 were interpolated. Due to lack of information, emission factors for machinery coke ovens were adopted for all other countries. As described in Sect. S3.2.3, emission factors were assumed to include contribution from combustion emissions.

Default emission factors of NMVOC was taken from Klimont et al. (2002a) and that of $CO_2$ was obtained from IPCC (2006) as follows:
- NMVOC: 1.44 t/kt-produced
- $CO_2$: 560 t/kt-produced

For PM species, default emission factors of $PM_{10}$, $PM_{2.5}$, BC, and OC t/kt-produced were assumed as follows:

- China: 8.79, 5.22, 1.57, and 1.83 were taken from Lei et al. (2011).
- Others: 3.36, 2.00, 0.75, and 0.54 were taken from Klimont et al. (2002b) and Kupiainen and Klimont (2004).

As described in Sect. S3.2.4, emission factors were assumed to include contribution from combustion emissions. For emission controls of PM species, the same settings for combustion emissions in iron and steel industry were adopted except for China. (See Table 3.15 in Sect. S3.2.4.) For China, reduction rates were assumed as follows:

- Referring Zhao et al. (2014), reduction rates of $PM_{10}/PM_{2.5}$ in 2010 and 2015 were estimated for machinery and indigenous coke ovens as follows:
  - Machinery: 0.773/0.560 and 0.803/0.624 in 2010 and 2015, respectively.
  - Indigenous: 0.193/0.140 and 0.200/0.156 in 2010 and 2015, respectively.
  - Values between 2010 and 2015 were interpolated.
- Trends of reduction rates between 1985 and 2010 were assumed to be the same as settings for combustion emissions in other industries. (See Table 3.15 in Sect. S3.2.4.)

**Petroleum refineries**

In REASv3, emissions of $SO_2$, NMVOC and PM species were estimated using consumption amounts of crude oil in oil refinery industry. Default emission factors were derived from Kato and Akimoto (1992) for $SO_2$, Klimont et al. (2002a) for NMVOC, Klimont et al. (2002b) and Kupiainen and Klimont (2004) for PM species as follows:

- $SO_2$: 0.46S t/kt (S: Sulfur contents in fuel in wt%)
- NMVOC: 2.34 t/PJ
- $PM_{10}$: 1.20 t/kt
- $PM_{2.5}$: 0.96 t/kt
- BC: 0.00015 t/kt
- OC: 0.00 t/kt

For emission controls of $SO_2$ and PM species, the same settings for combustion emissions in industry sectors were adopted for all countries. (See Table 3.15 in Sect. S3.2.4.)

**S4.2.9 Speciation of NMVOC emissions**

Emission factors described in Sect. S4.2 were for total NMVOC. In REASv3, total NMVOC emissions were allocated to 19 NMVOC species categories defined in Sect. S2.1. The speciation was conducted based on speciation profiles for each sub-sector provided by D. G. Streets (private communication) generally based on Klimont et al. (2002a) used for REASv1 and REASv2. The speciation profiles were commonly used for all countries and periods.

**S5. Non-combustion sources of NMVOC**

In this section, activity data, emission factors, and their sources used to estimate evaporative NMVOC emissions in REASv3 are described. See Sect. S2.4.3 for sub-sector categories defined in REASv3. For Japan, NMVOC emissions from evaporative sources were derived from the Ministry of the Environment Japan (MEOJ, 2017a) and thus, activity data and emission factors of Japan were not compiled in REASv3 (see Sect. S5.3.1 for Japan).

**S5.1 Activity data**

In REASv3, activity data of REASv2 during 2000-2008 estimated based on Klimont et al. (2002a) were used as "default".

**S5.1.1 Extraction processes**

In REASv3, emissions from gas production and distribution, oil production and handling, petroleum refineries, service stations, and transport and depots are included in those from extraction processes. Data sources and treatments of activity data for each sub-sector category used in REASv3 were summarized in Table 5.1.

**Table 5.1.** Data sources and treatments of activity data for sub-sectors of extraction processes.

| Sub-sector categories | Data sources and treatments of activity data |
|---|---|
| Gas production and distribution | ● Activity data: Natural gas production
 ● Data sources and treatments:
 ➢ China: Regional data during 1985-2015 were taken from the China Energy Statistical Yearbook (CESY) (National Bureau of Statistics of China, 1986, 2001-2017). Before 1985, data were extrapolated to 1971 using the International Energy Agency (IEA) World Energy Balances (IEAWEB) (IEA, 2017) and to 1950 using Mitchell (1998).
 ➢ India: National total data were obtained from IEAWEB and extrapolated to 1950 using Mitchel (1998). For regional distribution, weighting factors were calculated using regional data taken from TERI (2013, 2018).
 ➢ Other countries: National total data were derived from IEAWEB or the United Nations (UN) Energy Statistics Database (UN, 2016) and |

| | extrapolated to 1950 using Mitchel (1998). |
|---|---|
| Crude oil production and handling | • Activity data: Crude oil production
• Data sources and treatments:
➢ China: Regional data during 1950-2015 were derived from China Data Online.
➢ India: National total data were obtained from IEAWEB and extrapolated to 1950 using Mitchel (1998). For regional distribution, weighting factors were calculated using regional data taken from TERI (2013, 2018).
➢ Other countries: National total data were derived from IEAWEB or the UN Energy Statistics Database (UN, 2016) and extrapolated to 1950 using Mitchel (1998). |
| Petroleum refineries | • Activity data: Consumption of crude oil in petroleum refineries
• Data sources and treatments: See Sect. S3.1. |
| Service stations | • Activity data: Consumption of gasoline in road transport sector
• Data sources and treatments: See Sect. S3.1. |
| Transport and depots | • Activity data: Consumption of gasoline and diesel in road transport sector
• Data sources and treatments: See Sect. S3.1. |

**S5.1.2 Solvent use**

In this sub-section, activity data of NMVOC evaporative emissions from solvent use except for printing (See Sect. S5.1.3) and paint application (See Sect. S5.1.4) were described. Data sources and treatments of activity data for each sub-sector category used in REASv3 were summarized in Table 5.2. (See Sect. S4.1.7 for data sources of GDP used in this sub-section.)

**Table 5.2.** Data sources and treatments of activity data for sub-sectors of solvent use.

| Sub-sector categories | Data sources and treatments of activity data |
|---|---|
| Dry cleaning | • Activity data: Textiles cleaned
• Data sources and treatments:
➢ China: National total data in 2012 were taken from Wu et al. (2016) and extrapolated during 1950-2015 using trends of GDP. For regional distribution, urban population (see descriptions for domestic use of solvents in this table) were used as weighting factors. |

| | |
|---|---|
| | ➢ India: National data in 2010 were based on Sharma et al. (2015) and extrapolated during 1950-2015 using trends of GDP. For regional distribution, urban population were used as weighting factors.
➢ Other countries: Default data were used and extrapolated during 1950-2015 using trends of GDP. |
| Degreasing operation | ● Activity data: Solvent used
● Data sources and treatments:
➢ China: National total data in 2005 were taken from Wei et al. (2008). Regional distribution and extrapolation during 1950-2015 were conducted based on GDP.
➢ Other countries and regions: Default data were used during 2000-2008 and extrapolated during 1950-2015 using trends of GDP. |
| Vehicle treatment | ● Activity data: Cars registered
● Data sources and treatments: See Sect. S6.1.1. |
| Domestic use of solvents | ● Activity data: Urban and rural population
● Data sources and treatments:
➢ China: National and regional total population were obtained from China Data Online. Regional urban population data were calculated using proportion of urban population during 2005-2015 in China Statistical Yearbook (National Bureau of Statistics of China, 1986–2016) and the proportion data in 2005 for each region were used to estimated urban population before 2004. Then rural population in each region during 1950-2015 were calculated.
➢ India: National total population were taken from UN (2018). Regional ratios and proportion of urban population during 1951-2011 were estimated using data in Indiastat. Then, urban and rural population in each region were calculated.
➢ Other countries: National urban and rural population during 1950-2015 were derived from UN (2018). For Taiwan, population data were taken from Worldometer (https://www.worldometers.info/). |
| Asphalt blowing | ● Activity data: Asphalt produced
● Data sources and treatments:
➢ China: National total data in 2012 were taken from Wu et al. (2016) and extrapolated to 1950 using trends of Bitumen consumption in IEAWEB and GDP. Regional distribution was based on GDP.
➢ Other countries and regions: National and regional data were taken |

| | |
|---|---|
| | from default and extrapolated to 1950 using trends of Bitumen consumption in IEAWEB and GDP. |
| Paint production | ● Activity data: Paint produced

● Data sources and treatments:

➢ China: National total data during 2011-2013 were taken from Zheng et al. (2017).

➢ Other countries and regions: National data were taken from Industrial Commodity Statistical Yearbook.

➢ All countries and regions: Extrapolation for missing data and regional distribution were based on GDP. |
| Ink production | ● Activity data: Ink produced

● Data sources and treatments:

[remaining 187,993 characters of this post omitted]

---

## Author Response (AR2)

Dear Editor,

The authors would like to appreciate Editor for taking your precious time to handle our manuscript. Because there waw no comment from Referee #2, the author's responses are only for comments from Referee #1. We revised the main manuscript and one supplementary material entitled "Supplementary information and data related to methodology for REASv3" (hereafter "the revised Supplement") based on comments from Referee #1. Therefore, other supplementary materials not revised this time are not included in this Author's Response. For distribution of the updated data sets, as explained in the previous Author's Response, the final version will be opened at the REAS download site as REASv3.2, when the revision process has been completed.

The structure of this document is as follows:

(1) Comments, author's responses, and author's changes in manuscript related to Referee #1
(2) The revised main manuscript where changed parts were yellow highlighted
(3) The revised main manuscript with track changes
(4) The revised supplementary material (the revised Supplement) where changed parts were yellow highlighted
(5) The revised supplementary material (the revised Supplement) with track changes

Sincerely Yours,
Jun-ichi Kurokawa
Asia Center for Air Pollution Research
kurokawa@acap.asia
TEL: +81-25-263-9558
FAX +81-25-263-0567

(1) Comments, author's response, and author's changes in manuscript related to Referee #1

**No. 1**
*Referee comments*
*Line 10: "The average total emissions in Asia during 1950-1955 and from 2010-2015 (growth rates in these 60 years)"*

*1950 - 2015 is 65 years?*
*(I see now on line 470 this is defined. Define this on first use of the 60-year period in the main text or, better yet, in the first paragraph of section "3.1 Trends of Asian and national emissions".)*

Author's response to the Referee comments

"60 years" here means from middle of 1950-1955 to that of 2010-2015 (from about 1953 to 2013). To avoid confusion, corresponding sentences (L10 and L436) were revised as follows (underlined parts were added):

L10: The average total emissions in Asia during 1950-1955 and from 2010-2015 (growth rates in these 60 years estimated from the two averages) are ….

L436: Average total emissions in Asia during 1950-1955 and 2010-2015 (growth rates in these 60 years estimated from the two averages) are …

Author's changes in manuscript

- Line numbers (Page numbers) including corresponding revisions in the revised main manuscript are as follows: L11 (P1) and L438 (P14).

**No. 2**
*Referee comments*
*"which were relatively large even in past years in Asia."*
*did the authors perhaps mean to say "even in recent Asia"?*
*(Otherwise I'm not quite sure what this means. In earlier times residential emissions dominate over other sectors in general before widespread industrialization)*

Author's response to the Referee comments

For clarification, based on the advice, the corresponding part was revised as follows:

… which dominated over other sectors in earlier times and were relatively large even in recent years in Asia.

Author's changes in manuscript
- Line numbers (Page numbers) including corresponding revision in the revised main manuscript are as follows: L453-454 (P15).

**No. 3**

*Referee comments*

*It would be useful to get the author's perspective in section "3.4 Uncertainty" on uncertainty in emission trends given their extensive work with emissions data for this region. I realize this was not qualitatively estimated, but at least a qualitative discussion would be useful. In particular, as the author's noted in their response to reviewer comments, detailed information on emission control (and technology changes) were not available for all regions. So for some regions one would presume that recent trends might be more uncertain as a result due to lack of information. Similarly, for SO2, as mentioned in the current text, sulfur content in fuels is not known for the entire time period, which could impact trends.*

Author's response to the Referee comments

We appreciate the comments. Based on the advice, we added discussion to the latter half of second paragraph in Sect. 3.4 as follows (underlined parts were added):

For $SO_2$ emissions in China, uncertainties in 2015 were estimated to be slightly larger than those in 1985 due to uncertainties for removal efficiencies which were not considered in 1985. The same situation was found in uncertainties of $NO_x$ emissions from power plants in China between 1985 and 2015. Lack of detailed information for changes of technologies such as combustion burners and abatement equipment affect uncertainties of recent emission trends in Asia. For South and Southeast Asia, uncertainties of $SO_2$ emissions in 1985 were slightly smaller than those in 2015. This is because settings of sulfur contents in fuels were based on surveys conducted in 1990 (Kato and Akimoto, 1992) and thus, the uncertainties in 1985 were assumed to be smaller than those in 2015. In REASv3, information of temporal variations of sulfur contents in fuels including low-sulfur fuel regulations was limited which were also causes of uncertainties of emission trends. In general, uncertainties of emissions in REASv3 were smaller in later years because activity data are more accurate in recent years. However, detailed surveys for recent changes of technologies and information of emission controls are essential in future studies.

Author's changes in manuscript
- Line numbers (Page numbers) including corresponding revisions in the revised main manuscript are as follows: L887-890 (P28) and L892-896 (P28).

**No. 4**

*Referee comments*

*Comments on Supplement: Kurokawa_and_Ohara_Supplement_Methodology*

*Page 36, Section on SO2 Emission Factors*

*It is not clear what the units are here. Are these the fraction of Sulfur in the fuel that is emitted as SO2? Please clarify.*

Author's response to the Referee comments

To clarify the unit, the corresponding sentence was revised as follows:

Settings of REASv3 for the fraction of sulfur in the fuel that is emitted as $SO_2$ were taken from …

Author's changes in manuscript

- The corresponding revision (yellow highlighted) is in 12th line from the bottom of Page 36 of the revised Supplement.

**No. 5**

*Referee comments*

*Page 37 and forward, "Settings of emission controls".*

*In the China section, please clarify if "In 2015, reduction rates of SO2 emissions were assumed to be 75%, 63%, and 52% for (A), (B), and (C), respectively."*

*It is not clear if this is the assumed reduction per FGD unit, the total reduction as a result of the FGD deployment, or some other percentage (FGD penetration in 2015? Although this seems low.). Please clarify.*

Author's response to the Referee comments

These values were total reduction as a result of the FGD deployment. To avoid confusion, "total" were inserted before "reduction rates of $SO_2$ emissions" in the above sentence.

Author's changes in manuscript

- The corresponding revision (yellow highlighted) is in the last bullet for China in Table 3.8 on Page 38 of the revised Supplement as follows (the underlined part was added): In 2015, total reduction rates of $SO_2$ emissions were assumed to be 75%, 63%, and 52% for (A), (B), and (C), respectively.

**No. 6**

*Referee comments*

*Throughout this section, where not otherwise noted (it is in some places), please indicate what the assumed reduction fraction for FGD units are (since penetration rates are already given in general).*

Author's response to the Referee comments

The assumption of removal efficiencies of FGD units were added to Table 3.8

Author's changes in manuscript

- The corresponding revisions (yellow highlighted) are in Table 3.8 for China, Japan, and Republic of Korea on Pages 38-39 of the revised Supplement as follows (underlined parts were added):

  ➢ China:

  P37: … considered as point sources and 90% for other power plants. Removal efficiencies of FGD units were assumed to be 0.75 before 2003 and 0.90 after 2010 and the values were interpolated during 2004-2009.

  P38: … assumed to be smaller than (A) by 10% and 15%, respectively. It was assumed that removal efficiencies of FGD units were 0.75 for (A), 0.70 for (B) and 0.65 for (C).

  ➢ Japan

  P38: In 1990 and after 2000, introduction rates of FGD in power plants as point sources were assumed to be 95% and 100%, respectively. It was assumed that removal efficiencies of FGD units were 0.95 after 1990. Trends of total reduction rates during 1968 and 1990 were assumed based on MOEJ (2000) and those between 1990 and 2000 were interpolated.

  P38: Other sectors: Referring Kato et al. (1991), total reduction rates of SO2 emissions were assumed as follows:

  ➢ Republic of Korea

  P39: … area sources were assumed to be 5% lower than point sources. Removal efficiencies of FGD units were roughly assumed to be 0.90 in 1990 and 0.95 after 2000 and the values were interpolated during 1991-1999.

  P39: … 1990, 2005, and 2010 were interpolated and data in 2010 were used after 2011. It was assumed that removal efficiencies of FGD units were 0.95 for large industries and half of the values were adopted for other industries.

**No. 7**

*Referee comments*

*Page 51 "Settings of emission controls" for primary particulate emissions.*

*Please clarify how these assumptions impact BC/OC. Were the PM2.5 reduction assumptions applied to BC and OC, or were some other assumptions used?*

Author's response to the Referee comments

For BC and OC, the same reduction rats for $PM_{2.5}$ were applied in REASv3. It was clarified in the first paragraph of page 51.

Author's changes in manuscript

- The corresponding revision (yellow highlighted) in the revised Supplement is as follows: The sentence ("Note that the reduction rates of PM2.5 were applied to BC and OC.") was added to the end of first paragraph on Page 52 of the revised Supplement.

**No. 8**

*Page 55*

*It appears that CO2 emissions are a mix of fossil CO2 emissions and short-cycle CO2 emissions (e.g. from biomass sources, etc.). If so please make sure in the data release that these are reported separately.*

Author's response to the Referee comments

For $CO_2$ emissions, following revisions were conducted both in the main manuscript and the revised Supplement:

- In the first paragraph of Sect. 3.1, it was clarified that $CO_2$ emissions in this paper include contribution from biofuel combustion.
- For $CO_2$ emission values in the main manuscript including Table 3, emissions excluding those from biofuel combustion were also provided.
- For gridded data of $CO_2$ emissions from power plants, industry, and domestic sectors, data excluding contribution from biofuel and those from biofuel combustion are developed separately. Table 2.9 of the revised Supplement providing sector codes for gridded data in REASv3 were revised correspondingly.
- For table data of $CO_2$ emissions from major sectors released from the REAS web site, emissions excluding contribution from biofuel and those from biofuel combustion are presented independently.

Author's changes in manuscript
- Line numbers (Page numbers) including corresponding revisions in the revised main manuscript are as follows: L13 (P1); L432 (P14); L439-440 (P14); L473-474 (P15); L538-539 (P17); and $CO_2$ emissions excluding biofuel combustion were added to Table 3.
- In Table 2.9 of the revised Supplement, the corresponding footnote was added.
- When the final version will be opened at the REAS download site as REASv3.2 (after the revision process has been completed), $CO_2$ gridded data excluding biofuel combustion and those from biofuel combustion will be provided independently. Furthermore, for the table data released from the REAS download site, $CO_2$ emission excluding biofuel combustion and those from biofuel combustion will be both reported in the table separately.

**No. 9**

*Referee comments*

*On page 142 it says: "(Note that uncertainties for SO2 here were only for ratios of sulfur in fuels emitted as SO2 and influences of uncertainties in sulfur contents in fuels were not included.)"*

*While on page 143 it says "For SO2, in addition to uncertainties for ratios of sulfur emitted as SO2, those in sulfur contents in fuels need to be taken into considered. "*

*This was a bit confusing. Please clarify.*

Author's response to the Referee comments

We agree that descriptions of two pointed out sentences were confusing and inappropriate. The sentences were revised as follows:
- (Note that uncertainties of $SO_2$ estimated here were both for ratios of sulfur in fuels emitted as $SO_2$ ($U_{ERS}$ in equation (3) in Sect. 10.1) and for emission factors in the case of not using sulfur contents in fuels (see Sect. 3.2.1).)
- For $SO_2$, uncertainties for sulfur contents in fuels including effects of regulation (i.e. usage of low sulfur fuels) need to be taken into considered.

Author's changes in manuscript
- The corresponding revisions (yellow highlighted) are in 4th-7th lines from the bottom of Page 143 and 12th-14th lines from the bottom of Page 144 of the revised Supplement.

**No. 10**

*Referee comments*

*Table 10.3. Settings of uncertainties of removal efficiencies*

*For removal efficiencies it would be useful to clarify how these uncertainties were applied. Given that this is used in an emission factor calculation as (1 - removal-efficiency), a multiplicative uncertainty could result in a removal efficiency larger than 1, but I assume something else was done (or max efficiency capped?).*

Author's response to the Referee comments

Uncertainties described in Table 10.3 were assumed for total uncertainties in effects of emission controls. Namely, the assumed values were used as $U_R$ in equations (2) and (3) in Sect. 10.1. For clarification, first, "Removal efficiencies" in page 144 were changed to "Effects of emission controls". Then, descriptions including those in Table 10.3 on pages 144-146 were revised. The same revisions were conducted for descriptions including those in Table 10.5 on pages 148-149 in Sect. 10.2.2.

Author's changes in manuscript
- The corresponding revisions (yellow highlighted) in the revised Supplement are as follows:
  - $8^{th}$ line from the bottom of Page 145: "Removal efficiencies" was changed to "Effects of emission controls".
  - $2^{nd}$ line from the bottom of Page 145: "uncertainties of removal efficiencies" was changed to "total uncertainties in effects of emission controls, namely $U_R$ in equations (2) and (3) in Sect. 10.1".
  - $1^{st}$ line of Page 146: "Uncertainties of removal efficiencies" was changed to "$U_R$".
  - $4^{th}$ line of Page 146: "uncertainties of corresponding removal efficiencies" was chanted to "corresponding $U_R$".
  - $6^{th}$ line of Page 146: "uncertainties of removal efficiencies" was changed to "$U_R$".
  - The caption of Table 10.3 was changed from "Settings of uncertainties of removal efficiencies adopted in REASv3. Note that uncertainties of removal efficiencies for sources without description here were assumed to be zero." to "Settings of total uncertainties in effects of emission controls ($U_R$) adopted in REASv3. Note that $U_R$ for sources without description here were assumed to be zero."
  - In Table 10.3, all "uncertainties of removal efficiencies" were changed to "$U_R$".
  - $1^{st}$ line of Page 149: "settings" was changed to "effects".
  - $3^{rd}$ line of Page 149: "settings of emission controls" was changed to "effects of emission controls ($U_R$)".
  - $6^{th}$ line of Page 149: "effects of" was inserted before "emission controls".

➢ In caption of Table 10.5, "emission controls" was changed to "effects of emission controls $(U_R)$".

➢ In Table 10.5, all "Emission controls" were changed to "$U_R$".

(2) The revised main manuscript where changed parts were yellow highlighted

From the next page, the revised main manuscript where changed parts were yellow highlighted is provided.

[revised manuscript text omitted]
[e] 2010 | 43635 | 46368 | 302562 | 52711 | 30621 | 17055 | 29880 | 21220 | 3233 | 6757 |
| Asia[e] 2011 | 45003 | 48868 | 304900 | 55136 | 30878 | 18047 | 30540 | 21559 | 3266 | 6652 |
| Asia[e] 2012 | 44227 | 48962 | 304396 | 57285 | 31283 | 18496 | 30414 | 21526 | 3254 | 6587 |
| Asia[e] 2013 | 42725 | 47561 | 304484 | 58971 | 31559 | 19200 | 30649 | 21627 | 3227 | 6485 |
| Asia[e] 2014 | 40864 | 46970 | 302718 | 60801 | 31770 | 19447 | 30469 | 21475 | 3219 | 6478 |
| Asia[e] 2015 | 37876 | 44835 | 296809 | 61627 | 31950 | 19423 | 29034 | 20644 | 3155 | 6422 |

[a] Gg NO$_2$ yr[-1].

[b] Tg yr[-1].

[revised manuscript text omitted]

(4) The revised supplementary material (the revised Supplement) where changed parts were yellow highlighted

From the next page, the revised supplementary material (the revised Supplement) where changed parts were yellow highlighted is provided. In addition to the revisions based on comments from Referees #1 described in (1), some additional revisions (yellow highlighted) were done in the revised Supplement mainly for correction of typos and English problems as follows:

- Contents: Number of pages for sections were revised.
- Table 2.1: "Organic" was changed to "organic"
- Table 2.2: Following changes were conducted:
  - ➢ "Other Alkanes" was changed to "Other alkanes"
  - ➢ "Terminal Alkenes" was changed to "Other alkenes"
  - ➢ "Internal Alkenes" was changed to "Internal alkenes"
  - ➢ "14 Other Aromatics" was changed to "14 Other aromatics"
  - ➢ "16 Other Aromatics" was changed to "16 Other aldehyde"
- P10 5th-6th lines: ", respectively" were inserted after "2.7".
- P10 7th line: "Sect. 5.1.7 and Sect. S6.3" was changed to "Sects. S5.1.7 and S6.3".
- P11 1st line: "," was inserted before "offset printing".
- P11 7th line: "for" was changed to "of".
- P13 10th line: "are" was changed to "is".
- P13 18th line: "not included in Table 3.1" was inserted after "other fuel types" for clarification.
- P25-P27 in Table 3.4: Typos were corrected in "Data sources and treatments" for Anhui, Chongqing, Shaanxi, Qinghai, Ningxia, and Xinjiang.
- P28 6th line from the bottom: "ratio" was changed to "ratios"
- P30 in Table 3.5: In "Industry and energy sectors (default)" and "Residential and other domestic sectors", typos were corrected and missing information was added.
- P30 2nd line from the bottom: "for" was changed to "to".
- P36 in Table 3.7: Typos were corrected in "Settings and assumptions" for Japan.
- P40 2nd line from the bottom: "," was inserted after "industry".
- P43-P44 in Table 3.10: Typos were corrected in "Settings and assumptions" for China and Others.
- P48 6th line: "," was inserted after "Industry".
- P50 in Table 3.14: A typo was corrected.
- P52 in Table 3.15: Following revisions were conducted:
  - ➢ "." was added to the end of caption.
  - ➢ "," was inserted after Vietnam in "Settings and assumption" for Others.

- ➤ Missing information was added to "Settings and assumption" for Thailand.

- P62 1st line from the bottom: "in" was changed to "after".

- P63 3rd line: "in" was changed to "after".

- P65 3rd, 5th, and 8th lines: All "2015" were chanted to "2005"

- P113-P114 in Table 6.7: Typos were corrected in "Settings and data sources" for India, Indonesia, Singapore, and Thailand.

- P145 in Table 10.2: missing footnote indicators [a] were added to Coal fuels and $PM_{10}/PM_{2.5}/BC/OC$.

- In the previous Supplement, a following paper was referred as Lei et al. (2011)

  Lei, Y., Zhang, Q., Nielsen, C., and He, K.: An inventory of primary air pollutants and $CO_2$ emissions from cement production in China, 1990–2020, Atmos. Environ., 45, 147-154, https://doi.org/10.1016/j.atmosenv.2010.09.034, 2011.

  However, there were several points where the following paper must be referred:

  Lei, Y., Zhang, Q., He, K. B., and Streets, D. G.: Primary anthropogenic aerosol emission trends for China, 1990–2005, Atmos. Chem. Phys., 11, 931–954, https://doi.org/10.5194/acp-11-931-2011, 2011.

  In the revised Supplement, first and second ones were referred as Lei et al. (2011a) and Lei et al. (2011b), respectively and following corrections were conducted:

  - ➤ In Reference, the year of first paper was changed to "2011" to "2011a" and information of the second paper was added and the year was defined as "2011b".

  - ➤ Changes from "Lei et al. (2011)" to "Lei et al. (2011a)" were conducted as follows:
    P42 17th-20th lines; P47 13th-16th lines; P51 10th line; P65 11th-13th lines

  - ➤ Corrections from "Lei et al (2011)" to "Lei et al. (2011b)" were done as follows:
    P48 footnote b of Table 3.12; P49 footnote b of Table 3.13; P50 footnotes a, b, e, and j of Table 3.14; P51 2nd line; P51 10th line; P51 4th line from the bottom; P52 in Table 3.15 for China; P62 footnote c of Table 4.2; P64 11th line; P66 15th line; P68 3rd line; P144 15th line; P149 in Table 10.5 for Cement production.

*Supplement of*

**Long-term historical trends in air pollutant emissions in Asia: Regional Emission inventory in ASia (REAS) version 3**

**Junichi Kurokawa and Toshimasa Ohara**

*Correspondence to*: Junichi Kurokawa (kurokawa@acap.asia)

**Supplementary information and data related to methodology of REASv3**

**Contents**

**S1. Introduction**

This document provides detailed information related to methodologies of Regional Emission inventory in ASia (REAS) version 3 (hereafter REASv3 in this document) developed as a supplementary material of the main manuscript entitled "Long-term historical trends in air pollutant emissions in Asia: Regional Emission inventory in ASia (REAS) version 3". In this document, first and second versions of REAS are often cited and expressed as REASv1 (Ohara et al., 2007) and REASv2 (Kurokawa et al., 2013), respectively. The framework of REASv3 such as target species, countries and regions, and emission sources was summarized in Sect. 2. Sects. 3, 4, 5, 6, and 7 provide details of activity data and emission factors including settings of emission controls for stationary combustion, industrial production, non-combustion sources of NMVOC, road transport, and other transport, respectively. The details related to methodology for non-combustion sources of $NH_3$ were given in Sect. 8. Grid allocation and monthly variation factors for spatial and temporal distribution were described in Sect. 9. In Sect. 10, details of methodology and settings for estimation of uncertainties were provided.

Note that this document is for REASv3.2 which is an updated version of REASv3.1 (Kurokawa et al., 2019). The differences between REASv3.2 and REASv3.1 and causes of the discrepancies were provided in another document entitled "Differences between REASv3.2 and REASv3.1" developed as an additional supplement of the main manuscript.

**S2. Framework of REASv3**

**S2.1 Target species**

Target species of REASv3 are summarized in Table 2.1. In REASv3, NMVOC species were divided into 19 chemical species categories as presented in Table 2.2. Codes of each species used in emission tables and gridded data of REASv3 are also provided in the tables.

**Table 2.1.** Target species of REASv3.

| Species code | Species |
| --- | --- |
| SO2 | Sulfur dioxide |
| NOX | Nitrogen oxides (as $NO_2$) |
| CO_ | Carbon monoxide |
| NMV | Non-methane volatile organic compounds |
| NH3 | Ammonia |
| CO2 | Carbon dioxide |
| PM10_ | Primary $PM_{10}$ |
| PM2.5 | Primary $PM_{2.5}$ |
| BC_ | Black carbon |
| OC_ | Primary organic carbon |

**Table 2.2.** NMVOC species categories defined in REASv3.

| Species number code | NMVOC species |
| --- | --- |
| 01 | Ethane |
| 02 | Propane |
| 03 | Butanes |
| 04 | Pentanes |
| 05 | Other alkanes |
| 06 | Ethylene |
| 07 | Propene |
| 08 | Terminal alkenes |
| 09 | Internal alkenes |
| 10 | Acetylene |
| 11 | Benzene |
| 12 | Toluene |
| 13 | Xylenes |

| | |
|---|---|
| 14 | Other aromatics |
| 15 | Formaldehyde |
| 16 | Other aldehyde |
| 17 | Ketones |
| 18 | Halocarbons |
| 19 | Others |
| 20 | Total |

**S2.2 Target years**

Target years of REASv3 are 1950-2015 (each year). In future updated versions, the oldest target year is basically fixed, but data in later years (after 2016) are planned to be added.

**S2.3 Target countries and regions**

Table 2.3 provides list of countries and sub-regions included in the inventory domain of REASv3. Codes of region, countries, and sub-regions used in the main manuscript, emission tables and gridded data of REASv3 are also provided in the table.

**Table 2.3.** Region, country, and sub-region included in the inventory domain of REASv3 with codes used in the main manuscript and files of emission tables and gridded data provided from the REAS website (https://www.nies.go.jp/REAS/).

| Region name/ Region code | Country name: Sub-region name | Country and sub-region code CCCRR CCC: Country code RR: Sub-region code |
|---|---|---|
| China/ CHN | China: Whole Country | CHNWC |
| | China: Beijing | CHNBJ |
| | China: Tianjin | CHNTJ |
| | China: Hebei | CHNHE |
| | China: Shanxi | CHNSX |
| | China: Inner Mongolia | CHNNM |
| | China: Liaoning | CHNLN |
| | China: Jilin | CHNJL |
| | China: Heilongjiang | CHNHL |

| | | |
|---|---|---|
| | China: Shanghai | CHNSH |
| | China: Jiangsu | CHNJS |
| | China: Zhejiang | CHNZJ |
| | China: Anhui | CHNAH |
| | China: Fujian | CHNFJ |
| | China: Jiangxi | CHNJX |
| | China: Shandong | CHNSD |
| | China: Henan | CHNHA |
| | China: Hubei | CHNHB |
| | China: Hunan | CHNHN |
| | China: Guangdong | CHNGD |
| | China: Guangxi | CHNGX |
| | China: Hainan | CHNHI |
| | China: Chongqing | CHNCQ |
| | China: Sichuan | CHNSC |
| | China: Guizhou | CHNGZ |
| | China: Yunnan | CHNYN |
| | China: Tibet | CHNXZ |
| | China: Shaanxi | CHNSN |
| | China: Gansu | CHNGS |
| | China: Qinghai | CHNQH |
| | China: Ningxia | CHNNX |
| | China: Xinjiang | CHNXJ |
| | China: Hong Kong | CHNHK |
| | China: Macau | CHNMC |
| India/ IND | India: Whole Country | INDWC |
| | India: Andhra Pradesh | INDAP |
| | India: Bihar, Jharkhand | INDBJ |
| | India: North East (Arunachal Pradesh/Assam/Manipur/ Meghalaya/Mizoram/Nagaland/Sikkim/Tripura) | INDAN |
| | India: Gujarat | INDGU |
| | India: Haryana | INDHA |
| | India: Karnataka/Goa | INDKG |
| | India: Kerala | INDKE |
| | India: Madhya Pradesh/Chhattisgarh | INDMC |

| | | |
|---|---|---|
| | India: Maharashtra | INDMA |
| | India: Orissa | INDOR |
| | India: Punjab/Chandigarh | INDPU |
| | India: Rajasthan | INDRA |
| | India: Tamil Nadu | INDTN |
| | India: Utter Pradesh/Uttaranchal | INDUU |
| | India: West Bengal | INDWB |
| | India: Himachal Pradesh/Jammu and Kashmir | INDHJ |
| | India: Delhi | INDDE |
| Japan/
JPN | Japan: Whole Country | JPNWC |
| | Japan: Hokkaido-Tohoku (Hokkaido/Aomori/Iwate/
Miyagi/Akita/Yamagata/Fukukshima) | JPNHT |
| | Japan: Kanto (Ibaraki/Tochigi/Gunma/Saitama/Chiba/
Tokyo/Kanagawa) | JPNKN |
| | Japan: Chubu (Niigata/Toyama/Ishikawa/Fukui/
Yamanashi/Nagano/Gifu/Shizuoka/Aichi) | JPNCB |
| | Japan: Kinki (Mie/Shiga/Kyoto/Osaka/Hyogo/Nara/
Wakayama) | JPNKK |
| | Japan: Chugoku-Shikoku (Tottori/Shimane/Okayama/
Hiroshima/Yamaguchi/Tokushima/Kagawa/Ehime/Kochi) | JPNCS |
| | Japan: Kyushu-Okinawa (Fukuoka/Saga/Nagasaki/
Kumamoto/Oita/Miyazaki/Kagoshima/Okinawa) | JPNKO |
| Other East Asia /
OEA | Democratic People's Republic of Korea, Whole Country | PRKWC |
| | Republic of Korea, Whole Country | KORWC |
| | Mongolia: Whole Country | MNGWC |
| | Taiwan: Whole Country | TWNWC |
| Southeast Asia/
SEA | Brunei: Whole Country | BRNWC |
| | Cambodia: Whole Country | KHMWC |
| | Indonesia: Whole Country | IDNWC |
| | Laos: Whole Country | LAOWC |
| | Malaysia: Whole Country | MYSWC |
| | Myanmar: Whole Country | MMRWC |
| | Philippines: Whole Country | PHLWC |
| | Singapore: Whole Country re | SGPWC |
| | Thailand: Whole Country | THAWC |

| | Vietnam: Whole Country | VNMWC |
|---|---|---|
| Other South Asia/ OSA | Afghanistan: Whole Country | AFGWC |
| | Bangladesh: Whole Country | BGDWC |
| | Bhutan: Whole Country | BTNWC |
| | Maldives: Whole Country | MDVWC |
| | Nepal: Whole Country | NPLWC |
| | Pakistan: Whole Country | PAKWC |
| | Sri Lanka: Whole Country | LKAWC |

**S2.4 Target emission sources**

**S2.4.1 Combustion sources**

Table 2.4 provides list of sub-sector categories of combustion sources defined in REASv3. Aggregated sector categories used in the main manuscript and emission tables of REASv3 are presented as "Sector code". IEA codes show relationships between sub-sector categories of REASv3 and the International Energy Agency (IEA) World Energy Balances (IEAWEB) (IEA, 2017). Fuel types defined in REASv3 are provided in Sect S3.1.1. See Sects. S3, S6, and S7 for details of stationary combustion, road transport, and other transport sectors, respectively.

Several emission sources related to transformation sectors except for power plants were included in Table 2.4. Sources categorized as energy sectors in IEAWEB are only considered as combustion sources. For coke ovens (not as the energy sector), emissions were estimated based on coal input for $SO_2$ and $NO_x$ and coke production for CO, NMVOC, $CO_2$, and PM species. In REASv3, for coke ovens as energy transformation sectors, contributions from both combustion and non-combustion processes were included in the emissions. In other words, their emissions were not estimated separately. Similarly, the following sources include both combustion and non-combustion emissions which were not estimated separately:

● Charcoal production plants
● Manufacture of other solid fuels
● Gas works

In addition, CO emissions from pig iron, crude steel, and sinter production for all countries, those from brick production except for China, Japan, Republic of Korea, and Taiwan, emissions of PM species from sinter and pig iron production for China, and those from brick production for all countries estimated based on their production amounts include contributions from both combustion and non-combustion sources (not estimated separately).

**Table 2.4.** Sub-sector categories of combustion sources considered in REASv3 with sector codes used in the main manuscript and emission tables of REASv3 and IEA codes showing relationships between sub-sector categories of REASv3 and the IEAWEB.

| Sector code | Sub-sector category | IEA code |
|---|---|---|
| Power Plants/ PP | Power plants (point sources/area sources) | MAINELEC/AUTOELEC/ MAINCHP/AUTOCHP/ MAINHEAT/AUTOHEAT/ THEA/TBOILER/TELE |
| | Power plants (energy) | EPOWERPLT |
| Industry/ IND | Coke ovens | TCOKEOVS |
| | Charcoal production plants | TCHARCOAL |
| | Manufacture of other solid fuels | TPATFUEL/TBKB/TNONSPEC |
| | Coke ovens (energy) | ECOKEOVS |
| | Charcoal production plants (energy) | ECHARCOAL |
| | Manufacture of other solid fuels (energy) | EMINES/EPATFUEL/EBKB/ ENONSPEC |
| | Petroleum refineries (energy) | EREFINER |
| | Manufacture of other liquid fuels (energy) | EOILGASEX/ECOALLIQ/EGTL |
| | Gas works | TGASWKS |
| | Gas works (energy) | EGASWKS |
| | Manufacture of other gaseous fuels (energy) | ELNG/EGTL |
| | Chemical and petrochemical industry | CHEMICAL |
| | Iron and steel industry | IRONSTL |
| | Blast furnace | TBLASTFUR |
| | Blast furnace (energy) | EBLASTFUR |
| | Non-ferrous metal industry | NONFERR |
| | Cement industry | NONMET |
| | Lime industry | |
| | Brick industry | |
| | Other non-metallic minerals industries | |
| | Construction industry | CONSTRUC |
| | Transport equipment industry | TRANSEQ |
| | Machinery industry | MACHINE |
| | Mining and quarrying industry | MINING |
| | Food and tobacco industry | FOODPRO |

| | Paper, pulp and printing industry | PAPERPRO |
| | Wood and wood products industry | WOODPRO |
| | Textile and leather industry | TEXTILES |
| | Other industries | INONSPEC |
| Road transport/ ROAD | Road transport | ROAD |
| Other transport/ OTRA | Rail | RAIL |
| | Pipeline transport | PIPLINE |
| | Other transport[*1] | TRNONSPE |
| Residential/ RESI | Residential | RESIDENT |
| Other domestic/ ODOM | Commercial and public services | COMMPUB |
| | Agriculture[*2] | AGRICULT |
| | Others | ONONSPEC |

[*1]Aviation and navigation (both for domestic and international) are not included.

[*2]Forestry is included, but fishing is not included.

**S2.4.2 Non-combustion sources: Industrial production and other transformation**

Table 2.5 provides list of sub-sector categories of non-combustion sources defined in REASv3 with target species and notes for each sub-sector category. See Sect. S4 for details of industrial processes and other transformation. See Sects. S5 and S8 for industrial processes related to NMVOC and $NH_3$, respectively. Note that, as described in Sect S2.4.1, non-combustion emissions from coke production, those of CO from pig iron, crude steel, and sinter productions (for all countries and regions) and from brick production (except for China, Japan, Republic of Korea, and Taiwan), and those of PM species from sinter and pig iron production (for China) and from brick production (for all countries) were not estimated separately. For these sources, estimated emission in REASv3 include contributions from both combustion and non-combustion processes.

**Table 2.5.** Sub-sector categories of non-combustion sources from industrial production and other transformation considered in REASv3.

| Sub-sector category | Target species | Notes |
| --- | --- | --- |
| Pig iron production | CO, PM species | Iron and steel industry |
| Crude steel production | CO, NMVOC, PM species | |

| | | |
|---|---|---|
| Sinter production | CO, PM species | |
| Rolled steel production | NMVOC | |
| Copper production | $SO_2$, PM species | Non-ferrous metal industry |
| Zinc production | $SO_2$, PM species | |
| Lead production | $SO_2$, PM species | |
| Almina production | $SO_2$, PM species | |
| Aluminium production | $SO_2$, PM species | |
| Cement production | $CO_2$, PM species | Non-metallic minerals industry |
| Lime production | $CO_2$, PM species | |
| Brick production | PM species | |
| Sulphuric acid production | $SO_2$ | Inorganic chemicals industry |
| Carbon black production | NMVOC, PM species | |
| Ethylene production | NMVOC | Organic chemicals industry |
| Polyethylene production | NMVOC | |
| Styrene production | NMVOC | |
| Polystyrene production | NMVOC | |
| Polyvinylchloride production | NMVOC | |
| Propylene production | NMVOC | |
| Polypropylene production | NMVOC | |
| Polyvinylchloride processing | NMVOC | |
| Polystyrene processing | NMVOC | |
| Bread production | NMVOC | Other industries considered for NMVOC |
| Beer production | NMVOC | |
| Asphalt production | NMVOC | |
| Pulp and paper production | NMVOC | |
| Ammonia | $NH_3$ | Synthetic fertilizer industry considered for $NH_3$ |
| Ammonium nitrate | $NH_3$ | |
| Urea | $NH_3$ | |
| Coke production | CO, NMVOC, $CO_2$, PM species | Manufacture of solid fuels |
| Petroleum refineries | $SO_2$, NMVOC, PM species | Manufacture of liquid fuels For NMVOC, contributions were included in extraction processes. See Sect. S2.4.3. |

**S2.4.3 Non-combustion sources of NMVOC**

Non-combustion sources for NMVOC emissions considered in REASv3 are extraction processes, solvent use, industrial processes, waste disposal and evaporative emissions from road vehicles. Sub-categories of extraction processes and solvent use are summarized in Tables 2.6 and 2.7, respectively. Definitions of the sub-sectors are the same as with those of Klimont et al. (2002a). See Table 2.5, Sects. S5.1.7 and S6.3 for industrial processes, waste disposal, and evaporative emissions from road vehicles, respectively. See Sect. S5 for details of non-combustion sources of NMVOC.

**Table 2.6.** Sub-sector categories of extraction processes considered in REASv3.

| Sub-category |
| --- |
| Gas production |
| Gas distribution |
| Crude oil production |
| Crude oil handling |
| Petroleum refineries[a] |
| Service station |
| Transport and depots |

a. Except for NMVOC, contributions were included in industrial processes. See Sect. S2.4.2.

**Table 2.7.** Sub-sector categories of solvent use considered in REASv3.

| Sub-category |
| --- |
| Dry cleaning |
| Decreasing operation |
| Vehicle treatment |
| Domestic use of solvents |
| Asphalt blowing |
| Paint production |
| Ink production |
| Tire production |
| Synthetic rubber production |
| Textile industry |
| Preservation of wood |
| Adhesive application |
| Printing[a] |
| Paint application[b] |

a. Contributions from following activities were included: packing, offset printing, publication, and screen printing. b. Contributions from following purposes were included: architecture, domestic usage, automobile manufacture, vehicle refinishing, and other industrial application.

**S2.4.4 Non-combustion sources of NH₃**

Non-combustion sources of NH₃ emissions considered in REASv3 are manure management of livestock, fertilizer application, industrial processes, human, and latrines as summarized in Table 2.8. See Sect. S8 for details of non-combustion sources of NH₃.

**Table 2.8.** Sub-sector categories of non-combustion sources of NH₃ considered in REASv3.

| Sub-category |
| --- |
| Manure management[a] |
| Fertilizer application[b] |
| Industrial processes[c] |
| Human[d] |
| Latrines |

a. Contributions from manure management including housing, storage and yards were included. Those from manure applied to soils were included in fertilizer application. b. Contributions from both synthetic fertilizer and animal manure used as fertilizer were included. c. See Sect. S2.4.2. d. Contributions from perspiration and respiration were included.

**S2.5 Spatial and temporal resolution**

In REASv3, only large power plants are treated as point sources and gridded data of other emission sources are provided with a horizontal resolution of $0.25° \times 0.25°$. For temporal resolution, monthly emissions are estimated in REASv3 by allocating annual emissions to each month using monthly proxy data. Details of methodologies and data used for spatial and temporal allocation are described in Sect. S9.

Table 2.9 provides sub-sector categories included in aggregated sector codes for gridded data in REASv3.

**Table 2.9.** Sector codes for gridded data in REASv3 and sub-sector categories included in each code.

| Sector categories code | Sub-sector categories included in each sector code |
|---|---|
| POWER_PLANTS_POINT | Power plants (points) in Table 2.4 |
| POWER_PLANTS_NON-POINT[a] | Power plants (area sources and energy) in Table 2.4 |
| INDUSTRY[a] | Combustion sources of industry sector in Table 2.4 |
| | Non-combustion sources of industrial production and other transformation sector in Table 2.5 |
| ROAD_TRANSPORT | Road transport sector in Table 2.4 |
| | Evaporative NMVOC emissions from road vehicles described in Sect. S6.3 |
| OTHER_TRANSPORT | Other transport sector in Table 2.4 |
| DOMESTIC[a] | Residential and other domestic sectors in Table 2.4 |
| EXTRACTION | NMVOC emissions from extraction processes in Table 2.6 |
| SOLVENTGS | NMVOC emissions from solvent use in Table 2.7 |
| WASTE | NMVOC emissions from waste disposal described in Sect. S5.1.7 |
| MANURE_MANAGEMENT | $NH_3$ emissions from manure management described in Sect. S8.1 |
| FERTILIZER | $NH_3$ emissions from fertilizer application described in Sect. S8.2 |
| MISC | $NH_3$ emissions from human and latrines described in Sects. S8.4 and S8.5. |

a. For $CO_2$ gridded data of POWER_PLANTS_NON-POINT, INDUSTRY, and DOMESTIC, emissions excluding biofuel (-NON-BF) and those from biofuel (-BF) are provided separately.

**S3. Stationary combustion**

**S3.1 Activity data**

**S3.1.1 Definition of fuel types**

Table 3.1 describes fuel types considered in stationary combustion sources of REASv3. Emissions of air pollutants were estimated individually for each fuel type. In Fig. 4 of the main manuscript and Figs. S2, S4, S6, S8, S10, and S12 of the supplement, fuel types are aggregated to several categories. Definition of the categories is also provided in Table 3.1. For each fuel type, definitions are mostly the same as those of the International Energy Agency (IEA) World Energy Balances (IEAWEB) (IEA, 2017). Exceptions are "Raw coal", "Cleaned coal", "Other washed coal", and "Other coking products" which are defined only for China in the China Energy Statistical Yearbook (CESY) (National Bureau of Statistics of China, 1986, 2001-2017). Definition of "Bituminous coal", "Kerosene", and "Diesel oil" in Table 3.1. is the same as that of "Other bituminous coal", "Other kerosene", and "Gas/diesel oil excl. biofuels" of IEAWEB, respectively. For hard (brown) coal, if there is no detailed information, corresponding fuel type is considered as "Bituminous coal" ("Lignite"). For other fuel types not included in Table 3.1, emissions from combustion were ignored in REASv3.

**Table 3.1.** List of detailed fuel types considered in REASv3 and definition of aggregated categories used in the main manuscript and the supplement.

| Aggregated categories (code) | Aggregated categories (description) | Detailed fuel types |
| --- | --- | --- |
| COAL | Primary coal | Coking coal |
| | | Anthracite |
| | | Bituminous coal |
| | | Raw coal |
| | | Cleaned coal |
| | | Other washed coal |
| | | Sub-bituminous coal |
| | | Lignite |
| DC | Secondary coal | Coke oven coke |
| | | Gas coke |
| | | Coal tar |
| | | Patent fuel |
| | | Brown coal briquettes (BKB) |
| | | Other coking products |
| NGAS | Natural gas | Natural gas |
| OGAS | Other gas fuels | Gas works gas |
| | | Coke oven gas |
| | | Blast furnace gas |
| | | Other recovered gases |
| LF | Light oil fuels | Refinery gas |
| | | Liquefied petroleum gas (LPG) |
| | | Natural gas liquids |
| | | Motor gasoline |
| | | Naphtha |
| | | Kerosene |
| MD | Diesel oil | Diesel oil |
| HF | Heavy oil fuels | Crude oil |
| | | Heavy fuel oil |
| | | Petroleum coke |
| | | Other oil products |
| BF | Biofuel | Fuelwood |
| | | Crop Residue |

| | | | |
|---|---|---|---|
| | | Animal waste | |
| | | Biogas | |
| | | Biogasoline | |
| | | Biodiesels | |
| | | Charcoal | |
| OTH | Other fuels | Municipal waste (renewable) | |
| | | Municipal waste (non-renewable) | |
| | | Industrial waste | |

**S3.1.2 Data sources of fuel consumption and assumptions to estimate missing historical data**

In REASv3, fuel consumption data were primarily obtained from IEAWEB, CESY, the United Nations (UN) Energy Statistics Database (UN, 2016), and UN data, which is a web-based data service of the UN (http://data.un.org/). However, all these sources do not include data for the entire target period of REASv3, that is from 1950-2015. Furthermore, past data for sectors do not contain as many categories. In this sub-section, data sources and assumptions for estimating missing historical data used in REASv3 are summarized in Table 3.2 including how to distribute total or sub-total data to detailed sub-sectors and how to extrapolate data until 1950. Note that descriptions for fuel consumption data in transport sector are also included in this sub-section.

**Table. 3.2.** Data sources and assumptions for estimating missing historical data used in REASv3 for each country and region.

**(a) China**

| | |
|---|---|
| Data sources and treatments | ● Fuel consumption for each region except for Tibet, Hong Kong and Macau were obtained from CESY during 1985-2015 and those before 1984 were extrapolated to 1950 using data for whole China during 1950-2015. See Sect. S3.1.3 for regional fuel consumption data in China.

● Data of whole country were taken from IEAWEB during 1971-2015 and extrapolated to 1950. Those of Tibet were taken from REASv2 (based on GAINS ASIA at that time) during 2000-2008 and extrapolated using data of whole country. See (n) and (o) of this sub-section for Hong Kong and Macau, respectively. |
| Assumptions for estimating missing historical data | ● Assumptions for modifying IEAWEB during 1971-2015 are as follows:
  ➤ Energy industry own use sector:
    ✧ Data of bituminous coal and natural gas before 1989 were distributed to sub-sectors based on relative ratios of fuel consumption data in 1990.
    ✧ Fuel consumption data of coke oven gas in 1990 were extrapolated to 1980 using trends of coke oven gas production in IEAWEB during 1980-1990 and then, extrapolated to 1971 based on trends of coke oven coke production in IEAWEB during 1971-1980.
  ➤ Industry sector:
    ✧ Data of coking coal, gas works gas, coke oven gas, refinery gas, and LPG/other bituminous coal and crude oil/natural gas, other kerosene, diesel oil, and heavy fuel oil before 1989/1984/1979 were distributed to sub-sectors based on relative ratios in 1990/1985/1980.
    ✧ Fuel consumption data of coke oven gas in 1980 were extrapolated to 1971 using trends of coke oven gas production in IEAWEB during 1971-1980.
  ➤ Transport sector:
    ✧ Data of diesel oil before 1989 were distributed to road transport, domestic navigation, and agriculture/forestry based on relative ratios of corresponding fuel consumption in 1990.
● See "Assumption for data extrapolation" in this sub-section how to extrapolate the data of IEAWEB to 1950. |

**(b) India**

| Data sources and treatments | • Data of whole country were taken from IEAWEB during 1971-2015 and extrapolated to 1950.
 • See Sect. S3.1.4 for regional fuel consumption data in India. |
|---|---|
| Assumptions for estimating missing historical data | • No major modifications were done for IEAWEB during 1971-2015.
 • See "Assumption for data extrapolation" in this sub-section how to extrapolate the data of IEAWEB to 1950. |

**(c) Japan**

| Data sources and treatments | • Data of whole country were taken from IEAWEB during 1960-2015 and extrapolated to 1950.
 • See Sect. S3.1.5 for regional fuel consumption data in Japan. |
|---|---|
| Assumptions for estimating missing historical data | • Assumptions for modifying IEAWEB during 1960-2015 are as follows:
   ➤ Industry sector:
     ✧ Data of hard coal and coke oven coke/natural gas and LPG/crude oil/heavy fuel oil before 1974/1981/1965/1969 were distributed to sub-sectors based on relative ratios of fuel consumption data in 1975/1982/1966/1970.
   ➤ Residential and other sectors:
     ✧ Data of heavy fuel oil before 1969 were distributed to sub-sectors based on relative ratios in 1970.
   ➤ Other kerosene and diesel oil:
     ✧ Data of total final consumption before 1969 were distributed to sub-sectors based on relative ratios in 1970.
 • See "Assumption for data extrapolation" in this sub-section how to extrapolate the data of IEAWEB to 1950 except for following procedures:
   ➤ Consumption of hard coal, brown coal, patent fuel, coke oven coke, gas works gas, natural gas, and primary solid biofuels in residential sector were extrapolated to 1950 using the Historical Statistics of Japan (Japan Statistical Association, 2006).
   ➤ Consumption of primary solid biofuels in paper, pulp and printing industry before 1981 were extrapolated to 1950 based on trends of production amounts of paper and pulp in Japan (Economy, Trade and Industry Statistics Association, 1998). |

**(d) Republic of Korea**

| Data sources and treatments | • Data of whole country were taken from IEAWEB during 1971-2015 and extrapolated to 1950. |
|---|---|
| Assumptions for estimating missing historical data | • Assumptions for modifying IEAWEB during 1971-2015 are as follows:
  ➢ Industry sector:
    ✧ Data of coke oven coke/other kerosene, diesel oil, and heavy fuel oil/natural gas before 2001/1980/1992 were distributed to sub-sectors based on relative ratios of fuel consumption data in 2002/1981/1993.
  ➢ Transport and other sectors:
    ✧ Data of diesel oil and heavy fuel oil before 1980 were distributed to sub-sectors based on relative ratios in 1981.
  ➢ Residential and other sectors:
    ✧ Data of primary solid biofuels before 1989 were distributed to sub-sectors based on relative ratios in 1990.
• See "Assumption for data extrapolation" in this sub-section how to extrapolate the data of IEAWEB to 1950. |

**(e) Taiwan**

| Data sources and treatments | • Data of whole country were taken from IEAWEB during 1971-2015 and extrapolated to 1950. |
|---|---|
| Assumptions for estimating missing historical data | • Assumptions for modifying IEAWEB during 1971-2015 are as follows:
  ➢ Residential and other sectors:
    ✧ Data of diesel oil/heavy fuel oil before 1979/1981 were distributed to sub-sectors based on relative ratios of fuel consumption data in 1980/1982.
• See "Assumption for data extrapolation" in this sub-section how to extrapolate the data of IEAWEB to 1950. |

**(f) Indonesia**

| Data sources and treatments | ● Data of whole country were taken from IEAWEB during 1971-2015 and extrapolated to 1950. |
|---|---|
| Assumptions for estimating missing historical data | ● Assumptions for modifying IEAWEB during 1971-2015 are as follows:
➤ Industry sector:
  ✧ Data of other bituminous coal and sub-bituminous coal before 1999 were distributed to sub-sectors based on relative ratios of consumption data of sub-bituminous coal in 2000.
  ✧ Data of natural gas/diesel oil and heavy fuel oil before 1980/1988 were distributed to sub-sectors based on relative ratios of fuel consumption data in 1981/1989.
  ✧ Fuel consumption data of primary solid biofuels in 1990 were extrapolated to 1971 using trends of primary solid biofuels consumption data in the other sector in IEAWEB during 1971-1990.
➤ Transport, residential and other sectors:
  ✧ Data of heavy fuel oil after 2000 were distributed to sub-sectors based on relative ratios in 1999.
● See "Assumption for data extrapolation" in this sub-section how to extrapolate the data of IEAWEB to 1950. |

**(g) Myanmar**

| Data sources and treatments | ● Data of whole country were taken from IEAWEB during 1971-2015 and extrapolated to 1950. |
|---|---|
| Assumptions for estimating missing historical data | ● Assumptions for modifying IEAWEB during 1971-2015 are as follows:
➤ Industry sector:
  ✧ Data of other bituminous coal/diesel oil before 2010/2011 were distributed to sub-sectors based on relative ratios of fuel consumption data in 2011/2012.
● See "Assumption for data extrapolation" in this sub-section how to extrapolate the data of IEAWEB to 1950. |

**(h) Philippines**

| Data sources and treatments | ● Data of whole country were taken from IEAWEB during 1971-2015 and extrapolated to 1950. |
|---|---|
| Assumptions for estimating missing historical data | ● Assumptions for modifying IEAWEB during 1971-2015 are as follows:
➢ Industry sector:
    ✧ Data of diesel oil and heavy fuel oil before 1979 were distributed to sub-sectors based on relative ratios of fuel consumption data in 1980.
● See "Assumption for data extrapolation" in this sub-section how to extrapolate the data of IEAWEB to 1950. |

**(i) Singapore**

| Data sources and treatments | ● Data of whole country were taken from IEAWEB during 1971-2015 and extrapolated to 1950. |
|---|---|
| Assumptions for estimating missing historical data | ● Assumptions for modifying IEAWEB during 1971-2015 are as follows:
➢ Residential and other sectors:
    ✧ Data of natural gas before 2005 were distributed to sub-sectors based on relative ratios of fuel consumption data in 2006.
● See "Assumption for data extrapolation" in this sub-section how to extrapolate the data of IEAWEB to 1950. |

**(j) Thailand**

| Data sources and treatments | ● Data of whole country were taken from IEAWEB during 1971-2015 and extrapolated to 1950. |
|---|---|
| Assumptions for estimating missing historical data | ● Assumptions for modifying IEAWEB during 1971-2015 are as follows:
➢ Industry sector:
    ✧ Data of other bituminous coal/natural gas before 1988/2001 were distributed to sub-sectors based on relative ratios of fuel consumption data in 1989/2002.
● See "Assumption for data extrapolation" in this sub-section how to extrapolate the data of IEAWEB to 1950. |

**(k) Vietnam**

| Data sources and treatments | ● Data of whole country were taken from IEAWEB during 1971-2015 and extrapolated to 1950. |
|---|---|
| Assumptions for estimating missing historical data | ● Assumptions for modifying IEAWEB during 1971-2015 are as follows:
➢ Industry
   ✧ Data of anthracite, diesel oil and heavy fuel oil during 1980-2009 were distributed to sub-sectors based on relative ratios of corresponding fuel consumption data in 2010.
   ✧ Data of natural gas before 2009 were distributed to sub-sectors based on relative ratios in 2010.
   ✧ Data of other bituminous coal and lignite before 2009 were distributed to sub-sectors based on relative ratios of anthracite consumption data in 2010.
   ✧ Data of other bituminous coal and sub-bituminous coal after 2011 were distributed to sub-sectors based on relative ratios of anthracite consumption data in corresponding years of 2011-2015.
➢ Hard coal, diesel oil, and heavy fuel oil
   ✧ Data of total final consumption before 1979 were distributed to sub-sectors based on relative ratios in 1980.
● See "Assumption for data extrapolation" in this sub-section how to extrapolate the data of IEAWEB to 1950. |

**(l) Mongolia**

| Data sources and treatments | ● Data of whole country were taken from IEAWEB during 1985-2015 and extrapolated to 1950. |
|---|---|
| Assumptions for estimating missing historical data | ● No major modifications were done for IEAWEB during 1985-2015.
● See "Assumption for data extrapolation" in this sub-section how to extrapolate the data of IEAWEB to 1950. |

**(m) Cambodia**

| Data sources and treatments | ● Data of whole country were taken from IEAWEB during 1995-2015 and extrapolated to 1950. |
|---|---|
| Assumptions for estimating missing historical data | ● No major modifications were done for IEAWEB during 1995-2015.
● See "Assumption for data extrapolation" in this sub-section how to extrapolate the data of IEAWEB to 1950. |

**(n) Hong Kong, Democratic People's Republic of Korea, Brunei, Malaysia, Bangladesh, Nepal, Pakistan, and Sri Lanka**

| Data sources and treatments | • Data of whole country were taken from IEAWEB during 1971-2015 and extrapolated to 1950. |
|---|---|
| Assumptions for estimating missing historical data | • No major modifications were done for IEAWEB during 1995-2015.
 • See "Assumption for data extrapolation" in this sub-section how to extrapolate the data of IEAWEB to 1950. |

**(o) Macau, Laos, Afghanistan, Bhutan, and Maldives**

| Data sources and treatments | • Data of whole country were taken from UN data during 1990-2015 and extrapolated to 1950. |
|---|---|
| Assumptions for estimating missing historical data | • No major modifications were done for UN data during 1990-2015.
 • Data before 1990 were extrapolated to 1950 using trends of fuel consumption estimated using UN Energy Statistics Database as follows:
 Consumption = Production + Import – Export + Changes in stocks
 • Biofuel consumption data before 1970 were extrapolated to 1950 using trends of population numbers. |

**Assumption for data extrapolation**

As described above, fuel consumption data before 1959 and 1970 were not included in IEAWEB for Japan and other countries, respectively. The missing historical fuel consumption data were estimated by extrapolation using trends of related data for each sub-sector. Trend factors used in REASv3 are summarized in Table 3.3.

**Table. 3.3.** Trend factors for extrapolating fuel consumption data to 1950 in each sub-sector.

| Sub-sectors | Trend factors and data sources |
|---|---|
| Power plants including energy sector | • Trend factors: Amounts of generated power for all fuel types
 • Data sources:
 ➢ Each region of China: China Data Online
 ➢ Other countries and regions: Mitchell (1998) |
| Coke ovens and blast furnace including energy sector | • Trend factors: Amounts of pig iron production for all fuel types
 • Data sources: See Sect. S4.1.1 |
| Charcoal production plants | • Trend factors: Amounts of charcoal production for all fuel types
 • Data sources: Data after 1961 were obtained from FAOSTAT |

| | |
|---|---|
| | (http://www.fao.org/faostat/en) and trends between 1950 and 1960 were assumed based on Fernandes et al. (2007). |
| Petroleum Refineries including energy sector | ● Trend factors: Amounts of total crude oil consumption for all fuel types
● Data sources: Total crude oil consumption was estimated using Mitchell (1998) as follow: Consumption = Production + Import – Export |
| Iron and steel | ● Trend factors: Total amounts of pig iron and crude steel production for all fuel types
● Data sources: See Sect. S4.1.1 |
| Non-ferrous metals | ● Trend factors: Total amounts of copper, lead, zinc, and primary aluminum production for all fuel types
● Data sources: See Sect. S4.1.2 |
| Non-metallic minerals industry (cement, lime, and brick) | ● Trend factors: Amounts of cement production for all fuel types
● Data sources: See Sect. S4.1.3 |
| Railway | ● Trend factors: Length of railway line for all fuel types
● Data sources: Mitchell (1998) |
| Road transport | ● Trend factors: Total annual mileages of vehicles for each fuel type
● Data sources: See Sect. S6.1.1 |
| Others | ● Trend factors and data sources:
➢ Coal fuels except for coke fuels: Total coal consumption estimated using Mitchell (1998) as follows: Consumption = Production + Import – Export
➢ Coke fuels and gas fuels except for natural gas: The same trends as those for coke ovens
➢ Natural gas: Total natural gas consumption estimated using Mitchell (1998)
➢ Oil fuels: The same trends as those for petroleum refineries
➢ Biofuels: See Sect. S3.1.8
➢ Charcoal: The same trends as those for charcoal production plants
➢ Other fuels: Fuel consumption data were not extended to 1950. |

**S3.1.3 Regional fuel consumption data in China**

REASv3 used CESY for fuel consumption data of regions in China defined in Table 2.1 except for Hong Kong and Macau. However, in CESY, only total data are available in industry and transport sectors which need to be distributed to sub-sectors. In REASv3, weighting factors for the distribution were prepared for each region. Basic methodology and data used for the weighting factors are described briefly in this sub-section. Note that all motor gasoline listed in both industry and transport sectors of CESY are assumed to be consumed in road transport sector based on IEAWEB.

**Industry sector**

For most regions, total consumption data in industry sector were divided into sub-sectors based on weighting factors prepared using energy data in statistical yearbook of each region. Availabilities of detailed data for the weighting factors are different among regions and summarized in Table 3.4 except for Shanghai, Jiangsu, Zhejiang, Shandong, Hainan and Sichuan where no energy data are available in statistical yearbook of each region.

**Table. 3.4.** Data sources and treatments of weighting factors for each region to distribute total fuel consumption in industry sector to each sub-sector.

| Regions | Data sources and treatments |
|---------|------------------------------|
| Beijing | • Data of major fuel types were taken from Beijing Statistical Yearbook.
• For the year when statistics are not available, data in 2001/2005/2007/2010/2014 were used before 2000/for 2004/for 2008/for 2011/for 2015. |
| Tianjin | • Data of major fuel types were taken from Tianjin Statistical Yearbook.
• For the year when statistics are not available, data in 2001/2010/2013 were used before 2000/for 2011/after 2012. |
| Hebei | • Consumption of main energy sources were taken from Hebei Statistical Yearbook and used for all fuel types.
• For the year when statistics are not available, data in 2005/2010/2013 were used before 2004/for 2011/after 2012. |
| Shanxi | • Data of coal, coke, and diesel oil were taken from Shanxi Statistical Yearbook. For other fuels, weighting factors were based on data of REASv2 (based on GAINS ASIA at that time). |

| | |
|---|---|
| | ● For the year when Shanxi Statistical Yearbook are not available, data in 2000/2010/2013/2014 were used before 1999/for 2011/for 2012/for 2015. For REASv2 (available during 2000-2008), data in 2000/2008 were used before 1999/after 2009. |
| Inner Mongolia | ● Data of major fuel types were taken from Inner Mongolia Statistical Yearbook.
● For the year when statistics are not available, data in 2001/2007/2010/2013 were used before 2000/for 2006/for 2011/after 2012. |
| Liaoning | ● Data of major fuel types were taken from Liaoning Statistical Yearbook.
● For the year when statistics are not available, data in 2001/2010/2013 were used before 2000/for 2011/after 2012. |
| Jilin | ● Data of major fuel types were taken from Jilin Statistical Yearbook.
● For the year when statistics are not available, data in 2000/2002/2005/2010/2013 were used before 1999/for 2001/for 2004/for 2011/after 2012. |
| Heilongjiang | ● Data of major fuel types were taken from Heilongjiang Statistical Yearbook.
● For the year when statistics are not available, data in 2005/2010/2013 were used before 2004/for 2011/after 2012. |
| Shanghai | See descriptions below this table. |
| Jiangsu | See descriptions below this table. |
| Zhejiang | See descriptions below this table. |
| Anhui | ● Data of major fuel types were taken from Anhui Statistical Yearbook.
● For the year when statistics are not available, data in 2000/2002/2005/2010/2013 were used before 1999/for 2001/for 2004/for 2011/after 2012. |
| Fujian | ● Data of major fuel types were taken from Fujian Statistical Yearbook.
● For the year when statistics are not available, data in 2001/2010/2013 were used before 2000/for 2011/after 2012. |
| Jiangxi | ● Data of major fuel types were taken from Jiangxi Statistical Yearbook.
● For the year when statistics are not available, data in 2000/2010/2013 were used before 1999/for 2011/after 2012. |
| Shandong | See descriptions below this table. |

| | |
|---|---|
| Henan | • Data of major fuel types were taken from Henan Statistical Yearbook.
• For the year when statistics are not available, data in 2001/2010/2013 were used before 2000/for 2011/after 2012. |
| Hubei | • Data of coal and diesel oil were taken from Hubei Statistical Yearbook. For other fuels, weighting factors were based on data of REASv2 (based on GAINS ASIA at that time).
• For the year when Hubei Statistical Yearbook are not available, data in 2000/2010/2013 were used before 1999/for 2011/after 2012. For REASv2 (available during 2000-2008), data in 2000/2008 were used before 1999/after 2009. |
| Hunan | • Data of major fuel types were taken from Hunan Statistical Yearbook.
• For the year when statistics are not available, data in 2001/2005/2010/2013 were used before 2000/for 2004/for 2011/after 2012. |
| Guangdong | • Data of coal were taken from Guangdong Statistical Yearbook. For other fuels, weighting factors were based on data of REASv2 (based on GAINS ASIA at that time).
• For the year when Guangdong Statistical Yearbook are not available, data in 2000/2010/2013/2014 were used before 1999/for 2011/for 2012/for 2015. For REASv2 (available during 2000-2008), data in 2000/2008 were used before 1999/after 2009. |
| Guangxi | • Data of total energy consumption were taken from Guangxi Statistical Yearbook for all fuel types.
• For the year when statistics are not available, data in 1995/2000/2014 were used before 1997/for 1998 and 1999/for 2015. |
| Hainan | See descriptions below this table. |
| Chongqing | • Data of major fuel types were taken from Chongqing Statistical Yearbook.
• For the year when statistics are not available, data in 2001/2010/2013 were used before 2000/for 2011/after 2014. |
| Sichuan | See descriptions below this table. |
| Guizhou | • Data of major fuel types were taken from Guizhou Statistical Yearbook.
• For the year when statistics are not available, data in 2000/2010/2014 were used before 1999/for 2011/for 2015. |
| Yunnan | • Data of coal, coke, and oil were taken from Yunnan Statistical |

| | |
|---|---|
| | Yearbook. For other fuels, weighting factors were based on data of REASv2 (based on GAINS ASIA at that time).
● For the year when Yunnan Statistical Yearbook are not available, data in 2000/2013 were used before 1999/after 2014. For REASv2 (available during 2000-2008), data in 2000/2008 were used before 1999/after 2009. |
| Tibet | ● Fuel consumption data were not from CESY. (See Sect. S3.1.2) |
| Shaanxi | ● Data of coal, coke, and diesel oil were taken from Shaanxi Statistical Yearbook. For other fuels, weighting factors were based on data of REASv2 (based on GAINS ASIA at that time).
● For the year when Shanxi Statistical Yearbook are not available, data in 2002/2005/2010/2013 were used before 2001/for 2004/for 2009 and 2011/after 2012. For REASv2 (available during 2000-2008), data in 2000/2008 were used before 1999/after 2009. |
| Gansu | ● Data of major fuel types were taken from Gansu Statistical Yearbook.
● For the year when statistics are not available, data in 2001/2010/2013/2014 were used before 2000/for 2011/for 2012/for 2015. |
| Qinghai | ● Data of coal were taken from Qinghai Statistical Yearbook. For other fuels, weighting factors were based on data of REASv2 (based on GAINS ASIA at that time).
● For the year when Qinghai Statistical Yearbook are not available, data in 2001/2010/2013 were used before 2000/for 2011/after 2012. For REASv2 (available during 2000-2008), data in 2000/2008 were used before 1999/after 2009. |
| Ningxia | ● Data of major fuel types were taken from Ningxia Statistical Yearbook.
● For the year when statistics are not available, data in 2000/2010/2013 were used before 1999/for 2011/after 2012. |
| Xinjiang | ● Data of major fuel types were taken from Xinjiang Statistical Yearbook.
● For the year when statistics are not available, data in 2001/2007/2009/2013 were used before 2000/for 2008/for 2010 and 2011/after 2012. |
| Hong Kong | Fuel consumption data were not from CESY. (See Sect. S3.1.2) |
| Macau | Fuel consumption data were not from CESY. (See Sect. S3.1.2) |

For Shanghai, Jiangsu, Zhejiang, Shandong, Hainan and Sichuan, weighting factors were assumed based on sub-sector level fuel consumption data developed using the China total data described in Sect. S3.1.2 and related regional data as follows:

- Weighting factors to distribute fuel consumption in whole China to each region were prepared for each sub-sector and commonly used for all fuel types. The weighting factors for each sub-sector used in REASv3 are as follows:

  ➢ Amounts of steel production in each region (see Sect. S4.1.1) were used for iron and steel sub-sector.

  ➢ Total amounts of copper, lead, zinc, and primary aluminum production in each region (see Sect. S4.1.2) were used for non-ferrous metals sub-sector.

  ➢ Amounts of cement production in each region (see Sect. S4.1.3) were used for non-metallic minerals sub-sector in IEAWEB. (Fuel consumption in non-metallic minerals were further distributed to cement, lime, and brick sub-sectors in REASv3. See Sect. S3.1.7.)

  ➢ Amounts of coal production in each region taken from China Data Online were used for coal mines (in energy sector) and mining and quarrying sub-sectors.

  ➢ Amounts of paper and paperboard production in each region taken from China Data Online were used for paper, pulp and prints sub-sector.

  ➢ Amounts of textile production in each region (see Sect. S5.1.2) were used for textile and leather sub-sector.

  ➢ GDP of each region taken from China Data Online were used for other sectors.

- Using the China total data and the weighting factors, the tentative regional fuel consumption data (TRFCD) were developed. Then, the fuel consumption ratio of each sub-sector to industry sector total was calculated for Shanghai, Jiangsu, Zhejiang, Shandong, Hainan and Sichuan using the TRFCD of each region. Finally, fuel consumption in industry sector of each region in CESY was distributed to sub-sectors using the corresponding ratios. When categories of fuel types are different between the TRFCD and CESY, following procedures were adopted:

  ➢ For raw coal, cleaned coal, and other washed coal in CESY, the ratio for total of anthracite, coking coal and other bituminous coal in the TRFCD were used.

  ➢ For other coking products and other petroleum products in CESY, the ratio for coke oven coke and heavy fuel oil in the TRFCD were used, respectively.

**Transport sector**

For transport sector, no detailed data are available even in statistical yearbook of each region. Therefore, weighting factors for each region were assumed in the similar procedure for industry sector as follows:

- As mentioned in the first paragraph of this sub-section, all motor gasoline consumption (including those in industry sector) is distributed to road transport sector.
- All solid coal fuels are assumed to be used in railway sector.
- Natural gas consumption before and after 1995 was distributed to pipeline transport and road transport sectors, respectively.
- All heavy fuel oil consumption is distributed to domestic navigation sector.
- For diesel oil, using the same methodology for industry sector, diesel oil consumption data in road transport, railway, and domestic navigation sectors in each region were developed and then, weighting factors were assumed. For regional diesel oil consumption data, those in railway and domestic navigation sectors were taken from REASv2 (based on GAINS ASIA at that time) during 2000-2008 and data in 2000 and 2008 were used before 1999 and 2009, respectively. See Sect. S6.1.2 for diesel oil consumption in each region in road transport sector.
- Consumption of all other fuels is distributed to non-specified transport sector.
- Assumptions of motor gasoline, solid coal fuels, natural gas and heavy fuel oil described above were based on IEAWEB.

**S3.1.4 Regional fuel consumption data in India**

As defined in Table 2.1, REASv3 has 17 sub-regions for India. Therefore, fuel consumption data of country total based on IEAWEB need to be divided for each sub-region. Table 3.5 provides weighting factors used to allocate country total data to the 17 sub-regions.

**Table. 3.5.** Weighting factors for allocating country total fuel consumption data to the 17 sub-regions in India.

| Sectors and fuel types | Weighting factors and data sources |
|---|---|
| Power plants including energy sector | • Weighting factors: Total generation capacities in each region
• Data sources: World Electric Power Plants Database (Platts, 2018) |
| Iron and steel | • Weighting factors: Amounts of crude steel production for all fuel types
• Data sources: See Sect. S4.1.1 |
| Non-ferrous metals | • Weighting factors: Total amounts of copper, lead, zinc, and primary |

| | aluminum production for all fuel types |
|---|---|
| | ● Data sources: See Sect. S4.1.2 |
| Non-metallic minerals industry (cement, lime, and brick) | ● Weighting factors: Amounts of cement production for all fuel types
● Data sources: See Sect. S4.1.3 |
| Road | ● See Sect. S6.1 |
| Rail | ● Weighting factors: Length of railway line for all fuel types
● Data sources: Factors after 2005 were estimated from TERI (2013, 2018) and those in 2005 were used before 2004. |
| Biofuels | ● See Sect. S3.1.8 |
| Industry and energy sectors (default) | ● Weighting factors and data sources:
➢ Factors for LPG, motor gasoline, kerosene, diesel oil, heavy fuel oil, and naphtha during 1998-2013 were estimated from TERI (2013, 2018) and those in 1998 and 2013 were used before 1997 and after 2014, respectively.
➢ Factors for other fuels after 1999 were estimated from "Fuel Consumed" in Annual Survey of Industries (Ministry of Statistics & Programme Implementation, http://www.csoisw.gov.in/cms/en/1023-annual-survey-of-industries.aspx) and those in 1999 were used before 1998. |
| Residential and other domestic sectors | ● Weighting factors and data sources:
➢ Factors for kerosene and LPG during 1983-1999 were estimated from TERI (2013, 2018) and those in 1983 were used before 1982. The factors in 2010 were estimated based on Census of India 2011 (Chandramouli, 2011) and used after 2011. Factors between 1999 and 2010 were interpolated.
➢ Data of LPG were also used for natural gas. For other fuels, those of kerosene were used. |

**S3.1.5 Regional fuel consumption data in Japan**

REASv3 has 6 sub-regions for Japan as defined in Table 2.1 and the same as the case of India, fuel consumption data of country total based on IEAWEB need to be divided to each sub-region. Table 3.6 provides weighting factors used to allocate country total data to the 6 sub-regions.

**Table 3.6.** Weighting factors for allocating country total fuel consumption data to the 6 sub-regions in Japan.

| Sectors and fuel types | Weighting factors and data sources |
|---|---|
| Power plants including energy sector | • Weighting factors: Total generation capacities in each region
• Data sources: World Electric Power Plants Database (Platts, 2018) |
| Non-ferrous metals | • Weighting factors: Total amounts of copper, lead, zinc, and primary aluminum production for all fuel types
• Data sources: See Sect. S4.1.2 |
| Road | • See Sect. S6.1 |
| Others | • Weighting factors:
  ➢ Factors for each sector and fuel type during 1990-2015 were estimated using energy consumption statistics of each prefecture in corresponding years of 1990-2015.
  ➢ Factors in 1990 were used for those before 1989.
• Data sources:
  ➢ Website of the Agency for National Resources and Energy https://www.enecho.meti.go.jp/statistics/energy_consumption/ec002/results.html (in Japanese) |

**S3.1.6 Fuel consumption in power plants**

**General methodology**

In REASv3, power plants with following criteria were treated as point sources:

• Power plants which were treated as point sources in REASv2 (see Kurokawa et al., 2013).

• Power plants which entered commercial operation after 2008 and whose total generating capacities of units in each power plant were larger than 300MW.

Then, fuel consumption in power plants sector was estimated as follows:

1) Fuel consumption in each power plant (point source) was estimated. (see "Fuel consumption in each power plant" below)

2) (A) Total of the fuel consumption in each power plant was calculated in each country and region.

3) If (A) was larger than (B) fuel consumption in total power plant sector in a corresponding country and region, data of each power plants prepared in 1) were adjusted by the ratio of (B) to (A). In this case, fuel consumption of power plants as area sources was assumed to be zero.

4) IF (A) was smaller than (B), the value of (B) minus (A) was assumed to be fuel consumption in area sources. In this case, there is no change for the data of each power plant developed in 1).

**Fuel consumption in each power plant**

In REASv2, power plants whose annual $CO_2$ emissions in the Carbon Monitoring for Action (CARMA) Database (Wheeler and Ummel, 2008) were more than 1 Mt in 2000 and/or 2007 were treated as point sources. Before 2007, REASv3 used the same power plants as point sources with some revisions for such as generation capacities, fuel types, etc. using the updated World Electric Power Plants Database (Platts, 2018). For fuel consumption, data between 2000 and 2007 were basically the same as those in REASv2. Before 2000, fuel consumption of each power plant in operation was assumed to be the same as that in 2000 which will be adjusted based on total fuel consumption in power plants sector as described in "General methodology" above. (Note that power plants which were constructed and retired before 2000 were not considered in REASv3.) After 2008, REASv3 included power plants which entered commercial operation after 2008 as new point sources based on the WEPP (see also "General methodology" above). Although major information was available including fuel types used in each power plant, there are no data of fuel consumption in the WEPP. Thus, in REASv3, annual fuel consumption per generation capacity for each fuel type was estimated first using data in 2000 and 2007 for each country. The data were estimated for power plants which started operation before 1999 and after 2000, separately. Then, using the generation capacities data obtained from the WEPP, fuel consumption in each power plant was estimated.

**S3.1.7 Fuel consumption in non-metallic minerals**

REASv3 defined cement, lime, brick, and non-specified sub-sectors in the non-metallic minerals category in stationary combustion sources. However, energy statistics used in REASv3 including IEAWEB and regional statistical yearbook of China provide fuel consumption in total non-metallic minerals industry which needs to be distributed to each sub-sector.

In REASv3, all primary coal fuels were assumed to be used in cement, lime, and brick production. For China, Hua et al. (2016), Wang et al. (2012), and Streets et al. (2006) give coal consumption in cement (1980-2012), brick (1950-2015), and lime (2001) industries, respectively. Using these data and production amounts of cement, lime and brick, coal consumption per unit of production of cement, lime, and brick was estimated, respectively. Then, coal consumption data in non-metallic minerals in each region were distributed to each sub-sector based on production amounts of cement, lime, and brick in each region and corresponding coal consumption per united of production. Similarly, Maithel (2013) provides coal consumption in cement and brick industries in Pakistan

during 2001-2010 and with production amounts of cement and brick, fuel consumption in non-metallic minerals industry were distributed to each sub-sector. For other countries, due to lack of information, averaged coal consumption per unit of production of cement, lime, and brick for China was used for other East and Southeast Asian countries. For other countries in South Asia, averaged coal consumption per unit of production of cement and brick for Pakistan and that of lime for China was used. Then, with production data of cement, lime, and brick, fuel consumption in non-metallic minerals were distributed to each sub-sector. See Sects. S4.1.3, S4.1.4, and S4.1.5 for production data of cement, lime, and brick, respectively.

For other fuels, in REASv3, coke oven coke and heavy fuel oil were assumed to be used in cement industry and others including gas fuels and diesel oil were allocated to the non-specified sub-sector.

**S3.1.8 Biofuels**

**China**

CESY provides biofuel consumption data of fuelwood, crop residue, and biogas in each region during 1998-2007 which were used in REASv3. Before 1997, data were extended to 1980 using trends of each fuel consumption data in REASv1 and then extended to 1950 based on trends of biofuel consumption in East Asia obtained from Fernandes et al. (2007). After 2007, fuelwood, crop residue, and biogas consumption in total China were extrapolated to 2015 using trends of primary solid biofuels consumption in IEAWEB. Then, consumption of each fuel in each region in 2007 were tentatively extrapolated to 2015 using trends of rural population numbers in each region. Finally, fuelwood, crop residue, and biogas consumption in total China estimated during 2008-2015 were distributed to each region using the tentatively extrapolated data in each region.

**India**

Primary solid biofuels in IEAWEB were assumed to be total of fuelwood, crop residue and animal waste in India during 1971-2015. Before 1970, the primary solid biofuels consumption was extrapolated to 1950 using trends of biofuel consumption in South Asia obtained from Fernandes et al. (2007). Then, relative ratios of fuelwood, crop residue, and animal waste consumption in 17 sub-regions to consumption of the primary solid biofuels in total India were calculated for 1990 and 2010 using data in Streets and Waldhoff (1998) and Census of India 2011 (Chandramouli, 2011), respectively and interpolated between 1991 and 2009. Before 1989 and after 2011, the ratios of 1990 and 2010 were assumed to be constant, respectively. Finally, fuel consumption of fuelwood, crop residue, and animal waste in each sub-region during 1950-2015 were calculated.

**Japan**

Primary solid biofuels consumption in IEAWEB were assumed to be fuelwood consumption in Japan during 1982-2015. Before 1981, as described in Sect. S3.1.2, fuel consumption in residential and paper, pulp and printing industry sectors was extrapolated to 1950 using the Historical Statistics of Japan (Japan Statistical Association, 2006) and trends of production amounts of paper and pulp in Japan, respectively.

**Macau, Laos, Afghanistan, Bhutan, and Maldives**

See Sect. S3.1.2 for methodology and data sources. Only fuelwood and charcoal were included for this group.

**Other countries**

Primary solid biofuels data in IEAWEB were assumed to be total of fuelwood, crop residue and animal waste consumption in each country and extrapolated to 1950 using trends of biofuel consumption in East or Southeast or South Asia obtained from Fernandes et al. (2007). For distribution to each fuel type, consumption ratios of fuelwood, crop residue, and animal waste in 1990 obtained from Streets and Waldhoff (1998) were used during 1950-2015.

**S3.2 Emission factors and settings of emission controls**

**S3.2.1 SO$_2$**

**Sulfur contents in fuels**

In REASv3, default settings were taken from those of REASv1 during 1980-2000 generally based on RAINS ASIA at that time, Streets et al. (2000), Kato and Akimoto (1992) and Kato et al. (1991). For countries using default settings, data in 1980 and 2000 were used before 1979 and after 2001, respectively. For China, India, Japan, Republic of Korea, and Taiwan, additional country-specific settings were considered as described in Table 3.7.

**Table 3.7.** Settings and assumptions of sulfur contents in fuels for China, India, Japan, Republic of Korea, and Taiwan.

| Countries | Settings and assumptions |
|-----------|--------------------------|
| China | <li>Coal:<li>During 1985-2000: Data were taken from REASv1 based on Kato and Akimoto (1992) in 1985 and China Coal Industry Yearbook 2002 (State Administration for Coal Safety, 2003) in 1990 and 1995. In 2000, data in 1995 were adjusted so that the national average sulfur contents were 1.08% after Lu et al. (2010). Data in other years were interpolated.</li><li>During 2001-2005: Data were taken from REASv2 where settings of power plants in 2005 were based on Zhao et al. (2008) and national average sulfur contents were adjusted to 1.02% after Lu et al. (2010). Data between 2000 and 2005 were interpolated.</li><li>Before 1984 and after 2006, settings in 1980 and 2005 were used, respectively.</li></li><li>Oil<li>Before 1985, data were obtained from Kato et al. (1991) and those in 1995 were based on information from Tsinghua University (1.5% for heavy fuel oil and 0.58%, 0.35%, and 0.163% for diesel oil in north, northeast, and other areas, respectively) for REASv1. Data between 1986-1994 were interpolated and after 1996, data in 1995 were used.</li></li> |
| India | <li>Data were taken from REASv1 based on Reddy and Venkataraman (2002) for coal, heavy fuel oil, and light fuels and Kato et al. (1991) for others. The same data were used for the entire target period of REASv3.</li> |
| Japan | <li>Coal: Data during 1960-1996 were taken from Li and Dai (2000). The value in 1960 was 1.06% and gradually decreased to 0.60% in 1996. It was assumed that the value was reduced by 10% from 1996 to 2010 referring a report of MOEJ (2012). Data between 1996 and 2010 were interpolated and those in 1960 and 2010 were used before 1959 and after 2011, respectively.</li><li>Heavy fuel oil and crude oil: Settings during 1965-2010 for power plants were based on Iwaya (2013). Those for industry were based on Kato et al. (1991), Streets et al. (2000), and Imura et al. (1999). Data</li> |

| | in 1965 and 2010 were used before 1964 and after 2011, respectively. |
| | ➢ Heavy fuel oil for power plants: The values before 1965 were 2.6% and decreased almost constantly to 0.80% in 1975. Then the values were gradually decreased to 0.75% in 1990 and the values was used after 1990. |
| | ➢ Heavy fuel oil for industry: The values before 1965 were 2.60% and assumed to be decreased gradually to 1.4% in 1975, 1.1% in 1985, and 1.0% in 2000. The values after 2000 were assumed to be constant. |
| | ➢ Crude oil for power plants: The value before 1965 were 2.8% and decreased almost linearly to 0.20% in 1975. After 1975, values were between 0.15% and 0.20%. |
| | ● Diesel: Settings were based on regulations of diesel oil in Japan as follows: 1.2% before 1975, 0.50% during 1976-1991, 0.20% during 1992-1996, 0.05% during 1997-2003, and 0.005% after 2004. |
| Republic of Korea and Taiwan | ● Data during 1980-2000 were taken from REASv1 based on Kato et al. (1991), RAINS ASIA, and Streets et a. (2000) and those in 1975 were obtained from Kato et al. (1991). Data between 1976-1981 were interpolated and those in 1975 and 2000 were used before 1974 and after 2001, respectively. |

**Emission factors**

SO$_2$ emissions from coal and oil fuels were calculated using sulfur contents in fuels and ratios of sulfur emitted as SO$_2$. Settings of REASv3 for the fraction of sulfur in the fuel that is emitted as SO$_2$ were taken from REASv1 and REASv2 based on Kato and Akimoto (1992), Kato et al. (1991) and RAINS ASIA as follows:

● Power plants (point sources): 0.95
● Power plants (area sources)): 0.90 for Japan, Republic of Korea, and Taiwan; 0.775 for other countries and regions.
● Industry sector: 0.775
● Coke ovens: 0.0685
● Iron and steel: 0.1483
● Transport sector: 0.775
● Domestic sector: 0.60
● Coke oven coke for all sectors: 0.885

- Oil fuels for all sectors: 1.0

For coke ovens, activity data are coal input and it is considered that the estimated $SO_2$ emissions include both combustion and non-combustion sources.

For gas fuels such as coke oven gas and blast furnace gas, light fuels such as LPG, and other fuels except for primary biofuels such as charcoal and municipal wastes, emission factors were derived from Kato and Akimoto (1991). Those for fuelwood and crop residue were taken from Garg et al. (2001) and those for animal waste were from Gadi et al. (2003).

In cement plants, effects of absorption of $SO_2$ by cements need to be considered. In REASv3, the absorption rates for China were obtained from Li et al. (2017) and those for other countries were based on Kato et al. (1991).

**Settings of emission controls**

Settings and assumptions for reduction of $SO_2$ emissions from combustion sources by abatement equipment adopted in REASv3 are summarized in Table 3.8. For other sources not described in Table 3.8, no emission controls were considered.

**Table 3.8.** Settings and assumptions of emission controls of $SO_2$

| Countries | Settings and assumption |
|-----------|-------------------------|
| China | <li>Power plants: Effects of flue-gas desulfurization (FGD) were considered after 2000 as follows:<li>Settings during 2000-2008 were taken from REASv2 based on national introduction rates of FGD from Lu et al. (2010) and those of each province from Zhao et al. (2008).</li><li>After 2008, increases of penetration of FGD were assumed referring Liu et al. (2015) and Li et al. (2017). In 2015, the introduction rates were assumed to be 100% in power plants considered as point sources and 90% for other power plants. Removal efficiencies of FGD units were assumed to be 0.75 before 2003 and 0.90 after 2010 and the values were interpolated during 2004-2009.</li></li><li>Industry: Effects of FGD were roughly assumed as follows:<li>Referring Li et al. (2017), it was assumed that regulations started from (A) Beijing and Shanghai, then (B) Shandong, Hebei, and Guangdong, and finally (C) other provinces.</li><li>Regulations of industrial boiler were strengthened after 2014</li></li> |

| | |
|---|---|
| | referring Zheng et al. (2018). |
| | ➢ For (A), it was assumed that introduction of FGD started from 2000 and penetration rates in 2010 were 40% which is a setting for China in 2020 in Business-as-usual scenario of Wang et al. (2014). For the penetration rates, linear trends were assumed during 2000-2013. |
| | ➢ For (B) and (C), it was assumed that penetration of FGD started 2 and 4 years after (A), respectively and reduction effects were assumed to be smaller than (A) by 10% and 15%, respectively. It was assumed that removal efficiencies of FGD units were 0.75 for (A), 0.70 for (B) and 0.65 for (C). |
| | ➢ In 2015, total reduction rates of $SO_2$ emissions were assumed to be 75%, 63%, and 52% for (A), (B), and (C), respectively. |
| Japan | ● Power plants: Referring MRI (2015), Kato et al. (1991), and MOEJ (2000), effects of FGD were considered after 1968 as follows: |
| | ➢ In 1990 and after 2000, introduction rates of FGD in power plants as point sources were assumed to be 95% and 100%, respectively. It was assumed that removal efficiencies of FGD units were 0.95 after 1990. Trends of total reduction rates during 1968 and 1990 were assumed based on MOEJ (2000) and those between 1990 and 2000 were interpolated. |
| | ➢ For introduction rates of FGD in power plants as area sources, it was assumed to be 95% after 2000 and the trends before 1990 were estimated based on those of point sources. |
| | ● Other sectors: Referring Kato et al. (1991), total reduction rates of $SO_2$ emissions were assumed as follows: |
| | ➢ For large industries including sulphuric acid plants, 80% of reduction rates of power plants as area sources were adopted. |
| | ➢ For other industries, reduction rates were assumed to be 50% of large industries. |
| | ➢ For commercial and public services, 50% of reduction rates of other industries were adopted. |
| Republic of Korea | ● Effects of FGD were roughly assumed as follows: |
| | ➢ Power plants: Referring Ebata et al. (1997) and Wang et al. (2014), it is assumed that introduction of FGD was from 1990. The penetration rates in power plants as point sources in 2000, |

| | |
|---|---|
| | 2005, and 2010 were 90%, 97%, and 98%, respectively. Data between 1990, 2000, 2005, and 2010 were interpolated and data in 2010 were used after 2011. Effects of FGD on power plants as area sources were assumed to be 5% lower than point sources. Removal efficiencies of FGD units were roughly assumed to be 0.90 in 1990 and 0.95 after 2000 and the values were interpolated during 1991-1999.

➢ Industry: It was assumed that introduction of FGD started from 1990 and penetration rates of FGD were 80% and 85% in 2005 and 2010, respectively based on Wang et al. (2014). Data between 1990, 2005, and 2010 were interpolated and data in 2010 were used after 2011. It was assumed that removal efficiencies of FGD units were 0.95 for large industries and half of the values were adopted for other industries. |
| Taiwan | ● Effects of FGD were roughly assumed as follows:

➢ Power plants: Due to lack of information, the same reduction rates of Republic of Korea were adopted after 1995. But according to Ebata et al. (1997), introduction of FGD started earlier than Republic of Korea. It was assumed that penetration rates in 10% and 30% in 1980 and 1990, respectively and data between 1980, 1990, and 1995 were interpolated.

➢ Industry: Similar to power plants, the same reduction rates of Republic of Korea were adopted after 2000 and it was assumed that introduction of FGD started from 1985. Data between 1985 and 2000 were interpolated. |
| Thailand | ● Effects of FGD were assumed as follows:

➢ Power plants as point sources: Referring UN Environment (2018), reduction rates were assumed for four power plants as follows: Mae Moh (0.8-0.97 in 1978-2015), BLCP Power (0.84 from 2006), National Power Supply (0.75 from 1999), and GHECO-One (0.952 from 2012). |
| Other countries | ● Effects of FGD were assumed as follows:

➢ Power plants as point sources: Reduction rets (0.7-0.9) were assumed if units have information of installed FGD equipment in World Electric Power Plants Database (Platts, 2018).

➢ Countries which have power plants with FGD and number of such |

| | power plants in 2015 (in parentheses) in REASv3 were as follows: India (10), Indonesia (5), Laos (1), Malaysia (4), Vietnam (10), and Sri Lanka (2). |
|---|---|

**S3.2.2 NO$_x$**

**Default emission factors**

Table 3.9 summarized default emission factors used in REASv3 for fuel combustion in power plants, industry, and residential sectors. Specific settings for coke ovens, iron and steel industry, cement industry, and emission controls were described below the table.

**Table 3.9.** Default emission factors of $NO_x$ from fuel combustion in power plants, industry and residential sectors. Unit is t/PJ expressed as $NO_2$.

| Fuel type | Power plants | Industry | Residential |
|---|---|---|---|
| Hard coal[h] | 345[a] | 260[e] | 78[g] |
| Raw coal[i] | See Table 3.10. | 203[f] | 61.1[g] |
| Cleaned coal[i] | | 162[f] | 48.5[g] |
| Other washed coal[i] | | 509[f] | 153[g] |
| Sub-bituminous coal | 524[a] | A | B |
| Lignite | 433[a] | A | B |
| Coke oven coke[j] | 345 | 260 | 78 |
| Natural gas | 105[b] | 53[b] | 37[b] |
| Gas works gas | 10.5[b] | 7.4[b] | 5.25[b] |
| Coke oven gas | 77.8[b] | 55[b] | 38[b] |
| Blast furnace gas | 10.5[b] | 7.4[b] | 38[b] |
| LPG | 79[b] | 56[b] | 33[b] |
| Kerosene | 485[b] | 167[b] | 25[b] |
| Diesel oil | 632[b] | 222[b] | 74[b] |
| Crude oil | 249[b] | 145[b] | 49[b] |
| Heavy fuel oil | 249[b] | 145[b] | 49[b] |
| Fuelwood | 45[c] | | |
| Crop residue | 91.1[c] | | |
| Animal waste | 91.1[c] | | |
| Charcoal | 100[d] | | |

a. AP-42 (US EPA, 1995). b. Kato and Akimoto (1992). c. Streets and Waldhoff (1998), d. Revised 1996 IPCC guidelines (IPCC, 1997). e. Estimated based on ratios of emission factors between power plants and industry in Kato and Akimoto (1992). f. Estimated referring Zhang et al. (2007). g. 30% of emission factors of industry were adopted based on Kato and Akimoto (1992). h. Emission factors were commonly used for coking coal, anthracite and bituminous coal. i. Only defined for China. j. Emission factors for hard coal were adopted. A. Estimated based on ratios of emission factors between power plants and industry in Kato and Akimoto (1992) considering differences of net calorific values. B. 30% of emission factors of industry were adopted.

**Coke ovens**

For coal input to coke ovens, emission factor was 1.0 t/kt taken from Kato and Akimoto (1992). It is considered that $NO_x$ emissions estimated using this emission factor include contributions from both combustion and non-combustion processes.

**Iron and steel industry**

In iron and steel industry, emission factors for cokes, coke oven gas, and blast furnace gas were taken from Kato and Akimoto (1992) as follows:

- Coke oven coke: 4.0 t/kt for China and 2.5 t/kt for other countries
- Coke oven gas: 141 t/PJ
- Blast furnace gas: 76.4 t/PJ

For other fuel types, default emission factors were used.

**Cement industry**

For China, emission factors of coal combustion in each cement kiln type were obtained from Lei et al. (2011a) as follows: 15.3 t/kt for precalciner kilns, 18.5 t/kt for other rotary kilns, and 1.7 t/kt for shaft kilns. Coal consumption in each cement kiln type were estimated based on Lei et al. (2011a) and Hua et al. (2016). For other fuel types, default emission factors in industry were used.

For Japan, $NO_x$ emissions were not estimated based on fuel consumption, but using amount of cement production in each kiln type. Emission factors (t/kt of clinker produced) were taken from AP-42 (US EPA, 1995) as follows: 3.7 for wet process kilns, 3.0 for long dry process kilns, 2.4 for preheater process kilns and 2.1 for preheater/precalciner kilns. Ratio of clinker to cement was assumed to be 0.85 based on Cement handbook (Japan Cement Association, 2019). (See Sect. S4.1.3 for production data by different kiln types.)

For other countries and regions, default emission factors in industry were used for all fuel types.

**Settings of emission controls**

Settings and assumptions for reduction of $NO_x$ emissions from combustion sources by abatement equipment adopted in REASv3 are summarized in Table 3.10. For other sources not described in Table 3.10, no emission controls were considered.

**Table 3.10.** Settings and assumptions of emission controls of $NO_x$

| Countries | Settings and assumption |
|---|---|
| China | • Power plants

➤ Referring Zhang et al. (2007) and Liu et al. (2015), emission factors [t/PJ] for coal fired power plants were assumed considering effects of low-$NO_x$ burner based on capacity and years as follows:

◇ 227: Larger than 300 MW or equal to 300 MW after 1995.

◇ 300: Smaller than 300 MW but equal to or larger than 100 MW after 1997.

◇ 393: Equal to 300 MW before 1995 or Smaller than 300 MW but equal to or larger than 100 MW before 1997.

◇ 369: Less than 100 MW.

◇ 300: Power plants as area sources (no information of capacity) before 2000. The values were assumed to be decreased by 10% until 2010 and by 15% until 2015.

➤ Penetration rates of selective catalytic reduction (SCR: efficiency 73%) and selective non-catalytic reduction (SNCR: efficiency 30%) for each province in 2011 were taken from Chen et al. (2014). Referring Chen et al. (2014), Li et al. (2017), and Zheng et al. (2018), national introduction rates were assumed to be 12%, 18%, and 75% in 2010, 2011, and 2015 and reduction rates for as point sources were estimated. For area sources, 50% of reduction rates of point sources were adopted.

• Industry

➤ Referring Li et al. (2017), effects of De-$NO_x$ system were considered for precalciner kilns in cement plants and penetration rates were roughly assumed to be 0% in 2010, 50% in 2014 and 90% in 2015. |
| Japan | • Power plants: Referring MRI (2015), JMF and ICETT (2003), and MOEJ (2000), effects of low-$NO_x$ burner and SCR were considered as follows:

➤ Effects of low-$NO_x$ burner were considered after 1970 and reduction efficiencies were assumed to be 15%, 35%, and 50% in 1975, 1980, and after 2005, respectively. Data between 1970, 1975, 1980, and 2005 were interpolated. |

| | |
|---|---|
| | ➤ Effects of SCR were considered after 1974 and introduction rates in coal, oil, and gas power plants as point sources were assumed to be 80%, 40%, and 72% in 2002 and 90%, 45%, and 80% after 2010, respectively. Trends of the introduction rates during 1974 -2002 were assumed based on MOEJ (2000) and reduction rates during 2002-2010 were interpolated. For power plants as area sources, reduction rates were assumed to be 85% of point sources.

● Industry: Effects of low-$NO_x$ burner and SCR were roughly assumed referring MRI (2015) and Kato et al. (1991) as follows:

➤ It was assumed that trends of introduction rates of low-$NO_x$ burner were the same as for those of power plants, but reduction efficiencies were 50% of those for power plants as point sources.

➤ For large industries such as cement, iron and steel, it was assumed that trends of penetration rates of SCR were the same as those of power plants, but reduction efficiencies were 50% of those for power plants as point sources. For other industries, reduction rates were assumed to be 50% of those for large industries. |
| Republic of Korea/Taiwan | ● For power plants, introduction rates of low-$NO_x$ burner were 84% and 86% in 2005 and 2010, respectively and those of SCR (SNCR) were 56% (5%) and 68% (5%) in 2005, and 2010, respectively based on Wang et al. (2014). It was roughly assumed that low-$NO_x$ burner, SCR, and SNCR were installed from 1990 and their penetration rates in 2015 were 90%, 73%, and 5%, respectively. Reduction rates between 1990, 2005, 2010, and 2015 were interpolated.

● Due to lack of information, the same settings for Republic ok Korea were adopted to Taiwan. |
| Others | ● Effects of low-$NO_x$ burner and De-$NO_x$ system were assumed as follows:

➤ Power plants as point sources: Reduction rets (0.3-0.5) were assumed if units have information of installed De-$NO_x$ system in World Electric Power Plants Database (Platts, 2018).

● Countries which have power plants with De-$NO_x$ equipment and number of such power plants in 2015 (in parentheses) in REASv3 were as follows: India (11), Indonesia (5), Malaysia (6), Philippines (4), Singapore (4), Thailand (9), Vietnam (4), Pakistan (1), and Sri Lanka (2). |

**S3.2.3 CO**

**Default emission factors**

Table 3.11 summarized default emission factors used in REASv3 for fuel combustion in power plants, industry and residential sectors. Specific settings for coal combustion and, iron and steel industry, cement and other non-metallic minerals industries were described below the table.

**Table 3.11.** Default emission factors of CO from fuel combustion in power plants, industry and residential sectors. Unit is t/PJ.

| Fuel type | Power plants | Industry | Residential |
|---|---|---|---|
| Hard coal[e] | 20[a] | See "Emission factors for coal combustion" below. | |
| Raw coal[f] | 20[a] | | |
| Cleaned coal[f] | 20[a] | | |
| Other washed coal[f] | 20[a] | | |
| Sub-bituminous coal | 20[a] | | |
| Lignite | 20[a] | | |
| Coke oven coke | 20[a] | 150[a] | 2000[a] |
| Natural gas | 20[a] | 30[a] | 50[a] |
| Gas works gas | 20[a] | 150[a] | 150[a] |
| Coke oven gas | 20[a] | 150[a] | 150[a] |
| Blast furnace gas | 20[a] | 150[a] | 150[a] |
| LPG | 15[a] | 10[a] | 326[a] |
| Kerosene | 15[a] | 15[a] | 179[a] |
| Diesel oil | 15[a] | 15[a] | 20[a] |
| Crude oil | 15[a] | 15[a] | 20[a] |
| Heavy fuel oil | 15[a] | 15[a] | 20[a] |
| Fuelwood | 255.5[b] | 2555[c] | 5110[d] |
| Crop residue | 354.5[b] | 3545[c] | 7090[d] |
| Animal waste | 330[b] | 3300[c] | 6600[d] |
| Charcoal | 400[b] | 4000[a] | 7000[a] |

a. The global atmospheric pollution forum air pollutant emission inventory manual (SEI, 2012). b. Emission factors of power plants were assumed to be 10% of industry sector. c. Emission factors of industry sector were assumed to be 50% of residential sector. d. Streets and Waldhoff (1999). e. Emission factors were commonly used for coking coal, anthracite and bituminous coal. f. Only defined for China.

**Emission factors for coal combustion**

(a) Industry sector except for cement and other non-metallic minerals industries

Due to lack of information of detailed boiler and furnace types in industry sub-sectors in each country, CO emission factors of industry sector were roughly assumed in REASv3 as follows:

- 5.75 t/kt: average of emission factors for fluidized bed furnace and automatic stoker boiler based on AP-42 (US EPA, 1995).
  - ➢ Default emission factors for Japan, Republic of Korea, and Taiwan
  - ➢ Emission factors for large industries in China
- 18.6 t/kt: Emission factors for other industries in China estimated referring Streets et al. (2006) and data for fluidized bed furnace, automatic stoker, and hand-feed stoker in AP-42 (US EPA, 1995).
- 8.5 t/kt: Emission factors based on automatic stoker in AP-42 (UE EPA, 1995) were adopted for large industries in other countries.
- 66.25 t/kt: Emission factors based on average of automatic stoker and hand-feed stoker in AP-42 (UE UPA, 1995) for other industries in other countries.
- It was assumed that emission factors in China were decreased by 25% from 2000 to 2015 linearly assuming improvement in combustion efficiency.

(b) Residential sector

Emission factors for China, India, and other countries were assumed as follows:

- 75 t/kt for China obtained from Streets et al. (2006) for stove in residential sector.
- 275 t/kt for India taken from Pandey et al. (2014) for traditional stove in residential sector.
- 2.61 kt/PJ for other countries as default emission factor derived from the global atmospheric pollution forum air pollutant emission inventory manual (SEI, 2012)

**Coke production and iron and steel industry**

In REASv3, CO emissions from coke production and iron and steel industry were also estimated using production amounts of coke oven coke, sinter, pig iron, and crude steel (see Sects. S4.2.1 and S4.2.8). CO emission factors for coal consumption in coke ovens, those for coal and coke fuels in blast furnace, and coke furls and gas fuels in iron and industry sectors were assumed to be zero assuming their contributions were included in the emissions estimated based on production amounts described in Sects S4.2.1 and S4.2.8. These mean that CO emissions from combustion sources in coke production and iron and steel industry were not estimated separately in REASv3.

**Cement industries**

For China, emission factors of coal combustion in each cement kiln type were obtained from Lei et al. (2011a) as follows: 17.8 t/kt for precalciner kilns, 17.8 t/kt for other rotary kilns, and 155.7 t/kt for shaft kilns. Coal consumption in each cement kiln type were estimated based on Lei et al. (2011a) and Hua et al. (2016). For other fuel types, default emission factors in industry were used.

For Japan, CO emissions were not estimated based on fuel consumption, but using amount of cement production in each kiln type. Emission factors (t/kt of clinker produced) were taken from AP-42 (US EPA, 1995) as follows: 0.06 for wet process kilns, 0.11 for long dry process kilns, 0.49 for preheater process kilns and 1.8 for preheater/precalciner kilns. Ratio of clinker to cement was assumed to be 0.85 based on Cement handbook (Japan Cement Association, 2019). (See Sect. S4.1.3 for production data by different kiln types.)

For other countries and regions, 63.8 t/kt were used for emission factors for coal consumption in cement industry based on average of emission factors for precalciner kilns, other rotary kilns, and shaft kilns taken from AP-42 (US EPA, 1995). For other fuel types, default emission factors in industry were used.

**Other non-metallic minerals industries**

For lime industry, 155.7 t/kt were commonly used for coal combustion in all countries and default emission factors were used for other fuel types. For brick industry, 150 t/kt were used for coal combustion in China and default emission factors were adopted for Japan, Republic of Korea, and Taiwan. For other countries, emissions from brick industry were not estimated based on fuel combustion, but using amount of brick production. Emission factor 2.0 t/kt of brick produced was assumed based on Weyant et al. (2014) (See Sect. S4.2.5). For other sources, default emission factors were used.

**S3.2.4 PM species**

**Default emission factors**

Tables 3.12-14 summarized default emission factors of $PM_{10}$, $PM_{2.5}$, BC, and OC used in REASv3 for fuel combustion in power plants, industry, and residential sectors (Note that emissions of PM species from gas fuels were neglected in REASv3). Specific settings for biofuels, iron and steel industry, cement and other non-metallic minerals industries were described below the table.

**Table 3.12.** Default emission factors of $PM_{10}$, $PM_{2.5}$, BC, and OC from fuel combustion in power plants. Unit is t/kt.

| Fuel type | $PM_{10}$ | $PM_{2.5}$ | BC | OC |
|---|---|---|---|---|
| Hard coal[f] | 12.0[a] | 5.08[c] | 0.072[a] | 0.0[a] |
| Raw coal[g] | 46.0[b] | 12.0[b] | 0.024[b] | 0.0[b] |
| Cleaned coal[g] | 46.0[b] | 12.0[b] | 0.024[b] | 0.0[b] |
| Other washed coal[g] | 46.0[b] | 12.0[b] | 0.024[b] | 0.0[b] |
| Sub-bituminous coal | 29.0[a] | 9.3[c] | 0.174[a] | 0.0[a] |
| Lignite | 29.0[a] | 9.3[c] | 0.174[a] | 0.0[a] |
| Coke oven coke[h] | 12.0 | 5.08 | 0.072 | 0.0 |
| Diesel oil | 0.49[a] | 0.186[d] | 0.147[a] | 0.0441[a] |
| Crude oil[i] | 1.1 | 0.775 | 0.088 | 0.033 |
| Heavy fuel oil | 1.1[a] | 0.775[d] | 0.088[a] | 0.033[a] |
| Fuelwood | 2.2[e] | 1.79[e] | 0.11[e] | 0.44[e] |
| Crop residue[j] | 2.2 | 1.79 | 0.11 | 0.44 |
| Animal waste[j] | 2.2 | 1.79 | 0.11 | 0.44 |
| Charcoal | 4.1[e] | 3.32[e] | 0.205[e] | 0.82[e] |

a. Bond et al. (2004). b. Lei et al. (2011b). c. $PM_{2.5}/PM_{10}$ ratios were estimated based on AP-42 (US UPA, 1995). d. $PM_{2.5}/PM_{10}$ ratios were estimated based on Klimont et al. (2002b). e. Emission factors of $PM_{10}$, BC, and OC for fuelwood and charcoal were taken from Bond et al. (2004). $PM_{2.5}/PM_{10}$ ratios were estimated based on the global atmospheric pollution forum air pollutant emission inventory manual (SEI, 2012). f. Emission factors were commonly used for coking coal, anthracite and bituminous coal. g. Only defined for China. h. Emission factors for hard coal were adopted. i. Emission factors for heavy fuel oil were adopted. j. Emission factors for fuelwood were adopted.

**Table 3.13.** Default emission factors of $PM_{10}$, $PM_{2.5}$, BC, and OC from fuel combustion in industry sector. Unit is t/kt.

| Fuel type | $PM_{10}$ | $PM_{2.5}$ | BC | OC |
|---|---|---|---|---|
| Hard coal[f] | 4.2[a] | 1.79[c] | 0.84[a] | 0.168[a] |
| Raw coal[g] | 7.21[b] | 2.17[b] | 0.412[b] | 0.0868[b] |
| Cleaned coal[g] | 7.21[b] | 2.17[b] | 0.412[b] | 0.0868[b] |
| Other washed coal[g] | 7.21[b] | 2.17[b] | 0.412[b] | 0.0868[b] |
| Sub-bituminous coal | 17.0[a] | 7.23[c] | 0.85[a] | 1.7[c] |
| Lignite | 17.0[a] | 7.23[c] | 0.85[a] | 1.7[c] |
| Coke oven coke[h] | 4.2 | 1.79 | 0.84 | 0.168 |
| Kerosene | 0.9[a] | 0.341[d] | 0.117[a] | 0.09[a] |
| Diesel oil | 0.49[a] | 0.186[d] | 0.147[a] | 0.0441[a] |
| Crude oil[i] | 1.1 | 0.775 | 0.088 | 0.033 |
| Heavy fuel oil | 1.1[a] | 0.775[d] | 0.088[a] | 0.033[a] |
| Fuelwood | 6.1[e] | 4.95[e] | 0.555[e] | 3.22[e] |
| Crop residue[j] | 6.1 | 4.95 | 0.555 | 3.22 |
| Animal waste[j] | 6.1 | 4.95 | 0.555 | 3.22 |
| Charcoal | 4.1[e] | 3.32[e] | 0.205[e] | 0.82[e] |

a. Bond et al. (2004). b. Estimated based on Lei et al. (2011b) and Streets et al. (2006). c. $PM_{2.5}/PM_{10}$ ratio was estimated based on the global atmospheric pollution forum air pollutant emission inventory manual (SEI, 2012). OC/BC ratio was assumed based on ABC Emission Inventory Manual (Shrestha et al., 2013). d. $PM_{2.5}/PM_{10}$ ratios were estimated based on Klimont et al. (2002b). e. Emission factors of $PM_{10}$, BC, and OC for fuelwood and charcoal were taken from Bond et al. (2004). $PM_{2.5}/PM_{10}$ ratios were estimated based on the global atmospheric pollution forum air pollutant emission inventory manual (SEI, 2012). f. Emission factors were commonly used for coking coal, anthracite and bituminous coal. g. Only defined for China. h. Emission factors for hard coal were adopted. i. Emission factors for heavy fuel oil were adopted. j. Emission factors for fuelwood were adopted.

**Table 3.14.** Default emission factors of $PM_{10}$, $PM_{2.5}$, BC, and OC from fuel combustion in residential sector. Unit is t/kt.

| Fuel type | $PM_{10}$ | $PM_{2.5}$ | BC | OC |
|---|---|---|---|---|
| Hard coal[i] | 7.4[a] | 4.49[a] | 1.02[a] | 2.15[a] |
| Raw coal[j] | 8.82[b] | 6.86[b] | 1.56[b] | 3.29[b] |
| Cleaned coal[j] | 8.82[b] | 6.86[b] | 1.56[b] | 3.29[b] |

| | | | | |
|---|---|---|---|---|
| Other washed coal[j] | 8.82[b] | 6.86[b] | 1.56[b] | 3.29[b] |
| Sub-bituminous coal | 4.6[c] | 2.79[c] | 0.636[c] | 1.334[c] |
| Lignite | 4.6[c] | 2.79[c] | 0.636[c] | 1.334[c] |
| Coke oven coke[k] | 7.4 | 4.49 | 1.02 | 2.15 |
| LPG | 0.52[d] | 0.197[d] | 0.0676[d] | 0.052[d] |
| Kerosene | 0.9[d] | 0.341[d] | 0.117[d] | 0.09[d] |
| Diesel oil | 0.49[d] | 0.186[d] | 0.147[d] | 0.044[d] |
| Crude oil[l] | 1.1 | 0.775 | 0.088 | 0.033 |
| Heavy fuel oil | 1.1[d] | 0.775[d] | 0.088[d] | 0.033[d] |
| Fuelwood | 5.76[e], 4.80[f] | 5.58[e], 4.60[f] | 1.12[e], 0.85[f] | 4.46[e], 3.20[f] |
| Crop residue | 7.21[e], 6.01[f] | 6.98[e], 5.75[f] | 1.05[e], 0.95[f] | 3.98[e], 3.70[f] |
| Animal waste | 9.8[g] | 9.8[g] | 0.4[g] | 3.1[g] |
| Charcoal | 4.1[h] | 3.32[h] | 0.205[h] | 0.82[h] |

a. Estimated based on $PM_{10}$ emission factors for residential sectors in Bond et al. (2004) and ratios of $PM_{2.5}$, BC, and OC to $PM_{10}$ in Lei et al. (2011b). b. Estimated based on emission factors for stove in Lei et al. (2011b). c. Emission factor for $PM_{10}$ derived from Bond et al. (2004) and ratios of $PM_{2.5}$, BC, and OC to $PM_{10}$ were from those for hard coal. d. Bond et al. (2004) for $PM_{10}$, BC, and OC and $PM_{2.5}/PM_{10}$ ratios were estimated based on Klimont et al. (2002b). e. Estimated based on Lei et al. (2011b) and used for East Asian countries. f. Estimated based on Pandy et al. (2014) and used for Southeast and South Asian countries. g. Estimated based on Pandy et al. (2014) and commonly used for all countries. h. Emission factors of $PM_{10}$, BC, and OC were taken from Bond et al. (2004). $PM_{2.5}/PM_{10}$ ratios were estimated based on the global atmospheric pollution forum air pollutant emission inventory manual (SEI, 2012). i. Emission factors were commonly used for coking coal, anthracite and bituminous coal. j. Only defined for China. Values were gradually decreased from 1990 until their two third by 2005 referring Lei et al. (2011b). k. Emission factors for hard coal were adopted. l. Emission factors for heavy fuel oil were adopted.

**Coke production and iron and steel industry**

The same as for CO, in REASv3, emissions of PM species from coke ovens were also estimated base on production amounts of coke oven coke (see Sect. S4.2.8). Emission factors of PM species for coal consumption in coke ovens were assumed to be zero assuming their contribution were included in the emissions estimated based on production amounts of coke described in Sect. S4.2.8. For China, emissions of PM species from iron and steel production were also estimated base on

production amounts of sinter, pig iron, and crude steel (see Sect. S4.2.1). It was assumed that emission factors for sinter and pig iron production obtained from Lei et al (2011b) include emissions from coal combustion. Therefore, emission factors of PM species for coal combustion in iron and steel industry were assumed to be zero for China.

**Cement industry**

Emissions of PM species in China and Japan were not estimated based on fuel consumption, but using amount of cement production in each kiln type. For China, emission factors (t/kt of cement produced) of $PM_{10}$/$PM_{2.5}$/BC/OC were estimated based on Hua et al. (2016) and Lei et al. (2011a, b) as follows: 44.8/19.2/0.115/0.192 for precalciner kilns, 37.3/14.9/0.0894/0.149 for other rotary kilns, and 8.9/3.2/0.0192/0.032 for shaft kilns. For Japan, emission factors of $PM_{10}$/$PM_{2.5}$/BC/OC (t/kt of clinker produced) were taken from AP-42 (US EPA, 1995) and Kupiainen and Klimont (2004) as follows: 15.6/4.55/0.0273/0.0455 for wet process kilns, 35.9/15.4/0.0924/0.154 for long dry process kilns, 54.6/23.4/0.140/0.234 for preheater process kilns and preheater/precalciner kilns. Ratio of clinker to cement was assumed to be 0.85 based on Cement handbook (Japan Cement Association, 2019). (See Sect. S4.1.3 for production data by different kiln types.). For other countries and regions, default emission factors in industry were used for all fuel types. See Sect. S4.2.3 for non-combustion emissions from cement production.

**Brick industry**

Emissions of PM species from brick production were not estimated based on fuel combustion, but using amount of brick production. Emission factors of $PM_{10}$/$PM_{2.5}$/BC/OC were assumed referring Lei et al. (2011b), Weyant et al. (2014), and Klimont et al. (2017) as follows:
- China: 0.71/0.27/0.108/0.0945 t/kt of brick produced
- Japan, Republic of Korea, and Taiwan: 0.473/0.18/0.002/0.0035 t/kt of brick produced
- Other countries: 0.5/0.19/0.15/0.007 t/kt of brick produced

**Settings of emission controls**

Settings and assumptions for reduction of emissions of PM species from combustion sources by abatement equipment adopted in REASv3 are summarized in Table 3.15. For other sources not described in Table 3.15, no emission controls were considered. Note that the reduction rates of $PM_{2.5}$ were applied to BC and OC.

**Table 3.15.** Settings and assumptions of emission controls of PM species.

| Countries | Settings and assumption |
|---|---|
| China | • Power plants
  ➢ Effects of control technologies by cyclones, wet scrubbers, electrostatic precipitators (ESP), and fabric filters during 1990-2015 were estimated based on their penetration rates in Lei et al. (2011b) and Zhao et al. (2014).
  ➢ Reduction rates of $PM_{10}$/$PM_{2.5}$ were assumed to be 0.84/0.62, 0.92/0.78, and 0.98/0.94, and in 1990, 2000, and 2015, respectively. It was assumed that reduction rates before 1970 were zero and the values between 1970 and 1990 were interpolated.
• Industry
  ➢ Iron and steel industry: See Sect. S4.2.1
  ➢ Coke ovens: See Sect. S4.2.8.
  ➢ Non-ferrous metals industry: See Sect. S4.2.2
  ➢ Cement industry: See Sect. S4.2.3.
  ➢ Lime industry: See Sect. S4.2.4.
  ➢ Brick industry: See Sect. S4.2.5.
  ➢ Other industries: Due to lack of information, reduction rates were roughly assumed as follows: Reduction rates of $PM_{10}$ and $PM_{2.5}$ in 1990 were 0.55 and 0.25 referring settings of cement industry. Those in 2015 were 0.77 and 0.53 referring Wang et al. (2014) for settings of industry in 2010. It was assumed that reduction rates before 1980 were zero and the values between 1980, 1990, and 2015 were interpolated. |
| India | • Due to lack of information, referring Sadavarte and Venkataraman (2014), Pandey et al. (2014), Guttikunda and Jawahar (2014), and Reddy and Venkataraman (2002), reduction rates of $PM_{10}$/$PM_{2.5}$ for power plants and industries during 1980-2015 were roughly assumed |

| | as follows: |
|---|---|
| | ➢ Power plants: 0.0/0.0, 0.45/0.40, 0.85/0.81, and 0.87/0.85 in 1980, 1985, 2000, and 2015, respectively. Values between 1980, 1985, 2000, and 2015 were interpolated. |
| | ➢ Iron and steel and cement industries: 0.0/0.0, 0.47/0.46, and 0.85/0.83 in 1980, 1995, and 2015, respectively. Values between 1980, 1995, and 2015 were interpolated. |
| | ➢ Other industries: 0.0/0.0, 0.40/0.30, and 0.45/0.40 in 1980, 1995, and 2015, respectively. Values between 1980, 1995, and 2015 were interpolated. |
| Japan | ● Referring MRI (2015) and other literatures such as Shimoda (2016), Suzuki (1990) and Goto (1981), following assumptions were considered for control equipment of PM species:
 ➢ Introduction of control equipment for power plants was expanded from 1957.
 ➢ Introduction of bag filter was expanded from 1960.
 ➢ From 1968, installation of ESP in power plants became mandatory.
 ➢ Introduction of high quality ESP was expanded from 1975.
 ➢ Regulations for PM species were strengthened from 1995.
 ● Based on above assumption, reduction rates of $PM_{10}/PM_{2.5}$ for power plants were assumed as follows: 0.37/0.27, 0.9/0.88, and 0.995/0.99 in 1960, 1975, and after 2000, respectively. It was assumed that reduction rates before 1956 were zero and the values between 1950, 1960, 1975, and 2000 were interpolated.
 ● For industry, reduction rates of $PM_{10}/PM_{2.5}$ after 2000 were assumed to be 0.99/0.985 for iron and steel and cement industries and 0.98/0.96 for other industries. Trends between 1950 and 2000 were assumed to be the same as for those of power plants. |
| Republic of Korea/Taiwan | ● Power plants: Based on Wang et al. (2014), reduction rates of $PM_{10}$ and $PM_{2.5}$ after 2005 were assumed to be 0.985 and 0.97, respectively. Referring Ebata et al. (1997), it was assumed that penetration rates of control equipment of PM species in 1990 were already high. Reduction rates in 1990 were assumed to be 0.9 and 0.88 for $PM_{10}$ and $PM_{2.5}$, respective and zero before 1970. Values between 1970, 1990, and 2005 were interpolated. |

| | |
|---|---|
| | - Industry: Based on Wang et al. (2014), reduction rates of $PM_{10}/PM_{2.5}$ in 2005 and in 2010 were assumed to be 0.944/0.905 and 0.948/0.910, respectively. It was roughly assumed that reduction rates of $PM_{10}/PM_{2.5}$ in 2015 were 0.968/0.935, respectively and zero before 1970. Values between 1970, 2005, 2010, and 2015 were interpolated.

- Due to lack of information, the same settings for Republic of Korea were adopted. |
| Thailand | - Power plants: Referring Thao Pham et al. (2008), reduction rates of $PM_{10}$ and $PM_{2.5}$ in 2000 were assumed to be 0.84 and 0.80, respectively. For trends of reduction rates, it was roughly assumed that reduction rates of $PM_{10}$ and $PM_{2.5}$ were increased to 0.90 and 0.88 in 2015, respectively and zero before 1980. Values between 1980, 2000, and 2015 were interpolated.

- Industry: Referring Thao Pham et al. (2008), for iron and steel and cement industries, reduction rates of $PM_{10}$ and $PM_{2.5}$ in 2005 were assumed to be 0.82 and 0.80, respectively. For trends of reduction rats, it was roughly assumed that reduction rates of $PM_{10}$ and $PM_{2.5}$ in 2015 were 0.85 and 0.83, respectively and zero before 1980. Values between 1980, 2000, and 2015 were interpolated. For other industries, 50% of reduction rates of iron and steel and cement industries were adopted. |
| Others | - Due to lack of information, settings of Thailand during 1980-2005 were adopted for those of Indonesia, Malaysia, Myanmar, Philippines, Vietnam, and Mongolia during 1990-2015 and the same settings of Thailand were used for Singapore.

- For Laos and Sri Lanka, reduction rates of 0.95/0.92 for $PM_{10}/PM_{2.5}$ were used for large power plants equipped with ESP based on information from World Electric Power Plants Database (Platts, 2018), |

**S3.2.5 Other species and sources**

**NMVOC**

Emission factors for fossil fuel combustion were taken from REASv2 based on Wei et al. (2008) for East Asian countries and the global atmospheric pollution forum air pollutant emission inventory manual (SEI, 2012) for Southeast and South Asian countries. For fuelwood, crop residue, and animal waste, emission factors were estimated as follows:

- Fuelwood
  - 3.13 t/kt based on Wei et al. (2008) for East Asian countries
  - 15.9 t/kt based on Sharma et al. (2015) for Southeast and South Asian countries
- Crop residue
  - 8.36 t/kt based on Wei et al. (2008) for East Asian countries
  - 13.3 t/kt based on Sharma et al. (2015) for Southeast and South Asian countries
- Animal waste
  - 10.4 t/kt based on Sharma et al. (2015) for all countries
- Charcoal
  - 100 t/PJ taken from IPCC (1997) for all countries

Emission factors described above were for total NMVOC. In REASv3, total NMVOC emissions were allocated to 19 NMVOC species categories defined in Sect. S2.1. The speciation was conducted based on speciation profiles for each sub-sector and fuel type provided by D. G. Streets (private communication) generally based on Klimont et al. (2002a) used for REASv1 and REASv2. The speciation profiles were commonly used for all countries and periods.

**NH₃**

Emission factors for fossil fuel combustion were taken from REASv1 based on EMEP/CORINAIR Emission Inventory Guidebook (EEA, 1996). For biofuel, 1.29 t/kt for fuelwood and 0.97 t/kt for charcoal were obtained from ABC Emission Inventory Manual (Shrestha et al., 2013). Due to lack of information, the emission factor for fuelwood was adopted to crop residue and animal waste.

**CO₂**

Emission factors for fuel combustion except for fuelwood, crop residue, and animal wastes were obtained from 2006 IPCC Guidelines for National Greenhouse Gas Inventories (IPCC, 2006). Default emission factors were used except for those of coal combustion in China where lower values were adopted referring Guan et al. (2012). Emission factors for fuelwood, crop residue, and animal wastes were 83.1, 87.0, and 76.9 kt/PJ derived from Streets and Waldhoff (1999).

**Agriculture**

For emissions from fuel combustion in agriculture sub-sector, emission factors of industry sector were used except for following settings for diesel oil referring Bond et al. (2004) and ABC Emission Inventory Manual (Shrestha et al., 2013):

- 50.3 t/kt for $NO_x$
- 16.0 t/kt for CO
- 2.0 t/kt for $PM_{10}$
- 1.72 t/kt for $PM_{2.5}$
- 1.14 t/kt for BC
- 0.36 t/kt for OC

**Charcoal production**

Activity data to estimate emissions from charcoal production as energy transformation sectors is wood input. Fuelwood consumption data developed based on methodologies described in Sect. S3.1 were used. Emission factors of $NO_x$, CO, and NMVOC were taken from Revised 1996 IPCC guidelines (IPCC, 1997) and those of others were based on Akagi et al. (2011).

**S4. Stationary non-combustion: Industrial production and other transformation**

Descriptions for evaporative NMVOC emissions and $NH_3$ emissions from non-combustion sources are provided in Sects. S5 and S8, respectively.

**S4.1 Activity data**

**S4.1.1 Iron and steel production**

Activity data to estimate non-combustion emissions from iron and steel production industry in REASv3 are production amounts of pig iron, crude steel, sinter, and hot rolled products. National total production of pig iron, crude steel, and hot rolled products were obtained from Steel Statistical Yearbook (World Steel Association, https://www.worldsteel.org/steel-by-topic/statistics/steel-statistical-yearbook.html) during 1968-2015 and extrapolated to 1950 using trends of pig iron and crude steel production in Mitchell (1998). For crude steel, production data by each process, oxygen-blown converter, electric furnace, and open-hearth furnace were separately obtained. Sinter production data were taken from Steel Statistical Yearbook during 1977-1992. For China, sinter production data were available during 2000-2015 and those between 1992 and 2000 were interpolated. Then, missing data between 1950 and 2015 were estimated based on trends of pig iron production in each country.

For regional distribution in China, production amounts of steel during 1950-2015 and pig iron during 1983-2015 in each region were available in China Data Online and China Statistical Yearbook (National Bureau of Statistics of China, 1986-2016), respectively. Pig iron data before 1982 were extrapolated for each region using the trends of steel production in China Data Online. Then, using the steel data, production amounts of crude steel and hot rolled products in China total were distributed to each region. Similarly using the regional pig iron data, sinter and pig iron production amounts in whole China were distributed to each region. For India, ratios of crude steel production in 17 sub-regions were estimated using Minerals Yearbook (United States Geological Survey (USGS)) and Indiastat during 2000-2015. Using the regional data, production amounts of pig iron, crude steel, singer, and hot rolled products in India total were distributed to each sub-region. For Japan, ratios of steel production amounts in 6 sub-regions during 2003 and 2011 were estimated using statistics of major factories (https://www.japanmetaldaily.com/statistics/crudemateworks/details/index.html) and production data of pig iron, crude steel, singer, and hot rolled products in India total were distributed to each sub-region.

**S4.1.2 Non-ferrous metal production**

In REASv3, non-combustion emissions from copper, zinc, lead, and aluminum production were considered in non-ferrous metal production processes. Activity data were production amounts of primary copper, zinc, lead, alumina, aluminum, and secondary aluminum obtained from Minerals Yearbook during 1960-2015 (USGS) and extrapolated to 1950 using trends of corresponding production data in Mitchell (1998). For China, India, and Japan, national total data need to be distributed to each sub-region. Weighting factors for the distribution were estimated during 1995-2015 using annual generation capacities of major plants in Minerals Yearbook (USGS). Before 1994, the weighting factors for 1995 were used.

**S4.1.3 Cement production**

Activity data for non-combustion emissions from cement industry are production amounts of cement. For China, regional data were basically available in China Data Online during 1950-2015. However, not all regions had complete data during the period and sometimes interpolation and extrapolation procedures were necessary. Therefore, in REASv3, regional data were used for weighting factors to distribute national total data of cement production to each sub-region. For Japan, national cement production during 1990-2015 were obtained from Minerals Yearbook (USGS) and extrapolated to 1950 using trends of corresponding data in the Historical Statistics of Japan (Japan Statistical Association, 2006). For the distribution to each sub-region, first, weighting factors in 2004 and 2018 were estimated using production amounts by major cement plants. Then, those during 2005-2015 were interpolated and data in 2004 were used before 2003. In addition to total amounts, production data by different kiln types were available in China (Hua et al., 2016) and Japan (Japan Cement Association, http://www.jcassoc.or.jp/cement/2eng/index.html). For other countries, national total production during 1960-2015 were obtained from Minerals Yearbook (USGS). For extrapolation to 1950, in REASv3, trends of national $CO_2$ emissions from cement production taken from CDIAC (Carbon Dioxide Information Analysis Center) (Marland et al., 2008). For regional data in India, weighting factors during 1984 and 2009 were estimated using regional production data in TERI Energy & Environment Data Diary and Yearbook (TERI, 2013, 2018). Before 1983 and after 2010, data in 1984 and 2009 were used, respectively.

**S4.1.4 Lime production**

Activity data for non-combustion emissions from lime industry are production amounts of lime. Data were obtained from Minerals Yearbook during 1960-2015 (USGS) and were extrapolated to 1950 using trends of cement production estimated in REASv3.

**S4.1.5 Brick production**

Activity data for non-combustion emissions from brick industry are production amounts of brick. However, unlike the other products in non-metallic minerals industry, brick production data were not available in most international and national statistics. For Japan, national production data during 1950-2007 were taken from Hiragushi (2009) and Japan Statistical Yearbook (Statistics Bureau, 2010-2018) and were distributed to 6 sub-regions using total fuel consumption in non-metallic minerals sector. For other countries, first, default data were prepared taken from REASv2 and GAINS ASIA at that time during 1990-2015 and extrapolated to 1950 using trends of cement production in each country. For China, Vietnam, Bangladesh, India, and Pakistan, national production data in 1990, 2000, 2005, and 2010 were obtained from Klimont et al. (2017) and interpolated during 1990-2010 and extrapolated to 2015 using trends of the default data. For China, data between 1980-1990 were extrapolated based on trends of production in Zhang (1997) and those before 1980 were extrapolated using trends of the default data. For regional distribution, fuel consumption data in brick production in each region (see Sects. S3.1.3 and S3.1.7) were used for weighting factors. For India, data between 1983-1990 were extrapolated based on trends of production in Industrial Commodity Statistical Yearbook taken from UN data, which is a web-based data service of the UN (http://data.un.org/) and those before 1983 were extrapolated using trends of the default data. For regional distribution, common weighting factors during 1950-2015 were estimated based on Maithel et al. (2012). For Vietnam, Bangladesh, and Pakistan, data before 1990 were extrapolated using trends of the default data. For Nepal, production data in 2006 were obtained from Maithel (2013) and extrapolated during 1950-2015 using trends of the default data. For Rep. of Korea, Indonesia, Myanmar, the default data were used during 1990-2015 and before 1990, data were extrapolated to 1985 using trends of production in Industrial Commodity Statistical Yearbook and then extended to 1950 using trends of the default data. For other countries, the default data were directly used.

**S4.1.6 Sulphuric acid production**

Activity data to estimate non-combustion emissions from sulphuric acid plants are amounts of total sulphuric acid production in each country and region. For China, national total production data during 1950-2015 were obtained from China Data Online and distributed to each region using regional data during 1983-2015 in China Statistical Yearbook (National Bureau of Statistics of China, 1986-2016). Before 1983, data in 1983 were used as weighting factors for the regional distribution. For Japan, national production data were taken from statistics provided by the Sulphuric Acid Association of Japan (http://www.ryusan-kyokai.org/) during 1983-2015 and extrapolated to 1950 using trends of sulphuric acid production in Mitchell (1998). Weighting factors for regional distribution were estimated using annual generation capacities of major plants in 2015 in Minerals Yearbook (USGS). For other countries, national total production data were provided by the Sulphuric Acid Association of Japan during 1980-2015 and extrapolated to 1950 using trends of sulphuric acid production in Mitchell (1998). For India, national total data were distributed to 17 sub-regions using data of REASv2 during 2000-2008 based on GAINS ASIA at that time. For the weighting factors, data in 2000 and 2008 were used before 2000 and 2008, respectively.

**S4.1.7 Carbon black production**

In REASv3, non-combustion emissions from carbon black production were only considered for China, India, Japan, and, Rep. of Korea. Similar to brick production, default data were prepared taken from REASv2 and GAINS ASIA at that time during 1990-2015 and extrapolated to 1950 using GDP in each country and region. For GDP, regional data in China during 1950-2015 were obtained from China Data Online. For other countries, data during 1970-2015 were derived from UN data, which is a web-based data service of the UN (http://data.un.org/) and extrapolated to 1960 using OECD Data (https://data.oecd.org/gdp/gross-domestic-product-gdp.htm) and then extrapolated to 1950 using trends of total population.

For China, national total production in 2010 were obtained from Wei et al. (2011) and were extrapolated during 1950-2015 and distributed to each region using the default data as weighting factors. For India, national production data during 1983-2003 were taken from Industrial Commodity Statistical Yearbook taken from the UN data and similar to China, the data were extrapolated during 1950-2015 and distributed to each region using the default data. For Japan and Rep. of Korea, national production data during 1964-2014 were obtained from Mineral Yearbook (USGS) and extrapolated during 1950-2015 and data in Japan were distributed to 6 sub-regions using the default data.

**S4.1.8 Other transformation sectors**

**Coke ovens**

In REASv3, activity data to estimate emissions from coke ovens as energy transformation sectors are coal input for $SO_2$ and $NO_x$ and coke production for CO, NMVOC, $CO_2$, and PM species. Coal consumption was taken from data developed based on methodologies described in Sect. S3.1. For coke production, national data were obtained from the International Energy Agency (IEA) World Energy Balances (IEA, 2017) during 1960-2015 for Japan and 1971-2015 for other countries. The data were extrapolated to 1950 based on trends of pig iron production before 1959 and 1970 for Japan and other countries, respectively. For China, regional production data during 1990-2015 were available in the China Energy Statistical Yearbook (CESY) (National Bureau of Statistics of China, 1986, 2001-2017) and used to distribute national total production data to each sub-region. Before 1990, data in 1990 were used. For India and Japan, weighting factors for the regional distribution were based on regional pig iron production data in each country.

**Petroleum refineries**

Activity data to estimate emissions from petroleum refineries as energy transformation sectors is crude oil input. Consumption data of crude oil developed based on methodologies described in Sect. S3.1 were used.

**S4.2 Emission factors and settings of emission controls**

**S4.2.1 Iron and steel production**

**Emission factors**

In REASv3, emissions of CO, NMVOC, $CO_2$, and PM species were estimated using production amounts of sinter, pig iron, crude steel, and rolled steel. Default emission factors are summarized in Table 4.1 and emission factors of PM species for China are provided in Table 4.2. Note that emission factors of CO for all countries and those of PM species for China include contributions from both combustion and non-combustion emissions. (See also Sects. S3.2.3 and S3.2.4.)

**Table 4.1.** Default emission factors of CO, NMVOC, $CO_2$, $PM_{10}$, $PM_{2.5}$, BC, and OC from production of sinter, pig iron, crude steel, and rolled steel. It was assumed that both combustion and non-combustion emissions are included in emission factors of CO. Unit is t/kt-produced.

| | Sinter | Pig iron | Crude steel/ OHF[a] | Crude steel/ BOF[a] | Crude steel/ EF[a] | Rolled steel |
|---|---|---|---|---|---|---|
| CO | 22.0[b] | 40.5[c] | 34.5[d] | 69.0[b] | 9.0[b] | - |
| NMVOC[e] | - | - | 0.055 | 0.055 | 0.055 | 0.025 |
| $CO_2$[f] | - | - | - | - | 80.0 | - |
| $PM_{10}$[g] | 1.555 | 0.490 | 8.760 | 14.63 | 10.18 | - |
| $PM_{2.5}$[g] | 0.691 | 0.300 | 6.330 | 10.45 | 7.550 | - |
| BC[h] | 0.005 | 0.018 | - | - | - | - |
| OC[h] | 0.026 | - | - | 2.090 | 0.180 | - |

a. OHF: Open-hearth furnace, BOF: Basic oxygen furnace, and EF: Electric furnace. b. AP-42 (US EPA, 1995), c. Streets et al. (2006), d. 50% of BOF was adopted. e. Klimont et al. (2002a). f. IPCC (2006). g. Klimont et al. (2002b). h. Kupiainen and Klimont (2004).

**Table 4.2.** Emission factors of $PM_{10}$, $PM_{2.5}$, BC, and OC from production of sinter, pig iron, crude steel, and rolled steel for China. It was assumed that both combustion and non-combustion emissions are included (except for emission factors of PM species for crude steel production). Unit is t/kt-produced.

| | Sinter | Pig iron | Crude steel/ OHF[a] | Crude steel/ BOF[a] | Crude steel/ EF[a] | Rolled steel |
|---|---|---|---|---|---|---|
| CO[b] | 22.00 | 40.50 | 27.10[d] | 54.20 | 9.000 | - |
| $PM_{10}$[c] | 6.050 | 9.650 | 19.10 | 14.63 | 8.120 | - |
| $PM_{2.5}$[c] | 2.620 | 6.000 | 13.80 | 10.45 | 6.020 | - |
| BC[c] | 0.0262 | 0.600 | 0.138 | - | - | - |
| OC[c] | 0.131 | 0.120 | 0.690 | 2.090 | 0.120 | - |

a. OHF: Open-hearth furnace, BOF: Basic oxygen furnace, and EF: Electric furnace. b. Streets et al. (2006). c. Lei et al. (2011b). d. 50% of BOF was adopted.

For CO, the gas from blast furnace and basic oxygen furnace is collected and recycled in modern factories (Streets et al., 2006) and in REASv1, corresponding CO emissions in Japan were neglected. In REASv3, following settings were roughly assumed:

- China: Emission factors in Table 4.2 were used during 1950-2000 and 50% of the value was adopted in 2015. Emission factors between 2000 and 2015 were interpolated.
- Japan: Default emission factors were used before 1960 and 10% of the value was adopted after

1990. Emission factors between 1960 and1990 were interpolated.

- Republic of Korea and Taiwan: Default emission factors were used before 1975 and 10% of the value was adopted after 2005. Emission factors between 1975 and 2005 were interpolated.

**Settings of emission controls**

For iron and steel production, emission controls were only considered for PM species. Settings and assumptions for reduction of emissions in China by abatement equipment adopted in REASv3 are summarized in Table 4.3. For other countries, the same settings for combustion emissions in iron and steel industry were adopted. (See Table 3.15 in Sect. S3.2.4.)

**Table 4.3.** Settings and assumptions of emission controls of PM species for iron and steel production in China.

| Countries | Settings and assumption |
|---|---|
| China | • Referring Wu et al. (2017), reduction rates of $PM_{10}/PM_{2.5}$ for sinter production, pig iron, BOF, and EF in 2000, 2005, 2010, and 2015 were assumed as follows
➢ Sinter: 0.780/0.592, 0.892/0.809, 0.946/0.916, and 0.956/0.939
➢ Pig iron: 0.850/0.715, 0.910/0.844, 0.954/0.936, and 0.961/0.945
➢ BOF: 0.850/0.715, 0.870/0.758, 0.955/0.937, and 0.959/0.943
➢ EF: 0.782/0.568, 0.834/0.678, 0.900/0.815, and 0.977/0.968
• It was assumed that reduction rates were zero in 1980 and values between 1980, 2000, 2005, 2010, and 2015 were interpolated. |

**S4.2.2 Non-ferrous metal production**

In REASv3, emissions of $SO_2$, $PM_{10}$, and $PM_{2.5}$ were estimated using production amounts of copper, zinc, lead, and aluminum.

**$SO_2$**

Default emission factors were taken from Kato and Akimoto (1992) as follows:

- Copper: 2.0 kt/kt- produced
- Zinc: 1.0 kt/kt-produced
- Lead: 0.32 kt/kt-produced

In some countries, $SO_2$ emitted from non-ferrous metal plants were collected and used for

materials of sulphuric acid. In that case, the amounts of collected $SO_2$ need to be reduced from $SO_2$ emissions calculated by default emission factors. In REASv3, amounts of sulphuric acid produced using $SO_2$ collected from non-ferrous metal plants were obtained from the Sulphuric Acid Association of Japan based on reports of International Fertilizer Industry Association, the British Sulphur Cooperation Limited, Sulphuric Acid Notebook of Japan, and Kato et al. (1991). In addition, the same reduction rates of $SO_2$ by emission control equipment for non-ferrous metal industry were adopted.

**$PM_{10}$ and $PM_{2.5}$**

Default emission factors t/kt-produced were obtained from Lei et al. (2011b) for China and Klimont et al. (2002b) for other countries as follows:

China:
- Copper, Zinc, and Lead: 276.0 for $PM_{10}$ and 246.0 for $PM_{2.5}$
- Aluminum (primary): 26.51 for $PM_{10}$ and 18.28 for $PM_{2.5}$
- Aluminum (secondary): 6.98 for $PM_{10}$ and 5.20 for $PM_{2.5}$

Other countries:
- Copper, Zinc, and Lead: 13.8 for $PM_{10}$ and 12.3 for $PM_{2.5}$
- Aluminum (primary): 27.26 for $PM_{10}$ and 18.5 for $PM_{2.5}$
- Aluminum (secondary): 6.97 for $PM_{10}$ and 5.195 for $PM_{2.5}$

For emission controls, the same settings for combustion emissions in industry sectors were adopted except for China. (See Table 3.15 in Sect. S3.2.4.) For China, reduction rates were assumed as follows:

- Referring Zhao et al. (2014), reduction rates of $PM_{10}/PM_{2.5}$ in 2010 and 2015 were 0.910/0.882 and 0.945/0.906, respectively and values between 2010 and 2015 were interpolated.
- Trends of reduction rates between 1980 and 2010 were assumed to be the same as settings for combustion emissions in other industries. (See Table 3.15 in Sect. S3.2.4.)

**S4.2.3 Cement production**

In REASv3, emissions of $CO_2$ and PM species for all countries and those of $NO_x$ and CO for Japan were estimated using production amounts of cement. For emission of $NO_x$ and CO in Japan and those of PM species in China and Japan, emission factors for combustion emissions were described in Sects. S3.2.2, S3.2.3 and S3.2.4, respectively. In this sub-section, emission factors for non-combustion emissions were described.

Default emission factor of $CO_2$ was 0.52 t/t-clinker produced based on IPCC (2006). Clinker to

cement ratios were roughly assumed as follows:

- China: 0.72 before 2005 and 0.6 in 2015 based on Gao et al. (2017). Values between 2005 and 2005 were interpolated.
- India: 0.83 before 1990 and 0.77 after 2005 based on Barcelo (2014). Values between 1990 and 2005 were interpolated.
- Japan: 0.85 base on Cement handbook (Japan Cement Association, 2019)
- Others: 0.9 before 1990 and 0.85 after 2005 based on Barcelo (2014). Values between 1990 and 2005 were interpolated.

For PM species, default emission factors of $PM_{10}$, $PM_{2.5}$, BC, and OC t/kt-produced were assumed as follows:

- China: 34.3, 9.8, 0.0588, and 0.098 were taken from Hua et al. (2016) and Lei et al. (2011a).
- Others: 16.0, 4.64, 0.0278, and 0.0464 were derived from AP-42 (US EPA, 1995) and Lei et al. (2011a).

For emission controls, the same settings for combustion emissions in cement industry were adopted except for China. (See Table 3.15 in Sect. S3.2.4.) For China, reduction rates were assumed as follows:

- Referring Hua et al. (2016), reduction rates of $PM_{10}/PM_{2.5}$ during 1980-2012 were estimated for each year. Values were 0.565/0.218, 0.586/0.250, 0.746/0.527, and 0.973/0.916 in 1980, 1990, 2000, and 2012, respectively.
- It was roughly assumed that reduction rates of $PM_{10}/PM_{2.5}$ in 2015 were 0.98/0.97 and zero in 1975. Values between 1975 and 1980 and those between 2010 and 2015 were interpolated.

**S4.2.4 Lime production**

In REASv3, emissions of $CO_2$ and PM species were estimated using production amounts of lime. Default emission factors of $CO_2$ were taken from IPCC (2006) and those of PM species were derived from Klimont et al. (2002b) and Kupiainen and Klimont (2004) as follows:

- $CO_2$: 750 t/kt-produced
- $PM_{10}$: 12.0 t/kt-produced
- $PM_{2.5}$: 1.4 t/kt-produced
- BC: 0.028 t/kt-produced
- OC: 0.014 t/kt-produced

For emission controls of PM species, the same settings for combustion emissions in industry sectors were adopted except for China. (See Table 3.15 in Sect. S3.2.4.) For China, reduction rates were assumed as follows:

- Referring Zhao et al. (2014), reduction rates of $PM_{10}/PM_{2.5}$ in 2010 and 2015 were 0.766/0.670

and 0.782/0.697, respectively and values between 2010 and 2015 were interpolated.

- Trends of reduction rates between 1985 and 2010 were assumed to be the same as settings between 1980 and 2005 for combustion emissions in other industries. (See Table 3.15 in Sect. S3.2.4.)

**S4.2.5 Brick production**

In REASv3, emissions of CO and PM species were estimated using production amounts of brick.

For CO, note that emissions in China, Japan, Republic of Korea, and Taiwan were estimated using fuel consumption as described in Sect. S3.2.3. For other countries, emissions were estimated with production amounts of brick and emission factor 2.0 t/kt-produced was taken from Weyan et al. (2014).

For PM species, default emission factors of $PM_{10}$, $PM_{2.5}$, BC, and OC t/kt-produced were assumed as follows:

- China: 0.71, 0.27, 0.108, and 0.0945 were taken from Lei et al. (2011b).
- Japan, Republic of Korea, and Taiwan: Emission factors of tunnel kiln 0.4773, 0.18, 0.002, and 0.0035 were obtained from Klimont et al. (2017).
- Others: Emission factors of Bull's trench kiln 0.5, 0.19, 0.15, and 0.007 were based on Weyant et al. (2014).

For emission controls of PM species, the same settings for combustion emissions in industry sectors were adopted except for China. (See Table 3.15 in Sect. S3.2.4.) For China, reduction rates were assumed as follows:

- Referring Zhao et al. (2014), reduction rates of $PM_{10}$/$PM_{2.5}$ in 2010 and 2015 were 0.425/0.208 and 0.362/0.143, respectively and values between 2010 and 2015 were interpolated.
- Trends of reduction rates between 1985 and 2010 were assumed to be the same as settings for combustion emissions in other industries. (See Table 3.15 in Sect. S3.2.4.)

**S4.2.6 Sulphuric acid production**

In REASv3, emissions of $SO_2$ were estimated using production amounts of sulphuric acid. Default emission factors were taken from Kato et al. (1991) as follows:

- 20.0 t/kt-produced for China, Japan, Republic of Korea, and Taiwan
- 33.0 t/kt-produced for other countries.

For emission controls, the same settings for combustion emissions in large industries were adopted for Japan, Republic of Korea, and Taiwan and those for other industries were applied for China. For other countries, no emission controls were considered.

**S4.2.7 Carbon black production**

In REASv3, emissions of NMVOC and PM species were estimated using production amounts of carbon black. Default emission factor of NMVOC was taken from Klimont et al. (2002a) and those of PM species were derived from Klimont et al. (2002b) and Kupiainen and Klimont (2004) as follows:

- NMVOC: 90 t/kt-produced
- $PM_{10}$: 1.60 t/kt-produced
- $PM_{2.5}$: 1.44 t/kt-produced
- BC: 1.10 t/kt-produced
- OC: 0.00 t/kt-produced

For emission controls of PM species, the same settings for combustion emissions in industry sectors were adopted for all countries. (See Table 3.15 in Sect. S3.2.4.)

**S4.2.8 Other transformation sectors**

**Coke ovens**

In REASv3, emissions of CO, NMVOC, $CO_2$, and PM species were estimated using production amounts of coke oven coke.

For CO, emission factors were taken from Streets et al. (2006) as follows:
- 1.6 t/kt-produced for machinery coke ovens
- 15.6 t/kt-produced for indigenous coke ovens

Production amounts of coke oven coke in different technologies were only considered for China. Ratios of production amounts between machinery and indigenous coke ovens in each province in 2005 and 2006 were taken from China Industrial Economy Statistics Yearbook (National Bureau of Statistics, 2006-2007) and were extrapolated based on national ratios during 1990-2011 obtained from Huo et al. (2012a). It was roughly assumed that ratios of machinery coke ovens in 1970 were zero and gradually increased from 2011 to 2015. Data between 1970 and 1990 were interpolated. Due to lack of information, emission factors for machinery coke ovens were adopted for all other countries. As described in Sect. S3.2.3, emission factors were assumed to include contribution from combustion emissions.

Default emission factors of NMVOC was taken from Klimont et al. (2002a) and that of $CO_2$ was obtained from IPCC (2006) as follows:
- NMVOC: 1.44 t/kt-produced
- $CO_2$: 560 t/kt-produced

For PM species, default emission factors of $PM_{10}$, $PM_{2.5}$, BC, and OC t/kt-produced were assumed as follows:

- China: 8.79, 5.22, 1.57, and 1.83 were taken from Lei et al. (2011b).
- Others: 3.36, 2.00, 0.75, and 0.54 were taken from Klimont et al. (2002b) and Kupiainen and Klimont (2004).

As described in Sect. S3.2.4, emission factors were assumed to include contribution from combustion emissions. For emission controls of PM species, the same settings for combustion emissions in iron and steel industry were adopted except for China. (See Table 3.15 in Sect. S3.2.4.) For China, reduction rates were assumed as follows:

- Referring Zhao et al. (2014), reduction rates of $PM_{10}/PM_{2.5}$ in 2010 and 2015 were estimated for machinery and indigenous coke ovens as follows:
  - Machinery: 0.773/0.560 and 0.803/0.624 in 2010 and 2015, respectively.
  - Indigenous: 0.193/0.140 and 0.200/0.156 in 2010 and 2015, respectively.
  - Values between 2010 and 2015 were interpolated.
- Trends of reduction rates between 1985 and 2010 were assumed to be the same as settings for combustion emissions in other industries. (See Table 3.15 in Sect. S3.2.4.)

**Petroleum refineries**

In REASv3, emissions of $SO_2$, NMVOC and PM species were estimated using consumption amounts of crude oil in oil refinery industry. Default emission factors were derived from Kato and Akimoto (1992) for $SO_2$, Klimont et al. (2002a) for NMVOC, Klimont et al. (2002b) and Kupiainen and Klimont (2004) for PM species as follows:

- $SO_2$: 0.46S t/kt (S: Sulfur contents in fuel in wt%)
- NMVOC: 2.34 t/PJ
- $PM_{10}$: 1.20 t/kt
- $PM_{2.5}$: 0.96 t/kt
- BC: 0.00015 t/kt
- OC: 0.00 t/kt

For emission controls of $SO_2$ and PM species, the same settings for combustion emissions in industry sectors were adopted for all countries. (See Table 3.15 in Sect. S3.2.4.)

**S4.2.9 Speciation of NMVOC emissions**

Emission factors described in Sect. S4.2 were for total NMVOC. In REASv3, total NMVOC emissions were allocated to 19 NMVOC species categories defined in Sect. S2.1. The speciation was conducted based on speciation profiles for each sub-sector provided by D. G. Streets (private communication) generally based on Klimont et al. (2002a) used for REASv1 and REASv2. The speciation profiles were commonly used for all countries and periods.

**S5. Non-combustion sources of NMVOC**

In this section, activity data, emission factors, and their sources used to estimate evaporative NMVOC emissions in REASv3 are described. See Sect. S2.4.3 for sub-sector categories defined in REASv3. For Japan, NMVOC emissions from evaporative sources were derived from the Ministry of the Environment Japan (MEOJ, 2017a) and thus, activity data and emission factors of Japan were not compiled in REASv3 (see Sect. S5.3.1 for Japan).

**S5.1 Activity data**

In REASv3, activity data of REASv2 during 2000-2008 estimated based on Klimont et al. (2002a) were used as "default".

**S5.1.1 Extraction processes**

In REASv3, emissions from gas production and distribution, oil production and handling, petroleum refineries, service stations, and transport and depots are included in those from extraction processes. Data sources and treatments of activity data for each sub-sector category used in REASv3 were summarized in Table 5.1.

**Table 5.1.** Data sources and treatments of activity data for sub-sectors of extraction processes.

| Sub-sector categories | Data sources and treatments of activity data |
|---|---|
| Gas production and distribution | <li>Activity data: Natural gas production</li><li>Data sources and treatments:<li>China: Regional data during 1985-2015 were taken from the China Energy Statistical Yearbook (CESY) (National Bureau of Statistics of China, 1986, 2001-2017). Before 1985, data were extrapolated to 1971 using the International Energy Agency (IEA) World Energy Balances (IEAWEB) (IEA, 2017) and to 1950 using Mitchell (1998).</li><li>India: National total data were obtained from IEAWEB and extrapolated to 1950 using Mitchel (1998). For regional distribution, weighting factors were calculated using regional data taken from TERI (2013, 2018).</li><li>Other countries: National total data were derived from IEAWEB or the United Nations (UN) Energy Statistics Database (UN, 2016) and</li></li> |

| | extrapolated to 1950 using Mitchel (1998). |
|---|---|
| Crude oil production and handling | ● Activity data: Crude oil production
● Data sources and treatments:
  ➢ China: Regional data during 1950-2015 were derived from China Data Online.
  ➢ India: National total data were obtained from IEAWEB and extrapolated to 1950 using Mitchel (1998). For regional distribution, weighting factors were calculated using regional data taken from TERI (2013, 2018).
  ➢ Other countries: National total data were derived from IEAWEB or the UN Energy Statistics Database (UN, 2016) and extrapolated to 1950 using Mitchel (1998). |
| Petroleum refineries | ● Activity data: Consumption of crude oil in petroleum refineries
● Data sources and treatments: See Sect. S3.1. |
| Service stations | ● Activity data: Consumption of gasoline in road transport sector
● Data sources and treatments: See Sect. S3.1. |
| Transport and depots | ● Activity data: Consumption of gasoline and diesel in road transport sector
● Data sources and treatments: See Sect. S3.1. |

**S5.1.2 Solvent use**

In this sub-section, activity data of NMVOC evaporative emissions from solvent use except for printing (See Sect. S5.1.3) and paint application (See Sect. S5.1.4) were described. Data sources and treatments of activity data for each sub-sector category used in REASv3 were summarized in Table 5.2. (See Sect. S4.1.7 for data sources of GDP used in this sub-section.)

**Table 5.2.** Data sources and treatments of activity data for sub-sectors of solvent use.

| Sub-sector categories | Data sources and treatments of activity data |
|---|---|
| Dry cleaning | ● Activity data: Textiles cleaned
● Data sources and treatments:
  ➢ China: National total data in 2012 were taken from Wu et al. (2016) and extrapolated during 1950-2015 using trends of GDP. For regional distribution, urban population (see descriptions for domestic use of solvents in this table) were used as weighting factors. |

| | |
|---|---|
| | ➢ India: National data in 2010 were based on Sharma et al. (2015) and extrapolated during 1950-2015 using trends of GDP. For regional distribution, urban population were used as weighting factors. |
| | ➢ Other countries: Default data were used and extrapolated during 1950-2015 using trends of GDP. |
| Degreasing operation | ● Activity data: Solvent used |
| | ● Data sources and treatments: |
| | ➢ China: National total data in 2005 were taken from Wei et al. (2008). Regional distribution and extrapolation during 1950-2015 were conducted based on GDP. |
| | ➢ Other countries and regions: Default data were used during 2000-2008 and extrapolated during 1950-2015 using trends of GDP. |
| Vehicle treatment | ● Activity data: Cars registered |
| | ● Data sources and treatments: See Sect. S6.1.1. |
| Domestic use of solvents | ● Activity data: Urban and rural population |
| | ● Data sources and treatments: |
| | ➢ China: National and regional total population were obtained from China Data Online. Regional urban population data were calculated using proportion of urban population during 2005-2015 in China Statistical Yearbook (National Bureau of Statistics of China, 1986–2016) and the proportion data in 2005 for each region were used to estimated urban population before 2004. Then rural population in each region during 1950-2015 were calculated. |
| | ➢ India: National total population were taken from UN (2018). Regional ratios and proportion of urban population during 1951-2011 were estimated using data in Indiastat. Then, urban and rural population in each region were calculated. |
| | ➢ Other countries: National urban and rural population during 1950-2015 were derived from UN (2018). For Taiwan, population data were taken from Worldometer (https://www.worldometers.info/). |
| Asphalt blowing | ● Activity data: Asphalt produced |
| | ● Data sources and treatments: |
| | ➢ China: National total data in 2012 were taken from Wu et al. (2016) and extrapolated to 1950 using trends of Bitumen consumption in IEAWEB and GDP. Regional distribution was based on GDP. |
| | ➢ Other countries and regions: National and regional data were taken |

| | |
|---|---|
| | from default and extrapolated to 1950 using trends of Bitumen consumption in IEAWEB and GDP. |
| Paint production | ● Activity data: Paint produced
● Data sources and treatments:
➤ China: National total data during 2011-2013 were taken from Zheng et al. (2017).
➤ Other countries and regions: National data were taken from Industrial Commodity Statistical Yearbook.
➤ All countries and regions: Extrapolation for missing data and regional distribution were based on GDP. |
| Ink production | ● Activity data: Ink produced
● Data sources and treatments:
➤ China: National total data during 2011-2013 were taken from Zheng et al. (2017).
➤ Other countries and regions: National data were taken from Industrial Commodity Statistical Yearbook.
➤ All countries and regions: Extrapolation for missing data and regional distribution were based on GDP. |
| Tire production | ● Activity data: Tire produced
● Data sources and treatments:
➤ China: National total data during 2011-2013 were taken from Zheng et al. (2017).
➤ India: National data in 2010 were derived from Sharma et al. (2015).
➤ Other countries: National data were taken from Industrial Commodity Statistical Yearbook.
➤ All countries and regions: Extrapolation for missing data and regional distribution were based on GDP. |
| Synthetic rubber production | ● Activity data: Synthetic rubber produced
● Data sources and treatments:
➤ China: National total data during 2011-2013 were taken from Zheng et al. (2017).
➤ India: National data in 2010 were derived from Sharma et al. (2015).
➤ Indonesia: National data in 2010 were obtained from Permadi et al. (2017).
➤ Other countries: National data were taken from Industrial Commodity Statistical Yearbook. |

| | |
|---|---|
| | ➤ All countries and regions: Extrapolation for missing data and regional distribution were based on GDP. |
| Textile industry | ● Activity data: Textile produced
● Data sources and treatments:
➤ China: National total data during 2011-2013 were derived from Zheng et al. (2017).
➤ Other countries and regions: National and regional data were taken from default.
➤ All: Extrapolation for missing data and regional distribution for China were based on GDP. |
| Preservation of wood | ● Activity data: Wood treated
● Data sources and treatments:
➤ All: National and regional data were taken from default and extrapolated during 1950-2015 using trends GDP. |
| Adhesive application | ● Activity data: Adhesive consumed
● Data sources and treatments:
➤ China: National total data in 2005 and 2010 were taken from Wei et al. (2008; 2011).
➤ India: National data in 2010 were derived from Sharma et al. (2015).
➤ Indonesia: National data in 2010 were obtained from Permadi et al. (2017).
➤ Other countries: National data were taken from default.
➤ All countries and regions: Extrapolation for missing data and regional distribution were based on GDP. |

**S5.1.3 Printing**

In REASv3, NMVOC evaporative emissions from following four printing activities are considered: packing, offset printing, publication, and screen printing. Activity data are ink consumption for each purpose. In this sub-section, data sources and treatments of activity data used in REASv3 were described.

National total ink consumption data were calculated as default for this sub-section using production, export, and import amounts taken from Industrial Commodity Statistical Yearbook and missing data were extrapolated based on GDP. For China, national total ink consumption in 2005, 2010, and 2012 were derived from Wei et al. (2008, 2011) and Wu et al. (2016) and interpolated during 2005 and 2012. Before 2005 and after 2012, the data were extrapolated based on the default

data. For Indonesia, national total ink consumption data in 2010 were obtained from Permadi et al. (2017) and extrapolated during 1950-2015 based on the default data. For India, national ink consumption amounts in 2010 are available for packing, offset printing, publication, and screen printing in Sharma et al. (2015). The data were extrapolated during 1950-2015 based on the default data. For distribution of total ink consumption to each purpose except for India and regional distribution of national total data in China and India, activity data of REASv2 during 2000-2008 were used as weighting factors. Before 1999 and 2009, data in 2000 and 2008 were used respectively.

**S5.1.4 Paint application**

In REASv3, NMVOC evaporative emissions from paint application were considered for following purposes: architecture, domestic usage, automobile manufacture, vehicle refinishing, and other industrial applications. In this sub-section, data sources and treatments of activity data used in REASv3 were described.

National total paint consumption data during 2000-2009 were taken from a report of Information Research Limited and missing data were extrapolated during 1950-2015 based on GDP. For China, national total paint application data in 2005, 2010, and 2012 were derived from Wei et al. (2008, 2011) and Wu et al. (2016) and interpolated during 2005 and 2012. Before 2005 and after 2012, the data were extrapolated based on GDP. For India and Indonesia, national total paint consumption data in 2010 were obtained from Sharma et al. (2015) and Permadi et al. (2017), respectively and extrapolated during 1950-2015 based on GDP. The total paint consumption data were distributed to each purpose described above except for automobile manufacture using activity data of REASv2 during 2000-2008 as weighting factors. Before 1999 and after 2010, data in 2000 and 2008 were used respectively.

For automobile manufacture, activity data are production number of small and large vehicles. Production data of passenger vehicles (treated as small vehicles), bus and trucks (considered as large vehicles) in Asian countries during 2013-2015 were derived from the Japan Automobile Manufacture Association, Inc. (http://www.jama-english.jp/). Data of India and Republic of Korea were extrapolated to 1999 using data taken from Global Note (https://www.globalnote.jp/). Production number of passenger and duty vehicles were obtained from Michell (1998) and missing data were interpolated. For China, regional data during 1980-2015 were obtained from China Statistical Yearbook (National Bureau of Statistics of China, 1986–2016) and extrapolated to 1950 using national data in China Data Online.

**S5.1.5 Chemical industry**

Activity data of NMVOC evaporative emissions from chemical industry were described in this sub-section. Data sources and treatments for each sub-sector category used in REASv3 were summarized in Table 5.3. (See Sect. S3.1 for energy consumption in chemical industry sub-sector and Sect. S4.1.7 for data sources of GDP used in this sub-section.)

**Table 5.3.** Data sources and treatments of activity data for sub-sectors of Chemical industry.

| Sub-sector categories | Data sources and treatments of activity data |
|---|---|
| Ethylene production | ● Activity data: Ethylene produced
● Data sources and treatments:
➢ China: Regional data during 2004-2015 were extrapolated to 1978 using national data both obtained from China Statistical Yearbook (National Bureau of Statistics of China, 1986–2016). The data were extrapolated to 1950 based on total energy consumption in chemical industry sub-sector.
➢ India: National data in 2010 were derived from Sharma et al. (2015) and Industrial Commodity Statistical Yearbook during 1983-2003. Data between 2003 and 2010 were interpolated and missing data were extrapolated based on total energy consumption in chemical industry sub-sector. For regional distribution, the default data were used as weighting factors.
➢ Other countries and regions: National data before 1983 were taken from Industrial Commodity Statistical Yearbook and TOZAI BOEKI TSUSHINSHA (2014a). Missing data were interpolated and extrapolated based on total energy consumption in chemical industry. |
| Polyethylene production | ● Activity data: `Polyethylene produced
● Data sources and treatments:
➢ China: National data before 1985 were taken from Industrial Commodity Statistical Yearbook and TOZAI BOEKI TSUSHINSHA (2014b). For regional distribution, data of ethylene were used as weighting factors.
➢ Other countries and regions: National data before 1983 were taken from Industrial Commodity Statistical Yearbook and TOZAI BOEKI TSUSHINSHA (2014a). For regional distribution in India, the default |

| | |
|---|---|
| | data were used as weighting factors. |
| Styrene production | ● Activity data: Styrene produced |
| | ● Data sources and treatments: |
| | ➢ National data during 2008-2013 in China and those during 2009-2015 were obtained from TOZAI BOEKI TSUSHINSHA (2014b; a). Extrapolation during 1950-2015 and regional distribution for China and India were conducted based on data of ethylene. |
| Polystyrene production | ● Activity data: Polyethylene produced |
| | ● Data sources and treatments: |
| | ➢ China: National data in 2010 were obtained from Wei et al. (2011). The data were extrapolated to 1950 and distributed to each region using data of ethylene. |
| | ➢ India: National data in 2010 were derived from Sharma et al. (2015). The data were extrapolated to 1950 and distributed to each region using data of ethylene. |
| | ➢ Other countries and regions: National data before 1983 were taken from Industrial Commodity Statistical Yearbook and TOZAI BOEKI TSUSHINSHA (2014a). Missing data were interpolated and extrapolated based on data of ethylene. |
| Polyvinylchloride production | ● Activity data: Polyvinylchloride produced |
| | ● Data sources and treatments: |
| | ➢ China: National data during 2008-2013 were obtained from TOZAI BOEKI TSUSHINSHA (2014b). The data were extrapolated to 1950 and distributed to each region using data of ethylene. |
| | ➢ India: National data in 2010 were derived from Sharma et al. (2015). The data were extrapolated to 1950 and distributed to each region using data of ethylene. |
| | ➢ Other countries and regions: National data before 1983 were taken from Industrial Commodity Statistical Yearbook and TOZAI BOEKI TSUSHINSHA (2014a). Missing data were interpolated and extrapolated based on data of ethylene. |
| Propylene production/ Polypropylene production | ● Activity data: Propylene produced/Polypropylene produced |
| | ● Data sources and treatments: |
| | ➢ China: National data during 2008-2013 were obtained from TOZAI BOEKI TSUSHINSHA (2014b) and extrapolated to 1950 using data of ethylene. |

| | ⮞ Other countries and regions: National data before 1983 were taken from Industrial Commodity Statistical Yearbook and TOZAI BOEKI TSUSHINSHA (2014a). Missing data were interpolated and extrapolated based on data of ethylene. Regional distribution for China and India were conducted also based on data of ethylene. |
|---|---|
| Storage of organic chemicals | • Activity data: Total production of organic chemicals
• Data sources and treatments: See descriptions for organic chemicals in this table. |
| Polyvinylchloride processing | • Activity data: Polyvinylchloride produced
• Data sources and treatments: The same as for "Polyvinylchloride production" |
| Polystyrene processing | • Activity data: Polyethylene produced
• Data sources and treatments: The same as for "Polystyrene production" |
| Carbon black | • Activity data: Carbon black produced
• Data sources and treatments: See Sect. S4.1.7. |

**S5.1.6 Other industry**

In this sub-section, activity data of NMVOC evaporative emissions from other industrial processes were described. Data sources and treatments for each sub-sector category used in REASv3 were summarized in Table 5.4. (See Sect. S4.1.7 for data sources of GDP used in this sub-section.)

**Table 5.4.** Data sources and treatments of activity data for sub-sectors of other industry.

| Sub-sector categories | Data sources and treatments of activity data |
|---|---|
| Bread production | • Activity data: Bread produced
• Data sources and treatments:
  ⮞ China: National total data in 2012 were taken from Wu et al. (2016).
  ⮞ India: National data in 2010 were derived from Sharma et al. (2015).
  ⮞ Other countries: National data were taken from Industrial Commodity Statistical Yearbook.
  ⮞ All countries and regions: Extrapolation for missing data were based on population (see descriptions for domestic use of solvents in Sect. S5.1.2). For regional distribution of China and India, the default data were used as weighting factors. |
| Beer production | • Activity data: Beer produced |

| | |
|---|---|
| | ● Data sources and treatments:
➢ China: Regional data during 1983-2015 were obtained from China Statistical Yearbook (National Bureau of Statistics of China, 1986–2016) and extrapolated to 1950 using Mitchell (1998).
➢ Other countries: National data after 2006 were taken from Brewers Association of Japan (http://www.brewers.or.jp/english/index.html) and before 1993 were obtained from Mitchell (1998). For regional distribution of India, the default data were used as weighting factors. |
| Coke production | ● Activity data: Coke produced
● Data sources and treatments: See Sect. S4.1.8. |
| Asphalt production | ● Activity data: Asphalt produced
● Data sources and treatments: See Sect. S5.1.2 (Asphalt blowing). |
| Crude steel production | ● Activity data: Crude steel produced
● Data sources and treatments: See Sect. S4.1.1. |
| Hot rolled steel production | ● Activity data: Hot rolled steel produced
● Data sources and treatments: See Sect. S4.1.1. |
| Pulp and paper production | ● Activity data: Paper pulp produced
● Data sources and treatments:
➢ China: Regional data during 1983-2015 were obtained from China Statistical Yearbook (National Bureau of Statistics of China, 1986–2016) and extrapolated to 1950 using China Data Online.
➢ Other countries: National data were taken from FAOSTAT (http://www.fao.org/faostat/en/). For regional distribution of India, the default data were used as weighting factors. |

**S5.1.7 Waste disposal**

In REASv3, evaporative NMVOC emissions from disposal of municipal wastes were considered and those of industrial wastes were not included due to lack of information. Activity data are amounts of municipal wastes. Data sources and treatments of activity data used in REASv3 were summarized in Table 5.5. (See Sect. S5.1.2 (Domestic use of solvents) for data sources of population used in this sub-section.)

**Table 5.5.** Data sources and treatments of activity data for waste disposal.

| Countries and regions | Data sources and treatments of activity data |
|---|---|
| China | Regional amounts of municipal wastes after 2003 were derived from China Statistical Yearbook (National Bureau of Statistics of China, 1986–2016) and extrapolated to 1950 using number of population. |
| India | National total data in 2000, 2005, 2010, and 2015 were taken from Niyati (2015) and those in 2012 were obtained from UN Environment Programme (2017). The data were interpolated, extrapolated during 1950-2015, and distributed to each region based on number of population. |
| Rep. of Korea | National data during 1994-2004 were taken from Shragge and An (2014) and those in 2012 were obtained from UN Environment Programme (2017). The data were interpolated and extrapolated during 1950-2015 based on number of population. |
| Taiwan | National data during 2003-2015 were taken from Environmental Protection Administration (https://www.epa.gov.tw/eng/2C04F91E41A2000B/) and extrapolated during 1950-2015 using number of population |
| Thailand | National data during 1993-2002 were taken from Chiemchaisri et al., (2007) and extrapolated during 1950-2015 using number of population |
| Other countries | National data were obtained from UN Environment Programme (2017) and missing data were extrapolated during 1950-2015 based on number of population. |

**S5.2 Emission factors**

In this section, emission factors for non-combustion sources of NMVOC for each sub-category are described. Note that emission controls were not considered for non-combustion emissions of NMVOC in REASv3.

**S5.2.1 Extraction processes**

Emission factors for following sub-sectors were taken from Klimont et al. (2002a) and the same settings were used for all countries and regions as well as for all target years of REASv3:

- Gas production
- Gas distribution
- Oil production

- Oil handling
- Petroleum refinery
- Service stations
- Transport and depots (gasoline/diesel)

**S5.2.2 Solvent use**

In this sub-section, emission factors for solvent use except for printing and paint use are described. Sources and settings of emission factors are summarized in Table 5.6.

**Table 5.6.** Sources and settings of emission factors for sub-sectors of solvent use.

| Sub-sector categories | Sources and settings of emission factors |
|---|---|
| Dry cleaning | • Sources: Data for existing and new installations in Klimont et al. (2002a)
• Settings: The value for existing installations was commonly used for all target countries and periods except for Rep. of Korea and Taiwan where the same value was used before 2000. For Rep of Korea and Taiwan, it was assumed that all installations in 2020 are new and ratios of existing and new installations were changed linearly between 2000 and 2020. Based on the assumption emission factors during 2001 and 2015 were calculated. |
| Degreasing operation | • Sources and settings are the same as those "Dry cleaning". |
| Vehicle treatment | • Sources: Default data and settings until 2030 in Klimont et al. (2002a)
• Settings: The Default value was used before 2000. After 2001, data in 2000 and those assumed in 2030 in Klimont et al. (2002a) were interpolated. These settings are commonly adopted for all countries. |
| Domestic use of solvents | • Sources: Default emission factors and settings until 2030 for rural and urban population in Klimont et al. (2002a)
• Settings: Emission factors for rural and urban population were estimated by the same methodology for "Vehicle treatment" and adopted for all countries. |
| Asphalt blowing | • Sources: Klimont et al. (2002a)
• Settings: The value was used for all target countries and periods. |
| Paint production | • Sources: Klimont et al. (2002a)
• Settings: The value was used for all target countries and periods. |

| | |
|---|---|
| Ink production | ● Sources: Klimont et al. (2002a) |
| | ● Settings: The value was used for all target countries and periods. |
| Tire production | ● Sources: Klimont et al. (2002a) |
| | ● Settings: The value was used for all target countries and periods. |
| Synthetic rubber production | ● Sources: Klimont et al. (2002a) |
| | ● Settings: The value was used for all target countries and periods. |
| Textile industry | ● Sources: Klimont et al. (2002a) |
| | ● Settings: The value was used for all target countries and periods. |
| Preservation of wood | ● Sources and settings are the same as those "Dry cleaning". |
| Adhesive application | ● Sources: EEA (2016) |
| | ● Settings: The value was used for all target countries and periods. |

**S5.2.3 Printing**

Klimont et al. (2002a) provides emission factors of packaging, offset printing, publication, and screen printing for existing and new installations. The same assumption for sub-sectors such as dry cleaning described in Sect. S5.2.2 was used in RESv3.1. as follows:

● The values for existing installations were commonly used for all target countries and periods except for Rep. of Korea and Taiwan where the same value was used before 2000.

● For Rep of Korea and Taiwan, it was assumed that all installations in 2020 are new and ratios of existing and new installations were changed linearly between 2000 and 2020. Based on the assumption emission factors during 2001 and 2015 were calculated.

**S5.2.4 Paint use**

In this sub-section, emission factors for paint use for architecture, domestic usage, automobile manufacture, vehicle refinishing, and other industrial applications are described. Sources and settings of emission factors are summarized in Table 5.7.

**Table 5.7.** Sources and settings of emission factors for sub-sectors of paint use.

| Sub-sector categories | Sources and settings of emission factors |
|---|---|
| Architecture | ● Sources: Klimont et al. (2002a) |
| | ● Settings: The value was used for all target countries and periods. |
| Domestic use | ● Sources: Klimont et al. (2002a) |

| | ● Settings: The value was used for all target countries and periods. |
|---|---|
| Vehicle refinishing | ● Sources: Data for existing and new installations in Klimont et al. (2002a) |
| | ● Settings: The value for existing installations was commonly used for all target countries and periods except for Rep. of Korea and Taiwan where the same value was used before 2000. For Rep of Korea and Taiwan, it was assumed that all installations in 2020 are new and ratios of existing and new installations were changed linearly between 2000 and 2020. Based on the assumption emission factors during 2001 and 2015 were calculated. |
| Automobile manufacturing | ● Sources: Range of emission factors depending on the proportion of vehicle types in Klimont et al. (2002a) |
| | ● Settings: The lowest and highest values of the range were used for small and large vehicles, respectively. See Sect. S5.1.4 for the definitions of vehicle sizes here. |
| Other industrial application | ● Sources: Klimont et al. (2002a) |
| | ● Settings: The value was used for all target countries and periods. |

**S5.2.5 Chemical industry**

In this sub-section, emission factors for chemical industry are described. Sources and settings of emission factors are summarized in Table 5.8.

**Table 5.8.** Sources and settings of emission factors for sub-sectors of chemical industry.

| Sub-sector categories | Sources and settings of emission factors |
|---|---|
| Ethylene production | ● Sources: Klimont et al. (2002a) |
| | ● Settings: The value was used for all target countries and periods. |
| Polyethylene production | ● Sources: Klimont et al. (2002a) |
| | ● Settings: Average of emission factors for low and high-density polyethylene production were used for all target countries and periods. |
| Styrene production | ● Sources: EEA (2016) |
| | ● Settings: The value was used for all target countries and periods. |
| Polystyrene production | ● Sources: EEA (2016) |
| | ● Settings: The value was used for all target countries and periods. |
| Polyvinylchloride production | ● Sources: Klimont et al. (2002a) |
| | ● Settings: The value was used for all target countries and periods. |

| Propylene production | ● Sources: Klimont et al. (2002a)
● Settings: The value was used for all target countries and periods. |
|---|---|
| Polypropylene production | ● Sources: Klimont et al. (2002a)
● Settings: The value was used for all target countries and periods. |
| Storage of organic chemicals | ● Sources: Klimont et al. (2002a)
● Settings: Emission factors of EEA (2016) include contribution from the storage. In REASv3, 10 percent of the value was used for all target countries and periods. |
| Polyvinylchloride processing | ● Sources: Klimont et al. (2002a)
● Settings: The value was used for all target countries and periods. |
| Polystyrene processing | ● Sources: EEA (2016)
● Settings: The value was used for all target countries and periods. |
| Carbon black | ● Sources: Klimont et al. (2002a)
● Settings: The value was used for all target countries and periods. |

**S5.2.6 Other industry**

In this sub-section, emission factors for non-combustion emissions from other industry are described. Sources and settings of emission factors are summarized in Table 5.9.

**Table 5.9.** Sources and settings of emission factors for sub-sectors of other industry.

| Sub-sector categories | Data sources and treatments of activity data |
|---|---|
| Bread production | ● Sources: Klimont et al. (2002a)
● Settings: The value was used for all target countries and periods. |
| Beer production | ● Sources: Klimont et al. (2002a)
● Settings: The value was used for all target countries and periods. |
| Coke production | ● Sources: Klimont et al. (2002a)
● Settings: The value was used for all target countries and periods. |
| Asphalt production | ● Sources: Klimont et al. (2002a)
● Settings: The value was used for all target countries and periods. |
| Crude steel production | ● Sources: Klimont et al. (2002a)
● Settings: The value for steel production was used for all target countries and periods. |
| Hot rolled steel production | ● Sources: Klimont et al. (2002a)
● Settings: The value for rolling mills was used for all target countries and |

| | periods. |
|---|---|
| Pulp and paper production | ● Sources: Klimont et al. (2002a) |
| | ● Settings: The value was used for all target countries and periods. |

**S5.2.7 Waste disposal**

In REASv3, the emission factor for landfills for waste disposal in Klimont et al. (2002a) were adopted for all activity data (amounts of municipal wastes) described in S5.1.7.

**S5.2.8 Speciation of NMVOC emissions**

Emission factors described in Sect. S5.2 were for total NMVOC. In REASv3, total NMVOC emissions were allocated to 19 NMVOC species categories defined in Sect. S2.1. The speciation was conducted based on speciation profiles for each sub-sector provided by D. G. Streets (private communication) generally based on Klimont et al. (2002a) used for REASv1 and REASv2. The speciation profiles were commonly used for all countries and periods.

**S5.3 Other emission inventories included in REASv3**

**S5.3.1 Japan**

In REASv3, evaporative emissions of individual NMVOC species from sub-sectors in Japan during 2000-2015 were obtained from the Ministry of the Environment of Japan (MOEJ, 2017a). Information for regional distribution was also available in MOEJ (2017a). Emissions of the individual species were aggregated to 19 NMVOC species categories defined in Sect S2.1. Before 1999, data in 2000 were extrapolated based on trend factors related to each sub-sector as described in Table 5.10.

**Table 5.10.** Sources and treatments of trend factors for sub-sectors of NMVOC evaporative emissions in Japan

| Sub-sector categories | Data sources and treatments of trend factors |
|---|---|
| Natural gas production | ● Trend factors: Natural gas production |
| | ● Data sources and treatments: Data during 1960-2000 were derived from IEAWEB and extrapolated to 1950 using trends taken from the Historical Statistics of Japan (Japan Statistical Association, 2006). |

| | |
|---|---|
| Coke production | • Trend factors: Coke produced
• Data sources and treatments: See Sect. S4.1.8. |
| Petroleum refinery | • Trend factors: Consumption of crude oil in petroleum refineries
• Data sources and treatments: See Sect. S3.1. |
| Service stations | • Trend factors: Consumption of gasoline in road transport sector
• Data sources and treatments: See Sect. S3.1. |
| Transport and depots | • Trend factors: Consumption of gasoline and diesel in road transport sector
• Data sources and treatments: See Sect. S3.1. |
| Dry cleaning | • Trend factors: Number of facilities
• Data sources and treatments: Data during 1963-2000 were taken from Japan Cleaning Journal (http://www.nicli.co.jp/stat-sisetu.html) and extrapolated to 1950 using values of shipments for industrial organic chemicals obtained from the Historical Statistics of Japan (Japan Statistical Association, 2006) were used as trend factors. |
| Detergents usage in industry | • Trend factors: Values of shipments of detergents for industries
• Data sources and treatments: Data during 1960-2000 were obtained from Yearbook of Chemical Industry Statistics (Ministry of Economy, Trade and Industry, Japan, https://www.meti.go.jp/statistics/). Before 1960, values of shipments for industrial organic chemicals obtained from the Historical Statistics of Japan (Japan Statistical Association, 2006) were used as trend factors. |
| Adhesive application | • Trend factors: Adhesive produced
• Data sources and treatments: Data during 1960-2000 were obtained from Yearbook of Chemical Industry Statistics (Ministry of Economy, Trade and Industry, Japan, https://www.meti.go.jp/statistics/). Before 1960, values of shipments of industrial organic chemicals obtained from the Historical Statistics of Japan (Japan Statistical Association, 2006) were used as trend factors. |
| Asphalt blowing | • Trend factors: Asphalt produced
• Data sources and treatments: Data during 1950-2000 were derived from the Historical Statistics of Japan (Japan Statistical Association, 2006). |
| Rubber production | • Trend factors: Rubber produced
• Data sources and treatments: Production amounts and values of shipments for rubber products were taken from the Historical Statistics of Japan (Japan Statistical Association, 2006). |

| | |
|---|---|
| Synthetic leather production | • Trend factors: Synthetic leather produced
• Data sources and treatments: Data during 1985-2000 and those for all leather products before 1984 were obtained from the Historical Statistics of Japan (Japan Statistical Association, 2006). |
| Protection of fishing net | • Trend factors: Fishing net produced
• Data sources and treatments: Data were obtained from Yearbook of Current Production Statistics (Ministry of Economy, Trade and Industry, Japan, https://www.meti.go.jp/statistics/) and the Historical Statistics of Japan (Japan Statistical Association, 2006). |
| Ink application | • Trend factors: Values of shipments by publishing, printing and allied industries
• Data sources and treatments: Data were obtained from Yearbook of Chemical Industry Statistics (Ministry of Economy, Trade and Industry, Japan, https://www.meti.go.jp/statistics/) and the Historical Statistics of Japan (Japan Statistical Association, 2006). |
| Paint application | • Trend factors: Values of shipments by paint industries of manufacturing
• Data sources and treatments: Data during 1960-2000 were obtained from Yearbook of Chemical Industry Statistics (Ministry of Economy, Trade and Industry, Japan, https://www.meti.go.jp/statistics/). Before 1960, production of synthetic paints obtained from the Historical Statistics of Japan (Japan Statistical Association, 2006) were used. |
| Other solvent use | • Trend factors: Values of shipments of industrial organic chemicals
• Data sources and treatments: Data during 1950-2000 were obtained from the Historical Statistics of Japan (Japan Statistical Association, 2006) |
| Chemical industry | • Trend factors: Petrochemicals produced
• Data sources and treatments: Data during 1960-2000 were obtained from Yearbook of Chemical Industry Statistics (Ministry of Economy, Trade and Industry, Japan, https://www.meti.go.jp/statistics/) were extrapolated to 1950 using values of shipments of industrial organic chemicals obtained from the Historical Statistics of Japan (Japan Statistical Association, 2006) were used as trend factors. |
| Food production | • Trend factors: Values of shipments by food industries of manufacturing
• Data sources and treatments: Data during 1950-2000 were obtained from the Historical Statistics of Japan (Japan Statistical Association, 2006). |
| Pesticide application | • Trend factors: Pesticide produced
• Data sources and treatments: Data during 1950-2000 were taken from |

| | Japan Crop Production Association (https://www.jcpa.or.jp/qa/a5_12.html). |
| --- | --- |
| Others | ● Trend factors: GDP |
| | ● Data sources and treatments: See Sect. S4.1.7 |

**S5.3.2 Republic of Korea**

For Republic of Korea, first, NMVOC (including 19 individual species) emissions from evaporative sources were tentatively estimated using activity data and emission factors described in Sects. S5.1 and S5.2, respectively. Then, emissions from extraction processes, solvent use including printing and paint application, and industrial processes in both chemical and other industries, and waste disposal were obtained from the National Institute of Environmental Research (http://airemiss.nier.go.kr/mbshome/mbs/airemiss/index.do) during 1999-2015. Finally, the tentatively estimated emissions for each sub-sector were adjusted by ratios between the aggregated emissions of the National Institute of Environmental Research and those of the tentative estimation. For example, tentative emissions from dry cleaning were adjusted by factors calculated for solvent use. Before 1999, the tentative emissions were adjusted using the factors for the year 1999. Note that emissions from combustion sources for Republic of Korea were originally estimated in REASv3.

**S6. Road transport**

**S6.1 Activity data**

**S6.1.1 Annual mileage**

In REASv3, exhaust emissions from road vehicles were estimated based on annual distances vehicles are driven (annual mileage) and corresponding emission factors (amounts of air pollutants per distance driven). The annual mileages were calculated by number of vehicles and annual distances traveled for each vehicle type. The number of vehicles was obtained from national and international statistics and related literatures. However, available vehicle categories in the data are different among countries and regions. In addition, information for categories of different fuel types such as gasoline and diesel and annual distances traveled for each vehicle type is limited. In Table 6.1, data sources and assumptions to estimate historical annual mileage data are provided.

**Table 6.1.** Data sources and settings of number of vehicles and annual distance travelled for each country and region in REASv3.

**(a) China**

| Number of vehicles | <li>Data sources:</li><li>Regional data of large/medium/small/mini passenger vehicles and heavy/medium/light/mini trucks during 1985-2015 were taken from China Statistical Yearbook and extrapolated to 1950 using number of civil motor vehicles in each region in China Data Online.</li><li>For motorcycles, national total during 1991-2015 were taken from IRF (1990-2018) and distributed to each region and extrapolated to 1950 using the number of civil motor vehicles in each region.</li><li>Vehicle categories:</li><li>For data based on China Statistical Yearbook, large/medium and small/minicar passenger vehicles were treated as buses and cars, respectively. For trucks, heavy/medium and light/mini vehicles were treated as heavy and light trucks, respectively. For distribution of fuel types, data in He et al. (2005) were used for cars and those in Yan and Crookes (2009) were used for buses and trucks.</li><li>No classification was done for motorcycles and it was assumed that only gasoline was used in motorcycles.</li> |
|---|---|
| Annual distance travelled | <li>Settings of annual distance travelled for each vehicle type were based on Huo et al. (2012b).</li> |

**(b) Hong Kong**

| Number of vehicles | <li>Data sources:</li><li>Data of passenger cars, buses, trucks, and motorcycles during 1964-2015 were obtained from IRF (1976-2018) and extrapolated to 1950 using trends of number of vehicles for aggregated vehicle types in Mitchell (1998).</li><li>Vehicle categories:</li><li>Vehicle types include gasoline, diesel, and LPG passenger cars, taxis, buses, and light and heavy trucks, and motorcycles. For relative ratios of vehicles numbers of each fuel type, in addition to data of Streets et al. (2003) and REASv2 generally based on GAINS ASIA at that time, data in A clean air plan for Hong Kong (Environment Bureau, 2013) and consumption amounts of LPG in road transport sector in the International Energy Agency (IEA) World Energy Balances (IEAWEB) (IEA, 2017) were used.</li> |
|---|---|

| Annual distance travelled | ● Settings of Singapore were used in REASv3. |
|---|---|

**(c) Macau**

| Number of vehicles | ● Data sources:
➤ Data of passenger cars, buses, trucks, and motorcycles during 1994-2015 were obtained from IRF (1976-2018) and extrapolated to 1950 using trends of fuel consumption in the United Nations (UN) Energy Statistics Database (UN, 2016).
● Vehicle categories:
➤ Vehicle types include gasoline and diesel passenger cars, buses, light and heavy trucks, and motorcycles. For relative ratios of gasoline and diesel vehicle numbers, data of Hong Kong in REASv2 generally based on GAINS ASIA at that time were used. |
|---|---|
| Annual distance travelled | ● Settings of Singapore were used in REASv3. |

**(d) India**

| Number of vehicles | ● Data sources:
➤ Regional data of passenger cars, taxis, jeeps, buses, light trucks, heavy trucks, trailers, light motor vehicles, and motorcycles during 2001-2015 were taken from TERI (2013, 2018) and extrapolated to 1950 using trends of national data for cars & jeeps & taxis, buses, trucks, and motorcycles obtained from Indiastat.
● Vehicle categories:
➤ In general, passenger cars, taxis, jeeps, light motor vehicles, and motorcycles assumed to consume gasoline and for buses, trucks, and trailers, the fuel type is assumed to be diesel. For Delhi and Mumbai (in Maharashtra), number of CNG cars, taxis, and buses in 2010 were assumed based on Sahu et al. (2014) and extrapolated using IEAWEB.
➤ According to Baidya and Borken-Kleefeld (2009), there are large differences between registered number of vehicles and those actually circulating on the road. Relative ratios of vehicle numbers in operation to registered ones were taken from Prakash and Habib (2018) and Baidya and Borken-Kleefeld (2009). |
|---|---|

| Annual distance travelled | • Settings of annual distance travelled for each vehicle type were based on Prakash and Habib (2018) and Pandey and Venkataraman (2014). |
|---|---|

**(e) Japan**

| Annual mileages | • Data sources:
   ➤ National annual mileages for each vehicle type (including different fuel types) among different vehicle speed categories were derived from reports of Pollutants Release and Transfer Register (METI, 2003-2017) during 2001-2015 and extrapolated to 1950 using trends of annual distances travelled for aggregated vehicle types in the Historical Statistics of Japan (Japan Statistical Association, 2006). Vehicle types were further divided into detailed categories using number of vehicles provided in the report of the Japan Auto-Oil Program (JATOP) Emission Inventory-Data Base (JEI-DB) (JPEC 2012a).
   ➤ For regional distribution of national data, weighting factors during 1960-2015 were calculated using annual distances travelled of aggregated vehicle types in annual reports of road transport statistics (MLIT, 1961-2016). Before 1960, data in 1960 were used.
• Vehicle categories:
   ➤ Vehicle types include passenger cars (gasoline and LPG), light, medium and heavy trucks (gasoline and diesel), buses (gasoline and diesel), special purpose vehicles (gasoline and diesel), and several sizes of motorcycles. Trucks, buses, and special purpose vehicles were further divided into different weight categories. |
|---|---|

**(f) Republic of Korea**

| Number of vehicles | • Data sources:
   ➤ National data of passenger cars, buses, trucks, and motorcycles during 1976-2015 were obtained from IRF (1976-2018) and extrapolated to 1950 using trends of number of vehicles for aggregated vehicle types in Mitchell (1998).
   ➤ Number of LPG and CNG vehicles in 2010 were taken from a report of European Commission (Alternative fuels and infrastructure in seven non-EU markets) and the Gas Vehicles Report, respectively and extrapolated using trends of fuel |
|---|---|

| | consumption in IEAWEB. |
| --- | --- |
| | ● Vehicle categories: |
| | ➢ Vehicle types include passenger cars (gasoline, diesel, and LPG), buses (gasoline, diesel, and CNG), light and heavy trucks (gasoline and diesel), rural vehicles, and several sizes of motorcycles. For relative ratios of number of gasoline and diesel vehicles, data of Streets et al. (2003) and REASv2 generally based on GAINS ASIA at that time were used. |
| Annual distance travelled | ● Settings of Singapore were used in REASv3 except for motorcycles which were taken from Jang et al. (2010). |

**(g) Democratic People's Republic of Korea**

| Number of vehicles | ● Data sources and vehicle categories: |
| --- | --- |
| | ➢ Number of gasoline and diesel vehicles for passenger cars, buses, light and heavy trucks, rural vehicles, and motorcycles in 2000 were taken from REASv1 generally based on Streets et al. (2003) and extrapolated using trends of gasoline and diesel oil consumption in road transport in IEAWEB. |
| Annual distance travelled | ● Annual vehicle kilometer travelled per vehicle type averaged in Asia provided in Clean Air Asia (2012) were used. |

**(h) Mongolia**

| Number of vehicles | ● Data sources: |
| --- | --- |
| | ➢ National data of passenger cars, buses, trucks, and motorcycles during 1950-2015 were obtained from National Statistics Office of Mongolia (https://www.en.nso.mn/). |
| | ● Vehicle categories: |
| | ➢ Vehicle types include gasoline and diesel passenger cars, buses, light and heavy trucks, and motorcycles. For relative ratios of gasoline and diesel vehicle numbers, data of Streets et al. (2003) and REASv2 generally based on GAINS ASIA at that time were used. |
| Annual distance travelled | ● Annual vehicle kilometer travelled per vehicle type averaged in Asia provided in Clean Air Asia (2012) were used. |

**(i) Taiwan**

[remaining 430,515 characters of this post omitted]

---

## Author Response (AR3)

Dear Editor,

The authors would like to appreciate Editor for taking your precious time to handle our manuscript. Because there were no additional comments from Referees, authors did a final proofread of the main manuscript and supplementary materials. We found several typos in the main manuscript and figure captions in the supplementary material entitled "Supplementary figures". In this letter, the changes in the main manuscript and the supplementary material by correction of the typos are reported. For distribution of the updated data sets, as explained in the previous Author's Response, the final version will be opened at the REAS download site as REASv3.2, when the revision process has been completed.

The structure of this document is as follows:

(1) Author's changes in the main manuscript by corrections of typos
(2) The revised main manuscript where changed parts were yellow highlighted
(3) The revised main manuscript with track changes
(4) The revised supplementary material ("Supplementary figures") where changed parts were yellow highlighted
(5) The revised supplementary material ("Supplementary figures") with track changes

Sincerely Yours,
Jun-ichi Kurokawa
Asia Center for Air Pollution Research
kurokawa@acap.asia
TEL: +81-25-263-9558
FAX +81-25-263-0567

(1) Author's changes in the main manuscript by corrections of typos

Revisions of the main manuscript by corrections of typos are as follows:

- P3L67: "Thao Pham" was changed to "Pham".
- P3L75; L79; L80: "REASv3.1" was changed to "REASv3".
- P3L77; L79; L80; L81: "Section" was changed to "Sect.".
- P4L120: "2.6 and 2.4.1" was changed to "2.4.1 and 2.6".
- P5L139: "EF$_{i,j,k}$" was changed to "$EF_{i,j,k}$".
- P5L158: "S.3" was changed to "S3".
- P6L179; L191: "Section" was chanted to "Sect.".
- P7L213: "REASv3.1" was changed to "REASv3".
- P8L250: "." was added after "Sect". ("Sect" was changed to "Sect.")
- P10L303: "." was added after "Sect". ("Sect" was changed to "Sect.")
- P10L312: "and the regions and" was changed to "and". ("and the regions" was deleted.)
- P10L315: "regions" was changed to "region".
- P11L341: "Sections" was changed to "Sects.".
- P14L421: "Thao Pham" was changed to "Pham".
- P17L533: "Fig.3" was changed to "Fig. 3".
- P17L547: "Trend so" was changed to "Trends of".
- P22L687: "Section" was changed to "Sect.".
- P22L703: "Figures" was changed to "Figs.".
- P24L758; L759-760; L768; L774; P25L781: "Sadavarte and Venkataraman (2014) and Pandey et al. (2014)" was changed to "Pandey et al. (2014) and Sadavarte and Venkataraman (2014)".
- P24L765: "Venkataraman (2014) and Pandey et al. (2014)" was changed to "Pandey et al. (2014) and Sadavarte and Venkataraman (2014)".
- P24L777: "Jiang et al., 2017" was changed to "Jiang et al. (2017)".
- P26L837: "rats" was changed to "rates".
- P29L914; L919: "Section" was changed to "Sect.".
- P29L918: "REASv3.1" was changed to "REASv3".
- P29L928: "NH" was changed to "NH$_3$".
- P29L936-937: "estimated from the two averages" was added after "… in these 60 years".
- P32L1016: "2" was changed to "49".
- P33L1056: "…India2005-2016" was changed to "…India 2005-2016".
- P39L1248: "PM2.5" was changed to "PM$_{2.5}$".
- P48L1341: "Red lines in the upper panels are total CO$_2$ emissions." was added.
- P53L1358: "Pandey and Venkataraman (2014)" was changed to "Pandey et al. (2014)".

- P56 How utilized in REASv3 for REASv2.1 (Kurokawa et al., 2013; JPEC 2012a, b, c; 2014): "… road transport sectors for Japan[d]" was changed to "… road transport[d] sectors for Japan".
- P56 Footnote of the Table 2: "S8.1" was changed to "S9.1".

(2) The revised main manuscript where changed parts were yellow highlighted

From the next page, the revised main manuscript where changed parts were yellow highlighted is provided.

[revised manuscript text omitted]

(4) The revised supplementary material ("Supplementary figures") where changed parts were yellow highlighted

From the next page, the revised supplementary material ("Supplementary figures") where changed parts were yellow highlighted is provided. Revisions of the supplementary materials by corrections of typos are as follows:

- P1L20: "NH3" was changed to "$NH_3$".
- P15L79: "(a, c)" was changed to "(a, b)".
- P15L79: "(b, d)" was changed to "(c, d)".
- P16L83: "BC" was changed to "OC".
- P18L92: "Pandey and Venkataraman (2014)" was changed to "Pandey et al. (2014)".
- P32L142: "and" was inserted before "(f)".
- P35L208: "https://doi.org/10.1007/978-3-319-59489-7-8" was changed to "https://doi.org/10.1007/978-3-319-59489-7".
- P36L244: "CO2" was changed to "$CO_2$".

*Supplement of*

**Long-term historical trends in air pollutant emissions in Asia: Regional Emission inventory in ASia (REAS) version 3**

**Junichi Kurokawa and Toshimasa Ohara**

*Correspondence to*: Junichi Kurokawa (kurokawa@acap.asia)

**Supplementary figures**

**Figures S1, S3, S5, S7, S9, and S11** show emissions of (a) $SO_2$, (b) $NO_x$, (c) CO, (d) $CO_2$, (e) $PM_{10}$, (f) $PM_{2.5}$, (g) BC, (h) OC, (i) NMVOC, and (j) $NH_3$ from major sectors during 1950-2015 in China, India, Japan, Southeast Asia (SEA), East Asia other than China and Japan (OEA), and South Asia other than India (OSA), respectively. See Fig. 1 for the definitions of SEA, OEA, and OSA. (Sectors for (a)-(h): PP = Power plants, IND = Industry, ROAD = Road transport, OTRA = Other transport, RESI = Residential, and ODOM = Other domestic; Sectors for (i): CMB = Combustion, ROAD = Road transport (including both tail pipe and evaporative emissions), INDPRC = Industrial processes, EXT = Extraction processes, PAINT = Paint use, SLV = Solvent use, and WST = Waste treatment; Sectors for (j): CMB = Combustion, MM = Manure management, FER = Fertilizer application, HUMAN = Human perspiration and respiration, LTRN = Latrines, and INDPRC = Industrial processes.)

**Figures S2, S4, S6, S8, S10, and S12** provide emissions of (a) $SO_2$, (b) $NO_x$, (c) CO, (d) $CO_2$, (e) $PM_{10}$, (f) $PM_{2.5}$, (g) BC, (h) OC, (i) NMVOC, and (j) $NH_3$ from each fuel type during 1950-2015 in China, India, Japan, SEA, OEA, and OSA, respectively. See Fig. 1 for the definitions of SEA, OEA, and OSA. Note that emissions from non-combustion sources are not included in (i) NMVOC and (j) $NH_3$ to show contributions from fuel types clearly because majority of their emissions are from non-combustion sources. (Fuel types: COAL = Primary coal, DC = Secondary coal, NGAS = Natural gas, OGAS = Other gas fuels, LF = Light oil fuels, MD = Diesel oil, HF = Heavy oil fuels, BF = Biofuels, OTH = Other fuels, NCMB = Non-combustion sources, and CEMK = combustion emissions from cement kilns (only for Japan). Notes: For CO emissions from pig iron, crude steel, and sinter production for all countries, those from brick production except for China, Japan, Republic of Korea, and Taiwan, emissions of PM species from sinter and pig iron production for China, and those from brick production for all countries and regions estimated based on their production amounts, both combustion and non-combustion emissions are included in NCMB here. In Japan, emissions from cement production were estimated not by fuel consumption, but based on production amounts of cement in each kiln type. Therefore, contributions from total emissions from cement kiln combustion are included in CEMK.)

**Figure S13** illustrates grid maps of annual emissions of $CO_2$ and $PM_{10}$ in 1965 and 2015.

**Figures S14 and S15** compare CO, NMVOC, $NH_3$, $PM_{10}$, $PM_{2.5}$, and OC emissions in REASv3 with other published estimates for China and India, respectively. Note that IM means estimates by inverse modeling and [A][B][C] of Jiang et al. (2017) are estimates based on A: MOPITT Column, B: MOPITT Profile, and C: MOPITT Lower Profile, respectively.

35 **Figures S16-S19** compare emissions of $SO_2$, $NO_x$, BC, CO, NMVOC, $NH_3$, $PM_{10}$, $PM_{2.5}$, and OC in REASv3 with other published estimates for Japan, SEA, OEA, and OSA, respectively. See Fig. 1 for the definitions of SEA, OEA, and OSA. Note that IM means estimates by inverse modeling and [A][B][C] of Jiang et al. (2017) are estimates based on A: MOPITT Column, B: MOPITT Profile, and C: MOPITT Lower Profile, respectively.

**Figures S20** compares Asia total emissions and relative ratios of those from China, India, Japan, SEA, OEA, and OSA for CO, NMVOC, $NH_3$, $PM_{10}$, $PM_{2.5}$, and OC among REASv3, CEDS, and EDGARv4.3.2. See Fig. 1 for definitions of SEA,

40 OEA, and OSA.

**China Sector**

**Figure S1.** Emissions of (a) $SO_2$, (b) $NO_x$, (c) CO, (d) $CO_2$, (e) $PM_{10}$, (f) $PM_{2.5}$, (g) BC, (h) OC, (i) NMVOC, and (j) $NH_3$ from major sectors in China from 1950 to 2015.

[Figure]

**Figure S2.** Emissions of (a) SO$_2$, (b) NO$_x$, (c) CO, (d) CO$_2$, (e) PM$_{10}$, (f) PM$_{2.5}$, (g) BC, (h) OC, (i) NMVOC, and (j) NH$_3$ from each fuel type in China from 1950 to 2015.

[Figure]

**Figure S3.** Emissions of (a) $SO_2$, (b) $NO_x$, (c) CO, (d) $CO_2$, (e) $PM_{10}$, (f) $PM_{2.5}$, (g) BC, (h) OC, (i) NMVOC, and (j) $NH_3$ from major sectors in India from 1950 to 2015.

**India Fuel**

**Figure S4.** Emissions of (a) SO₂, (b) NOₓ, (c) CO, (d) CO₂, (e) PM₁₀, (f) PM₂.₅, (g) BC, (h) OC, (i) NMVOC, and (j) NH₃ from each fuel type in India from 1950 to 2015.

[Figure]

**Figure S5.** Emissions of (a) $SO_2$, (b) $NO_x$, (c) CO, (d) $CO_2$, (e) $PM_{10}$, (f) $PM_{2.5}$, (g) BC, (h) OC, (i) NMVOC, and (j) $NH_3$ from major sectors in Japan from 1950 to 2015.

**Japan Fuel**

[Figure]

**Figure S6.** Emissions of (a) SO$_2$, (b) NO$_x$, (c) CO, (d) CO$_2$, (e) PM$_{10}$, (f) PM$_{2.5}$, (g) BC, (h) OC, (i) NMVOC, and (j) NH$_3$ from each fuel type in Japan from 1950 to 2015.

**SEA Sector**

[Figure]

**Figure S7.** Emissions of (a) SO₂, (b) NOₓ, (c) CO, (d) CO₂, (e) PM₁₀, (f) PM₂.₅, (g) BC, (h) OC, (i) NMVOC, and (j) NH₃ from major sectors in SEA from 1950 to 2015. See Fig. 1 for the definitions of SEA.

[Figure]

**Figure S8.** Emissions of (a) $SO_2$, (b) $NO_x$, (c) CO, (d) $CO_2$, (e) $PM_{10}$, (f) $PM_{2.5}$, (g) BC, (h) OC, (i) NMVOC, and (j) $NH_3$ from each fuel
type in SEA from 1950 to 2015. See Fig. 1 for the definitions of SEA.

**OEA Sector**

**Figure S9.** Emissions of (a) SO₂, (b) NOₓ, (c) CO, (d) CO₂, (e) PM₁₀, (f) PM₂.₅, (g) BC, (h) OC, (i) NMVOC, and (j) NH₃ from major sectors in OEA from 1950 to 2015. See Fig. 1 for the definitions of OEA.

**OEA Fuel**

**Figure S10.** Emissions of (a) $SO_2$, (b) $NO_x$, (c) CO, (d) $CO_2$, (e) $PM_{10}$, (f) $PM_{2.5}$, (g) BC, (h) OC, (i) NMVOC, and (j) $NH_3$ from each fuel type in OEA from 1950 to 2015. See Fig. 1 for the definitions of OEA.

**Figure S11.** Emissions of (a) $SO_2$, (b) $NO_x$, (c) CO, (d) $CO_2$, (e) $PM_{10}$, (f) $PM_{2.5}$, (g) BC, (h) OC, (i) NMVOC, and (j) $NH_3$ from major sectors in OSA from 1950 to 2015. See Fig. 1 for the definitions of OSA.

[Figure]

**Figure S12.** Emissions of (a) SO2, (b) NOx, (c) CO, (d) CO2, (e) PM10, (f) PM2.5, (g) BC, (h) OC, (i) NMVOC, and (j) NH3 from each fuel type in OSA from 1950 to 2015. See Fig. 1 for the definitions of OSA.

[Figure]

**Figure S13.** Grid maps of annual emissions of (a, b) $CO_2$ (kt year$^{-1}$ per grid cell) and (c, d) $PM_{10}$ (t year$^{-1}$ per grid cell) in 1965 (left) and 2015 (right).

[Figure]

**Figure S14.** Comparison of (a) CO, (b) NMVOC, (c) NH₃, (d) PM₁₀, (e) PM₂.₅ and (f) OC emissions in China between REASv3 and other studies. Emissions from domestic and fishing ships were excluded from REAS series, CEDS, and EDGARv4.3.2. Error bars indicate the uncertainty range of REASv3 in 1955, 1985, and 2015.

[Figure]

**Figure S14.** Continued.

[Figure]

90    **Figure S15.** Comparison of (a) CO, (b) NMVOC, (c) NH₃, (d) PM₁₀, (e) PM₂.₅ and (f) OC emissions in India between REASv3 and other studies. Emissions from domestic and fishing ships were excluded from REAS series, CEDS, and EDGARv4.3.2. Note that values of "Pandy+Sadavarte" are calculated from Pandey et al. (2014) and Sadavarte and Venkataraman (2014). Error bars indicate the uncertainty range of REASv3 in 1955, 1985, and 2015.

[Figure]

**Figure S15.** Continued.

[Figure]

**Figure S16.** Comparison of (a) SO₂, (b) NOₓ, (c) BC, (d) CO, (e) NMVOC, (f) NH₃, (g) PM₁₀, (h) PM₂.₅ and (i) OC emissions in Japan between REASv3 and other studies. Emissions from domestic and fishing ships were excluded from REAS series, CEDS, EDGARv4.3.2, Kannari et al. (2007), and Fukui et al. (2013). Error bars indicate the uncertainty range of REASv3 in 1955, 1985, and 2015.

[Figure]

**Figure S16.** Continued.

[Figure]

**Figure S16.** Continued.

[Figure]

**Figure S17.** Comparison of (a) SO₂, (b) NOₓ, (c) BC, (d) CO, (e) NMVOC, (f) NH₃, (g) PM₁₀, (h) PM₂.₅ and (i) OC emissions in SEA between REASv3 and other studies. Emissions from domestic and fishing ships were excluded from REAS series, CEDS, and EDGARv4.3.2. See Fig. 1 for the definitions of SEA. Error bars indicate the uncertainty range of REASv3 in 1955, 1985, and 2015.

[Figure]

**Figure S17.** Continued.

[Figure]

**Figure S17.** Continued.

[Figure]

120 **Figure S18.** Comparison of (a) SO₂, (b) NOₓ, (c) BC, (d) CO, (e) NMVOC, (f) NH₃, (g) PM₁₀, (h) PM₂.₅ and (i) OC emissions in OEA between REASv3 and other studies. Emissions from domestic and fishing ships were excluded from REAS series, CEDS, and EDGARv4.3.2. See Fig. 1 for the definitions of OEA. Error bars indicate the uncertainty range of REASv3 in 1955, 1985, and 2015.

[Figure]

125 **Figure S18.** Continued.

[Figure]

**Figure S18.** Continued.

[Figure]

**Figure S19.** Comparison of (a) SO₂, (b) NOₓ, (c) BC, (d) CO, (e) NMVOC, (f) NH₃, (g) PM₁₀, (h) PM₂.₅ and (i) OC emissions in OSA between REASv3 and other studies. Emissions from domestic and fishing ships were excluded from REAS series, CEDS, and EDGARv4.3.2. See Fig. 1 for the definitions of OSA. Error bars indicate the uncertainty range of REASv3 in 1955, 1985, and 2015.

[Figure]

**Figure S19.** Continued.

[Figure]

**Figure S19.** Continued.

[Figure]

**Figure S20.** Comparison of trends of (a) CO, (b) NMVOC, (c) NH$_3$, (d) PM$_{10}$, (e) PM$_{2.5}$, and (f) OC emissions in Asia and relative ratios of emissions from China, India, Japan, SEA, OEA, and OSA for (g, m, q) CO, (h, n, r) NMVOC, (i, o, s) NH$_3$, (j, t) PM$_{10}$, (k, u) PM$_{2.5}$, and (l, p, v) OC among (g, h, i, j, k, l) REASv3, (m, n, o, p) CEDS, and (q, r, s, t, u, v) EDGARv4.3.2. See Fig. 1 for the definitions of SEA, OEA, and OSA.

[Figure]

**Figure S20.** Continued.

[revised manuscript text omitted]

(5) The revised supplementary material ("Supplementary figures") with track changes

From the next page, the revised supplementary material ("Supplementary figures") with track changes is provided.

*Supplement of*

**Long-term historical trends in air pollutant emissions in Asia: Regional Emission inventory in ASia (REAS) version 3**

**Junichi Kurokawa and Toshimasa Ohara**

*Correspondence to*: Junichi Kurokawa (kurokawa@acap.asia)

**Supplementary figures**

**Figures S1, S3, S5, S7, S9, and S11** show emissions of (a) $SO_2$, (b) $NO_x$, (c) CO, (d) $CO_2$, (e) $PM_{10}$, (f) $PM_{2.5}$, (g) BC, (h) OC, (i) NMVOC, and (j) $NH_3$ from major sectors during 1950-2015 in China, India, Japan, Southeast Asia (SEA), East Asia other than China and Japan (OEA), and South Asia other than India (OSA), respectively. See Fig. 1 for the definitions of SEA, OEA, and OSA. (Sectors for (a)-(h): PP = Power plants, IND = Industry, ROAD = Road transport, OTRA = Other transport, RESI = Residential, and ODOM = Other domestic; Sectors for (i): CMB = Combustion, ROAD = Road transport (including both tail pipe and evaporative emissions), INDPRC = Industrial processes, EXT = Extraction processes, PAINT = Paint use, SLV = Solvent use, and WST = Waste treatment; Sectors for (j): CMB = Combustion, MM = Manure management, FER = Fertilizer application, HUMAN = Human perspiration and respiration, LTRN = Latrines, and INDPRC = Industrial processes.)

**Figures S2, S4, S6, S8, S10, and S12** provide emissions of (a) $SO_2$, (b) $NO_x$, (c) CO, (d) $CO_2$, (e) $PM_{10}$, (f) $PM_{2.5}$, (g) BC, (h) OC, (i) NMVOC, and (j) $NH_3$ from each fuel type during 1950-2015 in China, India, Japan, SEA, OEA, and OSA, respectively. See Fig. 1 for the definitions of SEA, OEA, and OSA. Note that emissions from non-combustion sources are not included in (i) NMVOC and (j) $NH_3$ to show contributions from fuel types clearly because majority of their emissions are from non-combustion sources. (Fuel types: COAL = Primary coal, DC = Secondary coal, NGAS = Natural gas, OGAS = Other gas fuels, LF = Light oil fuels, MD = Diesel oil, HF = Heavy oil fuels, BF = Biofuels, OTH = Other fuels, NCMB = Non-combustion sources, and CEMK = combustion emissions from cement kilns (only for Japan). Notes: For CO emissions from pig iron, crude steel, and sinter production for all countries, those from brick production except for China, Japan, Republic of Korea, and Taiwan, emissions of PM species from sinter and pig iron production for China, and those from brick production for all countries and regions estimated based on their production amounts, both combustion and non-combustion emissions are included in NCMB here. In Japan, emissions from cement production were estimated not by fuel consumption, but based on production amounts of cement in each kiln type. Therefore, contributions from total emissions from cement kiln combustion are included in CEMK.)

**Figure S13** illustrates grid maps of annual emissions of $CO_2$ and $PM_{10}$ in 1965 and 2015.

**Figures S14 and S15** compare CO, NMVOC, NH$_3$, PM$_{10}$, PM$_{2.5}$, and OC emissions in REASv3 with other published estimates for China and India, respectively. Note that IM means estimates by inverse modeling and [A][B][C] of Jiang et al. (2017) are estimates based on A: MOPITT Column, B: MOPITT Profile, and C: MOPITT Lower Profile, respectively.

35     **Figures S16-S19** compare emissions of SO$_2$, NO$_x$, BC, CO, NMVOC, NH$_3$, PM$_{10}$, PM$_{2.5}$, and OC in REASv3 with other published estimates for Japan, SEA, OEA, and OSA, respectively. See Fig. 1 for the definitions of SEA, OEA, and OSA. Note that IM means estimates by inverse modeling and [A][B][C] of Jiang et al. (2017) are estimates based on A: MOPITT Column, B: MOPITT Profile, and C: MOPITT Lower Profile, respectively.

    **Figures S20** compares Asia total emissions and relative ratios of those from China, India, Japan, SEA, OEA, and OSA for CO, NMVOC, NH$_3$, PM$_{10}$, PM$_{2.5}$, and OC among REASv3, CEDS, and EDGARv4.3.2. See Fig. 1 for definitions of SEA,

40     OEA, and OSA.

**China Sector**

**Figure S1.** Emissions of (a) $SO_2$, (b) $NO_x$, (c) CO, (d) $CO_2$, (e) $PM_{10}$, (f) $PM_{2.5}$, (g) BC, (h) OC, (i) NMVOC, and (j) $NH_3$ from major sectors in China from 1950 to 2015.

**China Fuel**

[Figure]

**Figure S2.** Emissions of (a) SO$_2$, (b) NO$_x$, (c) CO, (d) CO$_2$, (e) PM$_{10}$, (f) PM$_{2.5}$, (g) BC, (h) OC, (i) NMVOC, and (j) NH$_3$ from each fuel type in China from 1950 to 2015.

[Figure]

**Figure S3.** Emissions of (a) SO₂, (b) NOₓ, (c) CO, (d) CO₂, (e) PM₁₀, (f) PM₂.₅, (g) BC, (h) OC, (i) NMVOC, and (j) NH₃ from major sectors in India from 1950 to 2015.

**India Fuel**

**Figure S4.** Emissions of (a) SO₂, (b) NOₓ, (c) CO, (d) CO₂, (e) PM₁₀, (f) PM₂.₅, (g) BC, (h) OC, (i) NMVOC, and (j) NH₃ from each fuel type in India from 1950 to 2015.

**Japan Sector**

[Figure]

55   **Figure S5.** Emissions of (a) $SO_2$, (b) $NO_x$, (c) CO, (d) $CO_2$, (e) $PM_{10}$, (f) $PM_{2.5}$, (g) BC, (h) OC, (i) NMVOC, and (j) $NH_3$ from major sectors in Japan from 1950 to 2015.

[Figure]

**Figure S6.** Emissions of (a) $SO_2$, (b) $NO_x$, (c) CO, (d) $CO_2$, (e) $PM_{10}$, (f) $PM_{2.5}$, (g) BC, (h) OC, (i) NMVOC, and (j) $NH_3$ from each fuel type in Japan from 1950 to 2015.

**SEA Sector**

[Figure]

**Figure S7.** Emissions of (a) SO₂, (b) NOₓ, (c) CO, (d) CO₂, (e) PM₁₀, (f) PM₂.₅, (g) BC, (h) OC, (i) NMVOC, and (j) NH₃ from major sectors in SEA from 1950 to 2015. See Fig. 1 for the definitions of SEA.

**SEA Fuel**

[Figure]

**Figure S8.** Emissions of (a) SO$_2$, (b) NO$_x$, (c) CO, (d) CO$_2$, (e) PM$_{10}$, (f) PM$_{2.5}$, (g) BC, (h) OC, (i) NMVOC, and (j) NH$_3$ from each fuel type in SEA from 1950 to 2015. See Fig. 1 for the definitions of SEA.

**OEA Sector**

**Figure S9.** Emissions of (a) $SO_2$, (b) $NO_x$, (c) CO, (d) $CO_2$, (e) $PM_{10}$, (f) $PM_{2.5}$, (g) BC, (h) OC, (i) NMVOC, and (j) $NH_3$ from major sectors in OEA from 1950 to 2015. See Fig. 1 for the definitions of OEA.

**OEA Fuel**

**Figure S10.** Emissions of (a) $SO_2$, (b) $NO_x$, (c) CO, (d) $CO_2$, (e) $PM_{10}$, (f) $PM_{2.5}$, (g) BC, (h) OC, (i) NMVOC, and (j) $NH_3$ from each fuel type in OEA from 1950 to 2015. See Fig. 1 for the definitions of OEA.

**Figure S11.** Emissions of (a) $SO_2$, (b) $NO_x$, (c) CO, (d) $CO_2$, (e) $PM_{10}$, (f) $PM_{2.5}$, (g) BC, (h) OC, (i) NMVOC, and (j) $NH_3$ from major sectors in OSA from 1950 to 2015. See Fig. 1 for the definitions of OSA.

**OSA Fuel**

[Figure]

**Figure S12.** Emissions of (a) $SO_2$, (b) $NO_x$, (c) CO, (d) $CO_2$, (e) $PM_{10}$, (f) $PM_{2.5}$, (g) BC, (h) OC, (i) NMVOC, and (j) $NH_3$ from each fuel type in OSA from 1950 to 2015. See Fig. 1 for the definitions of OSA.

[Figure]

**Figure S13.** Grid maps of annual emissions of (a, be) $CO_2$ (kt year$^{-1}$ per grid cell) and (cb, d) $PM_{10}$ (t year$^{-1}$ per grid cell) in 1965 (left) and 2015 (right).

[Figure]

**Figure S14.** Comparison of (a) CO, (b) NMVOC, (c) NH₃, (d) PM₁₀, (e) PM₂.₅ and (f) OBC emissions in China between REASv3 and other studies. Emissions from domestic and fishing ships were excluded from REAS series, CEDS, and EDGARv4.3.2. Error bars indicate the uncertainty range of REASv3 in 1955, 1985, and 2015.

[Figure]

**Figure S14.** Continued.

[Figure]

90   **Figure S15.** Comparison of (a) CO, (b) NMVOC, (c) NH₃, (d) PM₁₀, (e) PM₂.₅ and (f) OC emissions in India between REASv3 and other studies. Emissions from domestic and fishing ships were excluded from REAS series, CEDS, and EDGARv4.3.2. Note that values of "Pandy+Sadavarte" are calculated from Pandey et al. (2014) and Sadavarte and Venkataraman (2014). Error bars indicate the uncertainty range of REASv3 in 1955, 1985, and 2015.

[Figure]

**Figure S15.** Continued.

[Figure]

**Figure S16.** Comparison of (a) SO$_2$, (b) NO$_x$, (c) BC, (d) CO, (e) NMVOC, (f) NH$_3$, (g) PM$_{10}$, (h) PM$_{2.5}$ and (i) OC emissions in Japan between REASv3 and other studies. Emissions from domestic and fishing ships were excluded from REAS series, CEDS, EDGARv4.3.2, Kannari et al. (2007), and Fukui et al. (2013). Error bars indicate the uncertainty range of REASv3 in 1955, 1985, and 2015.

[Figure]

**Figure S16.** Continued.

[Figure]

**Figure S16.** Continued.

[Figure]

**Figure S17.** Comparison of (a) SO₂, (b) NOₓ, (c) BC, (d) CO, (e) NMVOC, (f) NH₃, (g) PM₁₀, (h) PM₂.₅ and (i) OC emissions in SEA between REASv3 and other studies. Emissions from domestic and fishing ships were excluded from REAS series, CEDS, and EDGARv4.3.2. See Fig. 1 for the definitions of SEA. Error bars indicate the uncertainty range of REASv3 in 1955, 1985, and 2015.

[Figure]

**Figure S17.** Continued.

[Figure]

**Figure S17.** Continued.

[Figure]

120  **Figure S18.** Comparison of (a) SO₂, (b) NOₓ, (c) BC, (d) CO, (e) NMVOC, (f) NH₃, (g) PM₁₀, (h) PM₂.₅ and (i) OC emissions in OEA between REASv3 and other studies. Emissions from domestic and fishing ships were excluded from REAS series, CEDS, and EDGARv4.3.2. See Fig. 1 for the definitions of OEA. Error bars indicate the uncertainty range of REASv3 in 1955, 1985, and 2015.

[Figure]

125    **Figure S18.** Continued.

[Figure]

**Figure S18.** Continued.

[Figure]

**Figure S19.** Comparison of (a) SO₂, (b) NOₓ, (c) BC, (d) CO, (e) NMVOC, (f) NH₃, (g) PM₁₀, (h) PM₂.₅ and (i) OC emissions in OSA between REASv3 and other studies. Emissions from domestic and fishing ships were excluded from REAS series, CEDS, and EDGARv4.3.2. See Fig. 1 for the definitions of OSA. Error bars indicate the uncertainty range of REASv3 in 1955, 1985, and 2015.

[Figure]

**Figure S19.** Continued.

[Figure]

**Figure S19.** Continued.

[Figure]

**Figure S20.** Comparison of trends of (a) CO, (b) NMVOC, (c) $NH_3$, (d) $PM_{10}$, (e) $PM_{2.5}$, and (f) OC emissions in Asia and relative ratios of emissions from China, India, Japan, SEA, OEA, and OSA for (g, m, q) CO, (h, n, r) NMVOC, (i, o, s) $NH_3$, (j, t) $PM_{10}$, (k, u) $PM_{2.5}$, and (l, p, v) OC among (g, h, i, j, k, l) REASv3, (m, n, o, p) CEDS, and (q, r, s, t, u, v) EDGARv4.3.2. See Fig. 1 for the definitions of SEA, OEA, and OSA.

[Figure]

**Figure S20.** Continued.

[revised manuscript text omitted]